

# Bias and error in modelling thermochronometric data: resolving a potential increase in Plio-Pleistocene erosion rate

Sean D. Willett[1], Frederic Herman[2], Matthew Fox[3], Nadja Stalder[2], Todd A. Ehlers[4], Ruohong Jiao[5], and Rong Yang[6]

[1] Department of Earth Sciences, ETH-Zurich, Zurich, 8092, Switzerland

[2] Institute of Earth Surface Dynamics, University of Lausanne, Switzerland

[3] Department of Earth Sciences, University College London, UK

[4] Department of Geosciences Everhard Karls University Tübingen, Germany

[5] School of Earth and Ocean Sciences, University of Victoria, Canada

[6] School of Geosciences, Zhejiang University, China

*Correspondence to*: Sean Willett (swillett@erdw.ethz.ch)

**Abstract.** Thermochronometry provides one of few methods to quantify rock exhumation rate and history, including potential changes in exhumation rate. Thermochronometric ages can resolve rates, accelerations, and complex histories by

exploiting different closure temperatures and path lengths using data distributed in elevation. We investigate how the resolution of an exhumation history is determined by the distribution of ages and their closure temperatures through an error analysis of the exhumation history problem. We define the sources of error, defined in terms of resolution, model error and methodological bias in the inverse method used by Herman et al. (2013) which combines data with different closure temperatures and elevations. The error analysis provides a series of tests addressing the various types of bias, including addressing criticism that

there is a tendency of thermochronometric data to produce a false inference of faster erosion rates towards the present day because of a spatial correlation bias (Schildgen et al., 2018). Tests based on synthetic data demonstrate that the inverse method used by Herman et al. (2013) has no methodological or model bias towards increasing erosion rates. We do find significant resolution errors with sparse data, but these errors are not systematic, tending rather to leave inferred erosion rates at or near a Bayesian prior. To explain the difference in conclusions between our analysis and that of Schildgen et al. (2018), we examine

their paper and find that their model tests contained an error in the geotherm calculation, resulting in an incorrect age prediction. We also found that Schildgen et al. (2018) applied a biased operator to the results of Herman et al. (2013) thereby distorting the original results producing a bias that was falsely attributed to the original inverse model. Our reanalysis and interpretation show that the original results of Herman et al. (2013) are correct and there is no evidence for a systematic bias.



## 1.0 Introduction

Thermochronometry provides one of few methods to quantify rock exhumation histories. Over the last 30 years, it has been extensively applied to understand tectonics and landscape evolution. Part of its success stems from the large number of thermochronometer systems available (Reiners and Brandon, 2006; Reiners et al., 2005) as well as the development of numerical models able to convert thermochronometric data into constraints on cooling associated with exhumation by surface or tectonic processes (e.g., Braun, 2003; Ehlers and Farley, 2003; Ketcham, 2005; Gallagher, 2012; Braun et al., 2012; Willett

and Brandon, 2013; Fox et al., 2014). Where exhumation occurs by surface processes, the exhumation rate is equivalent to a surface erosion rate and we will use these two terms interchangeably in this paper. Models are integral part of thermochronometric data interpretation as they are needed for computing cooling histories from parent-daughter loss relationships with a complex thermal history, as well as for converting cooling histories into exhumation histories. Cooling or exhumation histories provide direct constraints on kinematics or tectonic processes and rates of surface erosional processes in

orogens (e.g., Grasemann and Mancktelow, 1993; Seward and Mancktelow, 1994; Brandon et al., 1998; Batt et al., 2000; Moore and England, 2001; Willett and Brandon, 2002; Ehlers et al., 2003; Lock and Willett, 2008; Campani et al., 2010; Herman et al., 2010; Barnes et al., 2012; McQuarrie and Ehlers, 2015).

Surface erosional processes are closely linked to climate through parameters and processes such as temperature, precipitation and biological activity (e.g., Antonelli et al., 2018; Starke et al., 2020). Temperature determines the dominant

surface processes, for example glacial, fluvial or hillslope, and precipitation often determines efficiency as well as its spatial distribution. Thermochronometry holds the potential to provide a measure of past erosional conditions and therefore how these conditions may be related to past climate, and how the coupled system has evolved (e.g., Shuster et al., 2005, 2011; Burbank et al., 2003; Reiners et al., 2003; Wobus et al., 2003; Vernon et al., 2008; Thomson et al., 2010a; Thiede and Ehlers, 2013; Fox et al., 2015, 2016; Herman and Brandon, 2015).

One of the fundamental questions of paleoclimate and its links to Earth processes is how the onset of Quaternary glaciations affected solid Earth processes including surface erosional efficiency. Glaciers can be very efficient agents of surface erosion, and regions of heavy continental or Alpine glaciation have clearly been morphologically modified. Whether geomorphic change has a corresponding change in net erosion rates is a more difficult question, although support for a Quaternary increase in global erosion rate comes from sedimentological evidence (Molnar and England, 1990; Zhang et al.,

2001; Molnar, 2004). A second open question is whether regions that have experienced Quaternary climate change, but have remained too warm for active glaciation, have also experienced an increase in average erosion rate, for example through increases in precipitation or climate variability (Zhang et al., 2001; Molnar, 2004). Controversy surrounds this question in part because many orogens experienced a decrease in precipitation in the Quaternary due to cooler air temperatures holding less moisture during glacial periods (e.g., Mutz et al., 2018; Mutz and Ehlers, 2019)

Although numerous single site thermochronometry studies have shown increases in Pliocene or Quaternary erosion rates (e.g., Zeitler et al., 1982; Tippett and Kamp, 1993; Farley et al., 2001; Shuster et al, 2005, 2011; Berger et al., 2008a,b,c; Vernon et al. 2008; Glotzbach et al., 2011; Valla et al., 2011, 2012; Sutherland et al., 2009; Thomson et al., 2010a,b; Avdeev





et al., 2011; Thiede and Ehlers, 2013; Fox et al., 2015, 2016; Michel et al., 2018), the first attempt to conduct a global scale analysis was carried out by Herman et al. (2013). Herman et al. (2013) compiled about 17,000 thermochronometric ages from

around the world primarily from four low-temperature thermochronometric systems, apatite and zircon (U-Th)/He (AHe and ZHe), and apatite and zircon fission track dating (AFT and ZFT)). In some cases, they augmented these systems with higher closure temperature thermochronometers. These data were interpreted to quantify the exhumation rate history of select regions distributed globally. A key objective of that study was to apply a uniform treatment of the data using a transient thermal model embedded in a Bayesian inverse model (Fox et al., 2014). Using this approach, Herman et al. (2013) found that of sites that

had enough thermochronometric data to resolve a recent change in erosion rate, over 80% were dominated by an increase in erosion rate over the past 4 Myrs. These sites included both tectonically active and inactive settings, glaciated and unglaciated regions and locations in the northern and southern hemispheres. Although Herman et al. (2013) did not attempt to attribute cause to any single locality, they argued that the strong skew towards an increase in erosion rate at such varied localities supported a common global phenomenon, mostly likely related to climate change (Zhang et al., 2001).

However, a recent paper by Schildgen et al. (2018) challenged this conclusion, arguing that the Bayesian inversion method employed by Herman et al. (2013) incorrectly interpreted spatial variability as temporal variability, resulting in a systematic bias towards an apparent temporal acceleration in exhumation. This contention suggests that not only are the conclusions of Herman et al. (2013) incorrect, but also calls into question the results of numerous other studies using the same method.

The objective of this paper is to test the hypothesis of Schildgen et al. (2018) that the interpretations of Herman et al. (2013) contain a bias. We test this hypothesis by conducting a complete error analysis of the method of Herman et al. (2013). The current manuscript is structured into three main parts. First, we provide a review of concepts associated with bias and resolution inherent to thermochronometry, as well as to methods of data treatment, including the method proposed by Herman et al., (2013) and Fox et al., (2014) (Sections 2 and 3). This is necessary to explain potential sources of error and how to test

for them. Second, we conduct a series of tests based on synthetic data and a specified, therefore known, erosion rate history in order to isolate and identify the sources and magnitude of errors (Section 4). Third, we conduct an interpretation and assessment of selected examples from the original Herman et al. (2013) results in order to explain which data were responsible for resolving the various erosion rate histories, and how these were averaged in space and time (Section 5). To summarize our conclusions, we find that our synthetic data tests confirm the initial analyses (Fox et al., 2014) which showed no significant spatial averaging

errors. Subsequently, we review the analysis of Schildgen et al. (2018) to identify the source of the discrepancy, and determine that Schildgen et al. (2018) made a series of self-reinforcing errors that combined to create the appearance of a bias in the original analysis of Herman et al. (2013) that does not exist.



## 2. Bias and Resolution in Thermochronometry-Derived Exhumation Rates

### 2.1 Resolution of erosion rate from thermochronometric ages

The problem of determining an exhumation history from thermochronometric ages is relatively simple. Expressed in terms of a closure temperature, $T_c$, which is an approximation to temperature-dependent diffusional loss of a daughter product, and knowing the geotherm, $T(z)$, the rate of exhumation, or erosion rate, $e$, can be expressed as:

$$\int_0^\tau e(t)\, dt = z_c, \tag{1}$$

where $\tau$ is the measured age and $z_c$ is the depth to the closure temperature below a sample in one dimension. Complications to
this approach arise from transient geotherms and the cooling rate dependence of the closure temperature (e.g., Graseman and Mancktelow, 1993; Mancktelow and Grasemann, 1997; Batt and Braun, 1997; Harrison et al., 1997; Moore and England, 2001; Braun et al., 2006; Reiners and Brandon, 2006; Willett and Brandon, 2013), but the principle holds for most cases of monotonic cooling. A single age can resolve only a single rate of cooling between its time of closure and sampling at the surface. Resolving more complex (e.g., transient) exhumation histories requires more information than a single age.

Several methods have been proposed for calculation of either cooling rates or exhumation rates from thermochronometric data. The simplest method is by using the relationship between age and elevation (Figure 2a) (Wagner and Reimer, 1972; Wagner et al., 1979; Parrish, 1983; Fitzgerald and Gleadow, 1988; Fitzgerald et al., 1995). This relationship exploits the fact that path length from the depth of closure to the surface increases with elevation. Provided that the closure isotherm is horizontal, and the exhumation rate is spatially constant over the sampling domain, the relationship is monotonic
with the slope giving the exhumation rate. Without a vertical wall, sampling across elevation also requires sampling horizontally, so an important assumption is that sample points are closely located in space, typically within a few kilometres of each other, so that exhumation rates and the depth to the closure isotherm do not differ between points. Application of this method also requires that surface relief has not significantly perturbed the depth to the closure isotherm (e.g., Lees, 1910; Stüwe et al., 1994; Mancktelow and Graseman, 1997; Braun 2002a, b; Ehlers, 2005).

The second method for estimating exhumation rate is by calculation of a cooling rate using more than one thermochronometric system (e.g., Harrison et al., 1979; Dodson and McClelland-Brown, 1983; Hurford, 1986). This method has a long history of use for metamorphic cooling histories in orogenic belts, where it is referred to as the two mineral, or mineral pair method and has identical application to low-temperature systems and erosional cooling to the surface. By using two thermochronometric systems and taking the difference between closure temperatures and ages, one can calculate a cooling
rate (Figure 2b). To convert the cooling rate to an exhumation rate requires a thermal model or knowledge of the geotherm, and if the geotherm is not linear, variations in cooling rate may not have corresponding changes in exhumation rate. For example, if the geotherm is convex upward due to upward advection of heat, a constant rate of exhumation may manifest itself as an increase in cooling rate as rock nears the surface (as in Figure 2b). Cooling rates should thus never be directly converted to exhumation rates, without considering variations in the thermal gradient with depth (Moore and England, 2001). Different



closure temperatures are used to resolve the time taken to pass from one closure depth to another closure depth or from the shallowest closure depth to the surface (Reiners and Brandon, 2006). For example, with an apatite fission track age and a (U-Th)/He age obtained from a single sample, two exhumation rates can be obtained, one from the difference in $z_c$ for closure of the fission track and (U-Th)/He systems and a second based on the passage from the closure depth for the (U-Th)/He age to the surface. An independent cooling rate can be determined for each additional thermochronometric system, provided the

closure temperatures are different.

Resolution of an exhumation rate history from thermochronometric ages is thus determined by the range of ages, the number and range of the closure temperature and the distribution of elevation of samples (Figure 1). This description of resolution differs from other problems in parameter estimation in that the resolution is determined by the value measured, i.e., the age, in addition to the location of a measurement. The estimation problem is to determine an exhumation rate in both space

and time. However, the ages measured at a particular point in space determine how much time can be resolved and the number of thermochronometers determines how many time intervals can be resolved. Age-elevation data extend the information considerably as they permit a single thermochronometer to resolve a time interval over the age range of the data.

A third method for determining a cooling or exhumation rate uses a thermal model that calculates the thermal history of exhumed rocks by solving the advection/diffusion equation for prescribed parameters (e.g., rate of exhumation,

thermophysical properties, and boundary and initial conditions). Application of a thermal model permits estimation of the geotherm, allowing multiple thermochronometers and elevations to be combined onto a single age-elevation relationship, greatly increasing the time span and details of the exhumation history resolved by data (Figure 2c) (Reiners and Brandon, 2006).

## 2.2 Spatial correlation bias

In order to increase the applicability of thermochronometers, it is necessary to amalgamate data spatially. Even an age-elevation profile is a collection of data across some spatial region. Important questions in these methods are over what distance is this valid, and are there unintended consequences to spatial averaging? In particular, it has been suggested that spatial variations in age can be mistakenly combined to produce an inference of temporal change. For example, if elevation and erosion rate covary over some region, amalgamating ages might produce a linear elevation trend, but not one that reflects

the average erosion rate. With spatial variation in erosion rates, it is possible to combine the ages in a way that would mimic an increase in erosion rate with time (Willenbring and Jerolmack, 2016), a point taken up by Schildgen et al. (2018). However, there are an unlimited number of ways in which thermochronometric ages can be averaged or combined to construct an erosion rate history, so it is important to evaluate methods to establish potential bias from specific assumptions.

To frame this problem, we show a thought experiment in Figures 2d, e, similar to the one set up by Willenbring and

Jerolmack (2016) and Schildgen et al. (2018). Figure 2d shows ages from 3 thermochronometers, exhumed at a constant rate, and set at 3 different closure depths. We also show two other sets of ages, each of which experienced exhumation at a rate different from the average by a value of $\sigma$, but in opposite directions. These data thus represent 3 regions, each exhuming at a



different, but steady rate, and having a common, regional mean value that is also steady in time. For simplification, we will keep the closure depth constant and assume no elevation differences for the ages, although a range in elevation acts much like

ages with independent closure temperatures. The ages define 3 lines, the slope of each gives the local exhumation rate (Figure 2e). Willenbring and Jerolmack (2016) (their Figure 3) defined a deviation in age, rather than exhumation rate, but their example is otherwise equivalent. Schildgen et al. (2018) presented a similar example (their Figure 2), but included only one thermochronometric system, and applied arbitrary elevations and geothermal gradient to each age, so their plot shows three independent effects with only five points.

There are many ways in which these data can be analysed and erosion rates derived, mostly depending on how much variability in the exhumation history one attempts to resolve and with what spatial resolution. If one recognizes that these data come from three independent areas, data from each area, i.e., a,b,c or d,e,f, or g,h,i define an exhumation history that is steady in time and any regression of age and depth to the closure isotherm gives the correct erosion rate value, with temporal resolution controlled by how many data are available at each location. If one fails to recognize that these data have independent

exhumation rates, they will be averaged, but there are many ways that this can be done. The simplest way is to average the rates from all 9 points shown as *Average 1* in Figure 2b. This is an unbiased, i.e., correct, measure of the regional median erosion rate, but with no time resolution. To resolve time variation requires averaging subsets of the data, for example, by taking a regression of the data in Figure 2d with a moving window or with a piecewise constant slope. Willenbring and Jerolmack (2016) argued that there is a limit to resolution at short times (shown as a fine dashed line in Figure 2d,e) so that

the region below this limit is unsampled and therefore introduces bias into the average. They calculated the average exhumation rate as the expected value of the measured rates, integrated at a fixed time, but excluding the region below this limit; their average is shown as *Average 2* on Figure 2e. Schildgen et al. (2018) showed a similar example in their Figure 2c; they showed data from regions with independent erosion rates, selecting points equivalent to data a, b, c, h, and i in our Figure 2d. However, they seemingly adjusted either the geothermal gradient or the closure temperature until points h and i were nearly collinear

with point c. They gave no explanation for this adjustment in closure depth, but the change removes the generality of the example and obscures the difficulties of obtaining a monotonic exhumation function from data distributed as in Figure 2d.

Using a constant time window to average rates (Figure 2d), or to regress closure depth against time (Figure 2d) presents some serious statistical problems. First, time is not the independent variable in this problem. The time information is entirely from measured thermochronometric ages, which makes time a dependent variable, so one should not regress depth

against age. In fact, if one regresses age against depth (Figure 2d) with a moving depth window, one obtains the correct, unbiased regional mean erosion rate (Figure 2e, *Average 1*). This is equivalent to using the ages corresponding to a common closure temperature to determine the average time of passage through the closure temperature (Figure 2f). If all regions are sampled adequately, this is the correct average time and the regional erosion rate is recovered with no false increase in erosion rate. Any change in the average erosion rate will be detected with resolution determined by the number of closure depths.

A second problem is that Willenbring and Jerolmack (2016) misinterpreted the meaning of the limit to resolution, or what they called "precision". They took this concept from measurements of sediment layer thickness, which can be biased by



the inability to measure layer thicknesses below a specific precision (Anders et al., 1987). Thermochronometric data are fundamentally different in that an age does not represent an estimate of rate at a point or across a closed interval of time such as represented by a stratigraphic layer, but rather, represents the integral of the exhumation rate (Equation 1) from the time of

closure to the present day. Thus, rather than being unsampled, the region left of the resolution limit in Figure 2e is actually the most heavily sampled part of parameter space. The better definition of the "limit to resolution" is that no change of exhumation rate can be resolved between this limit and time zero, because there is no sampling internal to this part of the parameter space, but the average rate across this interval is sampled and can be determined.

Recognizing the integral nature of a thermochronometric age, to determine a regional median, all data older than a

time of interest should be included in the determination of the rate. For example, Schildgen et al. (2018) presented a second averaging method that correctly recognized this integral nature of a thermochronometric age, and so calculated the mean exhumation rate at a point in time, including all ages older than that point, but excluding all ages younger than the integration point. Applying this method gives *Average 3* in Figure 2e. This average is also biased with respect to the regional average, but it is biased downward, and only at times early in the exhumation history. Times closer to the modern have the least bias from

the regional average. The bias is due to the loss of the information contained in the youngest ages as they are systematically excluded from the moving average. In practice, the rate of increase of this bias will depend on the number of higher-temperature thermochronometric ages in the analysis. Schildgen et al. (2018) used only a single thermochronometer in their example, so the bias in their example appears large, but had they used a more realistic case with multiple systems, or even more ages, the young part of the history would be only slightly biased.

The third problem with the type of analysis shown in Figure 2 is that it ignores the constraints placed on young erosion rates by older ages and on older erosion rates by young ages. If a suite of ages is inappropriately averaged, it very quickly leads to cases where it is impossible to fit all data. For example, if data with a common closure temperature but variable ages are combined, a physical model, based on exhumation down a temperature gradient, cannot fit all the data. If an old age is fit well, the younger age cannot be fit simply by increasing the cooling rate. The sample has already closed with respect to that

chronometer and as an integral quantity, it cannot go back and close again. Any good statistical or inverse model will penalize this scenario as indicating a problem, whereas the simple moving average models do not do this. For example, if one takes data points b, e, and h in Figure 2e and interprets these as indicating a progressively higher erosion rate, the resultant temperature history can fit only one of these ages. If the youngest age, b, is fit, the exhumation history will badly overpredict the temperature at the time of closure for points e and h. Where there is a common history, and ages with multiple closure

temperatures, all ages can be fit, but the cooling history of higher temperature systems must be consistent with the time of passage through the lower closure temperatures. Methods that average rates without regard for the temperatures predicted by those cooling rates will badly misfit ages. For example, *Average 2* and *Average 3* in Figure 2e do not satisfy this condition, so would provide poor predictions of the ages that they are intended to fit.

What should be clear from this exercise is that bias is not inherent to thermochronometric data, nor is it an inevitable

consequence of spatial averaging. Bias is inherent to the method used to analyse the data. For the data distribution of Figure





2d, an attempt to resolve a time-variable erosion rate using one of the methods proposed by Willenbring and Jerolmack (2016) or Schildgen et al. (2018) will give a biased result. The same data analysed by the method shown in Figure 2f gives an unbiased result.

It should also be clear that the problems with bias often arise from trying to derive too much information from too

few data. If an analysis attempts to derive a multistep cooling history from ages using a single thermochronometric dating method and ages have no path length differences from elevation, it will fail. If an analysis is used to derive a regional exhumation rate, constant over space and time, any number and value of ages can be used and most analyses will yield an unbiased estimate of the average rate. This is a classic trade-off problem in estimation theory. The number of data and their distribution across space and time must be matched to the complexity of the exhumation history that is to be inferred. In other

words, the distribution of the data determines the complexity of the solution that can be resolved as well as any potential bias. Analysis of bias cannot be separated from the questions of resolution.

Both Willenbring and Jerolmack (2016) and Schildgen et al. (2018) papers offered a critique of the use of thermochronometric data to sample and differentiate between spatial and temporal variations in erosion rate. In particular, they were critical of the approach used by Herman et al. (2013), with the suggestion that Herman et al. (2013) had, perhaps

inadvertently, used a biased method. However, each paper provided only examples, similar to those in Figure 2, without demonstrating that Herman et al. (2013) reverted to one of these biased methods, rather than one of the alternative, and unbiased, methods. In the following sections, we present a full bias and resolution analysis to address this question.

### 3.0  GLIDE Inversion Method and Error Analysis

The method of Fox et al. (2014), which we present in the following sections, was designed to combine the principles

of Figures 2a, 2b and 2c and was used in Herman et al. (2013). This approach is described in brief in this section, along with an analysis of the sources and propagation of error in the method. Finally, we examine the potential sources of error in post-processing treatment of the parameter estimates.

### 3.1 Inversion of spatially distributed data

Fox et al. (2014) introduced a method to invert spatially distributed thermochronometric data into maps of exhumation

rate histories. The method includes a thermal model to predict temperature, closure depths and thermochronometric ages, an inversion scheme based on inverse problem theory which minimizes Gaussian errors in observations, and a Bayesian parameter model in which model parameters are assumed to have Gaussian distributions about a prior value (e.g., Franklin, 1970; Jackson, 1979; Tarantola and Valette, 1982; Tarantola, 2005). This method was implemented in the program called GLIDE (Fox et al., 2014).

The method is based on a forward model to predict temperature, thermochronometric ages and closure depths. This includes a numerical solution to the transient advection-diffusion equation, including the upward advection due to erosion.



The solution is based on a Crank-Nickolson finite difference method to solve the advection-diffusion equation assuming a fixed temperature condition on the Earth's surface and a fixed heat flux applied at the base of the thermal lithosphere (Fox et al., 2014). The numerical solution is one-dimensional, but includes the 3-D lateral heat flow induced by surface topography

(Lees, 1910; Birch, 1950; Stüwe et al., 1994; Mancktelow and Grasemann, 1997; Braun 2002a,b), which is applied through a spectral method to solve for an equivalent flat surface with variable temperature. The second part of the forward model consists of deriving closure temperatures, which are then related to corresponding closure depths using the thermal model. Closure temperatures are estimated using the approach proposed by Dodson (1973), kinetic parameters for the Dodson equation from the literature (e.g., Reiners and Brandon, 2006), and a cooling rate predicted from the internal thermal model.

To simplify the estimation problem, Fox et al. (2014) cast the forward problem in a linear form, which is possible if the data are defined in terms of depth, rather than time. As a linear problem, it lends itself well to least-squares optimization methods (Legendre, 1805). This technique has been often used and significantly developed in geophysics, and it has been applied to complex problems (e.g., Franklin, 1970; Lawson and Hanson, 1974; Jackson, 1976, 1979; Menke, 1984; Tarantola and Valette, 1982; Tarantola, 2005). In the simple, linear form, one equation is defined for each age, but with the full timespan

resolved by the age broken into multiple time intervals. The exhumation rates for these time intervals are the unknown parameters. This results in an underdetermined problem, in that there are multiple time intervals but only a single age at each point. For a single age, the forward model is expressed as:

$$Ae = z_c \tag{2}$$

where $\mathbf{A}$ is simply a vector containing multiple entries of the time interval length, with enough of them to sum to the age,

including a partial time interval (e.g., Fig 1), $e$ is a vector of unknown exhumation rate values over those intervals and $\mathbf{z_c}$ is a vector for the depth of the closure temperature as determined from the thermal model. The inclusion of the thermal solution makes the problem nonlinear, as the advection term of the temperature equation includes the erosion rate inferred from the data. However, because the geotherm is always monotonic, this nonlinearity is weak and convergence occurs rapidly with a direct iteration, particularly if erosion rates do not vary much from the prior estimate which is used as the starting condition.

The measured age appears in the problem only as the sum of the vector components of $\mathbf{A}$. To address the argument made above that a single age can only resolve a single time interval, we include a probability constraint on each parameter in the form of Bayesian prior information included as a Gaussian prior model, comprising an expected mean value and variance for each exhumation rate on each time interval. For the simple, single age problem, this has the effect of providing equal weighting to each time interval so that the value of $e$ for each time interval is equal and the multiple time intervals act as a

single time interval. For a multiple age problem, each age provides one row vector that is combined into a matrix $A$, where we link all the individual solutions by assuming a spatial correlation structure that links the exhumation rate for a specific time interval, but retains no correlation across time. With the Bayesian prior constraints, the linear optimized solution is defined as

$$e_{post} = e_{prior} + (CA^t (ACA^t + C_\epsilon)^{-1}) (z_c - Ae_{prior}) \tag{3}$$



where $\mathbf{A}$ is a matrix that includes a row for each age, $\mathbf{C}$ is the prior covariance matrix, or model covariance, that encapsulates
the spatial correlation structure, $\mathbf{C_\epsilon}$ is the data covariance (which is a diagonal matrix that represents errors on the measured
ages, converted to an equivalent uncertainty on closure depth), $\mathbf{z}_c$ is a vector of the sample closure depths, $\mathbf{e}_{prior}$ is a vector
that represents the mean prior exhumation rate, $\mathbf{e}_{post}$ is the inferred, i.e., posterior, exhumation rate, which is a vector of
estimated exhumation rates at each data location for each time interval in $\mathbf{A}$. The exhumation rates are shown as maps of $\mathbf{e}_{post}$
during each time interval.

From Equation 3, one can define the 'inverse operator', $\mathbf{H}$, as

$$\mathbf{H} = \mathbf{CA^t} \left(\mathbf{ACA^t} + \mathbf{C_\epsilon}\right)^{-1} \tag{4}$$

and each element of $\mathbf{C}$ is given by

$$C(i,j) = \sigma^2 exp\left(-\frac{d_{ij}}{L}\right) \tag{5}$$

where $\sigma^2$, $L$ and $d_{ij}$ are the prior variance, the correlation length scale and the separation distance between samples,
respectively. These parameters, together with the mean erosion rate for each parameter, constitute the prior model that must
be specified as part of the inverse model construction.

This inverse model formulation thus treats each age as an independent estimate of the closure depth at a single point.
If two or more ages from different thermochronometers are collocated, they independently constrain their corresponding
closure depth (Figure 2b). Elevation is taken into account as data through its effect on the distance between the closure depth
and the sample location (Figure 2c). Because there is an underlying thermal model, the relationship between erosion rate,
exhumation path and predicted age obeys the physical constraint that a rock can only pass through its closure temperature only
once. The imposed spatial correlation on erosion rates links the temperature solutions in space, thereby allowing nearby data
to be combined either as multiple realizations of the same closure depth, or, if from different thermochronometers, as estimates
of different closure depths, thereby resolving a variable-rate cooling history.

**3.2 Error Analysis**

There are a number of standard methods to assess the quality of a solution. The two metrics used by Fox et al. (2014) are the
posterior variance and the resolution. The posterior covariance matrix is given by

$$\mathbf{C_{post}} = \mathbf{C} - \mathbf{HAC} \tag{7}$$

and provides a measure of how the uncertainty on a parameter is reduced from its prior value. The prior is regarded as an
estimate of the parameter, so following the addition of data, the uncertainty, expressed through this variance, must be lower.
The diagonal terms of $\mathbf{C_{post}}$ are the variance of a given parameter, with off-diagonals giving the covariances. As a quality
measure, we will show the normalized posterior variance, which is the posterior variance normalized by the prior variance, so
that the mapped quantity for the i[th] parameter is:





$$(8)$$


The other quality metric, resolution, needs more explanation. If we take the true solution, the forward model (Equation 2), and assume that there is uncorrelated measurement noise associated to each datum, the depth to the closure temperature can be expressed as:

$$\mathbf{z_c} = \mathbf{A}e_{true} + \boldsymbol{\varepsilon},$$

where $\boldsymbol{\varepsilon}$ is a vector of noise values, and $e_{true}$ are the true values of the parameters.

With the inverse operator, $\mathbf{H}$ (Equation 4), the posterior estimate of the parameters from Equation 3 in terms of the true parameters is given as:

$$e_{post} = e_{prior} + \mathbf{HA}(e_{true} - e_{prior}) + \mathbf{H}\varepsilon,$$

or, subtracting the true parameters from each side, we obtain the error in the estimate as:

$$e_{post} - e_{true} = (\mathbf{HA} - \mathbf{I})(e_{true} - e_{prior}) + \mathbf{H}\varepsilon, \qquad (9)$$

where $\mathbf{I}$ is the identity matrix. There are two components to the error. The second term, $\mathbf{H}\varepsilon,$ represents the propagation of noise into the estimate. The first term is what is referred to as the 'resolution error' or 'resolution bias' in the parameter estimate (Jackson, 1979). The closeness of $\mathbf{HA}$ to the identity matrix determines the quality of the estimate. If $\mathbf{HA}$ is equal to $\mathbf{I}$, and the noise is negligible, the estimate is equal to the true parameters. If $\mathbf{HA}$ is null, or at least has all terms much less than 1, the

resolution goes to zero and $e_{post} = e_{prior}$, meaning the new, posterior estimate is the same as the prior value; we have not added information by including the data.

The matrix quantity, $\mathbf{HA}$, is thus fundamental to assessing resolution errors, and this is referred to as the resolution matrix:

$$\mathbf{R} = \mathbf{HA}. \qquad (10)$$

The resolution matrix represents how effectively the model can be inverted to recover the true parameters. Failure to recover the true solution is bias, which, if we neglect the error due to noise in the data, can be written as:

$$e_{post} - e_{prior} = \mathbf{R}(e_{true} - e_{prior}). \qquad (11)$$

As with the error expression above, if $\mathbf{R}$ is the identity matrix, every parameter is perfectly resolved. However, this will never be the case, nor is it even ideal. There remains the second term in Equation 9 for the propagation of noise, $\mathbf{H}\varepsilon$. To minimize

this term, $\mathbf{H}$ should be as small as possible and it is not possible to satisfy both $\mathbf{HA} = \mathbf{I}$ and $\mathbf{H} = 0$. Furthermore, as posed, the thermochronometry problem is massively under-constrained, and the only way to get a meaningful solution is to add additional information inherent to age-elevation relationships and multiple thermochronometers by spatial averaging of nearby data.



The resolution matrix has dimensions of *(m x m)*, where *m* is the number of parameters in the problem, i.e., the number of time intervals sampled by data times the number of data. With reference to Equation 11, there is one row corresponding to each parameter estimate. This row has *m* entries, each entry corresponding to one of the erosion rates at some other location and/or another time interval. Again, if that row has only the diagonal value equal to 1 and the remainder of the row were equal to zero, the estimate would be exactly correct. The other entries in a row corresponding to erosion rates within the same time interval, but at other spatial points, comprise a set of weighting factors that define the spatial averaging for that parameter. This corresponds to a spatial resolution kernel (Backus and Gilbert, 1968). The remaining row entries give the contributions from other time intervals, either at the same data location or at other locations. These values give a type of leakage of information across time intervals, or what Ory and Pratt (1995) referred to as contamination kernels. Both space and time averaging are unavoidable and not necessarily bad, but both kernels should be tracked. The spatial resolution kernel is relatively simple as it is dominated by local data, but supplemented by data within the spatial correlation structure. Resolution kernels typically resemble the correlation function, which falls off exponentially. The contamination kernels are much more complex as they reflect the number of data and the ages and thus the time intervals sampled. To simplify portrayal of these large matrices, Fox et al. (2014) suggested integrating over the spatial dimension. This collapses the resolution kernel onto the parameter point, while keeping the area of the kernel as the value of what we called the temporal resolution. The quantity mapped by Fox et al. (2014), Herman et al. (2013) and in this paper is thus not the diagonal of the resolution matrix, **R**, but this integrated value. As such, it does not give a measure of the spatial resolution, only a sense of the temporal resolution, that is, how well the time interval is resolved in the neighborhood of the point of interest, and how much contamination enters from the other time intervals.

Another important point regarding models with poor resolution is the nature of the bias. Again, this depends on the resolution matrix. The parameter estimate is given by Equation 3 as:

$$\boldsymbol{e_{post}} = \boldsymbol{e_{prior}} + \mathbf{H}(\mathbf{z_c} - A\boldsymbol{e_{prior}}),$$

which implies that the prior information is being treated as data. Expanding, it can be expressed in terms of the resolution matrix as

$$\boldsymbol{e_{post}} = \mathbf{H}\mathbf{z_c} + (\mathbf{I} - \mathbf{R})\boldsymbol{e_{prior}}. \tag{12}$$

This shows that the parameter estimate is a weighted average of the data and the prior model, with the weights dependent on the resolution. At high resolution, **R** is close to **I** and the prior plays no role. At low resolution, **R** is close to zero, **H** is close to zero, and the estimate will go to the prior.

Perhaps the simplest illustration of how resolution in time reflects parameterization is the effect of time interval length. With reference to Figure 1, consider the case of inversion of a single age with no neighbours. If the age falls within the youngest time interval, the resolution of that time interval will be equal to 1.0. For example, with time intervals of 10 Ma and an observed age of 8 Ma, the time interval of 10 to 0 Ma at that location will be well resolved and equal to 1. If, however, the same age is inverted with a parameterization including a time interval length of 5 Ma, there are now two relevant time intervals





(0 to 5 Ma and 5 to 10 Ma) and the resolution of exhumation in each time interval will be 0.5. The estimated exhumation rate with each parameterization will be identical; only the resolution value will change. A similar result is obtained with larger inversions involving spatial correlation. A larger correlation length will always increase the numerical values of resolution, whereas the time interval number will always decrease the resolution. This does not imply that the solution is "better" as the

correlation length increases, nor is it because more data are averaged. It is simply reflecting the fact that the true parameter is no longer a local exhumation rate, but is a regional mean, and is therefore easier to resolve, much as the standard error on the slope of a linear regression is always smaller than the standard deviation of the data about the regressed line.

In addition to the resolution errors and the noise propagation errors, there is a third type of error in inverse models, model error. This refers to errors in the model parameterization and its lack of ability to represent the physical processes giving

rise to the data. Model error is perhaps the most difficult error to estimate as it emerges as the result of complexities in the real world that are intentionally simplified or ignored to make the problem tractable. Often these complexities are unknown, so the model error cannot easily be treated through error analysis. For example, if we build a thermo-kinematic model with a structural block that is uplifting with a constant rate, but in reality, the region of the model has three independent blocks with different uplift rates, our model has error. We might obtain the correct average uplift rate with our single-block model, but it is

impossible to find 3 independent uplift rates with 1 parameter, so our model is biased. Models constructed so as to have a small number of parameters, e.g., a few tectonic blocks, with fault orientations specified, etc., tend to have a large bias because they cannot express the generality of the behaviour needed to fit to data. Underdetermined models, such as implemented in GLIDE, where there are more parameters than data (each data point has several time intervals contributing to its value) tend to have little or no model error from the spatial parameterization. However, model error can arise from other sources, such as the

thermal model.

There are a number of potential model errors in the thermochronometry inversion problem. First is the thermal physics underlying the geotherm calculation. Boundary conditions and initial conditions for the geotherm are important unknowns, but are specified, not parameterized, so if specified wrongly they can introduce model error. The effect of advection on a geotherm is included, but this is also unknown for the parts of the space and time domain where the upward velocity is not resolved by

age data. A second model error comes through the kinetics of thermochronometric closure to daughter product loss. We have used a closure temperature formulation, which is an approximation. Closure temperature has implicit assumptions of monotonic cooling and first-order Arrhenius kinetics. Fission track annealing can be approximated by first-order kinetics, but not particularly well, and this introduces error. The third potential model error is the statistical spatial correlation model implicit to GLIDE. Spatial correlation imposes a smoothness in space on the erosion rate and this could fail to capture all the variability

in nature and thus introduce bias into the solution. This correlation bias is, however, not a simple model error, as it is a "soft" constraint. Smoothness is only through a correlation, not a parameterization, so it must be balanced against fitting the data. Depending on the number and quality of the data, where quality is relative to the strength of the spatial correlation, specified by the prior parameter variance, smoothness may or may not be forced on the solution. This is a much weaker constraint than, for example, in, thermo-kinematic models with rigid blocks, where the parameterization imposes equivalence instead of





correlation within each block. The model bias is thus linked to the resolution bias and they must be analyzed as a coupled
system.

    To summarize for purposes of our analysis in the following section, the complete error has 3 components, model
errors, resolution errors, and random noise errors,

$$e_{post} - e_{true} = \varepsilon_M + \varepsilon_R + \varepsilon_N.$$    (13)

The first two, model and resolution errors, constitute bias and the third is the stochastic noise propagation. The latter two are
defined in Equation 9. The first two will be analysed in section 4 of this paper.

### 3.3 Errors in Post-Estimation Parameter Analysis

    Results of the inversion method described above are typically presented as maps of erosion rate across particular time
intervals along with the corresponding normalized posterior variance on the exhumation rate and the diagonal value of the
resolution matrix, integrated over the spatial dimension as described above. In many cases, these model results are subjected
to additional analysis to increase their utility, and it can be important to document the propagation of error through any post
inversion analysis.

    For example, the study of Herman et al. (2013) was directed towards the question of erosion rate change during the
Quaternary, so results were summarized by taking the ratio of values across two time intervals, 6 to 4 Ma and 2 to 0 Ma.
Herman et al. (2013) intended this to be only an illustrative presentation of the results. However, this analysis was generalized
by Schildgen et al. (2018) into a metric that they called 'normalized erosion rate difference', denoted **NR** in this paper, such
that

$$\textbf{NR} = \frac{(e_2 - e_1)}{max(e_1, e_2)} \ ,$$    (14)

where $e_2$ is the erosion rate in the 6-4 Ma time interval and $e_1$ the final erosion rate (i.e., 2-0 Ma), which they then presented in
map form for both models and field studies. For this to serve as an interpretation tool, it is important that the error and resolution
analysis are propagated through this operation, otherwise this information is lost. A ratio of two variables containing error is a
biased operation that will not necessarily return the expected value if there is variance in the quantities, so both bias and
variance should be tracked. To demonstrate why a ratio is biased, assume that $e_2$ is larger than $e_1$, let x=$(e_2-e_1)$ and y=$e_2$, the
expected value of the ratio is defined as the expected value of the product:


$$E\left\{\frac{x}{y}\right\} = E\{x\}E\left\{\frac{1}{y}\right\}.$$

However, this is not equivalent to the ratio of the expected values of the terms, i.e.,

$$E\left\{\frac{x}{y}\right\} \neq \frac{E\{x\}}{E\{y\}}.$$    (15)



The erosion rates that come from the inversion are uncertain, that is they contain errors. Even if these errors are unbiased and random, if one takes the ratio of the estimated erosion rates, or the **NR** as in Equation 15, this is not the expected value of the ratio, thereby introducing bias. The bias is difficult to calculate as it depends on both the erosion rates and their variances and covariances, which are not negligible, but the expressions and some examples can be found in Marsaglia (1965). Importantly, the bias on the ratio is always towards higher values and becomes larger as the denominator approaches zero. This is intuitive

if one considers error in the denominator of a ratio. Because a $1/y$ function is nonlinear, going to infinity as $y$ goes to zero, a perturbation towards a positive value has a smaller effect than a perturbation towards a smaller, i.e., closer to zero, number. The effect becomes larger as the mean $y$ becomes smaller. The overall ratio is thus biased towards high numbers, with an effect most pronounced where $y$ is small.

In the context of the changing erosion rate problem this means that the normalized erosion rate difference is

introducing a bias towards an erosion rate increase, which is not a good characteristic for a metric when the point of the exercise is to search for bias towards erosion rate increases in the inversion methodology. Nonetheless, we will calculate this quantity, **NR**, throughout this paper for direct comparison to the Schildgen et al. (2018) analysis. However, we will provide a corresponding quality measure by propagating the posterior parameter variance into this metric. This can be done as follows. Given 2 Gaussian, correlated variables, $E_1$ and $E_2$, with expected values, $e_1$ and $e_2$, variances, $\sigma_1^2$, $\sigma_2^2$, and covariance, $\sigma_{1,2}$,

one can find an approximation of the variance and covariance of a function of the variables from the Taylor Series expanded about the expected value of the function. Provided that the expected value of the function is the function evaluated at the expected value of the variable, the variance is approximated by

$$\sigma^2 = E\{[f_2'(e_2, e_1)(E2 - e_2) + f_2'(e_2, e_1)(E1 - e_1)]^2\}$$
$$= f_2'^2(e_2, e_2)\sigma_2^2 + 2f_2'(e_2, e_1)f_1'(e_2, e_1)\sigma_{1,2}^2 + f_1'^2(e_2, e_1)\sigma_y^2$$

We will drop the maximum value evaluation for simplicity, using only the second value in the denominator and use the function:

$$f(E2, E1) = \frac{E_2 - E_1}{E_2}$$

With the first order Taylor expansion for $f$ about the mean, and partial derivatives

$$f_2' = \frac{E_1}{E_2{}^2},$$

$$f_2' = \frac{-1}{E_2}.$$

We obtain the variance of the function (Equation 14) as:

$$\sigma^2 = e_1^2 e_2^{-4}\sigma_2^2 - 2e_1 e_2^{-3}\sigma_{1,2}^2 + e_2^{-2}\sigma_1^2 \tag{16}$$

As is the case with the bias in the expected value of the ratio, the variance of the normalized erosion rate difference scales with the mean erosion rates raised to some negative power. Thus, it is very sensitive to the estimated exhumation rates, and will





grow rapidly in regions where the function values are small. The variance goes to infinity as the erosion rate goes to zero. This analysis does not provide an estimate of the bias, but at least the variance estimate will give a sense as to where the **NR** has introduced the largest bias.

## 4.0 Synthetic Data Tests for Resolution and Model Errors

          One of the fundamental tools for analysis of an inverse model is to generate synthetic data using a forward model and
invert those data to investigate how effectively the original parameters can be recovered. A careful model construction with a suite of experiments is capable of evaluating sources as well as magnitudes of error. In this section, we present a comprehensive set of tests, designed to isolate and identify sources of error in the GLIDE method for erosion history estimation.

### 4.1 Synthetic Data Tests and Sources of Error

          We demonstrated above (Equation 13) that errors can be classified as one of three types, (1) model bias, (2) resolution
bias or (3) propagated noise from observed data. In this paper, we will ignore (3), although these errors can be important for estimation problems, especially for low quality data. Model bias (1) is difficult to identify in natural cases, but in a synthetic test, it can be identified by eliminating resolution errors and noise errors, for example, by using many, accurate data. The most important error is likely to be resolution bias (2). Resolution bias is a function of the data distribution in space and time and so can be assessed by constructing and comparing multiple data sets, as well as by examining the resolution matrix structure.

Synthetic data models have been used in two studies to analyse errors in the GLIDE inversion method. Fox et al. (2014) showed one model with two contrasting zones of uplift and showed very good recovery of the original parameters. In contrast, Schildgen et al. (2018) presented three forward-inverse tests (their Figures ED 2, 4, 5) in which they found very large errors identified as a difference between the input and inferred parameters. They attributed these errors to inappropriate averaging across regions with differential exhumation rates, as a type of "spatial correlation bias". However, aside from the
results of these synthetic data tests, Schildgen et al. (2018) presented no error analysis. They did not attempt to identify the source of the error in the synthetic data models, nor did they precisely define what they meant by spatial correlation bias, using it in reference to an outcome (a false inference of erosion rate increase) but not to source of the error, i.e., whether its source was in the model, was inherent to the distribution across time of the ages, or reflected the sparsity of data. Note that the lack of definition could lead to some confusion in that our usage of terms such as "model bias" and "resolution bias" follows the
definitions in estimation theory and may not correspond to Schildgen et al.'s (2018) usage. Our use of the term "bias" will be interchangeable with "error", and will be independent of sign.

          The definition of bias is not just a semantic issue. It is important to determine whether the errors identified by Schildgen et al. (2018), and referred to as spatial correlation bias, are model errors or resolution errors, because the tests for each type of error are different. Model errors should not be dependent on the quantity or quality of the data. On the other hand,
if errors are resolution errors, one must analyse the characteristics of the data distribution in space and time to assess under



what conditions these errors arise. The presence of large resolution errors in one model or one data set cannot be generalized to other models or other natural data sets, because resolution errors are unique to the distribution of the data in space, elevation and, foremost in the thermochronometry problem, the values of the ages.

Schildgen et al. (2018) constructed synthetic data models using a single distribution of ages, which suggests that they regarded spatial correlation bias as a model error and independent of the data distribution. If they had intended for their hypothesis to be that the inversion method used in GLIDE is biased with poor resolution data, they would need to define "poor resolution". They did not do this, suggesting that in their view, the GLIDE model has bias, regardless of data number, quality or distribution. If true, this makes the spatial correlation bias a model bias, not a resolution bias. Nevertheless, in the following sections, we conduct a complete error analysis, separate model bias from resolution bias, and establish the importance of each.

## 4.2 Experimental Error

Before we can address either model error or resolution error, we need to address a different type of error. This is a type of methodological error that arises from incorrect experiment implementation. The purpose of constructing a test using synthetic data and an inverse model to estimate parameters is to isolate and identify sources of error. By using a forward model with known parameters, we eliminate other, unknown sources of error such as differences in physics between model and the real-world processes. It is thus essential that the forward model be identical in its implementation to the model used in the inverse model. Unfortunately, Schildgen et al. (2018) did not adhere to this principle, and used one model to generate the synthetic data (Pecube, Braun et al. 2012), and a second model to invert those data in their model tests (internal GLIDE thermal model, Fox et al., 2014). Both codes include advection, conductive heat transport and a three-dimensional representation of surface topography, and so should be similar enough not to create a problem. However, Schildgen et al. (2018) applied different boundary conditions in the forward and inverse models. Specifically, the Pecube model, as a default condition, uses a constant temperature basal boundary condition, and GLIDE by default, uses a constant flux boundary condition, meaning that the temperature gradient is constant, but the temperature at the base of the model is free to vary with time. These are fundamentally different conditions and result in different geotherms.

To demonstrate the importance of constant temperature vs. flux boundary conditions, we apply the parameters Schildgen et al. (2018) used for their first example, the western Alps, although the conclusions apply to all tests in their analysis. In the Alps example, the forward model used a fixed temperature boundary condition at a depth of 30 km. The consequence is a large difference in the predicted geotherm for the forward and inverse models at the time of closure. The differences in a 1-D version of these two models are shown in Figure 3, which includes the same initial geothermal gradient and exhumation rate of 1 mm/yr, appropriate for the high erosion rate region of the model. With such a large exhumation rate, the geotherm in the Pecube model has reached steady state by 20 Ma, but at an erroneously low average gradient caused by the fixed temperature boundary condition. In contrast, the flux boundary condition results in a higher average geothermal gradient and remains in a transient state as the temperature at the lower boundary has not yet reached its final value. The





difference in the depth to the closure temperature for all low-temperature systems is up to several kilometers, which would therefore influence any subsequent exhumation rate calculation.

To illustrate the age disparities that result from such different basal boundary conditions, we regenerated the ages using the exhumation rates of the Alps synthetic model, but using the thermal model assumptions made in GLIDE; assuming a flux boundary condition and a total length of time of 36 Myr. We calculated the ages for both high- and low-uplift rate regions. Not surprisingly, the predicted ages are significantly different (Figure 4) with the Schildgen et al. (2018) ages systematically older, by up to a factor of 2. This difference is due to the lower thermal gradient and deeper closure temperature

depths predicted from a constant temperature boundary condition (compare blue and red curves in Figure 3). In addition, the systems with higher closure temperatures are affected more strongly and the magnitude of the error increases. Depths to the closure temperature vary by more than 1 km even for the apatite systems which leads to errors in erosion rates of up to 40%. Additional (typically smaller) differences in predicted ages between the two approaches can also arise from the different kinetic parameters used in the age-prediction models, although we have not investigated this effect.

The use of different thermal models for the forward and inverse model invalidates this exercise as conducted by Schildgen et al. (2018). The inverse model will do a good job in fitting the ages, but it can only do so by producing the "wrong" erosion rate. Furthermore, because the errors in the synthetic ages have increased bias, that is they get systematically larger with increasing closure temperature, the inversion will infer an increase in erosion rate with time in order to fit these ages. From this, it is no surprise that the results presented in Schildgen et al. ED Figures 3, 5 and 6 show large errors and false

accelerations. However, rather than being an artefact of the inversion methodology, or an effect of spatial averaging of erosion rates, this is an inevitable consequence of introducing a large methodological error into the experiment design. Had Schildgen et al. (2018) correctly calculated the synthetic ages, with a flux basal boundary condition consistent with the inverse model, a completely different result would have been obtained, as we show in the following section. Note that it is not important which thermal model is correct; it is important only that the forward and inverse models are consistent. Given this error, these models

should be disregarded and we will not discuss them further.

### 4.3 Synthetic Data Tests for Model Errors in Correlation Structure

Model errors are any errors arising from the construction of the forward model and its inability to correctly predict the data. For example, the correlation structure implicit to GLIDE could produce an overly smooth erosion rate function that does not fit the age data; this would constitute a model error. Errors in the thermal model would be another type of model error,

and we will investigate this in section 4.5. For brevity, we will not investigate systematics in the closure age calculations or other potential model errors.

Testing model errors in GLIDE is complicated by the fact that GLIDE is not strictly linear (section 3.1). The geotherm calculated in GLIDE depends on the erosion rate for the advective component to heat transfer. Its inclusion requires an iteration in order to solve simultaneously for the temperature, the advection and the erosion rate. The non-linearity is not strong because

the geotherm is always monotonic and at low erosion rates advection is negligible. However, in a synthetic data test, it creates



a complex response to errors from other causes, complicating interpretation and isolation of errors, particularly because the main test of our analysis includes an extreme erosion condition with 36 Myr of erosion at 1 mm/yr. To simplify the problem, we start our analysis with a suite of models that remove this nonlinearity by fixing the geotherm for the full model simulation. Later in this paper, we will restore the transient geotherm calculation to show how the nonlinearity affects the error. By breaking the models into these two sets of experiments, we can more accurately isolate the source of errors.

To evaluate errors, we construct the same test presented in Figure ED2 of Schildgen et al. (2018), referred to as the Western Alps case. This model has two regions, each with different, but constant and steady exhumation rate and separated by a vertical fault crossing the domain diagonally. The true solution is shown by the colour of the inset boxes in Figure 6a, d and is 1.0 mm/yr in the NW and 0.25 mm/yr in the SE.  Five synthetic data sets were constructed using up to 5 thermochronometer systems and either the fixed geothermal gradient model (Figure 5a,b,c) or the full transient thermal model (Figure 5d,e). The model topography is taken from a region of the western Alps and the sample age distribution roughly corresponds in pattern to the Alpine data set.

To isolate model errors, it is necessary to remove resolution errors. With reference to Equation 9, resolution errors can be eliminated by one of two ways, either by obtaining a model with $\mathbf{R}=\mathbf{I}$ or by setting $e_{prior}=e_{true}$. Either of these will result in nullifying the entire contribution of the resolution errors. Because we have no data errors (measurement noise), the only remaining errors will be the model errors (Equation 13).

As a first test (Figure 6), we attempt to produce a model with $\mathbf{R}=\mathbf{I}$. This is not possible in practice as it would require an infinite number of data, but it will be adequate to simply use a large data set, with ages distributed across the time span of interest (Figure 5a). The first model results are shown in Figure 6. These show that the parameters are recovered very well with this data set. The largest errors are in the peripheral regions that have no data, suggesting that these errors are due to inadequate data coverage, and so constitute resolution errors. This indicates that the model data set is not complete enough to fully remove resolution error. Similarly, some smoothing of the solution is visible across the fault boundary, but this occurs primarily where there are no ages near the boundary on one side or the other of the fault. This smoothing extends less than one correlation length into the adjacent domain and occurs in all time intervals nearly equally, i.e., both the 2-0 Ma and 6-4 Ma time intervals have a small amount of smoothing, so that the net result with respect to inferred acceleration (Figure 6g) is zero. In fact, there is no trace of spurious acceleration in either the high uplift or the low uplift region throughout the model. We calculated the normalized erosion rate difference (**NR**) (Equation 14) of Schildgen et al. (2018); values are all very close to 0 throughout the domain (Figure 6g). Interestingly, the variance on the **NR** is very large in the lower right half of the domain. This is due to a combination of the lower resolution of this region and the low erosion rate given the dependence of the variance on the erosion rate (Equation 16). Also interesting is the lack of any acceleration in the low erosion rate domain. There are no ages younger than 6 Ma in this domain so it does not have ideal resolution, similar to the left side of Figure 1 with poor resolution for the last 6 Myr. Although we are mainly addressing model errors, this example indirectly addresses the resolution question. According to the hypothesis of Schildgen et al. (2018), the well resolved, high erosion rates of the upper left region





should be averaged into the late time intervals to the lower right region. This has not happened in spite of the low resolution,
refuting the hypothesis of Schildgen et al. (2018).

For this first test, the smoothing errors that do occur are a function of the spatial correlation, but only weakly. In Figure 7, we show another version of the model of Figure 6, using the same data set (Figure 5a) but with a longer correlation length of 100 km, compared to 30 km in Figure 6. The results are only slightly different. In fact, the solution in the peripheral regions is improved with the larger correlation length. There is likely more smoothing around the diagonal fault, but the
difference is so small that it is not readily visible in Figure 7. Resolution is higher, but this is because of the definition of the parameters as regional averages, not as local erosion rates. Based on these two models, it appears that smoothing across the boundary depends more on the data density than on the spatial correlation length, again implying that these are resolution errors, not model errors. Even with a strong spatial correlation, there is no inappropriate averaging of the high erosion rates into the region of low erosion rate. Therefore, there is no bias, *sensu* Schildgen et al. (2018), no inappropriate spatial averaging,
and no spurious acceleration.

As a second test for model error, we eliminate the resolution error by the second method, i.e., setting the prior model equal to the known, true solution. From Equation 9, this assures that the resolution errors are zero, and we are left with only model errors. The previous model used a data set with very high density in order to force the resolution matrix to be close to **I**. In this second test, high resolution is not needed, so we use a sparser, more realistic data set, shown in Figure 5c. This is
not a trivial test. There is no reason why a model with the correct prior will obtain a posterior solution that is exactly the same. In fact, this is one of the best tests for model bias, because if the test shows errors, these errors indicate a failure in the model, not errors or inadequacies in the data. For example, if the correlation structure is forcing excessive smoothness onto the solution, which is essentially the Schildgen et al. (2018) hypothesis, this will emerge as error in this test. Results are shown in Figure 8 and are conclusive; there is essentially zero error in this model. Even the peripheral regions with no data, that were
not well fit in Figure 6, show no error. The model shows that there is no model bias associated with the spatial correlation structure. If there are errors in the inversion methodology, they are linked to the resolution of the data and thus to the data distribution. This result establishes that the real issue in these and, likely, most thermochronometric models, is in determining how many data, and in what configuration, are required to constrain a solution and how to recognize and bound resolution errors. This is the subject of the next section.

**4.4 Synthetic Data Tests for Resolution Errors**

Schildgen et al. (2018) proposed that errors in the Herman et al. (2013) models were systematic and a consequence of the method, and so constitutes a type of model bias. The tests in section 4.3 showed conclusively that this hypothesis was false and that there is no model bias. However, an estimation problem with sparse data such as most thermochronometry problems will be dominated by resolution errors. For completeness, we conducted a suite of tests presented here to determine
any potential systematics in resolution errors in the GLIDE inversions.





Resolution errors are easy to calculate, but difficult to generalize, because each natural or model data set has a unique distribution in space, elevation and time and therefore unique resolution errors that must be evaluated individually. In the current problem, we are trying to estimate a field quantity (erosion rate) over two spatial dimensions and time, so this is a three-dimensional estimation problem. It is further complicated by the fact that time resolution is controlled entirely by the value of the thermochronometric ages, so we have no experimental control on time resolution, other than by application of different thermochronometers with different closure temperatures. Age variations with elevation increase the time range and are just as valuable as multiple thermochronometers in this regard, but we will not extensively investigate this aspect of the problem. Although we cannot fully generalize results of any one model, by conducting models with a variety of data sets reflecting a range of data sparsity, we can establish the general behaviour of the system and would certainly detect any sort of systematic errors such as, for example, a tendency towards acceleration.

There is a second purpose to these models. Now that we have established that errors will be errors of resolution, we should assess how well our posterior metrics of resolution and variance characterize this error. Knowing where we have error is in some ways more important than the error itself, as this determines the utility of the analysis in the real world where we do not know the "correct" answer. We would like to know how well we are able to bound our estimates and where they are unreliable.

The results in Figures 6 and 7 have already told us something of the importance of resolution. They provide evidence that with good data coverage, we will recover the proper solution with little bias. In fact, the models in Figures 6 and 7 are not particularly high data density. The SE corner has no ages under 6 Ma, which implies poor resolution for the 6 Ma to 0 Ma time interval. The high uplift region is well resolved because we generated ages for 5 independent thermochronometers with different closure kinetics. The synthetic data are well distributed across time to resolve the temperature history over the last 10 Ma (Figure 5a). There is also an increase in age range for each thermochronometer due to the elevation spread, and this provides additional resolution through the age-elevation relationship.

We illustrate the importance of age range by conducting inverse model experiments with subsets of the data in Figure 5a to test the impact on parameter recovery, error, and posterior statistics. Figure 9 shows an inversion experiment using Data Set B (Figure 5b and Table 1) which has a good age distribution, but very few data distributed across space. In spite of the sparsity of data, the solution recovery is very good, with the central region around the data accurately estimated. Peripheral regions are again poorly resolved, but there is no false acceleration emergent from these errors.

The importance of the posterior statistics becomes apparent with this model. The region of well-resolved erosion rate in the 2 to 0 Ma time interval is small and the high values of the resolution statistic surround the data well (Figure 9b). The maximum resolution is about 0.6, but values of about 0.4 define the accurate solution. The reduced variance plot (Figure 9c) is more conservative and indicates only a few points where the variance has been reduced to below 0.5. The erosion rate in the 6 to 4 Ma time interval is poorly resolved with almost the entire model under a value of 0.4.





We constructed another data set (Set C) with better spatial coverage, but poor temporal coverage (Fig 5c) to demonstrate how resolution is dominated by the values of the ages (Figure 10). Data Set C has no high temperature
thermochronometers, so all ages in the high uplift zone are less than 6 Ma. Losing the high temperature systems seriously degrades the quality of the solution. Time intervals older than 6 Ma are almost fully unresolved and we show an additional time interval to make this clear (Figure 10g, h, i). The 6 Ma to 4 Ma has only one point where the resolution is nearing 0.5. The 2 to 0 Ma time interval is resolved better with values approaching 0.8, and reduced variance of up to 0.5. The solution accuracy matches the resolution, with a reasonably good solution in the 2 to 0 Ma time interval and a poor solution in the other
time intervals. In the slow uplifting region, the solution is poor everywhere.

The poor resolution of this model (Figure 10) is useful to demonstrate another important characteristic of Bayesian models, such as GLIDE. As shown in Equation 12 as the resolution goes to zero, so does the inverse operator and the solution will revert to the prior solution. This is what we observe in Figure 10, where the prior model has a uniform value of 0.35 mm/yr. Everywhere in the model where the resolution is low, the erosion rate reverts to a value of 0.35 mm/yr. This is
particularly apparent in the earlier time intervals (e.g., Figure 10g) where there are no ages controlling the rate; the erosion rate here has nearly uniformly taken the prior value. The corresponding resolution is near zero and the variance is not reduced below its prior value, so it is clear that this is not a resolved parameter.

We confirm this finding by running the same inversion, with different values of the prior erosion rate. Figures 11 and 12 show models with Data Set C and all other parameters as in Figure 10, except that the prior erosion rate is set to 0.65 mm/yr
and 1.65 mm/yr, respectively. The differences between these models demonstrate the influence of the prior on the estimate. The regions of the model with poor resolution always take on values at or near the prior. The region with good resolution is in the central part of the high uplift region from 2 Ma to 0 Ma, and this region has little to no dependence on the prior. The solution in this area is robust in that time interval. However, in the high uplift region the earlier time intervals are poorly resolved; even the 6 to 4 Ma interval has no point with resolution higher than 0.4 and at this level of resolution, it still shows
sensitivity to the prior model. The low erosion rate zone is less sensitive to the prior because the ages, which are all over 10 Ma, constrain the average erosion rate over these time intervals. However, they are unable to resolve individual time intervals, which show more fluctuation, but low values of resolution and little variance reduction. Given the sensitivity to the prior and the low values of resolution and variance, the low-erosion rate region in these models should be regarded as unresolved, whereas parts of the high-erosion rate region could be regarded only as marginally acceptable.

The model of Figure 8 can be included with the models of Figures 10, 11 and 12 to define a suite of four examples where the only inversion parameter that has changed is the prior value of the erosion rate. Note that the resolution and posterior variance do not change between these four models, because these metrics depend on the data, and the prior variances, but not the prior erosion rate. The same is not true for the variance of the **NR**, because of its dependence on the value of the erosion rates. Surveying the solutions of all four of these models, we see that the range of outcomes for the **NR** is very wide from no
acceleration at all (Figure 8) to acceleration everywhere (Figures 10, 11) to neutral or deceleration in the high uplift zone with acceleration in the low uplift zone (Figure 12). This range of behaviour is the result of combining the poorly resolved 6 to 4





Ma interval with the well-resolved 2 to 0 Ma time interval. The former depends on the prior model and the latter is reasonably robust and accurate. Although interpretation of the **NR** with this variability would be problematic, this problem is largely avoidable by noting the error in the **NR**. The variance in the **NR** (panel k in all figures) is almost always large indicating that

the values entering the ratio have large uncertainty and so the **NR** itself is highly uncertain. The low uplift rate region never has a standard deviation under 0.5, indicating that the **NR** is never resolved in this region. The high uplift rate region has a lower standard deviation on the **NR** and could be considered to be on the edge of resolved, and in fact it is these regions that appear resolved by the other metrics as well. The solution is moderately accurate here, although there are still dependencies on the prior, so we would interpret this solution as marginally reliable.

**4.5 Synthetic Data Models for Errors in the Thermal Model**

All models above use a thermal model that has been simplified from the normal, transient model incorporated into GLIDE. This was done to remove the coupling between the thermal model and erosion rates that can lead to complex error propagation. At low erosion rates, heat advection is small and the effect on the geotherm is small, but the test model we are using includes a region with an erosion rate of 1 mm/yr for 36 Myr, so the advective component of the heatflux is large and

the geothermal gradient increases by a factor of nearly 4 (Figure 3). This is extreme compared to the real world where such large erosion rates are rarely sustained for long, but as a test case, extreme conditions are acceptable. However, unless the high uplift area has thermochronometers with very high closure temperatures, ages are unlikely to exceed 15 Ma (Figure 5) which leaves the first 20 Ma of the thermal history unconstrained. In the absence of data constraints, the erosion rate reverts to the prior value. For most of the tests of this paper, this prior value is 0.35 mm/yr, close to the low-erosion rate of the SE domain,

but low compared to the correct value for the NW corner. Thus for any synthetic test with a low prior erosion rate, the geothermal gradient will be badly in error, leading to errors in the erosion rate. Geotherm errors are primarily a non-linear, and often negative, multiplier to errors in the erosion rate. For example, if erosion rates early in a thermal history are too low because of a low prior, the geothermal gradient will be too low and the late history erosion rates will be too high to compensate and fit the data ages.

To demonstrate some of these effects, we generated two synthetic data sets using the full transient model internal to GLIDE (Set D, E, Figure 5d, e). Set D has good temporal coverage in the high uplift region, but poor coverage of the low uplift region. Set E has poor spatial coverage for the entire domain. To demonstrate that the thermal model does not introduce any model errors, we first invert Data Set D with the prior model set equal to the true solution. As above, and following Equation 12, this eliminates the resolution errors leaving only model error; results are shown in Figure 13. As in all other

models with the prior solution set equal to the true model, the true solution is recovered nearly perfectly, so there are no model errors or method bias, also with the full thermal model. By setting the prior model to the true model, the thermal model is now correct, meaning that it is equivalent to the model used to generate the ages, so there are no model errors arising from spatial correlation, data smoothing, the geotherm calculation or other sources related to the model. Any errors in subsequent models are due to data inadequacy, i.e., resolution, and resolution errors amplified by subsequent errors in the thermal model.





As a second experiment, we inverted Data Set D using a prior erosion rate of 0.35 mm/yr (Figure 14). The data distribution and number in this model is close to that of Data Set A, so we expect the resolution errors to be small (see Figure 6), but errors are considerably larger. The difference is the amplification due to model error in the geotherm. Errors are largest on the high uplift side of the fault, but are present on both sides. Estimated erosion rates are significantly larger than the true values. This is a consequence of an estimated geothermal gradient that is too low. By using a low prior, when the actual erosion

rate was high, insufficient advection occurred in the early time phase of the model and the consequence is a low gradient in the predicted model geotherm. Although the errors are large, they produce a false deceleration, not an acceleration. This indicates that, not only is the average gradient in the geotherm incorrect, but also the curvature is incorrect. Because of the error in the geothermal gradient and curvature, a low prior (as used in Herman et al., 2013) is likely to produce a false deceleration in regions of sustained high uplift, although we would not generalize this result given the non-linear response of

the thermal model to errors in the erosion rate.

        As an additional test of mixing resolution errors with model errors, we decimated the test-data set to make a sparse data set, although we kept the temporal range appropriate for the time range of interest (Figure 5e). Inverting with the low prior value (Figure 15) gave a result similar to Figure 14, although the spatial domain was much noisier, with highs and lows depending on the data locations and values. Where Figure 14 shows a uniform deceleration, Figure 15 shows a mix of

acceleration and steady regions, with the best resolution indicated for the steady regions. Resolution metrics show that the 6 to 4 Ma time interval is only marginally resolved, whereas the 2 to 0 Ma time interval is well resolved.

        Thermal model errors are difficult to identify in the posterior statistics. A systematic geotherm error does not appear in either resolution or posterior variance metrics and regions that these metrics indicate as well-resolved can have very large errors. This is different from the resolution errors, which were well characterized. The best solution to this problem in natural

problems is to minimize model errors by calibrating the final geothermal gradient against modern heat flow measurements (Willett and Brandon, 2013). In some cases, pressure-temperature-time constraints from metamorphic assemblages might serve the same purpose, but the time component is necessary if the geotherm is transient. In the synthetic data models of this paper, we could have calibrated each model to the final gradient used to generate the ages and we would have obtained much more accurate models, but following the methodological principle of isolating error and variables, we chose rather to hold the initial

condition constant for all models. Again, we would like to emphasize the point that the synthetic data models presented here are not intended to simulate actual applications; they are numerical experiments intended to give insight into the performance of the methodology and sources of errors.

## 5.0 Tectonics and Spatial Variability in thermochronometric Ages

        In this section, we turn our attention from error analysis to discuss spatial variability in thermochronometer ages in

tectonic settings as characterized by both thermo-kinematic models and natural data studies. The synthetic data experiments (Section 4.0) showed that spatial gradients, or spatial discontinuities, introduce no significant errors, systematic or otherwise, to the GLIDE model inversion, provided data coverage is adequate. We also showed that data adequacy is defined more by





the temporal coverage provided by multiple thermochronometers and the elevation distribution, rather than the spatial coverage of samples, so it is somewhat of a misdirection to stress spatial variability. Nonetheless, having extensive regions of spatially

uniform exhumation rate increases the probability that one can obtain good age-elevation profiles or can combine age data from independent thermochronometers that are not perfectly co-located. In addition, the resolving capability of data is determined by the complexity of the erosion rate field being sampled, so resolution should be evaluated in the context of tectonic variability. The following sections discuss spatial variability and age patterns from the modelling literature, make an evaluation of age variability and resolution in a set of natural examples, taken from the original study of Herman et al., (2013),

and, finally, provide a discussion on how one can, or cannot, differentiate between tectonic and climatic variability impacts on erosion rate.

## 5.1 Tectonic Settings and Age Gradients

There has been considerable work done with thermo-kinematic and thermo-dynamic models to provide prediction of the relationship between erosion rate and thermochronometric ages (e.g., Mancktelow and Grasemann, 1997; Stüwe and

Hintermüller, 2000; Ehlers and Farley, 2003; Braun, 2005; Braun et al., 2012; Nettesheim et al., 2018; Koptev et al., 2019). Many of these studies have demonstrated that, even in complex tectonic settings, there are broad regions characterized by constant rates of exhumation and thus constant thermochronometric age. It should be noted that when we speak of gradients in age in kinematic models, we refer only to gradients in cooling age, where the cooling is related to the last phase of orogeny and exhumation. In any tectonic setting, there is a contrast between active areas exhibiting young cooling ages and the

surrounding regions unaffected or weakly affected by current tectonics (e.g., Willett and Brandon, 2002). The "pre-orogenic" ages of these surrounding regions play no important role in a thermochronometric study targeting recent, i.e., less than 6 Ma, exhumation. Pre-orogenic ages are typically tens if not hundreds of millions of years old and such ages are either removed from the analysis if they are older than the model runtime, or, if left in, have minimal effect, given that an age whose exhumation spans multiple time intervals, devoid of other ages, has little influence on the young history, nor does it contribute

to resolution as its effect is dispersed across multiple time intervals. In addition, the fundamental concepts of the thermal model, including the definition of a closure temperature, are predicated on monotonic cooling, so multiple heating episodes, and multiple orogenies, cannot be resolved.

Some examples of tectonic settings and the predicted distribution of cooling ages are illustrated in Figure 16, taken directly from the literature cited in the caption. In extensional settings (Figure 16a), most exhumation occurs by erosion on

uplifted footwall blocks of normal faults. Footwalls are often tilted or flexed in response to unloading during fault slip, so one would expect a gradient in exhumation rate with the highest rates found near the fault. One of the most studied normal fault systems is the Wasatch Fault in Utah which has extensive thermochronometry data including both AFT and AHe data (Armstrong et al., 2003) and a complete suite of thermo-kinematic models (Ehlers et al., 2001; 2003). These studies demonstrated footwall tilt and a gradient of exhumation rate, but also showed that at high rates of exhumation, the gradients

in age were reduced to the point that they could barely be resolved. This was supported by the observations. Within 20 km of the fault, AFT ages were constant within error (Armstrong et al., 2003). AHe ages showed a small gradient, which, after





correction for elevation effects, resulted in a difference of about 3 Ma over a 20 km transect. The low spatial gradients are a consequence of the high rate of exhumation, which compresses the age-exhumation rate relationship. The Wasatch studies are most notable for their extensive sampling with two thermochronometers (AFT and AHe) and it is the age difference between

AFT and AHe that resolves a temporal change in erosion rate (Ehlers et al., 2003). Ehlers et al. (2003) found a small decrease in exhumation rate in the last 5 Ma, whereas Herman et al. (2013) found a small increase; this difference is due to the difference in the closure kinetics assumed by each of these studies.

Thrust ramps (Figure 16b) have also seen considerable attention due to the high exhumation rates and common occurrence of young ages. A number of thermo-kinematic models have been published based on ramp-flat, fold-fault

kinematics (Whipp et al., 2007; Lock and Willett, 2008; Herman et al., 2010; Coutand et al., 2014; McQuarrie and Ehlers, 2015; McQuarrie and Ehlers, 2017). These models consistently show that a constant dip ramp produces a broad zone of constant uplift rate and thus, with a steady surface elevation, a region of constant age. The hanging wall at the lower end of the ramp is typically bounded by a fault-bend fold whose geometry controls the wavelength of a transition from the ramp zone of constant age to the hinterland where there is no exhumation and thus inherited ages. If the up-dip limit of the ramp feeds

into a flat, there is a fault bend fold whose geometry controls the transition from the ramp zone of constant age to the foreland zone of inherited ages. If the ramp reaches the surface, and all remnants of the fault-propagation fold are eroded, there can be a narrow zone of older ages in the hanging wall exposed to cooling into the footwall, but otherwise, there is a sharp transition between the cooling zone and inherited ages in the foreland. Gradients in cooling age associated with fault motion are young to old from the center of the ramp to the thrust, but are steep, short and rarely observed in the field.

Orogenic or accretionary wedges have also been much studied, although the complexity of their internal deformation makes it difficult to generalize. One must also differentiate between models that assume a surface erosion rate and allow this to drive kinematics (Barr and Dahlen, 1989; Batt et al., 2001) and those that allow the surface erosion rate to be freely determined with the deformation (Willett, 1999; Batt and Braun, 1997; Fuller et al., 2006; Michel et al. 2018, 2019). Summarizing some of the latter, Willett and Brandon (2002) made the explicit point that within the deforming wedge, rock

uplift rates will be constant to maintain self-similar growth and that exhumation rates and ages will also be constant unless some surface process acts independent of the rock uplift associated with wedge growth. Their summary figure included nested reset zones of constant age (Willett and Brandon, 2002) (Figure 16c). The prowedge of an orogenic belt (right side of Figure 16c) is characterized by an unreset frontal zone where particle paths are too shallow to permit heating above closure temperatures. The transition to the reset zone can be in either prowedge or retrowedge and can be either discrete, associated

with a single structure, or diffuse, often exhibiting partially reset ages, but is rarely defined by a spatially extensive gradient of reset ages.

Schildgen et al. (2018) presented a summary of these same settings in their Figure 1, but with all figures depicting steep age gradients. This is odd, in that these gradients do not appear in the original studies cited above nor, as far as we can determine, in any of the studies cited in their paper. For the normal fault model, the shallow gradient reported by Ehlers et al.

(2003) is exaggerated to the maximum possible gradient with an isochron perpendicular to the surface, implying tilt, not of the





10 to 15° observed, but of 90° (Figure 16d), something that does not occur in this setting. For both the thrust ramp and the orogenic wedge, the zones of constant age have been replaced with a gradient younging towards the bounding ramp. The literature has no support for this gradient. With a thrust ramp there is either no gradient or a slight increase in age due to footwall cooling, but never a decrease in age. In fact, the Schildgen et al. (2018) versions of the ramp and wedge settings imply
a normal sense of shear across the ramp hanging wall or across the entire retrowedge (a rock at the ramp fault is moving faster than a rock in the orogen interior) and such a strain rate field has never been observed or emerged from a model. The natural settings indicated on their figure are similarly incorrect. New Zealand is not a simple thrust ramp, but rather more akin to the transpressional setting of Schildgen et al. (2018)'s Figure 1d, so the ages do young towards the foreland, but the kinematics are not those of a simple ramp and any thermo-kinematic model fit to those ages has had to include more complicated
kinematics with multiple blocks or internal deformation (e.g., Beaumont et al., 1996; Herman et al., 2009). The Central Himalaya have been much studied, with a number of papers arguing that the ages are constant across the major structural ramp (Burbank et al, 2003; Herman et al., 2010). As with rapidly moving normal faults, this is partially a consequence of compression of the age-erosion rate relationship at high erosion rate (Whipp et al., 2007; Thiede et al., 2009; Thiede and Ehlers, 2013). For the accretionary wedge model, rather than showing the youngest ages in the retrowedge (Figure 16f), the
Olympics have no reset ages (Batt et al., 2001), the Apennines have distinctly older ages towards the retro-deformation front with the youngest ages in the core of the mountain belt (Thomson et al., 2010b), Taiwan has constant ages (Willett et al., 2003; Fellin et al., 2017), and the Alborz as a transpressional system has no systematic pattern, with most ages unreset (Ballato et al., 2015). Thus, the age gradients depicted in Figure 1 of Schildgen et al. (2018) are perplexing because a figure intended to illustrate spatial patterns of ages, has altered those patterns from the original studies without explanation.

**5.2 Spatial Patterns of Exhumation and Acceleration: example from the western Alps**

Given the insights into potential sources of error in the inverse models (section 4.0) and tectonic causes for age gradients, or the lack thereof (section 5.1), we turn next to the natural examples to assess where there is potential systematic error. The synthetic models of section 4.0 were conclusive in showing that there is no model bias, but that resolution errors and their amplification through thermal model errors could lead to specific errors. We conduct an analysis of the resolution
errors for a number of the original sites, which are specific to each site and data set. In addition, we use these examples to demonstrate problems that arise through the use of the proposed **NR** metric as an interpretation tool.

The Alps have served as the principle example for the application of GLIDE (Fox et al., 2015, 2016), and played an important role in the study of Schildgen et al. (2018), so we will continue this focus here. Figure 17 shows the GLIDE inversion estimate of exhumation rate, together with resolution metrics and the normalized erosion rate difference. The ages used in the
inversion are shown in Figure 17l. They are also shown in Figures 17a-i, plotted within the time interval in which they fall, to give a better sense as to which time intervals are constrained and where. Exhumation rates are well constrained throughout the external Alps which lie northwest of the Penninic line fault (PLF in Figure 17). There are numerous ages that fall within the 6 to 0 Ma range, as well as by older ages in both external and internal Alps, southeast of the PLF.



There is a clear increase of erosion rate in the external Alps, closely associated with the young ages. NW of the
Penninic line, exhumation rates are notably higher in the 2 to 0 Ma time interval (Figure 17a), defining an acceleration with
respect to the previous time intervals. The high erosion rates are centered on the high external Alps where the youngest ages
are clustered. This zone of high erosion rate NW of the Penninic line has numerous ages in the 6 to 4 Ma time interval, which
is a typical age for the AFT samples from the region. The ZFT ages are older (8 to 16 Ma), and AHe ages are younger than 6
Ma. Erosion rates in this region are thus well-resolved across all time intervals.

Schildgen et al. (2018) claim that the increase in both the external and the internal Alps are spurious, with the only
argument presented being the comparison with the synthetic data inversion example. As discussed in section 4.2 (see also
Figures 3, 4), their version of this model was constructed incorrectly, so the comparison should be to the results of the models
shown in Figures 6 through 15. To properly make a comparison between the model and the natural example, one must estimate
the data density in both space and time and select the model with synthetic data distribution most similar to the natural example.
With the data density of the Alps, the closest models are those using data set A or D (Figure 5). Even here, the true age
distribution for the Alps is likely better than any age distribution of the models, because the extremely high and sustained
erosion rate in half of the synthetic model domain compresses the model ages into a narrow time range. Nonetheless, the
models all show that with the data coverage of the natural Alps, we expect only small resolution errors with a very well-
resolved history in the external Alps, a resolved average erosion rate with no resolution of late change in erosion rate in the
internal Alps and a smoothed transition between the two.

The region of accelerated erosion SE of the Penninic Line in Figure 17 is larger than the counterpart in the synthetic
models (e.g., Figure 14). There are several reasons for this. First, the data resolution of the synthetic models and the natural
data are not identical, so there are resolution errors throughout the natural example and direct comparison with the model will
never be perfect. Second, the uplift functions are very different and this affects the temporal resolution. The vertical fault with
13 km of relative offset in the synthetic data example makes a good synthetic test, but it should not be mistaken for the actual
uplift pattern in the Neogene Alps, which, at least today, has no major active structures. Finally, and most importantly, the
synthetic model and actual Alps differ in that the external Alps do have an acceleration in exhumation, whereas the synthetic
model does not. The increase in erosion rate in the external Alps is robust and well-resolved by its local ages. Smoothing of
this signal from the external Alps during each independent time interval into the surrounding regions gives the appearance of
an acceleration in these regions. The synthetic model had no acceleration in the high uplift region, so any smoothing of the
high uplift area into the low uplift area was equal over each time interval.

It should also be noted that even if there were a sharp boundary between young and old ages from the external to the
internal Alps, it would not be at the Penninic line as portrayed in this paper and Schildgen et al. (2018). Contrary to their
claims, there are a number of ages younger than 4 Ma SE of the Penninic Line faults (most clearly seen in Figures 17e and
17h) and these extend the high uplift zone to the south. The high uplift zone in Figure 17a continues SE into the internal Alps
for a distance of less than one correlation length beyond the southernmost young ages. This is the same smoothing effect we
observed in the synthetic data models (e.g., Figure 6 or Figure 14).





Earth **Surface**
**Dynamics**
Discussions

The normalized erosion rate difference (**NR**) map (Figure 17j) gives a very different story. The maximum acceleration is shown in the middle of the domain centered on a region with almost no age data younger than 6 Ma. Furthermore, the high acceleration region is large, covering the entire internal Alps at a value larger than the external Alps, where most of the young ages are found. It is this relationship that was the basis of the conjecture of Schildgen et al. (2018) for inappropriate spatial averaging, as there is no intuitive basis for this large acceleration zone centered on a data gap, rather than on the data. However, it is here that the shortcomings of a plot based on the **NR** ratio emerge. Comparison between the erosion rate maps and the normalized erosion rate difference map show that the two have very little in common. The offset in maximum values is the most obvious, but the extent of the large values of **NR** far to the south and east is peculiar given the data distribution and the lack of deviation in erosion rates from the prior value of 0.35 mm/yr through most of the south east of the study area. We agree with Schildgen et al. (2018) that this signal is peculiar and is a clear indication that something is wrong. However, the problem is not in the erosion rate maps; it is apparent only in the normalized erosion rate difference maps. The problem is further clarified by looking at the error associated with the **NR** (Figure 17k). It shows that the lowest variance in the **NR** is centered on the external Alps where the data density is highest and the erosion rates are best resolved. The variance in **NR** increases rapidly to the SE with high values encompassing most of the internal Alps. The shift of the locus of **NR**, relative to its error and its underlying parameters is a feature of a ratio expression where the quantity in the denominator has a wide range of values. Regions where the denominator is small, i.e., the internal Alps, have a large ratio, but also increasing sensitivity to error in the denominator. This effect creates a bias, a bias that is manifested as a false acceleration in regions of low erosion rate.

As with the synthetic data models, a test of the resolution can be provided by checking for sensitivity to the prior model. We reran the Alpine data inversion using a prior erosion of 1.35 mm/yr (Figure 18). As expected, the peripheral regions all have a high erosion rate reflecting the higher prior. However, the central external Alps are almost unchanged, reflecting the high resolution. The high-resolution contours (above 0.5 to 0.6) mark the region that is robust against changes in the prior. The internal Alps to the SE hold their lower erosion rate because of the older ages. There are also somewhat higher erosion rates at all time intervals in the external Alps. This is a model effect due to a difference in the geothermal gradient between the two models, in response to the increased advection of the early history in the high-prior model.

As an even more direct test of the hypothesis that the inferred acceleration is an artefact of mixing regions, we produced another inverse model for the Alps, but in this case, we omitted all the data from the internal Alps in the SE (Figure 19). Comparing the inverse model of all the data (Figure 19a,b,c) to the model of just the external Alpine data (Figure 19d,e,f), one can see that the solutions in the external Alps are essentially identical proving, once again, that there is no influence of the old ages to the SE on the erosion rate estimates in the NW.

These three models are conclusive; the acceleration found in the Alps is not spurious, and it is not the product of inappropriate spatial averaging. In the external Alps it is well resolved by the local data which span the timespan of interest. In the internal Alps, the erosion rates over this timespan are poorly resolved and so the solution from the NW is smoothed into this area, but not sufficiently that the older ages of this region are misfit. Schildgen et al. (2018) stated that the inferred



acceleration in both external and internal Alps was a product of averaging the ages from both regions. This hypothesis is clearly false, otherwise the acceleration would not be present in Figure 19.

Finally, we address the geologic evidence of Schildgen et al. (2018)'s hypothesis that the external and internal Alps are separated by an active normal fault along the Penninic Line. The potential of the Penninic Line to be an important recent tectonic feature has long been recognized on the basis of fission track dating (e.g., Seward and Mancktelow, 1994; Malùsa et al., 2005). As such, it has received a number of recent studies. First, as we pointed out above, there are multiple ages younger than 4 Ma (see data compilation Fox et al., 2016) as well as ages older than 6 Ma on both sides of the Penninic Line (Figure 17, and Fox et al., 2015, 2016). Recent field studies (Persaud and Pfiffner, 2004; Malùsa et al., 2005; Egli and Mancktelow,

2013) find no significant post-Miocene motion of the Penninic Line Fault of any sense (normal or thrust). Egli and Mancktelow (2013) concluded that: "*Our observations indicate that none of the major faults or shear zones around the Mont Blanc massif (i.e., Mont Blanc shear zone, Mont Blanc back-thrust, Penninic thrust) was active in Late Neogene times and that young exhumation is therefore not controlled by movements along these structures*". Likewise, dating of hydrothermal minerals in shear zones surrounding the Lepontine dome, including parts of the Penninic thrust, show that major fault activity ended about

5 Ma (Bergemann et al., 2020). Identified normal faults in the western and central Alps are small, steep, and interpreted to be gravity-driven features with small displacements, with the possible exception of the Simplon detachment, which has a local effect due to its orientation with respect to the regional transcurrent motions. Although geophysical data show distributed extension at low strain rate in the western Alps attributed to gravitational spreading (Sue et al., 1999; Delacou et al., 2004), seismicity is diffuse and mostly in the hanging wall southeast of the Penninic Line Fault. Seismicity thus corresponds to the

regions of low exhumation, not high exhumation. Independent constraints on the tectonic activity in the Alps also come from the peripheral foreland basins, the Molasse basin, and the Po Basin, both of which show Quaternary rock uplift, indicating that erosion has recently begun to outpace tectonic crustal thickening leading to flexural isostatic rebound (Champagnac et al., 2007, 2009; Willett, 2010). The frontal fold and thrust belts on both sides of the Central Alps ceased motion in the Pliocene, sometime prior to 3.4 Ma in the Jura (Bolliger et al., 1993; Becker, 2000), and in the early Pliocene for the Lombardic thrust

belt, as evident in growth strata and thrust-sealing post-tectonic sediments of the Po Basin (Fantoni et al., 2001). The geologic evidence thus points to the cessation of tectonic activity throughout the Alps over the last 5 Ma, at the same time that the erosion rates are increasing in the external Alps.

## 5.3 Analysis of other natural examples for an increase in exhumation rates

   According to the re-analysis of Schildgen et al. (2018), of the 32 sites identified in the Herman et al. (2013) study as

showing sufficient thermochronometric data to resolve an erosion rate history over the past 6 Ma, 23 of them were what they called "spurious", meaning that they arose as a result of inappropriate spatial averaging of age data. Furthermore, for those cases that were not spurious, or ambiguous, they interpreted the cause as either "tectonic" or "glacial", where glacial refers to Quaternary changes in erosion rate associated with the onset of glaciation. This second point will be discussed later (Section 6.2). The first issue, whether or not results are justified or spurious, will be assessed here.





To address the first question, we need to define what is meant by an artefact or a spurious result. We see this as related to the question of which data are responsible for a perceived increase in erosion rate and to the method of analysis. Are the local data sufficient to resolve an increase or are unrelated data from large distances being used, and is this increase robust with respect to the method of analysis? A spurious result should be limited to an artefact that arises only because of the inappropriate combination of data, model bias, or resolution errors. For the case of a spurious acceleration, a better analysis or

knowledge of the "true solution" would reverse the conclusion. This definition does not include smoothing errors if that smoothing includes an area that exhibits a true acceleration based on its local data, even if smoothing results in an acceleration zone expanding into regions without data. Smoothing errors are easily recognized and if removed, either by a better analysis or simply masking parts of the domain, the conclusions would be unchanged; an acceleration would be inferred in any case. In this sense, results are not regarded as being area dependent; the definition of an artefact must apply to the best resolved parts

of the study area, not just peripheral regions. Thus, the errors in Figure 10 would constitute a spurious acceleration, but the acceleration inferred for the transition between the external and internal Alps in Figures 17, 18, 19 would not be regarded as spurious, because it would not arise without the existence of a neighbouring region with a resolved increase in erosion rate.

        We cannot be certain that Schildgen et al. (2018) share our definition of a spurious result as they gave no definition as to how they arrived at their assessment. They present individual assessments for each site in their ED Table 1, but neither

this table, nor the text contain any methodology, that could be used to reproduce the result. The only data or analysis presented in the paper were the **NR** maps of each area and maps of the ages taken from the Herman et al. (2013) compilation. The **NR** map was interpreted over the full region where Herman et al. (2013) found resolution values above 0.25 in both the 2-0 Ma and 6-4 Ma time intervals, without consideration of the variance of the **NR** quantity itself. The typical interpretation seems to follow a pattern whereby, if they found large areas of high **NR** with data gaps or age gradients included within that region,

they stated that this proved averaging of the surrounding data resulting in a false acceleration and an assignment of "spurious".

        The problem with this assessment should be obvious from the preceding sections. Specifically, the synthetic data models as conducted by Schildgen et al. (2018) were incorrect, due to different boundary conditions in the forward and inverse models (Figure 3), and the correct models show none of the hypothesized acceleration pattern, at least in cases of good data temporal resolution, so analogies between models and natural sites are incorrect. Second, the **NR** maps are themselves

introducing bias that Schildgen et al. (2018) then hold up as diagnostic of an artefact of spatial averaging. Third, with poor resolution, the erosion rate estimate reverts to the prior, it does not pull in values from distal locations, so to associate a parameter value within a data gap to distal data is not correct. The site by site interpretation is simply repeating these three errors multiple times. This faulty diagnosis was illustrated in the Alpine example above (Figs 17, 18, 19).

        Part of the problem with these interpretations is that many of the regions interpreted by Schildgen et al. (2018) have

inadequate resolution and should not be interpreted at all. Part of the fault for this lies with Herman et al. (2013) who took a constant cutoff value of 0.25 in the resolution as adequately resolved for the last stage of their global analysis. This was a mistake as there is too much variability in the absolute values of resolution from site to site and a different, site-specific value should have been used for each location. We discuss this issue later in the paper.



As a second example of site interpretation, the Nanga Parbat region of the western Himalayan syntaxis is analysed with results shown in Figure 20. In order to understand how resolution arises, we have plotted the age data in different ways. First, the full age distribution is shown as a histogram (Figure 20l). This shows that there are 5 thermochronometers with ages bracketing the timespan of interest. It is important to show how many ages bracket, or fall within, each time interval, so these are shown on the resolution and variance plots. All ages greater than 6 Ma are shown in white on the erosion rate maps; values less than 6 Ma as black diamonds. Taken together, these maps give a good sense as to which data are constraining which erosion rates. The fact that the resolution and variance plots mirror the data reflects local support of resolution. These figures miss the additional resolution provided by elevation range and we do not show maps of the individual thermochronometers, but this information is implicit to the inverse model and can be partially visualized from the age histogram. Figure 20 shows that the core region of the syntaxis is very well resolved by the local data. With 5 thermochronometers yielding ages under 10 Ma, the resolution of the exhumation history is excellent. There are older ages outside the massif, but comparison with the models of Figures 6, 9 and 14, show that with data coverage of this density, old ages from other regions have no influence on the solution in well-resolved regions. The region with resolution above a value of 0.4 or perhaps 0.5 includes all the ages less than 6 Ma, none of the region with no ages under 6 Ma, and would be regarded as well resolved. This region also shows a clear increase in erosion rate with time. We also note that previous studies using different approaches found the same result, including this increase in rate (Zeitler, 1984; Thiede and Ehlers, 2013).

In contrast, the map of **NR** has lost all the information regarding resolution (Figure 20j). It shows simply a broad, long spatial wavelength increase. However, the error on **NR** shows that the resolved part of this map is also restricted to the Nanga Parbat core region (Figure 20k). Schildgen et al. (2018), not having the error information, made the statement: "*the linear inversion performed by ref. 6 [Herman et al., 2013] suggests a broad zone of late-Cenozoic erosion-rate increases both within and outside the massif (Figure S7b). This result is a consequence of combining data from inside and outside the NPHM [Nanga Parbat-Haramosh Massif], across the active bounding fault zones ...".* There is no additional support for this statement provided by Schildgen et al. (2018). It seems clear that they interpreted the **NR** map without knowledge of the error, without noting how local the high rates are in the erosion rate maps, and without the insights provided by correct synthetic data models. As a result, they came up with a new, but incorrect, interpretation of the results of Herman et al. (2013). The original result of Herman et al. (2013) is not a consequence of combining data from outside the massif. We could remove these data as we did in the Alps (Figure 19) and the result would be unchanged. We could mask or remove the regions where smoothing errors appear, and the conclusion would be unchanged. We could change the prior model or the correlation length and the conclusions would be unchanged. The acceleration is the result of effectively co-located data from 5 thermochronometric systems, and is not an artefact of the method or of inappropriate averaging, so by any meaningful definition, is not spurious.

A third example with good spatial and temporal coverage is the Olympic Mountains of NW Washington State (Figure 21). Here there are three thermochronometers providing coverage of the 10 Ma to present timespan and a fourth providing a longer timescale average rate. These data are dominantly from the central core of the mountain belt with older, largely unreset, ages found in the western region where it has been argued that the combination of lower erosion rates and shallow particle





trajectories through the orogenic wedge result in a frontal unreset zone (Brandon et al., 1998; Pazzaglia and Brandon, 2001).
The GLIDE inversion delineates a well-resolved circular region in the core of the mountain belt centered on the data, with

1040 ages falling into all three time intervals. The region is resolved by local data and as the synthetic models show, with resolution
at this level, old ages far removed from the resolved region have no influence on the erosion rate history (Figures 6, 9, 14).
The Olympics provide a good example of the distortion that occurs through application of the **NR** operator. The **NR** field
(Figure 21j) shows maximum values offset to the NW relative to the data. This is an artefact of the division operator, as shown
by the error on the **NR** which remains centered on the data. Schildgen et al. (2018) directly interpreted the **NR** field at face

value, claiming that: *"the global inversion found normalized increases in exhumation rates of 0.3 to 0.7 when comparing the
6-to-4 Ma and 2-to-0 Ma time bins, with the largest increases in the region with the steepest spatial gradient in ages (Figure
S11c). This difference likely arises because the vertical exhumation paths assumed in the linear inversion model used by ref.
6 [Herman et al. 2018] are inappropriate for this setting, and because data that experienced disparate exhumation histories
were combined."* This statement is not supported by any further evidence and we see the same error made as in the Alps and

Nanga Parbat. The largest, *well-resolved,* erosion rate increase occurs on top of the co-located ages from 4
thermochronometers, not where the steepest spatial gradients occur. The hypothesized combination of disparate exhumation
histories is conjecture based on comparison with the incorrect synthetic data models and interpretation of the statistically
insignificant parts of the **NR** map. More recent, and independent work modelling the Olympic thermochronometric data
(Michel et al., 2018, 2019) supported the Herman et al. (2013) result, finding an acceleration in exhumation. The Michel et al.

(2018, 2019) studies also included 3-D thermo-kinematic models and found that results did not differ significantly from 1-D
models. The acceleration is real, based on local data, not an artefact and there is no basis for referring to this result as spurious.

As a fourth example, the Marlborough region of New Zealand (NZ) is one of the most complex tectonic settings
considered in these studies. This complexity leaves much latitude for interpretation. In addition, the data are sparse with only
two thermochronometers applied in the region (Figure 22). The tectonic complexity includes changing rates of fault slip on a

system of oblique but dominantly strike-slip faults, which could result in local changes of erosion rate that are not characteristic
of the region as a whole. Schildgen et al. (2018) argued that: *"this region represents a clear case of combining samples that
experienced disparate histories, leading to a spurious erosion-rate increase."* We disagree with this. The youngest AFT ages
are all found south of the Wairau and Alpine Faults and north of the Hope or even the Clarence Faults (See Schildgen et al.,
2018, Figure S16). These are supplemented by a handful of ZFT ages in the 2 to 10 Ma range. In fact, there are three distinct

local areas where there are nearly co-located ZFT and AFT of the correct age to constrain the erosion rate history from 6 Ma
to the present; these are visible as the 3 nearly co-linear points in the 6 Ma to 4 Ma resolution or variance plots of Figure 22.
Not coincidentally, the best resolution and variance reduction occurs along an axis connecting these 3 points (e.g., Figure 22k).
The two southern points sit on a common tectonic block and the northern point on a second block. However, the data on each
of these blocks is sufficient to resolve an erosion rate history with no contribution from data points outside these blocks. This

case is thus analogous to the models of Figure 6 or Figure 14, where there are two adjacent blocks, one with a good distribution
of ages from multiple thermochronometers, one with only old ages. Just as there was no averaging across the fault in those
models, we see no evidence for averaging across the fault in this natural case. This is apparent from maps of the resolution and





variance, both of which define a resolved region centered about these three points. An appropriate threshold for the resolution and reduced variance would both be about 0.5 to capture these 3 regions and only these regions with no major inclusion of the

other tectonic blocks. Again, the **NR** is diffuse, broad and offset from its error, so it does not reflect the important characteristics of the inverse model result. In contrast, the correct threshold value of the resolution (0.5) or even better, a similar value of the reduced variance, would nearly follow the block limiting faults.

The important point illustrated by this NZ example is that it is the existence of collocated ages from different thermochronometers that increases the resolution to the point that we would interpret them as a well-constrained erosion rate

history. There are several regions to the north and south of the resolved region that also have a number of ages under 6 Ma, but these regions have exclusively AFT ages, and ages from a single thermochronometric system, regardless of their age range, are insufficient to bring resolution into an acceptable range. It is the sampling of different closure temperatures at different times that determines the resolution and the erosion rate history. It is not, as Schildgen et al. (2018) claim, a combination of ages from a single thermochronometer sampling different erosion rate histories from across the entire region. Instead, we see

most of the region at or near the prior value, with the exception of the tectonic block between the Wairu and Hope faults, which shows rapid uplift in the last 2 Ma. Again, we conclude that there is no spurious acceleration as an artefact of the method. The predicted model is consistent with all the ages, the predicted accelerated uplift is limited to one, or at most two tectonic blocks, and the data constraining that history are identifiable and have a common history. This result should not be regarded as spurious.

As a fifth and final example, we consider the data from the Fiordland region of New Zealand (Figure 23). In contrast to the Marlborough region, the data coverage here is excellent with widely distributed AFT and AHe ages, most of which are younger than 10 Ma, as well as some older ZFT and ZHe ages. The ages fall nearly uniformly across the center of the region within all three time intervals of interest (Figure 23), resulting in resolution values approaching 1.0 in many places within all three time intervals. A threshold resolution value of 0.5 would enclose the region with data; a slightly smaller number would

include more of the surrounding areas but would not change the interpretation. The **NR** is again offset from the data, with the locus of its largest values shifted to the southeast, so any direct interpretation would be misleading. There is some structure to the erosion rate pattern, for example higher rates in the north across the full 6 Ma interval, but in general the erosion rates are regionally consistent and show an acceleration in the last 2 Ma. There are no ages in distal regions to call upon for inappropriate spatial averaging. However, Schildgen et al. (2018) still assess this acceleration as spurious, stating: "*the broad zone of*

*increased exhumation rates is probably at least partly linked to correlating samples across active (or recently active) faults.*" This interpretation is partially linked to overreliance on the **NR** map, as they state that there are "*peak values frequently occurring in areas of relatively low relief*", so it is evident that they are putting weight on the large values of **NR** which appear offset from the high topography towards the SE, rather than focusing on the resolved values, which are directly over the high topography and young ages. However, the main culprits are apparently faults that are at the scale of the valleys and ridges and

are thought to be responsible for the lack of correlation of some ages with elevation. These are apparent faults, because, in fact, no faults have been mapped or described in the literature (Sutherland et al., 2009). They are inferred only from noise in





the thermochronometric data, for example, based on the lack of correlation between age and elevation (Sutherland et al., 2009). Even if these faults do exist, they would represent variations at a length scale much shorter than the correlation length. Ages in such close proximity are simply averaged as shown in Figure 2f and there is no reason to expect inappropriate artefacts. The

data are numerous enough for all three thermochronometers to provide an unbiased average of nearby data, so, although the short wavelength variations are not resolved, there is no reason to expect an artefact of false acceleration in the regional average. We find no justification for their assignment of "spurious" to the original finding of Herman et al. (2013).

We could continue to discuss each example covered by Schildgen et al. (2018) in their Table (1) and data supplement, but we have reviewed all sites and the results are the same as the four examples above. In each case, the only evidence presented

for the determination of a "spurious" outcome is the **NR** map and its relationship to the data distribution. In our view, we find none of the determinations of "spurious" in their Table 1 justified. In each case, there is a good explanation of the resultant acceleration in erosion in terms of co-located data from multiple thermochronometers or elevation relationships.

## 6.0 Discussion

### 6.1 Bias, Errors and Resolution in GLIDE

The central question investigated in this paper is whether there is a systematic error in the methodology of Fox et al. (2014) and Herman et al. (2013) that resulted in an apparent acceleration in inferred erosion rates. As we argue above, this question can only be addressed if one identifies the source of errors within the analysis. We have dissected the possible source of errors and shown that, neglecting measurement errors, which should not be systematic, there are three potential sources of error, (1) model errors due to the parameterization including spatial correlation smoothing, (2) model errors due to incorrect

prediction of the near-surface geotherm, and (3) resolution errors that result from inadequate data coverage of space and time. The synthetic data, forward-inverse models presented above show that (1) do not exist in any significant form, that (2) are a potentially serious source of error in erosion rates, but do not predict a systematic tendency towards acceleration, and that (3) can be large, but depend on the individual data distribution as well as the prior model parameters, and so also have no generalizable tendency towards acceleration.

Resolution errors are the result of data inadequacies, primarily in time, where multiple thermochronometric systems or elevation dispersion are needed in order to give a range of ages adequate to resolve a temperature history. Although every data set has a unique distribution and therefore unique resolution errors, we were able to make some generalizations through analysis of a range of data examples. First, we found that with a large number of data, distributed appropriately in age, all errors, including resolution errors, go to zero (Figure 6). In fact, it did not require a particularly large number of data to obtain

excellent estimates in our tests (Figure 9). Nor was there any evidence for inappropriate spatial averaging in high density models. There was smoothing of sharp boundaries, but even here, this occurred only where data were insufficient to resolve the boundary. As data density decreases, resolution errors become larger. If data are too few, there are large resolution errors, not surprising as it requires data to resolve complexity in either space or time. The distribution of data in time is much more





important than the distribution in space, at least with relatively simple patterns of erosion rate (compare Figures 9 and 10).
Errors in the analysis with sparse data followed a specific pattern. Areas with poor or no resolution revert to the erosion rate values of the Bayesian prior model. As data are added to the inversion, the solution is always a balance between the prior model and the erosion rates needed to fit the data. The resolution matrix and its integrated value reflects the weights given to the data relative to the prior. Resolution errors are always towards the prior values. In other words, if the prior erosion rate is less than the true erosion rate, and data are inadequate, the estimate of the erosion rate will be too low. If the prior is larger
than the true rate the estimate will be too high (Figure 12). If the prior is equal to the true rate, there is no error (Figures 8, 13). Distal data play a minimal role in this averaging process.

The idea that spatial differences in age, i.e., a combination of old and young ages from distinct regions, will always, or even frequently, combine to produce an apparent increase in erosion rate is false. Models in this paper were consistent in demonstrating this point. The argument that spatial variation maps into temporal variation was based on an intuitive argument
(Figure 2) that was never tested. The reason why this argument fails is that there is no temperature history that can fit multiple data that have the same closure temperature, but different ages. An inverse method based on optimization of age prediction will tend to favour spatial variability over temporal variability. With reference to Figure 2e, a model with 3 erosion rates distributed across space can fit all 9 ages perfectly. A exhumation history forced to combine disparate ages, but optimizing fit to the ages might tend towards lower rates in the early history and higher rates in the later history, dependent on fitting criterion,
but it will never fit the ages well. The GLIDE method, based on the soft constraint of spatial correlation, but designed to optimize age fit has a strong statistical tendency towards spatial variation rather than temporal variation.

With ages including multiple closure temperatures, the situation is different as a common cooling history can be found for a high closure temperature age from one region and a low closure-temperature age from another region. For example, the combination of data points a and e in Figure 2 would give a well-resolved, but false, acceleration. Fortunately, we do not
expect this case to arise often because high closure-temperature data tend to be collected in regions of fast exhumation, where they are the most useful and where erosion rates do not need to be sustained for long to bring them to the surface. It is far more common to see the application of high temperature dating methods applied in fast exhumation regions and low-temperature thermochronometers applied uniformly, or preferentially to low exhumation rate regions.

Given that resolution errors are the primary concern with thermochronometric data modelling, it is important to
quantify resolution for specific data sets and estimate resolution errors. For this reason, much of the study of Herman et al. (2013), and related papers (e.g., Fox et al., 2015, 2016; Jiao et al., 2017), have estimated resolution errors, investigating sensitivity to the prior model parameters (see supplement to Herman et al. (2013)), and developed new statistics such as the temporal resolution matrix as an integral of the resolution kernel (Fox et al., 2014). The synthetic models in this paper show that the resolution metrics do a good job in delineating relative resolution. Resolution remains a relative measure, and
determining a precise confidence level *a priori* is not possible, but can be estimated based on spatial patterns, relationship to sample locations, fit to the age data and sensitivity to the prior.

Although resolution errors do not tend towards an acceleration on their own, there are combinations of data and prior model that can lead to a false acceleration. For example, in the case of isolated young ages, less than 2 Ma, with a low prior,





the young ages give a high rate of erosion in the last time interval and if there are no higher closure temperature data nearby,

the earlier time intervals will take a value near the prior, leading to an apparent acceleration. However, the early time intervals will not be well resolved, so this should be recognized for what it is, a mix of resolved and unresolved time intervals. The danger is that the resolution does not have an absolute level for what is regarded as resolved, so there is a chance that early time intervals will be at the margin of acceptable, even though the constraining data are either too old, or too far away to constrain the erosion rate well. For this to occur it requires an unfortunate combination of ages in both space and time. Ages

must have a very specific distribution in order to raise the resolution value, but not pull the erosion rate away from the prior value. This can happen, but we have not encountered many examples of this occurrence. The danger of this combination was recognized by Herman et al., (2013) which is in part why Herman et al. (2013) made a comparison of the 6-4 Ma time interval and the 2-0 Ma time interval rather than the 4-2 Ma time interval and the 2-0 Ma time interval. Comparing the last two time intervals is easier and would still make the point that Quaternary climate change might have impacted erosion rates. However,

it is much more difficult to resolve two time intervals with a gap between them, so the 6-4 Ma time interval was used in order to make a more conservative estimate of potential accelerations by imposing a stricter condition on the resolution.

One case that we did not present in the modelling study was the extreme poor resolution that results when all data come from a single thermochronometric system. We have conducted simulations to evaluate this, but the results were as one would expect. It is impossible to find a resolved temperature history with a single thermochronometer, with the exception of a

fortuitous elevation distribution of age, in which case the estimate is both resolved and accurate. This is not surprising. When a thermal model is used as the basis of the analysis, and all ages have a common closure temperature, there is no means of resolving more than one point in time, as can easily be recognized by the principles shown in Figure 1 and Figure 2, and shown in Herman et al. (2013) Figure ED3. This is why all sites identified by Herman et al. (2013) as having sufficient resolution, have ages from more than one thermochronometer.

The other major source of error in forward and inverse modelling of thermochronometric data is in the determination of the crustal geotherm. Erosion rates derived from thermochronometric data depend on the geothermal gradient just as strongly as they depend on the measured age (Moore and England, 2001; Ehlers, 2005; Reiners and Brandon, 2006; Willett and Brandon, 2013), so it is essential that thermal models are accurate. By linking the thermal advection in GLIDE to the erosion rates derived from the ages, geotherm errors become part of the model errors in the data tests of this paper. Geotherm errors

were occasionally large in the synthetic data models, although this was in part a consequence of the extreme erosion rate scenario used in these tests and the fact that the true model was often far from the prior. This leads to a large error in the geotherm if old ages do not exist to constrain advective heating. Geotherm errors often have a smaller effect on inferred erosion accelerations than on inferred erosion rates. An acceleration or deceleration as determined by multiple thermochronometers depends, not on the gradient, but on the curvature of the geotherm. A false acceleration would be obtained if the predicted

geotherm has less curvature than reality (England and Molnar, 1990), but a systematic error in the gradient affects all rates equally and therefore does not lead to a false acceleration. This effect was demonstrated in Figure 14, where the gradient was too low, so the estimated erosion rates were too high, but there was no false acceleration. In fact, the error produced a false deceleration. In general, the transient geotherm calculated internal to GLIDE, based on a boundary condition at the base of the





lithosphere, should do a good job in predicting geotherm curvature. We would contrast these predictions to models that use a
fixed temperature at the base of the crust, as used for example by Schildgen et al. (2018) in their data summary figures. A fixed
temperature at the base of the crust suppresses advective increases in both gradient and curvature. The magnitude of these
errors was shown in Figure 3. As an error in the synthetic data models shown above, the problem was the lack of compatibility
between forward and inverse models, not the error itself. Yet the mistake was unlikely to have occurred were it not for an
increasing acceptance in the literature of fixed Moho-temperature models that are inappropriate for modelling thermo-
mechanical processes with high rates of erosion and a timescale of interest of over 10 Myrs. Fixed Moho-temperatures in
thermal models can lead to erroneous results because any perturbation to a geotherm (via erosion, basin sedimentation, faulting,
magmatism, etc) will perturb the entire lithosphere thermal field, including at the Moho boundary.  It is therefore preferable
in thermal modelling studies to apply either a fixed temperature or fixed flux boundary condition at the base of the thermal
lithosphere, or, if a simplification is required, apply a flux boundary condition at a shallower depth.
In practice, avoiding geotherm errors in GLIDE studies or other erosion rate estimations is best done by estimating
and incorporating the modern geothermal gradient obtained by heat flow studies and (if available) other geologic evidence for
P-T-t conditions at depth. If the final geotherm from the inverse model is calibrated to the observed final geotherm, errors at
the time of age closure will be small, even with a simplistic thermal model.

### 6.2 Climate or Tectonics – Chicken or Egg, Revisited

A secondary question of the Schildgen et al. (2018) site assessment was the determination of the underlying cause of
the increase in erosion rate. Is this the result of a tectonic process or a climatically modulated glacial erosion process? This
question is more important now that we have established that there are no spurious accelerations in erosion, only genuine ones.
Although this question is not central to the theme of bias in thermochronometry, it is of interest to a broader community (Kirby,
2018). Schildgen et al. (2018) found that, in cases where there is an increase in erosion rate, all but perhaps 1 or 2 cases have
a tectonic origin. Even many of the cases that they assessed as spurious were also assessed as having a tectonic "cause",
although it is not clear if they meant that tectonics was the cause of the spuriousness, or if they were just adding insurance to
the argument that glacial erosion was not important.

        This is a surprising conclusion, not so much that there is evidence for tectonics, but that this would even be attempted.
Unravelling the effects of tectonics and climate-modulated erosion has been regarded as one of the unsolved challenges in
Earth Science over the last 30 years and is commonly referred to as the "chicken or egg" paradox within the field (Molnar and
England, 1990; Zhang et al., 2001; Molnar, 2004). This problem is regarded as a paradox because of the feedback between
erosion and tectonic uplift. An increase in erosion rate due to climatic factors leads to an increase in rock uplift due to isostatic
unloading. If that uplift manifests itself through surface deformation, tectonics becomes the consequence of erosion, not the
cause. The observations would be essentially identical. Observations of erosion rates and tectonic activity are nearly always
correlated, but there is no basis to assign cause to one and response to the other because of the strong feedback.



For this reason, Herman et al. (2013) made no attempt to attribute cause to erosion rates or accelerations at any given site. This was regarded as an unanswerable question. Rather, the argument was that there were so many sites that showed an acceleration, a coincidence of a tectonic cause was not likely and Quaternary climate change leading to enhanced erosion rates was a more probable explanation for skewing the direction of the change (Molnar, 2004). Some sites almost certainly did have an increase in tectonic uplift rate; in particular in young mountain belts like Taiwan or the Southern Alps of New Zealand, where there was no topography 5 Myr ago. However, if one starts eliminating sites with determinable tectonic increases in uplift rate, where does one stop? There is too much subjectivity involved with such a complex problem, so Herman et al (2013) simply kept everything. In an unbiased and constantly changing world, half the sites would be expected to show an increase and half a decrease. Given the target timeframe of the last 6 My, young ages are needed and this gives a bias to tectonically young, active mountain belts. This was acknowledged by Herman et al. (2013). This presents a bias towards high erosion rates, but it does not follow that this leads to a bias towards recent acceleration.

In contrast, Schildgen et al. (2018) offered an assessment as to the cause of each change in exhumation rate, based on a reading of the literature. Complicating factors including the difficulty of establishing cause and effect in a system with feedback were not discussed. Nor do they explain how they established timing or rates of tectonic activity without referring back to the original thermochronometry data. Rather, the approach used was to search the literature for evidence of active tectonics and, if found, they attributed not just young ages, but also recent acceleration, to tectonics.

To be relevant to the chicken and egg debate, it is necessary to establish, not just tectonic activity, but accelerated rates of activity over the last 6 Ma. Determining rates of tectonic deformation in the past is very difficult; establishing rates of acceleration is nearly impossible. Add to this the fact that the primary means to date or establish rates of tectonic processes over million-year timescales is thermochronometry, and the problem is even greater. To use thermochronometry to establish tectonic deformation rates to explain thermochronometric cooling rates would be circular. The alternatives are few. There are a few isolated cases where geochronology can be used, for example on syn-tectonic volcanic flows, but really the only method capable of resolving tectonic deformation rates across a 5 Ma timescale with enough precision to identify changes in rate is sedimentary growth strata with established bio- or magneto-stratigraphy. These do not exist at any but a handful of the sites studied in both papers because most sites are in highly exhumed, often crystalline-cored mountain belts. In the two study sites where we are aware of relevant growth strata data (Alps and Apennines) Schildgen et al. (2018) did not make reference to these studies. Each of these studies show decreases in tectonic shortening rates over the same time frame that erosion rates are increasing (Willett et al., 2006; Boccaletti et al., 2010).

## 6.3 Summary of problems in Schildgen et al. (2018)

The primary hypothesis of Schildgen et al. (2018) is that there exists a "spatial correlation bias" in all the results of Herman et al. (2013), although they were vague as to exactly where that bias arose, from the methodology, or from the methodology as applied to sparse data. The analysis above found no evidence of systematic bias from either source, so it is



important to evaluate why Schildgen et al. (2018) came to such different conclusions. We summarize here why we think they came to a different outcome.

1275   Schildgen et al.'s (2018) first analysis, constructing forward-inverse models with synthetic data, in our view, was executed incorrectly. As we documented above (Figures 3, 4) they calculated the synthetic ages incorrectly, so these models show only that it is impossible to correctly recover model parameters given data from a different model. These models were the only test they provided of their hypothesis. Aside from the error in age calculation, the test would have been incomplete because of the vagueness of the hypothesis. By not specifying whether the spatial correlation bias was a methodological flaw

(model bias) or a consequence of data inadequacy (resolution bias), they paid little attention to the synthetic data distribution, constructing only a single data set for each setting, without specifying whether data were intended to be dense and therefore providing a test for model bias, or sparse, thereby testing for resolution bias. Nor did they demonstrate that the data distributions had the same resolution characteristics as any of the natural datasets. We conducted a more extensive set of synthetic data tests (Section 4) and demonstrated that there is no model bias (Figures 6, 7, 8, 9, 13), but that there are significant resolution errors

with sparse data. However, these resolution errors are not characterized by excessive spatial averaging of other data, but, rather, show that an estimated parameter remains close to, or at, the prior value specified for the erosion rate, which is the usual result of Bayesian models with poor resolution. This is a type of bias, but is a completely different mechanism from that proposed by Schildgen et al. (2018). Resolution errors of this type do not have a universal bias towards a false acceleration. Specific combinations of the prior model and the data distribution can result in an error with acceleration (Figures 10, 11, 15), but it

can also lead to errors with false decelerations (Figure 14).

   The second part of the Schildgen et al (2018) critique was a reinterpretation of the Herman et al. (2013) results through their proposed normalized erosion rate difference (**NR**) maps. This analysis is also problematic because the division of two random variables is a biased operation, and this bias is towards larger numbers as the denominator becomes small. The **NR** maps of Schildgen et al. (2018) took the results of Herman et al. (2013), reprocessed them using this biased operator (**NR**),

identified that the resultant map contains bias and concluded that the original analysis was flawed. We find this logic questionable. One can get a sense as to how much bias is introduced by the **NR** operator by propagating the posterior parameter variance through the normalization and plotting the corresponding variance to the **NR** (Equation 14). We showed that vast swaths of the maps including the largest values of **NR** being interpreted by Schildgen et al. (2018) were statistically insignificant. The **NR** bias has a tendency to shift the locus of largest values of acceleration towards regions with low average

erosion rate, because of the effect of division by small numbers. The result is a systematic shift and amplification of the signal found by Herman et al. (2013) (e.g., Figures 17, 20, 21, 23), so that the locus of high values always sits offset from the data locations and the original, resolved signal. This distortion played a major role in the Schildgen et al. (2018) interpretation of spatial averaging bias. Schildgen et al. (2018) repeatedly interpret these signals as the result of inappropriate spatial averaging. As far as we can determine, all these interpretations are incorrect. Every one of these is a case of interpreting an **NR** that is

mis-located with respect to the data and/or is statistically insignificant. These interpretation errors were compounded by the fact that Schildgen et al. (2018) seemed to be unaware of the tendency of the solution to go to the prior model in the case of poor resolution, although this was stated in Herman et al. (2013) and Fox et al. (2014). Their interpretation states multiple



times that in the absence of local data control, solutions were obtained by averaging data from distal locations. In most of these cases, the resolution is low and the erosion rate has simply gone to the prior value with no influence of distal data.

The third and final part of the Schildgen et al. (2018) study was the site by site analysis of the Herman et al. (2013) results along with their interpretation as to what processes were responsible for exhumation in specific settings. Their site-specific assessment is repeating the errors described above. It should be noted that this part of the Schildgen et al. (2018) paper is an interpretation, not an analysis. There is no reproducible methodology applied. It is simply an interpretation of past studies, including Herman et al. (2013), and an assessment of the raw data. This type of analysis is notoriously susceptible to another

type of bias, not yet discussed, which is known as confirmation bias. This emerges when some observations are selectively used, and others selectively ignored in order to support a preconceived hypothesis. We see ample evidence for this bias throughout the Schildgen et al. (2018) interpretation. Because their working hypothesis was that the increases in erosion rate found by Herman et al. (2013) were a consequence of spatial gradients in ages from single thermochronometric systems, often averaging data from large distances, these conditions were searched for and, if found, were taken as causal, even when much

simpler and better interpretations were available (Section 5.3). Even simple modelling studies, where tectonic kinematics and predictions are unambiguous, were represented incorrectly as demonstrating spatial gradients that were not present in the original studies (Figure 16). The modelling work presented above demonstrated that high resolution can only be obtained by multiple chronometers or age-elevation relationships crossing the key time interval, so the sites identified by Herman et al (2013) were significant for the existence of these critical characteristics of the data. Schildgen et al. (2018) never present or

discuss these aspects of the data. Age-elevation relationships are not systematically discussed or included in any figures, nor are the occurrence of higher closure-temperature thermochronometers systematically discussed. In addition, although many previous studies using a variety of other interpretation methods found results that support Herman et al. (2013) (e.g., Zeitler et al., 1982; Ehlers et al., 2006; Thiede and Ehlers, 2013; Michel et al., 2018; Vernon et al., 2008; Shuster et al., 2005; Sutherland et al., 2009; Thomson et al., 2010a,b; Avdeev et al., 2011; Shuster et al., 2011; Ballato et al., 2015; Bracciali et al., 2016), none

of these studies swayed an interpretation away from their "spurious" assessment.

A similar confirmation bias was shown in the attribution of tectonic causation over climatic change. Decades of research and numerous papers have addressed this topic without clear resolution, but Schildgen et al. (2018), solely on the basis of a literature review, conclude that evidence of high glacial erosion rates in the Quaternary is limited to a single locality. However, all assessments of tectonic cause listed in Schildgen et al. (2018)'s Table 1 are simply correlations between tectonic

activity and high erosion rates. Their paper joins quite a few others that have missed the difference between cause and correlation (Molnar, 1990), but it is surprising that Schildgen et al. (2018) give such a definitive answer to such an elusive question. Particularly given that none of the assessments provided in their paper present sufficient dating, independent of the thermochronometric data, to establish a tectonic *acceleration*, and none acknowledge the important feedback between tectonics and erosion that have defined this paradox for decades.




### 6.4 Summary of Problems in the study of Herman et al. (2013)

The study of Herman et al. (2013), like any study, could be improved. With the insights of more than 6 years of additional work using the inversion method, we have learned much that could be applied to the global assessment (Ballato et al., 2015; Herman and Brandon, 2015; Fox et al., 2015, 2016; Margirier et al., 2015; Yang et al., 2016; Bertrand et al., 2017; Jiao et al., 2017; Siravo et al., 2019; Vincent et al., 2019). However, the substantive changes that we would make would be to the way we handled the post-processing of the results. We see no major problems with the inverse method, but the treatment of the resultant erosion rate fields has some errors.

Herman et al. (2013) summarized the magnitude of the change in erosion rate by taking two time intervals (6-4 Ma and 2-0 Ma), taking their ratio on a point by point basis and compiling their distribution into a histogram (Herman et al. (2013), Fig 2). This approach is subject to the same ratio bias as the normalized difference used by Schildgen et al. (2018). A ratio analysis should never be used when the goal is to estimate the magnitude of a change, the range of values and uncertainty is large and some values are approaching zero.

This bias is avoided by using a difference instead of a ratio. We demonstrate this by recalculating the change of erosion rate found by Herman et al. (2013) replacing the ratio of the 6-4 Ma time interval and the 2-0 Ma interval with a simple difference between the two (Figure 24). We have done this for 3 values of the resolution cutoff, all larger than the .25 used in Herman et al. (2013). The principle effect of the change to a difference is to truncate the distribution reported in Herman et al. (2013) by removing the largest values. These large values were the consequence of small, but uncertain, numbers in the denominator of the ratio. However, the general form of the distribution and the positive mean remain the same as in Herman et al. (2013). The principle result of Herman et al. (2013) is thus unaffected by this bias.

As the resolution cut-off is increased, the number of resolved points is reduced (Figure 25). With a cut-off value of bias of 0.5, there are fewer than 200 points worldwide. However, these remain distributed globally, and a resolution cut-off of 0.4 gives over 500 points with a much broader spatial distribution. The mean change in erosion rate increases slightly with the increase in the resolution cut-off value. This indicates that the increase in erosion rate is defined by the best resolved points and the effect of a lower resolution cut-off is primarily to include more of the peripheral regions, where this signal is smoothed into surrounding regions.

Resolution and other posterior variables depend on a wide range of data parameters and inverse model parameters, so their absolute value is often scalable. In particular, they are subject to the well-known trade-off in inverse theory between resolution and variance reduction. To reduce the effects of noise in the data, resolution is sacrificed through spatial averaging. The tradeoff is between a well-resolved *average* parameter vs. a poorly-resolved *local* parameter. With reference to Figure 2d, e, data a, b, c, define a local erosion rate, whereas data a-i define the regional erosion rate (Figure 2e, *Average 1*). Neither estimate is biased, but the regional average is controlled by 9 points, whereas the local erosion rate is resolved by only 3 points, so has less variance reduction. Resolution and variance reduction scale against one another, so that high resolution corresponds to less reduction of variance. The degree of regularization of the model also comes into the absolute values of the resolution and variance. This comes primarily through the damping that occurs by setting a noise variance on the data (Equation 3). If data are regarded as infinitely accurate, the noise variance goes to zero. In this case the resolution goes up to nearly one





everywhere. If the data are regarded as less accurate and the noise variance is large, the resolution may never exceed some moderate value. The prior variance plays a similar role in scaling the posterior values. The point is that resolution and posterior variance should be regarded as relative values, not absolute values. A better approach to that taken by Herman et al. (2013) or our analysis in Figures 24 and 25 would be to use an individual value for each local data set. In addition, the correlation length

parameter should likely be calibrated to individual settings based on data density and tectonic setting. The use of a constant value is not inappropriate given that all inversions were run with the same parameters, but there is likely too much inclusion of peripheral regions in some examples. The assumption was that sceptical readers could use a more conservative resolution cut-off if they disagreed with the published selection. It was not anticipated that someone would focus on the poorly resolved peripheral regions in an attempt to discredit the method.

What we call resolution here is actually only the temporal resolution, as discussed above and in Fox et al. (2014). The spatial resolution has been integrated into this value. This was done because the spatial resolution is easily read from the data maps. It reflects the spatial distribution of the ages, with some complexities due to the ages themselves and which time intervals they are resolving. The best way to visualize the resolution in both space and time is by combining the information the way we have done in all the maps of this paper, showing the data that fall within a time interval on top of the temporal resolution

field. It might also be helpful to break this out by individual thermochronometer system, but this begins to be complicated. To also show elevation information is likely intractable. The other available method to establish resolution is to test for sensitivity to the prior model. This requires running multiple inverse models, but if resolution and bias to the prior are issues, this provides a good indication of the resolution provided by the age data.

A change in the resolution analysis would not change the conclusions of Herman et al. (2013). The best resolved parts

of the models are the ones with multiple thermochronometers or elevation gradients and it is these that serve as the basis for the signal of increasing erosion rate. Herman et al. (2013) did check for sensitivity to the prior (see supplement to their paper) and took into account data distributions and multiple posterior metrics, so the interpretation is fundamentally sound and we see no reason to alter the conclusions of the original paper.

**7.0 Conclusions**

An extensive error analysis of the thermochronometric age inversion method of Herman et al. (2013) (GLIDE) yielded a number of conclusions, many of which are relevant to other studies based on thermal modelling of thermochronometric ages for cooling or exhumation histories. We summarize as follows.

(1) The only significant model errors in GLIDE are associated with the calculation of the geotherm, demonstrating the importance of calibration against modern heat flow measurements or past P-T-t constraints and inclusion of

appropriate physical boundary conditions.

(2) Resolution errors are present in all thermochronometric age inversions, but vary from effectively zero with high data coverage to very important with sparse data. Temporal coverage, i.e., a wide range of ages with different closure



temperatures is much more important to constrain resolution than spatial distribution. At low resolution, solutions tend towards the prior value specified in the inversion.

(3)   Posterior error metrics including resolution and posterior variance provide an accurate measure of resolution errors, as does testing for sensitivity to the prior. Metrics are relative, not absolute measures. Posterior metrics do not accurately reflect model (e.g., geotherm) errors.

    (4)   The spatial correlation bias hypothesized in Schildgen et al. (2018) does not exist in any significant way. We found no false increases in erosion rate due to spatial averaging of ages from a single thermochronometric system that would

not be recognized as under-resolved.

    (5)   It is possible to obtain errors with false increases in erosion rate as resolution errors, but only for specific combinations of ages and the prior model. As resolution errors, they are characterized by bias to the prior, not spatial averaging, and are recognizable through the resolution statistics and sensitivity to the prior. With high data density, resolution errors are near zero.

(6)   The basis for the conclusions of the Schildgen et al (2018) paper was a combination of incorrect calculation of the ages used in their test models, post-processing of GLIDE inversion results using a biased operator, and a set of subjective interpretations, often incompatible with previous studies.

    (7)   The original conclusions of Herman et al. (2013) remain valid. The interpretation as to the underlying cause as tectonic or climate change is, as it was in 2013, open. The question as to the adequacy of the sample size to characterize global

changes also remains open. However, we find no evidence for a bias in the analysis method, or any reason to disregard the conclusions of that study.

**Code Availability**

All codes including input files used for the modelling are available at: https://github.com/cirederf13/glide/

**Author Contributions**

Research project and paper were conceived by SDW and FH. GLIDE code including resolution and error tracking was developed and written by FH, MF and RJ. Natural examples were modelled and researched by NS, TE, and RY. Error analysis was developed by SDW and FH. Models were constructed and executed by FH. Paper was written by SDW and FH, with contributions from TE, NS, and MF.

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





| Figure | Data | Correlation Length | Prior Erosion Rate | Notes | Results |
|---|---|---|---|---|---|
| Figure 6 | Set A 326 data 5 systems | 30 | .35 | Fixed Geotherm | No error |
| Figure 7 | Set A 326 data 5 systems | 100 | .35 | Fixed Geotherm High correlation length | No error |
| Figure 8 | Set C 143 data 3 systems | 30 | True solution | Fixed Geotherm Poor temporal coverage | No error |
| Figure 9 | Set B 48 data 4 systems | 30 | .35 | Fixed Geotherm Poor spatial coverage | No error |
| Figure 10 | Set C 143 data 3 systems | 30 | .35 | Fixed Geotherm Poor temporal coverage | Large errors |
| Figure 11 | Set C 143 data 3 systems | 30 | .65 | Fixed Geotherm Poor temporal coverage | Large errors |
| Figure 12 | Set C 143 data 3 systems | 30 | 1.65 | Fixed Geotherm Poor temporal coverage | Small to Moderate error |
| Figure 13 | Set D 326 data 5 systems | 30 | True Solution | Transient Thermal Model | No error |
| Figure 14 | Set D 326 Data 5 systems | 30 | .35 | Transient Thermal Model | No error |
| Figure 15 | Set E 42 Data 5 systems | 30 | .35 | Transient Thermal Model Sparse Data | Moderate error |

Table (1): Parameters and results for synthetic data models.



Earth **Surface**
**Dynamics**
Discussions



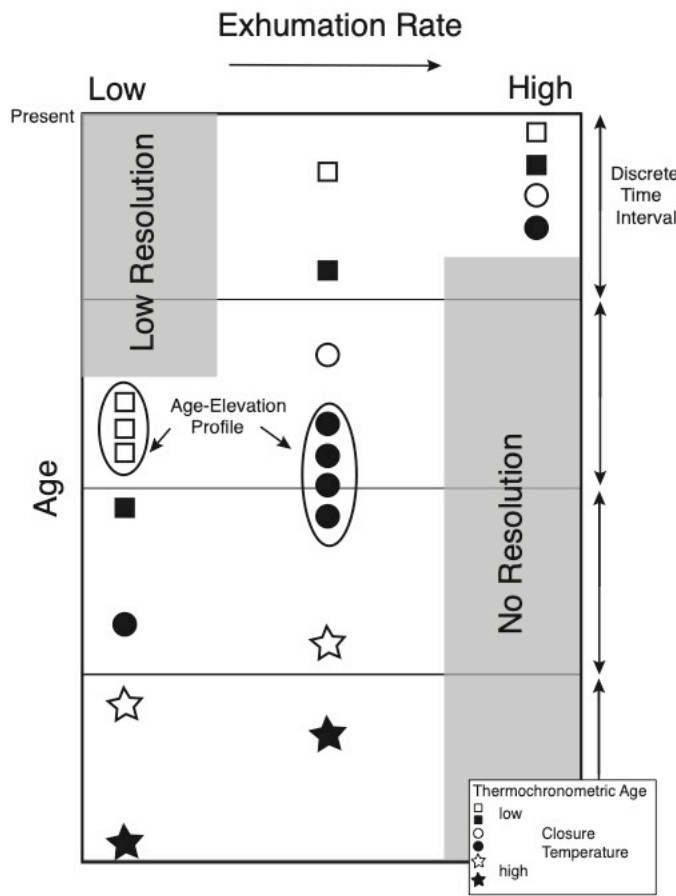

**Figure 1: Illustration of how the distribution of thermochronometric ages constrains exhumation rate in time and for different exhumation rates. Discrete time intervals as used in an inverse model are indicated. Hypothetical ages are shown for 6 thermochronometer systems with different closure temperatures. Age-elevation relationships extend age range for a single thermochronometer as shown by ellipses enclosing ages with the same symbol. Note that there is no resolution for times greater than the oldest age at a site. For time intervals younger than the youngest age, only a single (low resolution) time interval can be resolved.**


Earth **Surface**
**Dynamics**
Discussions

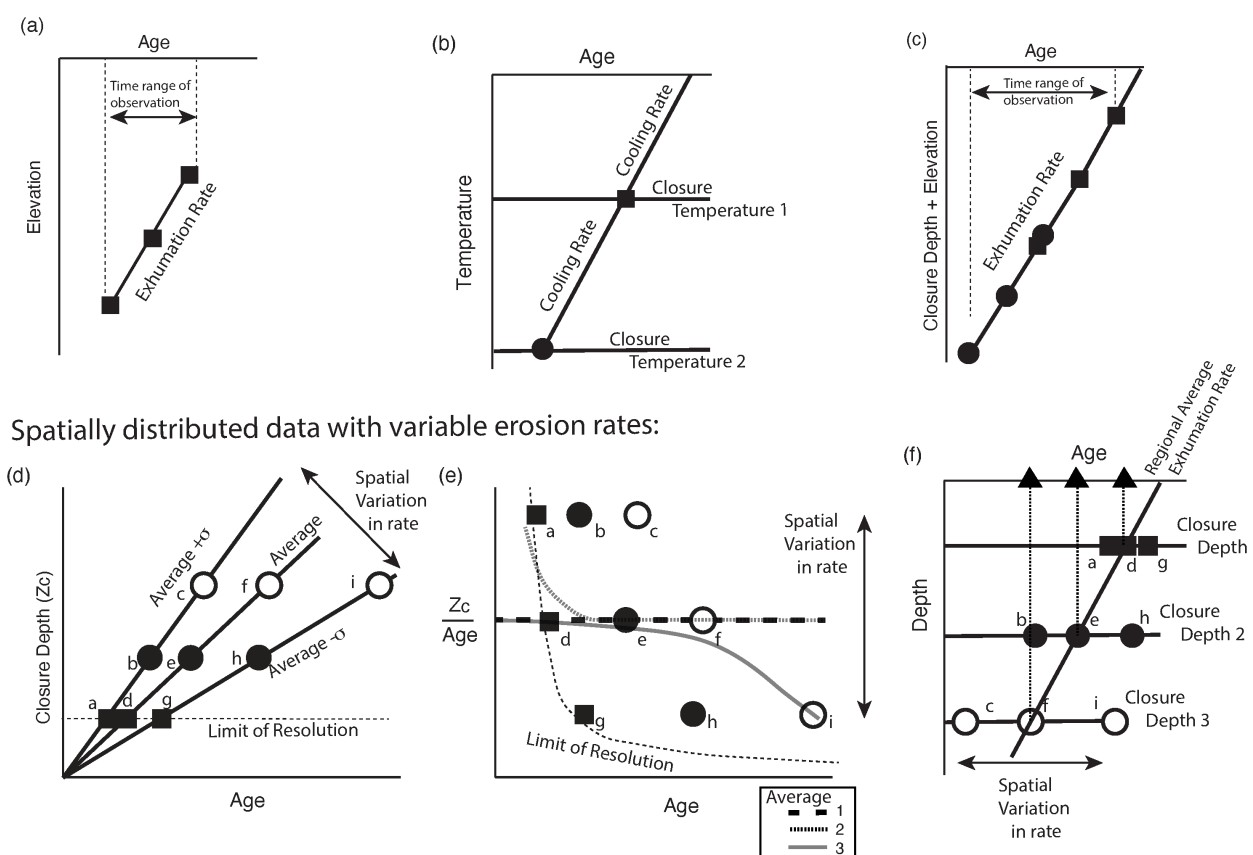

**Figure 2: Methods for combining thermochronometric ages with variability due to spatial variations in exhumation rate or elevation.**
**Squares and circles represent thermochronometers from independent systems as in Figure 1. (a) Plot of age against elevation for three, effectively co-located data. Slope is an estimate of exhumation rate. (b) multi-thermochronometer method for determining exhumation rate over two intervals by measuring cooling rate between time of passage of closure temperature. (c) Combining elevation and multi-thermochronometer methods, using a thermal model to calculate closure depths. (d) Hypothetical age distribution for three thermochronometers across a region including three spatial domains with different exhumation rates.**
**Exhumation rates are at the regional average and at plus or minus some deviation, $\sigma$, from the average. Limit of resolution shows the age below which no change in exhumation rate can be resolved. (e) Several proposed methods for averaging the ages of (d) to produce an exhumation rate. "Average 1" represents the average rate of all 9 ages and is an unbiased estimate of the regional median rate. Average 1 is also obtained by taking the median age for each closure temperature and converting this to a rate as in (f). "Average 2" is obtained by averaging only the ages that fall within a specific time window. "Average 3" is obtained by averaging the rates**
**from all ages older than a specific time. (f) Unbiased estimate of regional exhumation rate obtained by averaging all ages for a given closure depth from (d). Triangles give median age for passage through each closure temperature. Combining the median ages and the corresponding closure depths into an exhumation history corresponds again to "Average 1" in (e), but with temporal resolution corresponding to the number of closure depths. Methods in (a), (b), (c), (f) and "Average 1" in (e) are unbiased estimates of local or regional median exhumation rate, although the resolved timespan varies. Averaging methods 2 and 3 in (e) are biased, although over**
**different time spans.**

Earth **Surface**
**Dynamics**
Discussions

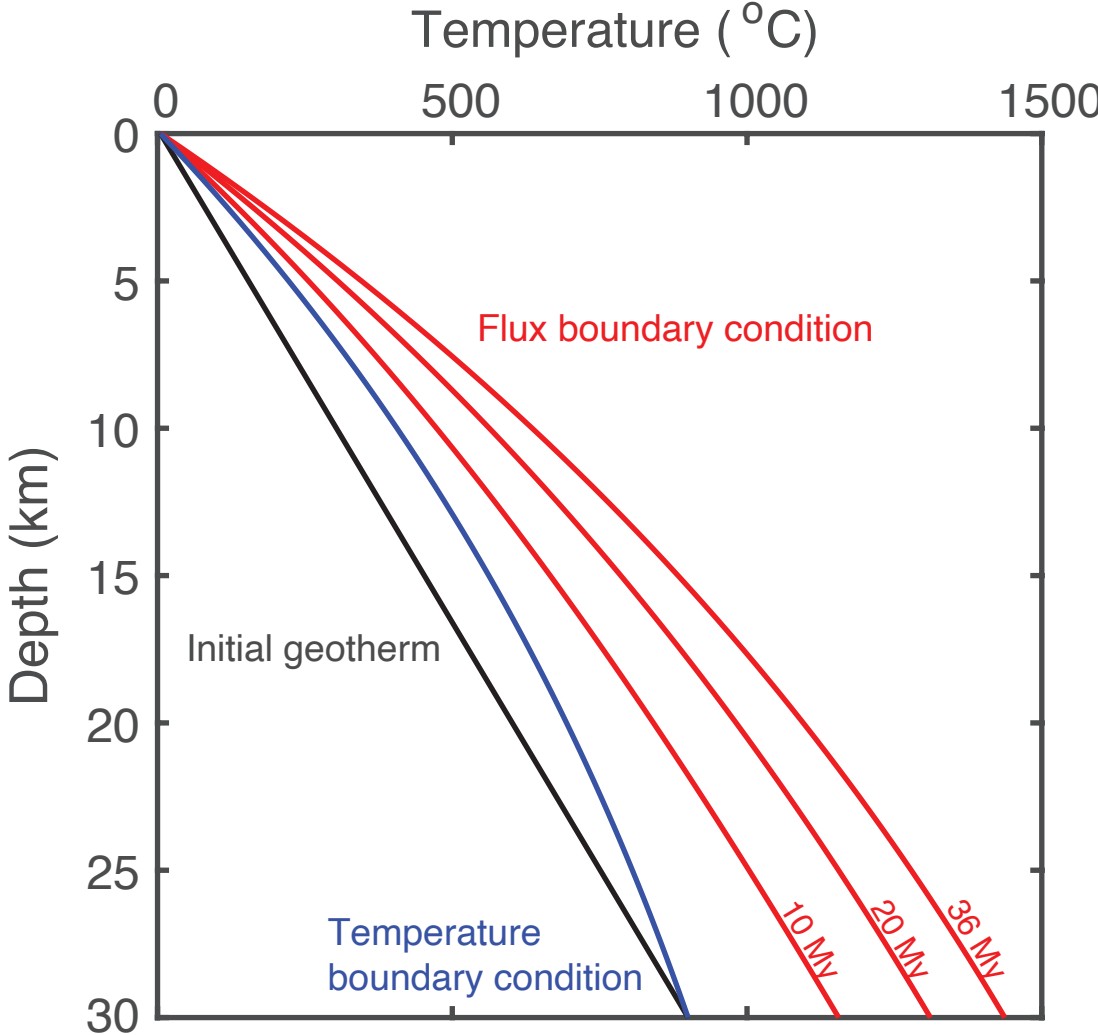

**Figure 3: Influence of basal boundary condition on the geothermal gradient. a) Two models are shown. Both initiate at a constant gradient (black), and with identical thermophysical properties. The blue line is the geotherm after 20 Myr assuming an exhumation rate of 1 mm/yr and using a fixed temperature at a depth of 30 km. By 20 Ma, the geotherm has reached a steady state. The red lines are the transient geotherms for an exhumation rate of 1 mm/yr and using a constant gradient boundary condition applied at 30 km. Steady state will not be reached for another ca. 50 Ma. In GLIDE, ages are predicted on one of the red curves through the course of the model run. Schildgen et al. (2018) used synthetic ages calculated using the blue geotherm, inverted them using temperatures predicted from the red geotherms, and concluded that the failure to recover the correct exhumation rate demonstrated a problem with spatial correlation in the GLIDE inversion method.**






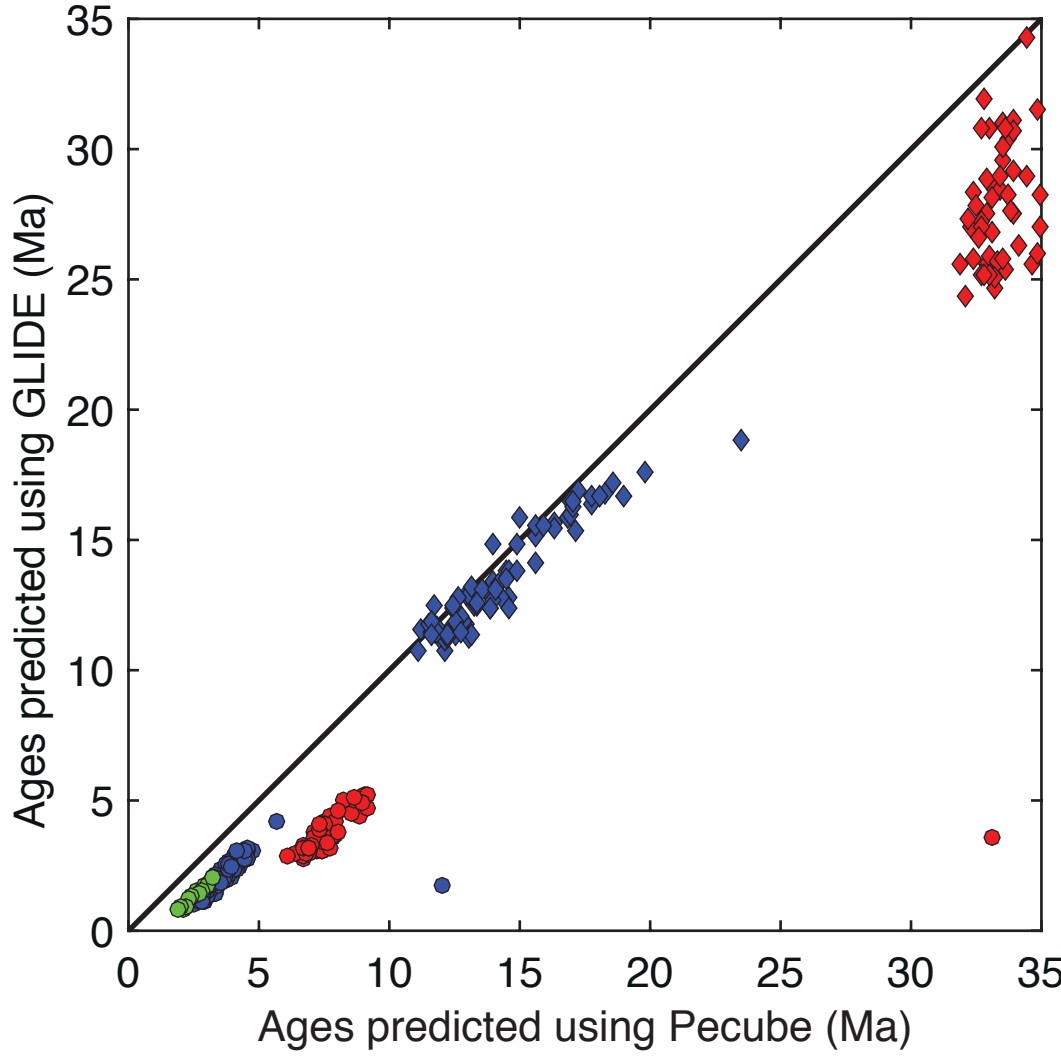


**Figure 4: Comparison between ages predicted with Pecube using a constant temperature boundary condition at 30 km (Braun et al., 2012) and GLIDE (Fox et al., 2014) using a flux boundary condition at 30 km. The ages were generated assuming the same exhumation rates (i.e., 1 mm/yr NW of the Penninic Line fault and 0.25 mm/yr SE of Penninic Line fault). Green points are AHe ages, blue points are AFT ages and red points are ZFT ages. The two exhumation rates zones are represented by circles (1 mm/yr)**
**and diamonds (0.25 mm/yr), respectively. Outliers in lower right corner were likely placed on wrong side of the fault in one of the models and were removed from the inverse analysis.**





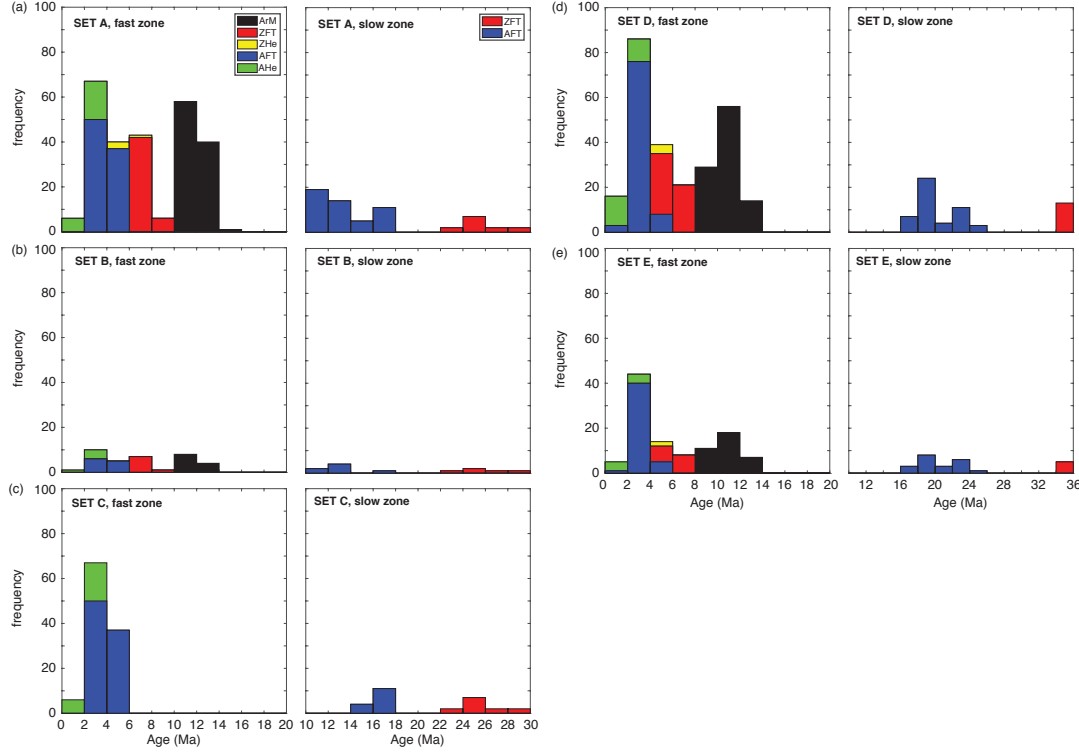

**Figure 5: Synthetic age data sets for model bias and resolution tests, comprising two zones with differing erosion rate. Colors represent ages from different thermochronometer systems corresponding in closure temperature to: green: AHe, blue: AFT, yellow: ZHe, red: ZFT and black: muscovite $^{40}$Ar/$^{39}$Ar. Data Sets A, B, C were generated using a constant geothermal gradient. Data sets D, E were generated using the GLIDE transient thermal model with a flux boundary condition.**


Earth **Surface**
Dynamics
Discussions

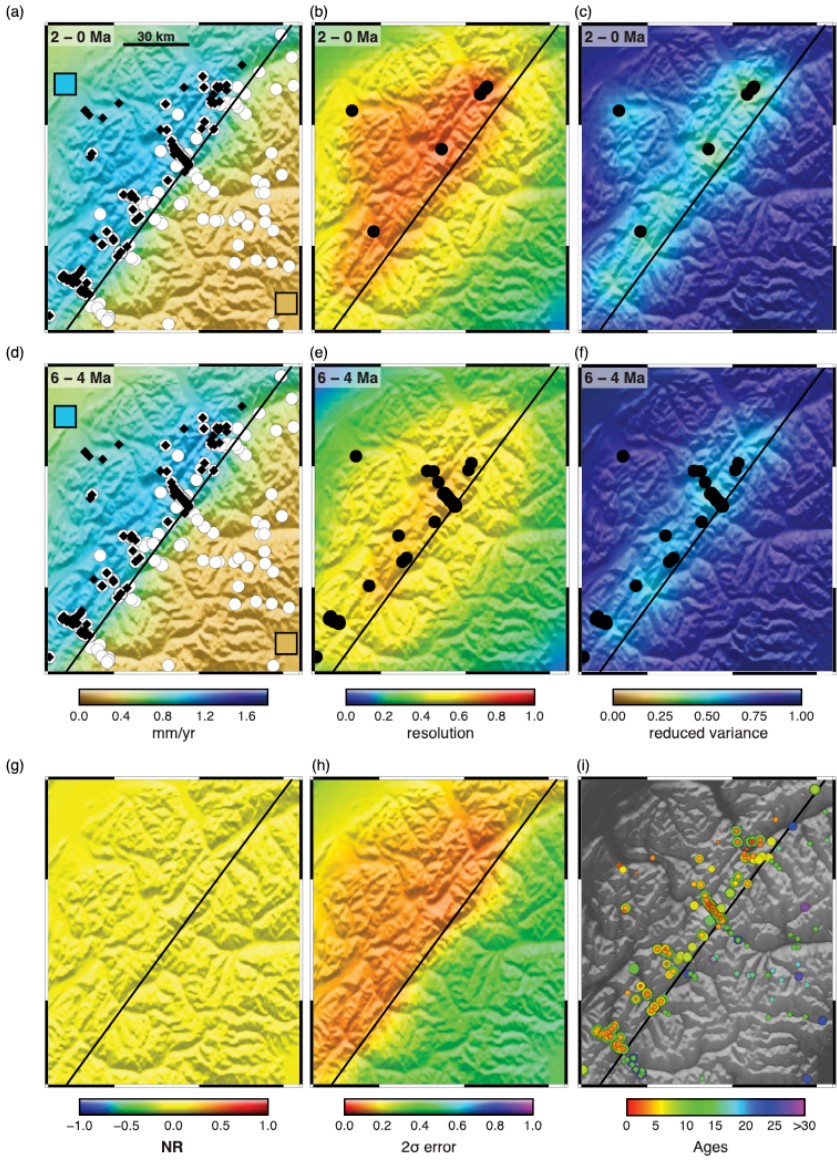

**Figure 6:** Synthetic data test for model errors, also referred to as model bias. True solution is constant exhumation rate through time, with high exhumation rate (1 mm/yr) in upper left corner and low exhumation rate (0.25 mm/yr) in lower right corner identical to the test proposed by Schildgen et al. (2018). Inset boxes in (a) and (d)show the true solution. Thermal model is replaced with a constant geothermal gradient to remove potential errors associated with the geotherm calculation. The correlation length scale (30 km) is given by the black bar in (a). Other parameters given in Table (1). Synthetic ages are shown in Figure 5a and (i) where the size of the point corresponds to one of four closure systems, AHe, AFT, ZHe, or ZFT; these are calculated with a constant geothermal gradient in both the forward thermal and inverse model. Data points in (a) and (d) show age locations with ages less than 6 Ma as black diamonds; ages greater than 6 Ma as white circles. Black data points in (b), (c), (e) and (f) show the locations of ages that fall inside the respective time interval. Estimated exhumation rate is shown with the temporal resolution and posterior variance for the time intervals indicated. (g) and (h) show the normalized exhumation rate difference (NR) and its posterior 2σ error. True solution is recovered very well in the center region where data density is high. No spurious acceleration is visible. Note that the lower right domain has no ages under 6 Ma, but still shows no spurious acceleration.




Earth **Surface**
**Dynamics**
Discussions

l780

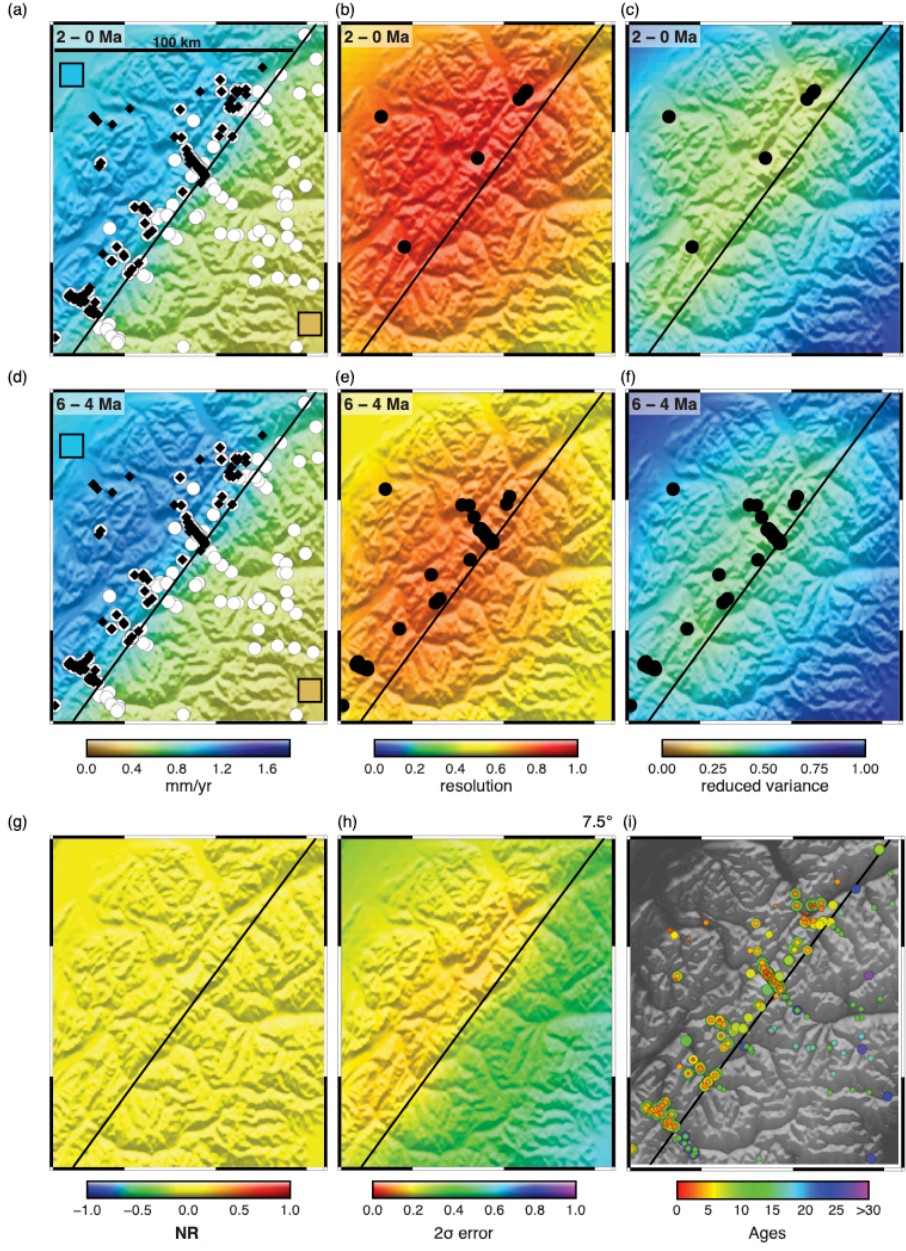

**Figure 7:** Synthetic data test for model error with a correlation length scale of 100 km. True solution is constant exhumation rate through time, with high exhumation rate (1 mm/yr) in upper left corner and low exhumation rate (0.25 mm/yr) in lower right corner identical to the test proposed by Schildgen et al. (2018). Inset boxes in (a) and (d) show the true solution. Thermal model is replaced with a constant geothermal gradient to remove potential error in the geotherm calculation. The correlation length scale is given by l785 the black bar in (a). Other parameters given in Table (1). Synthetic ages are shown in (i) where the size of the point corresponds to one of four closure systems, AHe, AFT, ZHe, or ZFT; these are calculated with a forward thermal model identical to the inverse model. See caption to Figure 6 for additional details. The correlation length scale is more than 3 times that used in Figure 6. True solution is recovered very well. No spurious acceleration is visible.

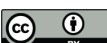



Figure 8: Synthetic data inversion test using reduced data density Set C (Figure 5) and a prior erosion rate of 1.0 mm/yr in the NW corner and 0.25 mm/yr in the SE corner. This is equivalent to the true erosion rate used to generate the synthetic ages. Other inversion parameters in Table 1. See Figure 6 caption for other formatting details.

790



**Figure 9: Synthetic data inversion test using reduced data density Set B (Figure 5b). Data have good coverage in time, but poor coverage in space. Prior erosion rate is 0.35 mm/yr. Other inversion parameters in Table 1. See Figure 6 caption for other formatting details.**

1795



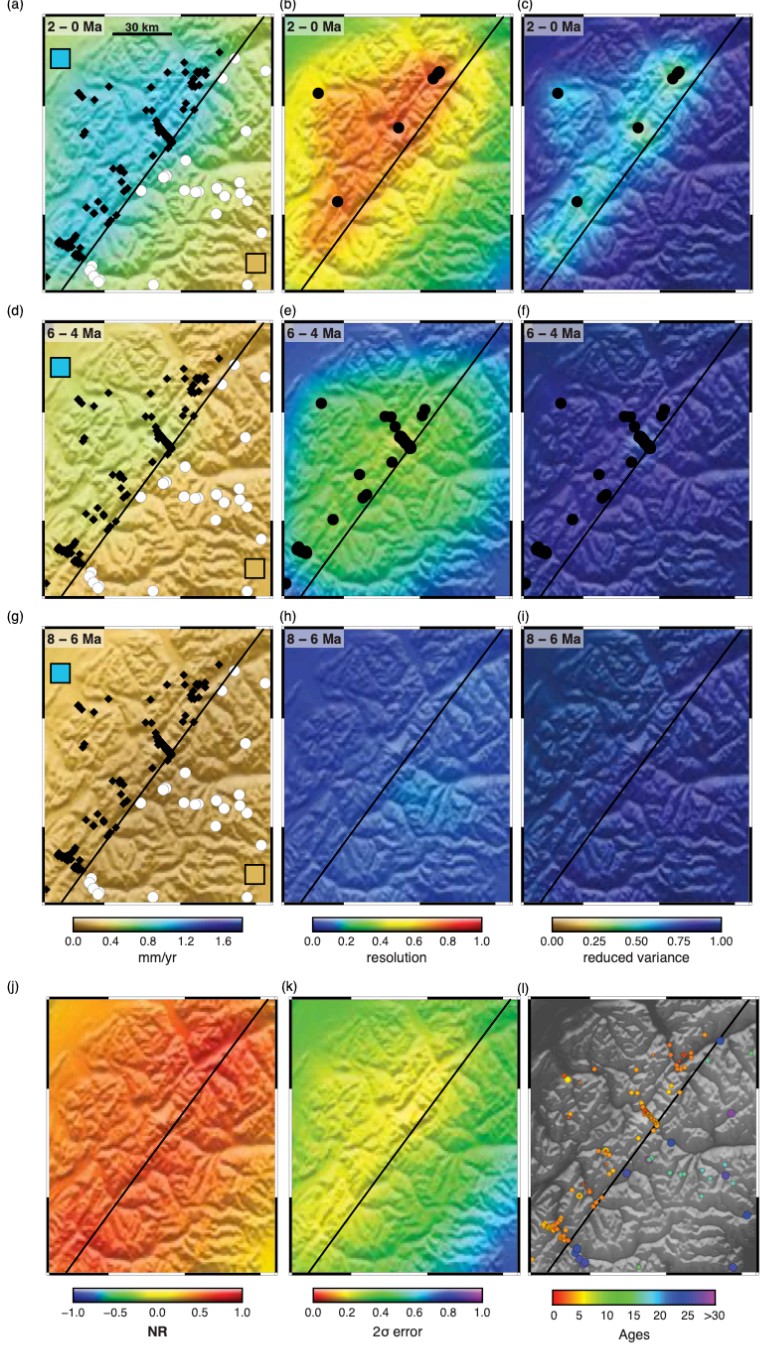

**Figure 10: Synthetic data inversion test using reduced data density Set C (Figure 5c). Data have moderately good coverage in space, but poor coverage in time. Prior erosion rate is 0.35 mm/yr. Other inversion parameters in Table 1. See Figure 6 caption for other formatting details.**

800



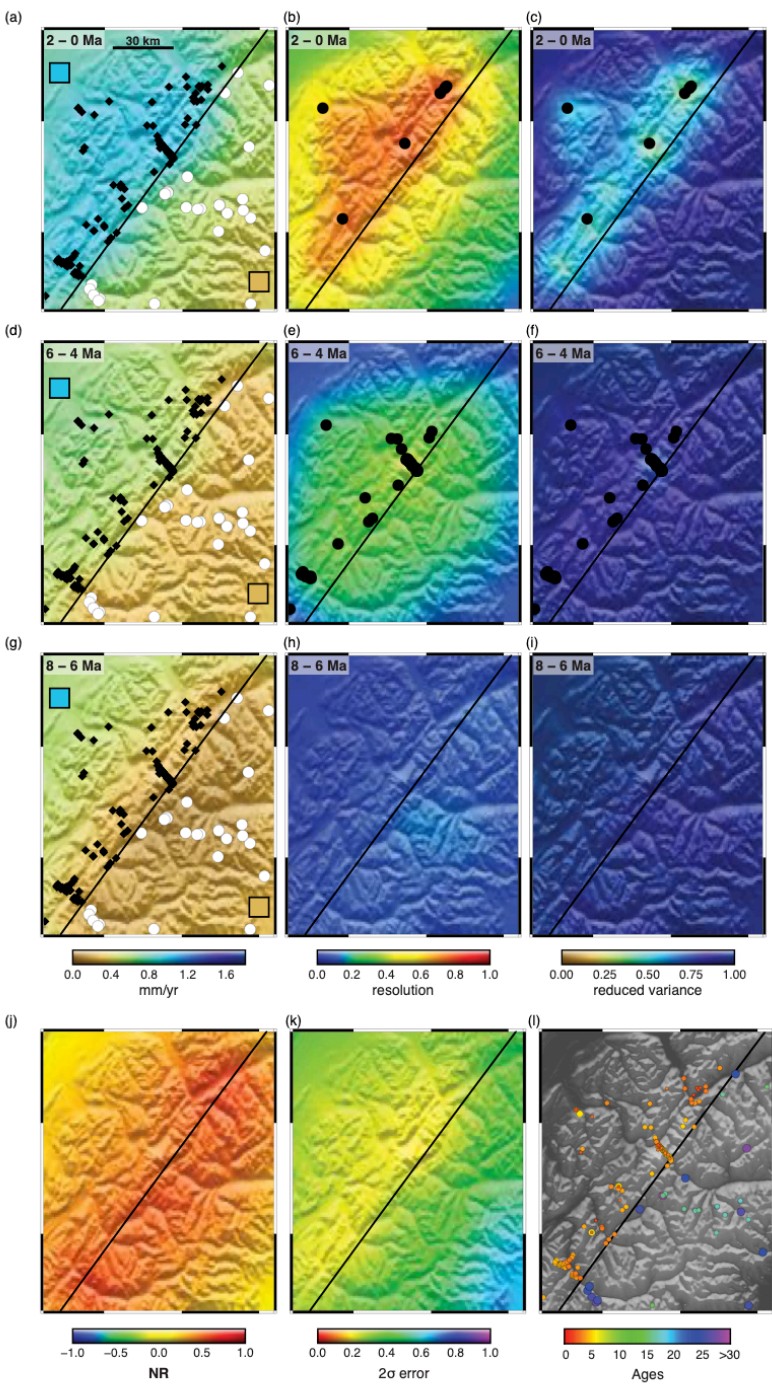

**Figure 11: Synthetic data inversion test using reduced data density Set C (Figure 5c). Prior erosion rate is 0.65 mm/yr. Other inversion parameters in Table 1. See Figure 6 caption for other formatting details.**





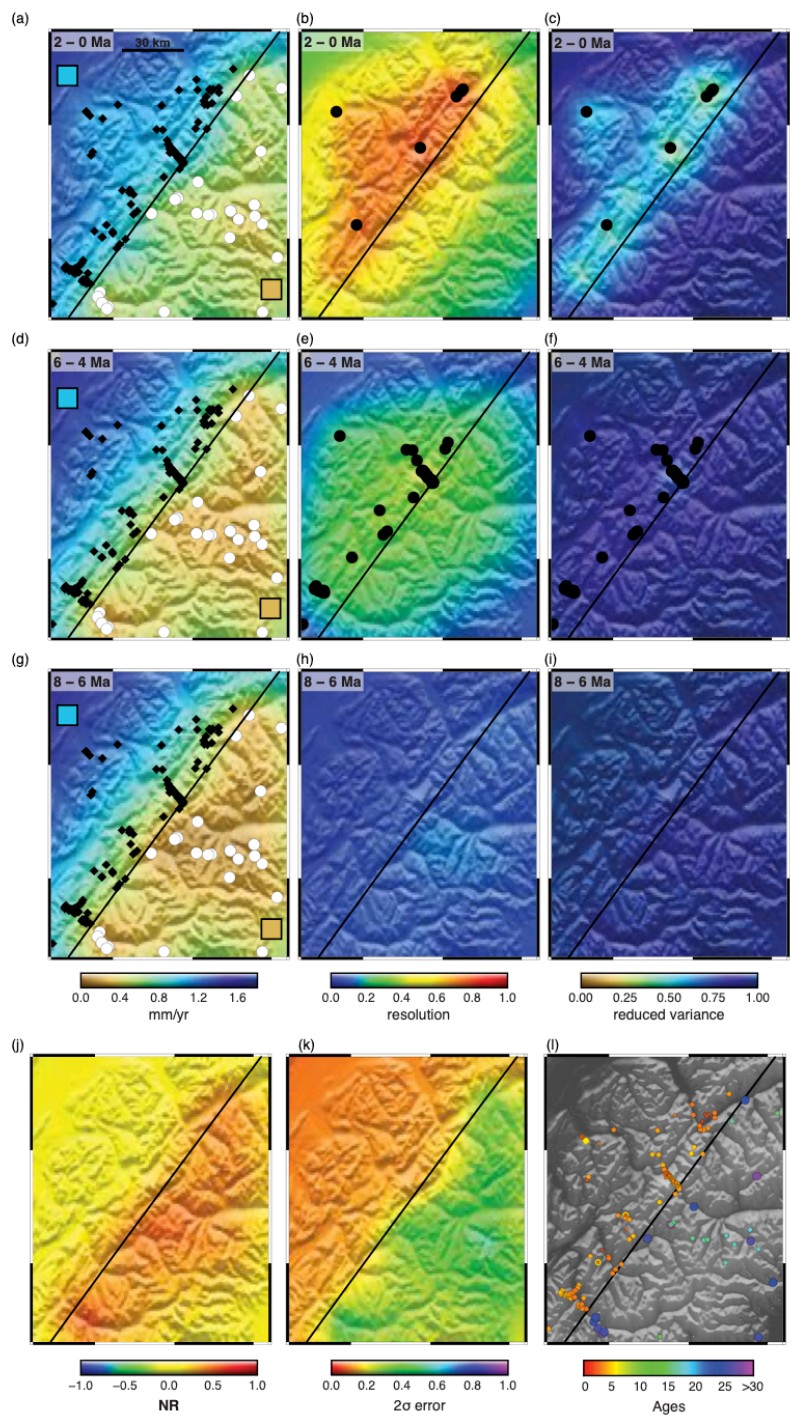

**Figure 12: Synthetic data inversion test using reduced data density Set C (Figure 5c) and a prior erosion rate of 1.65 mm/yr. Other inversion parameters in Table 1. See Figure 6 caption for other formatting details.**




Earth **Surface**
**Dynamics**
Discussions

EGU



**Figure 13: Synthetic data inversion test using synthetic Data Set D calculated using the full, transient thermal model internal to GLIDE (Figure 5). Prior erosion rates of 1.0 mm/yr in the NW corner and 0.25 mm/yr in the SE corner are used. This is equivalent to the true erosion rate, used to generate the synthetic ages. Other inversion parameters in Table 1. See Figure 6 caption for other formatting details.**





**Figure 14: Synthetic data inversion test using synthetic Data Set D calculated using the full, transient thermal model internal to GLIDE (Figure 5). The prior erosion rate is 0.35 mm/yr. Other inversion parameters in Table 1. See Figure 6 caption for other formatting details.**






**Figure 15: Synthetic data inversion test using synthetic Data Set E calculated using the full, transient thermal model internal to GLIDE, but sparse data (Figure 5). The prior erosion rate is 0.35 mm/yr. Other inversion parameters in Table 1. See Figure 6 caption for other formatting details.**





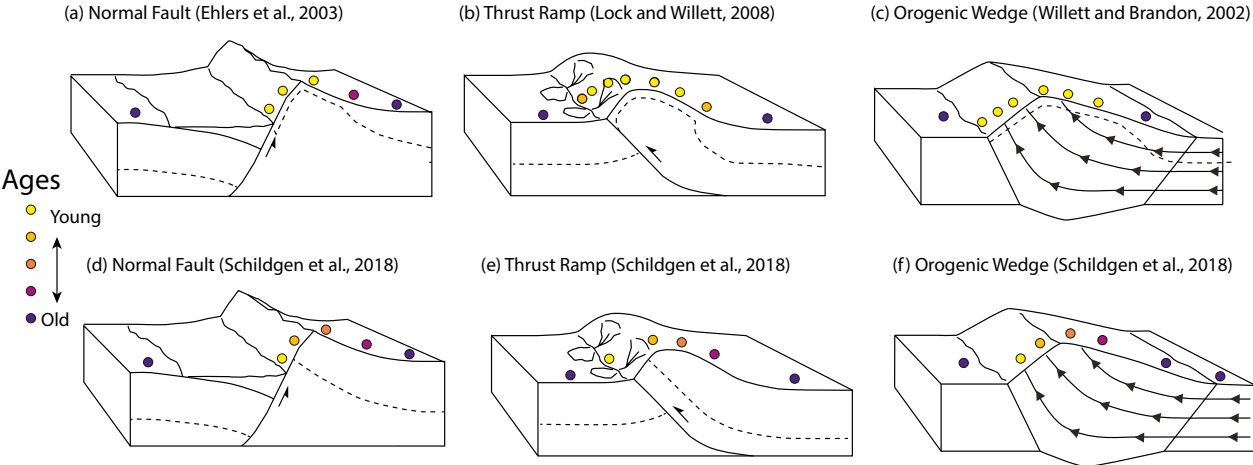


**Figure 16: Tectonic settings in which thermo-kinematic modelling has established extensive regions of nearly constant exhumation rate. Dashed line shows a line of constant age. (a) Normal fault footwall block (Ehlers et al., 2003). (b) Thrust ramp (Lock and Willett, 2008; McQuarrie and Ehlers, 2015, McQuarrie and Ehlers, 2017). (c) Orogenic wedge (Batt and Braun, 2000; Willett and Brandon, 2003; Fuller et al., 2006). (d), (e) and (f) show same settings as depicted by Schildgen et al., 2018, (their Figure 1). In (d)**
**and (e) the isochron is shown at approximately 90 degrees from the original publications, i.e., the maximum possible misrepresentation. In (f), the isochron was omitted, but surface age gradient is similarly misrepresenting the examples in the cited literature.**



Earth **Surface**
**Dynamics**
Discussions

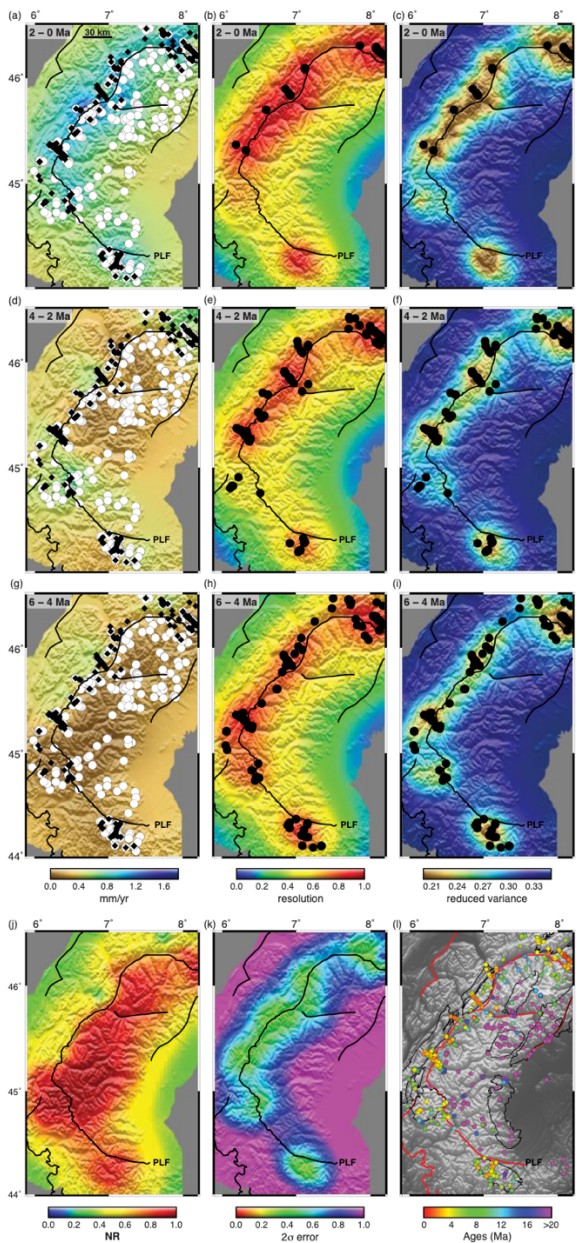

**Figure 17: Erosion rate history in the western Alps based on analysis from Herman et al. (2013) and Fox et al. (2015, 2016). (a-c)**
**Inferred erosion rate between 2 and 0 Ma, with temporal resolution and posterior variance. (d-f) Inferred erosion rate between 4**
**and 2 Ma, with temporal resolution and posterior variance. (g-i) Inferred erosion rate between 6 and 4 Ma, with temporal resolution**
**and posterior variance. (j) Normalized erosion rate difference (Equation 14), (k) 2σ error on the normalized erosion rate difference**
**(Equation 16). Inversion parameters are similar to Herman et al. (2013). A prior erosion rate of 0.35 mm/yr and a correlation length**
**scale of 30 km have been used. In (a), (d) and (g) the black diamonds indicate ages less than 6 Ma; white dots indicate ages older**
**than 6 Ma. In Resolution and variance plots the ages that fall within that time interval are shown as black circles. PLF is the Penninic**
**Line fault.**







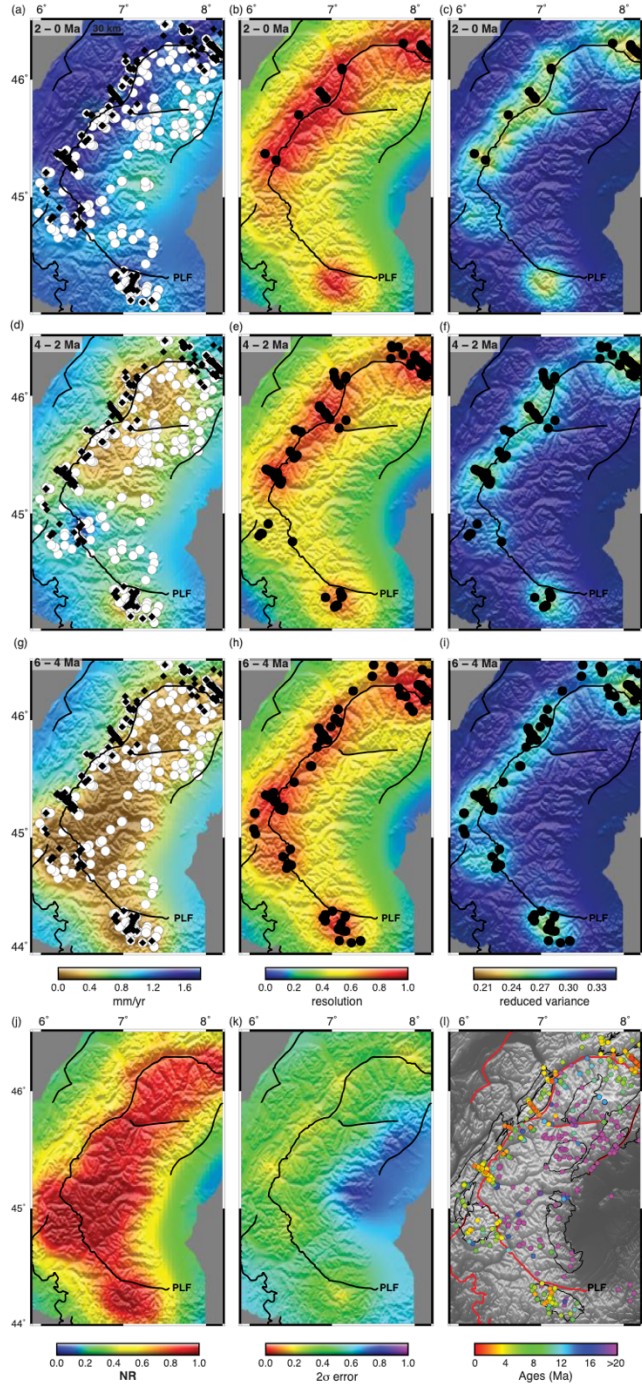

**Figure 18: Erosion rate history in the western Alps based on analysis from Herman et al. (2013). Model is identical to that of Figure 17, except that the prior erosion rate is taken to be uniform at a value of 1.35 mm/yr.**






**Figure 19: Erosion rate history in the western Alps excluding all data SE of the Penninic Line fault (PLF) (d-f), compared to history including all data (a-c) (identical to Figure 17). All inversion parameters are identical. The erosion rates and observed increase of erosion rates in the external Alps are virtually indistinguishable between the two models, confirming that the solution in the NW external Alps and the observed acceleration is determined by the local data.**

Earth **Surface**
**Dynamics**
Discussions




**Figure 20: Erosion rate history for the Western Himalayan syntaxis in the region of Nanga Parbat. Estimate based on analysis from Herman et al. (2013). Inferred erosion rates (a, d, g) for three time intervals, from 6 Ma to present, shown along with the temporal resolution (b, e, h) and reduced variance (c, f, i). (j) Normalized erosion rate difference (NR) and its associated error (k) are also shown. (l) Histogram shows age distribution broken out by thermochronometric system. Inversion parameters are similar to Herman et al. (2013). Erosion rate plots include ages less than 6 Ma as black diamonds and ages greater than 6 Ma as white dots. Resolution and variance plots include the ages that fall within the respective time interval.**








**Figure 21: Erosion rate history for the Olympic mountains of NW Washington State, USA. Estimate based on analysis from Herman et al. (2013). Inferred erosion rates (a, d, g) for three time intervals, from 6 Ma to present, shown along with the temporal resolution (b, e, h) and reduced variance (c, f, i). (j) Normalized erosion rate difference (NR) and its associated error (k) are also shown. (l) Histogram shows age distribution broken out by thermochronometric system. Inversion parameters are similar to Herman et al. (2013). Erosion rate plots include ages less than 6 Ma as black diamonds and ages greater than 6 Ma as white dots. Resolution and variance plots include the ages that fall within the respective time interval.**






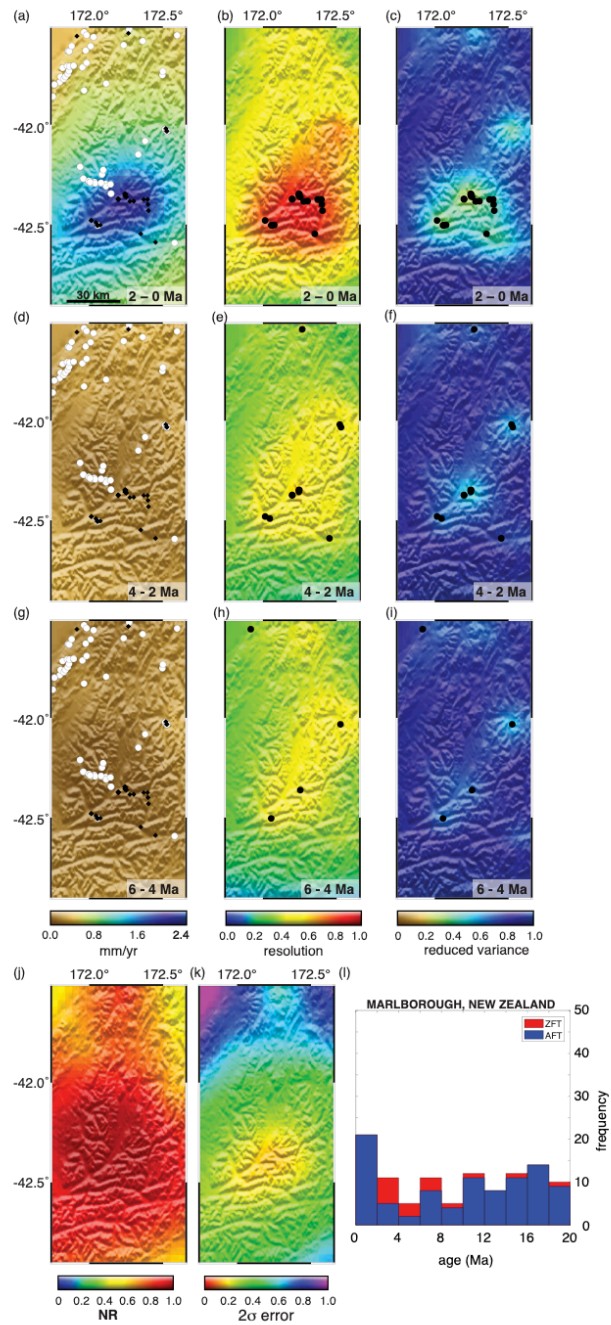

**Figure 22: Erosion rate history for the Marlborough region, New Zealand. Estimate based on analysis from Herman et al. (2013). Inferred erosion rates (a, d, g) for three time intervals, from 6 Ma to present, shown along with the temporal resolution (b, e, h) and reduced variance (c, f, i). (j) Normalized erosion rate difference (NR) and its associated error (k) are also shown. (l) Histogram shows age distribution broken out by thermochronometric system. Inversion parameters are similar to Herman et al. (2013). Erosion rate plots include ages less than 6 Ma as black diamonds and ages greater than 6 ma as white dots. Resolution and variance plots include the ages that fall within the respective time interval.**





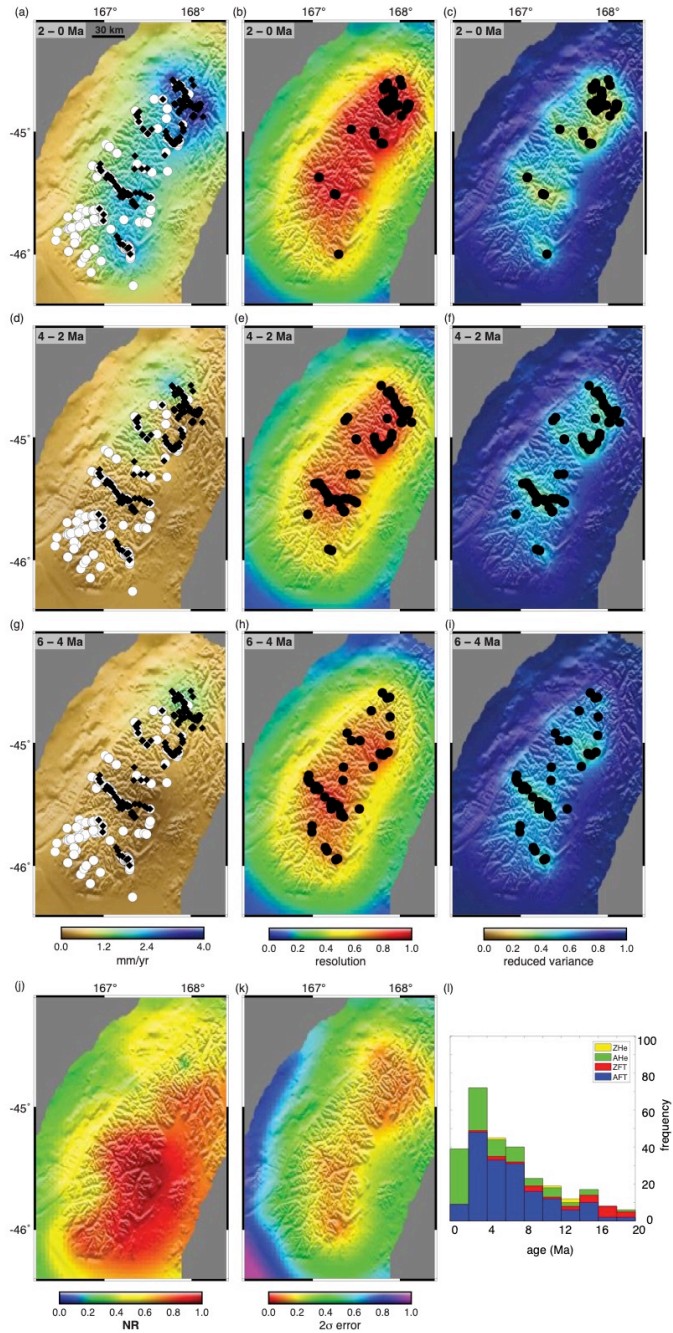

**Figure 23: Erosion rate history for the Fjordland region of New Zealand. Estimate based on analysis from Herman et al. (2013). Inferred erosion rates (a, d, g) for three time intervals, from 6 Ma to present, shown along with the temporal resolution (b, e, h) and reduced variance (c, f, i). (j) Normalized erosion rate difference (NR) and its associated error (k) are also shown. (l) Histogram shows age distribution broken out by thermochronometric system. Inversion parameters are similar to Herman et al. (2013). Erosion rate plots include ages less than 6 Ma as black diamonds and ages greater than 6 Ma as white dots. Resolution and variance plots include the ages that fall within the respective time interval.**






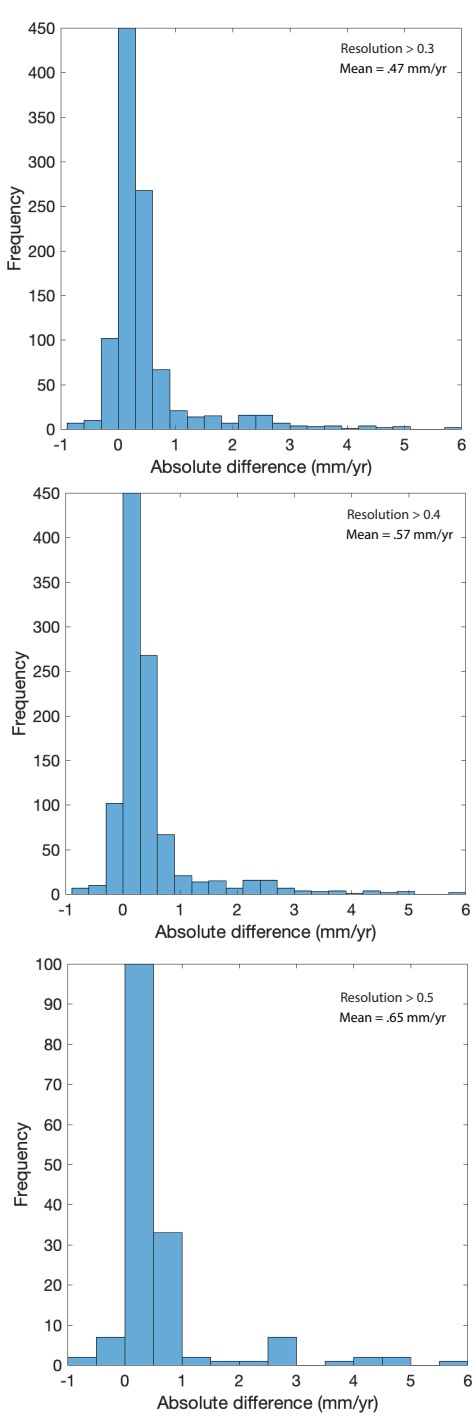

**Figure 24: Recalculation of the global change in erosion rate between 6-4 Ma and 2-0 Ma from the inversion results of Herman et al. (2013). Change is expressed as the difference between the erosion rate inferred for each of these time windows. Results are shown using three values for the resolution cutoff, from .3 to .5.**




Earth **Surface**
**Dynamics**
Discussions



**Figure 25: Distribution of points where thermochonometric ages permit resolution of a change in erosion rate from 6 ma to the present based on the inverse model analysis of Herman et al. (2013). Changes are expressed as a difference with distribution of values shown in Figure 24. Each frame contains points above a specified resolution cutoff.**
