# Peer review of "Bias and error in modelling thermochronometric data: resolving a potential increase in Plio-Pleistocene erosion rate"

_Earth Surface Dynamics, 2020_

## Short Comment (SC1) · 17 Sep 2020

Peter van der Beek1, Taylor Schildgen1,2, Rasmus Thiede3, Hugh Sinclair4

1Institute for Geoscience, Potsdam University, Potsdam, 14476, Germany
 2GFZ German Research Centre for Geosciences, Telegrafenberg, Potsdam, 14473, Germany
 3Institute for Geoscience, Christian Albrecht University Kiel, Kiel, 24118, Germany
 4School of Geosciences, The University of Edinburgh, Edinburgh, EH8 9XP, UK

10

Correspondence to: Peter van der Beek (vanderbeek@uni-potsdam.de)

The manuscript submitted by Willett et al. shows a welcome appreciation by the authors of the potential biases, errors, and resolution in inversions of thermochronometric data using GLIDE. However, it also contains several misrepresentations of

- 15 both the Herman et al. (2013) study and Schildgen et al.'s (2018) criticism of that study, and it overlooks some of the most important model and operator errors that affected the Herman et al. (2013) results. Based on additional analyses presented here, and our own critique of the material presented in Willett et al., we find that the spatial correlation bias remains a dominant bias in the results presented by Herman et al. (2013). Contrary to the suggestions by Willett et al., several of their synthetic tests demonstrate a clear spatial correlation bias, while others were carefully designed to avoid it. But, even in the
- 20 tests designed to avoid spurious accelerations in erosion rates due to the spatial correlation bias, we show that these accelerations still occur, but just a few million years earlier, in time windows not shown by Willett et al. We also show that with only minor modifications to the input data of those tests, strong accelerations in erosion rates caused by spatial correlation biases emerge within the last few million years. Issues raised by Willett et al. concerning geotherm differences among models used to create and invert synthetic ages in Schildgen et al. (2018) are insignificant, and do not produce the
- 25 spurious accelerations that Willett et al. claim they produce. Finally, the conclusion by Willett et al. that the results of Herman et al. (2013) remain valid is without support. We will discuss in detail below how we arrive at these conclusions. Before we do so, however, we feel it is useful to provide a short synthesis of these two preceding studies, to provide context for the current debate.

Herman et al. (2013) compiled global thermochronology data for four thermochronometers and inverted these data to generate erosion histories using the GLIDE code (Fox et al., 2014); the approach used globally uniform parameters, such as a correlation length scale of 30 km and an input background ("prior") erosion rate of 0.35 mm/yr. Herman et al. (2013) considered all predicted erosion rates, averaged over 2-Myr time-bins since 8 Ma, with a resolution (i.e., a measure of how well the erosion rate is resolved by the data at any given point in space and time, as defined by Fox et al., 2014) of > 0.25 to be "well resolved". Herman et al. (2013) found widespread accelerations in these "well resolved" erosion rates between the

35 time bins 6-4 Ma and 2-0 Ma, and they interpreted these accelerations as being linked to a global signal of cooling climate during that time. The analysis by Herman et al. (2013) depended heavily on: (1) the ratio of 2-0 Ma / 6-4 Ma erosion rates, and (2) inclusion of all predicted rates with a resolution > 0.25. Fig. 1C of Herman et al. (2013) shows these erosion-rate ratios for all points that meet their resolution threshold in both time windows and is the basis for arguing the "worldwide" nature of accelerated erosion rates; their Fig. 2 shows a histogram and cumulative density distribution of erosion-rate ratios,

- 40 as well as box plots of ratios split into latitude bins. Fig. 2 is the basis for the following assertions: "*The distribution of erosion rates shows that more than 80% of the regions with high-resolution values exhibit an increase, with an erosion rate ratio between >1 and 4 (Fig. 2). This increase is observed at all latitudes, but is more pronounced at latitudes outside the inter-tropical zone (inset of Fig. 2)*" (Herman et al., 2013; p. 424). These two figures and associated text thus constitute the core of their argument.
- 45 Disparities between the findings of Herman et al. (2013) and earlier interpretations of exhumation rates and patterns in many of the regions where they saw increases led Schildgen et al. (2018) to explore the robustness of the Herman et al. (2013) results. Several of these earlier studies employed thermo-kinematic modelling to quantitatively interpret the data, including some of our own studies (e.g., Bermúdez et al., 2011; Beucher et al., 2012; Glotzbach et al., 2011a; 2011b; Robert et al., 2009; Schildgen et al., 2009; Thiede and Ehlers, 2013) and some by co-authors of the Herman et al. (2013) paper (e.g.,
- 50 Ehlers et al., 2003; Fuller et al., 2006; Herman et al., 2007; 2010; 2009; Willett et al., 2003). Schildgen et al.'s (2018) approach was to examine the spatial patterns of the inferred erosion rates, compare these to the original data and mapped major structures, and review the literature to assess the original interpretations of these data. The approach was thus essentially *abductive*, which has been shown to be an efficient and even desirable logical approach in geomorphology (e.g., Baker, 1996). Schildgen et al. (2018) also developed synthetic tests to better understand what was likely driving the results
- of Herman et al. (2013). Schildgen et al. (2018) focused only on the predictions that Herman et al. (2013) deemed well resolved; only these points were shown on the maps of Schildgen et al. (2018).

A spatial correlation bias, as defined by Schildgen et al. (2018), creates spurious accelerations in erosion rates when data from areas with spatially variable exhumation histories are inappropriately combined. Although Schildgen et al. (2018) concluded that the spatial correlation bias was a common problem with the Herman et al. (2013) results, it was not the only

60 problem discussed. Other sources of error in the Herman et al. (2013) results arose from *model errors*, notably (1) assuming vertical rock exhumation in regions where lateral advection of rocks is important, and (2) ignoring the impacts of changes in topography on thermochronometer age patterns; as well as from *operator errors*, such as (3) the inclusion of samples reheated by volcanic flows or hydrothermal fluids, and (4) the inclusion of partially reset or unreset samples from sedimentary rocks in the inversions. A detailed analysis of the potential sources of error within the GLIDE inversion 65 procedure was outside the scope of the Schildgen et al. (2018) analysis, which focused rather on the implications and

robustness of the Herman et al. (2013) results. In other words, Schildgen et al. (2018) did not set out to test whether the

model could make robust predictions of erosion history, but whether it had done so in the Herman et al. (2013) analysis.

In the following, we will first focus on three main points made by Willett et al.:

70

- Model error due to variable geotherm calculation: We show that, although the geotherm calculation method in GLIDE is based on a poor choice of boundary conditions, Willett et al.'s dismissal of Schildgen et al.'s (2018) synthetic tests based on this difference is a red herring;
  - 2. **Bias to the prior versus the spatial correlation bias**: We show that bias to the prior erosion rate is an additional source of bias towards acceleration in the Herman et al. (2013) results, but new synthetic tests presented here imply it is less of a problem than the spatial correlation bias;
- Biased post-processing operator and resolution: We argue that Willett et al.'s criticism of the use of post-processing operators and lack of regard for resolution is misdirected at Schildgen et al. (2018), should instead be directed at Herman et al. (2013), and has no impact on the Schildgen et al. (2018) analysis.

We then continue with (4) a few comments on the **spatial variability in thermochronometric ages** and (5) on the **additional field examples** presented by Willett et al. We finish (6) with comments on the **definition of "spurious"** as used by Schildgen et al. (2018). In an appendix, we provide some line-specific comments on the Discussion section of Willett et al., which is replete with mistakes and mischaracterizations of the work presented by themselves, the work of Schildgen et al. (2018) and that of Herman et al. (2013).

**1 Model error due to variable geotherm calculation**

80

- Willett et al. claim that due to the differing boundary conditions between the model Schildgen et al. (2018) used to predict
  synthetic ages, Pecube (with a basal temperature boundary condition), and the model that performs the inversion, GLIDE (with a basal heat-flux boundary condition), *"the inversion will infer an increase in erosion rate with time in order to fit these ages"* (line 553). They use this argument to dismiss the synthetic tests presented in Schildgen et al. (2018) without further consideration or demonstration of the effect they argue exists. Willett et al. omit mentioning that the GLIDE input requires a basal temperature, not a flux. As the code is undocumented, it was assumed in Schildgen et al. (2018) that this was
  the temperature *at steady state*, but it appears to be the temperature for a stable geotherm (without advection).
- This choice of a stable geotherm with a flux basal boundary in GLIDE, together with vertical exhumation paths, is unsatisfactory, as it implies that the thermal effects of exhumation are felt throughout the thermal lithosphere, leading to both an excessively strong disturbance of the geotherm and excessively long response times, equivalent to the thermal response time of lithosphere. In natural collisional orogens, which are the sites of most rapid erosional exhumation, depths of
- 95 exhumation are generally limited to the upper and middle crust, above major mid-lower crustal detachments (e.g., Willett et al., 1993; Jamieson and Beaumont, 2013 and references therein). Material below this detachment is subducted and/or underthrusted, i.e., is advected *downward* instead of *upward*, with corresponding downward displacement of isotherms (e.g., Herman et al., 2010; Jamieson and Beaumont, 2013).
- Regardless whether the boundary condition incorporated into GLIDE is appropriate, if the differing boundary conditions between GLIDE and Pecube were significant in the synthetic tests presented by Schildgen et al. (2018), as argued by Willett et al., they should lead to consistent underestimations of exhumation rates in these inversions and consistent accelerations toward the present, as the effect is stronger for higher-temperature than for lower-temperature thermochronometers (Willett et al., lines 552-555). We tested this supposition by using GLIDE to invert synthetic data produced with a forward-run of Pecube, using Schildgen et al.'s (2018) synthetic "western Alps" case also utilised by Willett et al., but with data only from
- 105 the NW side of the fault, where exhumation rates are steady through time and space at 1 mm/yr (Fig. 1). In this test, we prevent any potential accelerations resulting from the spatial correlation bias by imposing spatially constant exhumation rates. If geotherm differences are important, GLIDE inversions of the Pecube-generated ages should produce accelerations. However, we predominantly see decreases in exhumation rates through time, or no change (Fig. 1). The underestimation of erosion rates results both from reverting to the prior erosion rate when this is set to < 1 mm/yr, and from the differing
- 110 boundary conditions between the forward and inverse models. The effect of the geotherm alone is best assessed in the case where the prior erosion rate is equal to the input erosion rate (1 mm/yr; Fig. 1 g-i): in this case, median predicted erosion rates are 0.84 ( $\pm$  0.09 at 1 $\sigma$ ) mm/yr for the 6-4 Ma age bin and 0.72 ( $\pm$  0.12) mm/yr for the 2-0 Ma age bin, i.e., they underestimate the "true" erosion rate by ~20%, which is within the range of the typical uncertainty associated with quantitative inferences of exhumation rates from thermochronology data. We thus conclude that the influence of differing
- 115 boundary conditions on these synthetic tests is minor and, importantly, cannot be invoked to explain the accelerations in the synthetic tests including spatially variable exhumation rates presented by Schildgen et al. (2018).

The reason for the minor influence of the differing boundary conditions on the synthetic tests likely relates to several factors. First, we note that Willett et al. also changed other thermal input parameters with respect to Schildgen et al.'s (2018) synthetic models, including the basal temperature and the crustal heat production. We are unable to reproduce Willett et al.'s

- 120 Figure 4, for instance, because the values of thermal parameters used in that calculation were not reported. Second, in contrast to Willett et al., we added realistic errors to our synthetic ages (see below), which are thus more scattered than what is shown in Willett et al.'s Figure 4. Third, the accelerations that we should see based on differing boundary conditions may be counteracted by the model error associated with the calculation of the geotherm in GLIDE described by Willett et al. (lines 739-741), which should lead to decelerations. Finally, while the effect is largest for deepest (highest-temperature)
- 125 thermochronometers, these systems close earlier in time, while the transient geotherm is still cooler (see Fig. 3 of Willett et al.). The exaggerated transience of the geotherm in GLIDE thus has the effect of mitigating the differences between predicted ages in Pecube and GLIDE.

**2 Bias to the prior versus spatial correlation bias**

**2.1 Bias to the prior versus spatial correlation bias in synthetic tests**

- 130 Willett et al. argue that the problematic results presented by Herman et al. (2013) are affected by a Bayesian bias to the prior erosion rate rather than a "spatial correlation bias". This argument is partly semantic; the inversion interpolates both temporally and spatially between incomplete data, which when done incautiously, introduces both types of bias. The spatial correlation bias causes spurious accelerations in inferred erosion due to the combination of areas that experienced rapid exhumation (and hence have young ages) with slowly exhumed areas that yield older ages. The Bayesian bias to the prior, in
- 135 contrast, returns the (input) prior erosion-rate estimate when the data do not constrain the erosion rate for a given time window. We illustrate the effects of these biases by a set of additional synthetic tests. We have run these tests in the same way as in Schildgen et al. (2018); as demonstrated in Section 1, the different calculation of the geotherm between the forward and the inverse model does not significantly affect these synthetic tests.

We illustrate the impact of the Bayesian prior bias alone with inversions that include data only from the NW side of the fault
(Fig. 1 a-i) or only from the SE side (Fig. 1 j-l). As discussed above, when including only data from the rapidly but constantly exhuming NW side, the model returns constant rates or minor decelerations since 6 Ma (Fig. 1 a-i). The absolute rates depend on the employed prior, especially toward the edges of the model, where the resolution is lowest. A similar result (no acceleration) is obtained when only inverting data from the more slowly exhuming SE side of the fault, although in that scenario, the low resolution (< 0.25 everywhere) implies that both the 6-4 Ma and 2-0 Ma time bins largely revert to the prior erosion rate (Fig. 1 j-l).</li>

prior crosion face (115. 1 j f).

We illustrate the combined effects of the Bayesian prior bias and the spatial correlation bias for the synthetic western Alps case by including data from both sides of the fault and varying the uniform prior erosion rate (Fig. 2). These tests show that spatially variable exhumation rates add substantial bias to the inversion results (compare Fig. 1 and. Fig. 2). The bias in Fig. 2 contains elements of the Bayesian prior bias; the two biases are impossible to disentangle in areas of spatially variable

150 exhumation. The compounded effects of both, nevertheless, are clear: accelerations occur when data from both sides of the fault are included in the inversion, regardless of the chosen prior erosion rate, suggesting a predominance of the spatial correlation bias. Furthermore, we see that by running an inversion with data from both sides of the fault, the resolution is higher on both sides compared to the resolution found when using data from only the NW or the SE (compare resolution

contours in Figs. 1 and 2). Thus, data are being combined from both sides to "better" constrain the exhumation history of each side, and that more highly resolved result produces the spurious increase we call the spatial correlation bias.

**2.2 Spatial correlation bias in the synthetic tests presented by Willett et al.**

At this point, we can ask why our synthetic results are so different from those of Willett et al. The answer to this question is twofold: removal of the spurious erosion-rate increase in Willett et al.'s inversions is achieved using spatially variable prior erosion rates and thus independent *a-priori* knowledge of this spatial distribution (their Fig. 8), the careful design of input data that have an idealized spatial and temporal distribution (their Figs. 6, 7, 9, 14 and 15), or both (their Fig. 13). The models of Willett et al. with a more realistic data distribution (i.e. Set C; their Figs. 10-12) show exactly the accelerations we expect (partly counteracted in Fig. 12 by the reversion to a very high prior erosion rate), and illustrate the dominance of the spatial correlation bias over the reversion to the prior bias similar to our Fig. 2. Although Willett et al. claim that their dataset A (and D, which has the same data distribution) "*roughly corresponds in pattern to the Alpine data set*" (line 581), it is in fact carefully designed to produce the desired result. In particular:

- Datasets A/D contain five thermochronological systems (and Sets B/E contain four), whereas the real Alps dataset only includes three systems and is dominated by apatite and zircon fission-track ages (290 out of 309 data, the remaining 19 being apatite (U-Th)/He ages). Most noticeably, 30% of the ages in Willett et al.'s Sets A/D (and 22% in Sets B/E) are mica 40Ar/39Ar ages. Neither Herman et al. (2013) nor Fox et al. (2016) used any mica 40Ar/39Ar cooling ages in their inversions, for the simple reason that these do not exist. The few 40Ar/39Ar dates available for this part of the Alps are
- crystallisation ages for minerals in fault zones (e.g., Egli et al., 2017; Rolland et al., 2008; Rossi and Rolland, 2014), and thus not representative of regional cooling related to exhumation.
  - The datasets by Willett et al. are characterised by a very high number of co-located data (120 out of 176 data locations in Sets A/D combine two thermochronometers, in general associated with mica 40Ar/39Ar; 15 have three or more
- thermochronometers). In the real dataset, in contrast, only 48 out of 251 data locations have two thermochronometers(all are apatite and zircon fission-track), and five locations have three (with additional apatite (U-Th)/He).
  - All datasets by Willett et al. consist of "perfect" ages, as predicted by the forward model. In contrast, we added a random scatter of up to 10% to our synthetic ages in order to better reflect imperfect natural data.
  - Sample locations in Sets A/D are more heavily weighted toward high elevations than the real data: 38% of the data from
- 180

185

170

the NW zone are from locations >2000 m, whereas only 28% of the real data are from such high elevations. The data locations in these datasets also have a much wider spatial spread than the real data.

To demonstrate the dependence of Willett et al.'s synthetic tests on the idealized data input, we reran the inversion of their Fig. 14 (Set D) both with and without the hypothetical  ${}^{40}$ Ar/ ${}^{39}$ Ar data (i.e., addressing only the first bullet point in the above list). Figure 3 shows our results: first, we reran the exact same inversion of dataset D using our version of GLIDE to make sure we obtained the same results; we do (compare Fig. 3 a-c with Willett et al.'s Fig. 14 d, a, g). In particular, the inversion strongly overestimates the erosion rate SE of the fault in the 6-4 Ma time bin, leading to a significant deceleration after this

- time, centred on the SE half of the model. However, Willett et al.'s explanation for this deceleration as being due to the sluggish transient response of the geotherm to exhumation rates that are higher than the prior rate (lines 735-745) can only apply to the rapidly exhuming NW side of the fault; that explanation cannot be applied to the slowly exhuming SE side of the model, where the decelerations are largest. Willett et al. did not show what happens in the earlier time bins of this test;
- Fig. 3 d-f shows that between the 10-8 Ma and 6-4 Ma time bins, predicted erosion rates show a significant spurious

increase, both to the NW and to the SE of the fault. In fact, the addition of the high-temperature mica  ${}^{40}$ Ar/ ${}^{39}$ Ar data (with older ages) has simply pushed the acceleration due to the spatial correlation bias back to these earlier time bins.

To demonstrate this effect, we removed the mica 40Ar/39Ar data from Set D and reran the inversion; the result shows major

- 195 accelerations from 6-4 Ma to 2-0 Ma on both the NW and SE sides of the fault (Fig. 3 g-i), similar to our previous test shown in Figure 2. Interestingly, removal of the 40Ar/39Ar data from Set D results in no substantial change in the resolution for the 6-4 Ma or 2-0 Ma time bins (compare resolution contours in Fig. 3 g-h and Fig. 3 a-b). The acceleration is again independent of the chosen prior erosion rate, as shown by our third test, in which we increased the prior erosion rate to 1 mm/yr (Fig. 3 j-1). Thus, the acceleration is predominantly due to a spatial correlation bias.
- 200 To show that this behaviour is independent of the chosen geotherm model, we repeated this test using Willett et al.'s dataset A and the version of GLIDE with a stable geotherm. The results, shown in Figure 4, are similar to those discussed above for dataset D and a transient geotherm. In this case, erosion rates are constant between the 6-4 Ma and the 2-0 Ma time bins (Fig. 4 a-c, compare to Willett et al.'s Fig. 6) but there is a similar spurious increase earlier in the history (Fig. 4 d-f). Like the case with Set D, when removing the high-temperature mica 40Ar/39Ar data from Set A, this spurious acceleration occurs
- 205 in the recent time bins (between 6-4 Ma and 2-0 Ma) despite no substantial loss of resolution in these time bins (Fig. 4 g-l). Thus, for both the stable (Fig. 4) and transient (Fig. 3) geotherm cases, even Willett et al.'s most highly temporally resolved dataset (A/D), with the densest data in terms of both spatial and temporal coverage, leads to spurious accelerations due to the spatial correlation bias, either somewhat earlier in the exhumation history than what Willett et al. showed, or in the most recent time bins after removal of the mica 40Ar/39Ar ages.
- 210 We agree with Willett et al. that temporal coverage of the data is important; the above tests show this dependence. However, the tests also show how, in the absence of temporal coverage in a particular time bin, information can be brought in from surrounding regions (creating a spatial correlation bias) rather than locally reverting to the prior. We acknowledge that the synthetic data in our tests have reduced temporal coverage compared to the real data. However, creating synthetic datasets that do not attempt to recreate the full age range of real datasets, but rather faithfully recreate real data distributions and the
- 215 available thermochronometers, is fully justified. As Willett et al. note, "the resolving capability of data is determined by the complexity of the erosion rate field being sampled, so resolution should be evaluated in the context of tectonic variability" (lines 771-773). For a synthetic site with a very simplified exhumation history (i.e., a steady exhumation rate through time), one would not expect the same distribution of ages that are obtained from a real site that has experienced a far more complicated exhumation history. By maintaining the full range of ages exhibited at a real site with a more complicated
- 220 history, Willett et al. are providing their inversions with far greater temporal coverage than could ever be reasonably expected to constrain a simple exhumation history. Yet, even in this case, spatial correlation biases occur, only at different times in the history.

We finally note that the western Alps dataset is one of the densest in the world, which is why we decided to explore this case

225

**in our synthetic test; if the inversion fails here (and our tests show that it does), it is likely to fail elsewhere. We also note that a relatively high resolution is no guarantee of an unbiased result, as our Figures 2 - 4 show spurious accelerations in erosion rates that have a resolution of > 0.5.**

**2.3 Bias in reversions to the prior in the Herman et al. (2013) results**

Willett et al. argue that, when "reversion to the prior" occurs, such a result will not have a "generalizable tendency toward acceleration" (lines 1128-1129). Although we agree with this general statement, in the application of GLIDE by Herman et

- al. (2013), reversions to the prior erosion rate will likely create spurious accelerations. This is because, as originally argued by Willenbring and Jerolmack (2016), all of the "resolved" regions include sites of rapid exhumation and therefore young thermochronological ages, which will tend to increase resolution in the most recent time bin. To demonstrate this point, we illustrate the distribution of resolution values from the results of Herman et al. (2013) in two different time bins, 2-0 Ma and 6-4 Ma (Fig. 5 a, b) and the distribution of erosion rates in those time bins (Fig. 5 c, d). These plots illustrate, first, the
- 235 tendency of the data to be better resolved in the 2-0 Ma time bin (median resolution of 0.48) compared to the 6-4 Ma time bin (median of 0.33). Moreover, 75% of all data points in the 6-4 Ma time bin have a resolution lower than 0.4, whereas only 32% of data points in the 2-0 Ma time bin have a resolution lower than 0.4. For co-located points, 90% show a higher resolution in the 2-0 Ma time bin compared to the 6-4 Ma time bin (Fig. 5 b). Hence, reversion to the prior, which Willett et al. suggest affects most of the results with resolution below 0.4, will much more likely affect points in the 6-4 Ma time bin
- 240 than in the 2-0 Ma time bin. Moreover, most of the erosion rates in the 2-0 Ma time bin are higher than the prior of 0.35 mm/yr: the median erosion rate in the 2-0 Ma time bin is 0.48 mm/yr, and 80% of the erosion rates in the 2-0 Ma time bin have an erosion rate higher than 0.35 mm/yr (Fig. 5 c, d). Given the relatively high erosion rates that characterize the 2-0 Ma time bin, reversion to the prior in the older time bin will commonly produce spurious accelerations in exhumation.
- The test by Herman et al. (2013) to explore the effect of the prior on the global compilation is an interesting counterexample to the argument by Willett et al. that most of the lower resolution results are affected by this reversion to the prior. Herman et al. (2013) reported that when choosing a prior erosion rate of 0.7 or 1.0 mm/yr (the latter of which is higher than many of the inferred erosion rates in the 2-0 Ma time bin), they still see a predominance of accelerations in their inversions. Whereas Herman et al. (2013) used this result to argue for the robustness of their conclusions, we argue instead that it illustrates that Herman et al.'s (2013) results are not dominated by a Bayesian prior bias, but rather they are predominantly affected by a spatial correlation bias, which creates spurious accelerations, but is less affected by the choice of the prior.

Our synthetic tests (Figs. 1 and 2) show that bias to the prior exists, but in most cases, it is overpowered by the spatial correlation bias. The fact that the results from Herman et al. (2013) are insensitive to the prior erosion rate is not a sign of robustness of the results, but rather the pervasiveness of the spatial correlation bias.

**3 Biased post-processing operator and resolution**

- 255 Willett et al.'s insistence that it is the post-processing operator, i.e. plotting normalized erosion-rate differences, rather than the inversion that creates the bias, is difficult to understand. Increased erosion rates through time appear as such, whether they are visualized from direct comparison of time-bin maps or plotted as differences, ratios, or normalized differences. The magnitude of the metric changes in each case, but the sign (positive or negative) does not. In contrast to the analysis of Herman et al. (2013), the interpretations of Schildgen et al. (2018) focused on the sign of the change, not its magnitude, and
- 260 on whether that change is reasonable considering (1) the spatial and temporal (age) distribution of the data, and (2) previous, more detailed analyses that include independent geologic data and more appropriate modelling of the data (e.g., with changing topography and/or lateral components of rock advection, and exclusion of data unrelated to exhumation). The ratios of erosion rates used by Herman et al. (2013), namely the erosion rate from the more recent time bin divided by that of the earlier time bin, are problematic, as the values tend to blow up when the earlier erosion rate is small. Although the
- normalized difference used by Schildgen et al. (2018) is arguably also imperfect, it has the benefit of tracking fractional changes in exhumation rates (i.e., a change from 0.5 to 1.0 mm/yr or from 0.05 to 0.1 mm/yr both result in a value of 0.5), and being symmetric (i.e., changes in erosion rates from 1.0 to 0.5 mm/yr or from 0.5 to 1.0 mm/yr yield respective

normalized differences of -0.5 and 0.5). We note that Willett et al. (1) make a mistake in their description of the normalised difference NR (their Eq. 14 returns a negative number in case of an acceleration); and (2), more critically, in their analysis of

- 270 the resolution of NR, they modify the definition of NR to maximise its value and minimise its resolution by dividing by  $e_2$  instead of max( $e_1$ ,  $e_2$ ), as done by Schildgen et al. (2018) (line 467 of Willett et al.; note that in their definition  $e_1 > e_2$  in case of an acceleration). By altering the definition of the NR in their error analysis, Willett et al. are analysing a metric that mixes Herman et al.'s (2013) original ratio and Schildgen et al.'s (2018) normalised difference.
- The absolute differences now advocated by Willett et al. are problematic in their own right, as areas with high erosion rates 275 tend to dominate any global "signal". Moreover, Willett et al. report mean values from their histograms of erosion-rate differences, neglecting to take into account how the dominance of extreme values is exacerbated by the use of the mean instead of the median as a measure of central tendency in these positively skewed distributions (their Fig. 24). For example, the median difference for a resolution cutoff of 0.5 is 0.42 mm/yr, whereas the mean is 0.65 mm/yr (Fig. 24c in Willett et al.). We can further illustrate the problem with absolute differences by considering the impact of removing one region of
- 280 rapid erosion rates from the compilation. Again using a resolution cut-off of 0.5, after excluding results from Taiwan, where extreme reported increases in erosion rates are related to the inclusion of unreset thermochronometer data in the inversion (Schildgen et al., 2018), the mean drops from 0.65 to 0.37 mm/yr and the median drops from 0.42 to 0.39 mm/yr. However, these numbers still cannot be taken at face value, as the results still suffer from spatial correlation biases, reversion-to-prior biases, model error, and operator error.
- Oddly, Willett et al. claim that the analysis of Schildgen et al. (2018) was focused on areas of poorly resolved results and that Schildgen et al. (2018) "never address resolution". We reiterate from Schildgen et al. (2018) that the analysed results were those that passed the threshold defined by Herman et al. (2013) as "well resolved", i.e., with a resolution > 0.25, and were used by these authors to support a "worldwide increase in erosion rates". Willett et al. show a welcome new appreciation for the importance of better resolved results (lines 1000-1003), but the repeated suggestion that Schildgen et al.
- 290 (2018) misdirected attention to areas of poorly resolved results is unfounded. The spurious increases documented in Schildgen et al. (2018) and here (Figs. 2 4) comprise from best to least resolved areas, and all are above the "well resolved" threshold of 0.25 used in Herman et al. (2013). Nevertheless, we agree that focusing on more highly resolved results is better practice, although doing so does not eliminate spurious increases (Schildgen et al., 2018 and Section 2 above). The implications of the selected cut-off resolution value are substantial: as noted in Schildgen et al. (2018),
- 295 increasing the resolution threshold to 0.5, which characterizes the "well resolved" region in the Alps that Willett et al. prefer to focus on, would eliminate 90% of the "resolved" erosion ratios reported by Herman et al. (2013), and would comprise data from only seven distinct regions, as shown in Willett et al.'s Fig. 25. One of those regions (central Himalaya) comprises a single resolved point that shows a decrease in exhumation rates through time, not an increase; five regions (Wasatch Mountains, Western Alps, Northern Apennines, Taiwan, Fiordland) suffer from spatial correlation bias, model or operator
- 300 errors and sometimes a combination of these, as discussed in the Supplementary Information of Schildgen et al. (2018); in three regions (Coast Mountains of British Columbia, External Massifs of Western Alps, Fiordland) glacial valley incision has been previously inferred (Shuster et al., 2005; 2011; Valla et al., 2011; see below). Restriction of the results to these few locations precludes any attempts at generalization to a global scale.

**4 Spatial variability of thermochronometer ages**

- 305 Willett et al. expend considerable effort in critiquing cartoons by Schildgen et al. (2018) that have no vertical or age scale, and only a rudimentary horizontal scale (lines 793-831; 1320-1322 in Willett et al.). We do not see much merit in this discussion, but note that Willett et al. appear to confuse *isotherms* (surfaces of equal temperature, which are sketched into the cartoons of their Figure 16 a-c) and *isochrones* (surfaces of equal thermochronologic age, which were drawn in the cartoons of Schildgen et al. (2018), reproduced in Willett et al.'s Figure 16 d-f). The point that Willett et al. appear to be
- 310 trying to make is that thermochronologic ages will be spatially constant over length scales larger than the correlation lengthscale used in the GLIDE inversions in most tectonic settings. To assess this point, we illustrate some original data that inspired the cartoons (Fig. 6). For the Wasatch Mountains (Fig. 6a), the AHe and ZFT ages shown in Ehlers et al. (2003) increase steadily with distance from the range-bounding Wasatch Fault, whereas AFT ages show only a slight increase until 17 km from the fault, where they start increasing more rapidly with distance. Likewise, in the Southern Alps of New Zealand
- 315 (Fig. 6b), all thermochronometer ages increase rapidly with distance from the Alpine Fault, starting from distances of 10-20 km from the fault (e.g., Herman et al., 2009). Importantly, in both these cases, the ages show significant variation over length scales of ~30 km, which was the correlation length scale used in Herman et al. (2013), making these settings prime examples of where that inversion was affected by the spatial correlation bias.

**5 Comments on natural examples**

**320 5.1 European Alps**

Willett et al. focus much of their analysis and discussion on the European Alps, presumably because, as they claim, the Alps "*play an important role in the study of Schildgen et al. (2018)*" (line 863). We consider the European Alps to be no more important than any of the other 32 locations where Herman et al. (2013) reported resolved changes in late-Cenozoic erosion rates. Schildgen et al. (2018) opted to highlight the Alps as one of three examples in the main text due to the very high density of available data, which gives GLIDE the best chance of performing well. Although Willett et al. have gone to great lengths to demonstrate the reality of the erosion-rate increase in the western external Alps, they neglect to consider that increased erosion in the External Crystalline Massifs, to the north and west of the Penninic thrust front, is limited to localized valley incision, as has been demonstrated with both apatite 4He/3He studies (Valla et al., 2011) and detailed

thermo-kinematic modelling that includes a temporally evolving topography (Glotzbach et al., 2011b). The latter study also

330 demonstrated how, when steady-state topography was assumed, the pattern could be mistaken for a generalized increase in exhumation rate, as happens in the GLIDE inversions. This difference is not trivial, as presuming a regional increase in exhumation rather than localized valley incision has major implications for sediment flux to the oceans, carbon cycle impacts, and landscape evolution.

Incidentally, on lines 939-940, Willett et al. state "*Finally, we address the geologic evidence of Schildgen et al. (2018)'s* 335 *hypothesis that the external and internal Alps are separated by an active normal fault along the Penninic Line.*" Although Willett et al. delve into considerable detail in the following paragraph to argue against this hypothesis, Schildgen et al. (2018) never suggested this. Willett et al. appear to have misunderstood the aim of the synthetic test presented in Schildgen et al. (2018): it was used simply to test for the occurrence of a spatial correlation bias across a densely sampled and strong gradient in thermochronologic ages, not to attempt a realistic simulation of the European Alps.

**340 5.2 Nanga Parbat**

It is difficult to assess the inversions Willett et al. present for Nanga Parbat, because they do not discriminate between ages inside and outside the massif. All apatite and zircon fission-track ages within the massif are < 3.4 Ma (Treloar et al., 2000; Zeitler, 1985; Zeitler et al., 1982), and all such ages outside the massif are > 3 Ma (see the Supplementary Information of Schildgen et al., 2018). Willett et al. now add mica  ${}^{40}$ Ar/ ${}^{39}$ Ar data, but it is unclear if Herman et al. (2013) used those data in

- their inversions. Herman et al. (2013) did not report any such data, and the resolution values they report for Nanga Parbat are considerably lower than the resolution values shown by Willett et al. (e.g., maximum resolution in the 6-4 Ma time bin of 0.4, rather than ca. 0.6 shown in Willett et al.; and maximum values in the 4-2 Ma time bin of 0.5, rather than ca. 0.7 shown in Willet et al.), implying that the inversion presented in Herman et al. (2013) used less data. Mica  $^{40}$ Ar/39Ar data from the core of the massif are  $\leq$  4 Ma, with one exception along the Indus River (Treloar et al., 2000; Zeitler et al., 2001); older ages
- are only encountered within the massif-bounding shear zones. Mica  ${}^{40}$ Ar/ ${}^{39}$ Ar ages outside the massif are > 10 Ma without exception, and they are mostly > 20 Ma. Moreover, careful interpretation of mica  ${}^{40}$ Ar/ ${}^{39}$ Ar data from Nanga Parbat is required, as excess Ar is a commonly reported problem, and several reported ages are crystallisation ages rather than cooling ages (Schneider et al., 2001). If any of these complications in the data were considered by Willet et al., they are not reported, raising the possibility of operator error.
- 355 Despite there being no information from in-situ data within the massif prior to 4 Ma, and there being no ages < 4 Ma outside the massif, the GLIDE solution shows reasonably resolved moderate erosion rates within the massif (< 0.8 km/Myr; resolution ~0.5) prior to 4 Ma and rapid recent rates "bleeding" outside the massif since 2 Ma (Willet et al.'s Fig. 20). Both inside and outside the massif, the inversion predicts large increases in erosion rate with time, which were included in the "worldwide pattern" of Herman et al. (2013). This example provides one of the clearest instances of the spatial correlation
- 360 bias, as data are combined across major massif-bounding faults that are generally considered to be active (see Butler, 2019 for a recent review).

It can be easily shown that the inferred exhumation history inside the massif presented by Willett et al. is erroneous, as U-Pb ages as young as < 2 Ma on metamorphic monazite and granite dikes imply much greater exhumation within the last 2 Myr than the < 8 km predicted by the GLIDE inversion (Zeitler et al., 2001 and references therein; Crowley et al., 2009; Butler, 2019). Moreover, < 4 Ma granites currently outcropping within the massif solidified at  $\sim$ 700 °C and 350-500 MPa (Crowley

365 2019). Moreover, < 4 Ma granites currently outcropping within the massif solidified at ~700 °C and 350-500 MPa (Crowley et al., 2009), or 13.0-18.5 km depth (assuming a crustal density of 2750 kg m-3), which is significantly higher than the < 10 km exhumation predicted by the GLIDE inversion since 4 Ma.

**5.3 Olympic Mountains**

The Olympic Mountains are a prime example of an orogenic wedge with curved particle paths (Willett et al.'s Fig. 16c). Any 1D analysis of such a system will infer a recent acceleration in exhumation; this acceleration is real, because for a constant particle velocity, as the particle path becomes more vertical closer to the surface, exhumation rates increase. However, this increased exhumation rate is not associated with an increased erosion rate at the surface, which is an important distinction when considering the possibility of climatically triggered erosion-rate increases. Thus, when considering the implications for surface erosion rates through time, we consider this increase to be spurious in the analysis of Herman et al. (2013), because it

375 assumes vertical exhumation pathways. Only models that incorporate curved particle pathways will potentially be able to distinguish between changes in exhumation rates related to the exhumation pathway versus those related to changes in surface erosion rates. The western flank of the Olympic Mountains has been glaciated and deep glacial valleys have been carved into the landscape (Montgomery, 2002; Adams and Ehlers, 2017). Thus, it is possible that samples from valley

bottoms on the western flank show an influence of valley incision, but this effect can only be assessed with models that 380 incorporate both realistic kinematics and changes in landscape morphology.

**5.4 Marlborough region of New Zealand**

The interpretation of thermochronological data in the Marlborough region by Herman et al. (2013) is even more problematic than described in Schildgen et al. (2018). Although Willett et al. have pointed to the co-located thermochronometers close to the Alpine Fault as evidence of increasing exhumation rates for a few data locations, they dismiss the increases inferred by
the model elsewhere as "non-resolved", even though these were included in the analysis of Herman et al. (2013). The zircon fission-track ages for the points highlighted by Willett et al. in close proximity to the Alpine Fault are < 6 Ma, whereas just a few km away, having crossed no major intervening structure, such ages jump to > 70 Ma (see Fig. S16 in Schildgen et al., 2018). In addition, several of the co-located apatite fission-track ages are reported as 0 Ma (Tippett and Kamp, 1993), and Herman et al. (2013) did not explain how they addressed such ages in their inversion. The clear implication is that most of the zircon fission-track ages from the sedimentary rocks in this region are unreset or only partially reset, and the young ages found only in close proximity to major mapped structures imply strong tilting of the individual fault blocks and/or local reheating due to hydrothermal fluid flow along the Alpine Fault. Although increasing the resolution threshold in this region would eliminate many of the clearly spurious erosion-rate increases illustrated in Fig. S16 of Schildgen et al. (2018), it will not eliminate the operator error associated with the inclusion of reheated or unreset samples.

**395 5.5 Fiordland**

400

Schildgen et al. (2018) argued that some of the well-resolved erosion-rate increases in Fiordland reported in Herman et al. (2013) are probably real, and could be linked to glacial valley incision (Shuster et al., 2011). However, like in other glaciated terrains, mistaking local valley incision for a regional increase in exhumation rates is a recurring issue with models like GLIDE, which do not consider modifications in surface morphology, and subsequently vastly overestimate the regional impact of thermochronometer age patterns controlled by localized valley incision. Apart from the clear localized influence of glaciers on valley incision, regional spatio-temporal patterns of exhumation have been argued to be linked to the evolving subduction zone (Sutherland et al., 2009; Jiao et al., 2017), an argument that was detailed in the Supplementary Information of Schildgen et al. (2018). Nevertheless, the largest increases reported by Herman et al. (2013) are spurious increases to the

SE of the main range, which Willett et al. now consider insufficiently resolved.

**405 6 On the definition of "spurious"**

On lines 964-966, Willett et al. state "According to the re-analysis of Schildgen et al. (2018), of the 32 sites identified in the Herman et al. (2013) study as showing sufficient thermochronometric data to resolve an erosion rate history over the past 6 Ma, 23 of them were what they called "spurious", meaning that they arose as a result of inappropriate spatial averaging of age data." In contrast, Schildgen et al. (2018) used the term "spurious" simply in its generally accepted meaning of "false"

- 410 or "fake", describing in detail why any given acceleration was deemed "spurious" for each region. In addition to inappropriate combination of data, the reasons also included models errors and operator errors that are not considered by Willett et al., such as (1) inappropriate assumptions of purely vertical exhumation in regions where lateral rock advection plays an important role (e.g., Southern Alps of New Zealand, Olympics, Apennines, Taiwan); (2) inappropriate assumptions of no change in surface morphology where such changes were shown to be critical for understanding thermochronometer age
- 415 patterns (e.g., Aconquija, Fiordland, western European Alps, southern Peru, Bolivia, Coast Range); (3) inappropriate

inclusion of samples that were reported to have been reheated by volcanic flows (e.g., San Juan Mountains, southern Peru) or hydrothermal fluids (Eritrea); and (4) inappropriate inclusion of partially reset or unreset data from sedimentary rocks (e.g., Taiwan, New Zealand). Some of the spurious increases may have arisen due to a reversion to the prior in some cases, but in reality, the reason for the spurious acceleration does not matter so much as the fact that it is fake. By strictly limiting the

420 definition of "spurious", the authors have sidestepped addressing the true extent of problems in the Herman et al. (2013) inversion results. Uniform application of a model that takes no account of changes in surface morphology, rock-exhumation pathways, or tectonic features to a global dataset that includes many data points unrelated to exhumation is bound to fail in some places. For the results presented by Herman et al. (2013), we conclude that it has failed in the majority of cases.

**Conclusions**

- 425 We have shown that the issues raised by Willett et al. in their criticism of the work by Schildgen et al. (2018) are either insignificant or unfounded. By reproducing the inversions that Willett et al. reported, and plotting results from time windows that they did not include, we have shown that (1) the spatial correlation bias is a common problem in the inversions shown by Willett et al., even in those designed in an attempt to avoid it; and (2) the effects of differing boundary conditions between Pecube and GLIDE, and consequent differences in the assumed geotherm, are insignificant when comparing
- 430 predicted ages from the former and inversion results from the latter. Our use of the synthetic data produced by Willett et al. in new synthetic tests presented here show a spatial correlation bias similar to that shown previously in Schildgen et al. (2018), demonstrating that Willet et al.'s dismissal of those earlier synthetic tests is unfounded. Other issues raised by Willett et al. concerning post-processing operators and critiques of cartoons are irrelevant with regards to the Schildgen et al. (2018) analysis.
- We also reaffirm the conclusions from Schildgen et al. (2018) that a great majority of the results reported by Herman et al. (2013) are unreliable due to a combination of spatial correlation biases, model error, and operator error. Reversion to the prior erosion rate may have also led to spurious results in some sites, but the spatial correlation bias is likely the most common issue in areas that were not significantly affected by model errors in GLIDE (e.g., assumption of vertical exhumation pathways and assumption of no changes in topography through time) or operator errors (e.g., inclusion of data
- 440 unrelated to exhumation, such as from samples reheated by hydrothermal fluids or volcanic flows, and inclusion of unreset or partially reset ages from sedimentary rocks). The small number of remaining regions where results from Herman et al. (2013) may be reliable are inadequate for any conclusions regarding the impact of late-Cenozoic cooling on worldwide erosion rates. We are in full agreement with Willett et al. that a resolution cut-off value much higher than the 0.25 value used by Herman et al. (2013) will lead to better results, but we have also shown that even the cut-off resolution value of 0.5
- 445 suggested by Willett et al. is insufficient to avoid the biases that we demonstrate.

**Appendix: Inaccuracies in the Discussion by Willett et al.**

The Discussion section by Willett et al. distorts much of what was presented in Schildgen et al. (2018) and in Herman et al. (2013). Although we consider the following comments somewhat minor relative to the main issues we raise earlier, we 450 highlight below several inaccurate points for the sake of completeness.

Willett et al. claim to have identified all sources of bias and error in their model, and conclude that they are either unimportant or do no create a tendency toward acceleration (lines 1120-1129). However, Willett et al. the authors have neglected to discuss the implications of the most important model errors as applied to several field settings, namely the assumption of vertical rock-exhumation pathways and no change in topography in the inversions. They also appear unconcerned with operator error, which takes the form of the inclusion of inappropriate data (e.g., samples reheated by volcanic flows or hydrothermal fluids, and unreset data from sedimentary rocks) for several field sites in the Herman et al. (2013) analysis. Willett et al.'s analysis instead misdirects readers toward trivial issues like geotherm differences, metrics used to illustrate erosion-rate changes, and cartoons. Willett et al. also neglected to consider how a reversion to the prior may constitute an additional bias, particularly when considering the analysis of Herman et al. (2013), as we illustrate in section 2.2, and they failed to recognize the spatial correlation bias in their own synthetic tests.

**A.1 Do spatial correlation biases occur?**

Willett et al. note that "The idea that spatial differences in age, i.e., a combination of old and young ages from distinct regions, will always, or even frequently, combine to produce an apparent increase in erosion rate is false. Models in this paper were consistent in demonstrating this point" (lines 1147-1149). These statements are odd for several reasons. First, it

- 465 is certainly possible to mistakenly combine data from regions with distinct exhumation histories to produce a spurious acceleration and, as we have pointed out, many of the synthetic tests presented by Willett et al. show a spatial correlation bias. The tests that do not show the spatial correlation bias were specifically designed to avoid it, at least in the time bins Willett et al. chose to report, and/or they were run in modes that are unrelated to the application in Herman et al. (2013), such as by setting spatially variable prior erosion rates. But even in the tests designed to avoid the spatial correlation bias
- 470 through highly temporally resolved datasets, spatial correlation biases occur; they simply occur earlier in time (Figs. 3, 4). The synthetic tests presented by Schildgen et al. (2018) and here (Figs. 2 4) further illustrate the common occurrence of the spatial correlation bias when the inversion is applied to realistic datasets with a setup equivalent to that applied by Herman et al. (2013). A model that combines real data based on a predefined correlation length, without regard for tectonic structure, will suffer from this bias whenever (1) there are insufficient data from a single tectonic block to fully constrain the
- 475 exhumation history, and data from an adjacent block, exhuming at a different rate, are available, or where (2) blocks are tilted such that exhumation rates and/or depths vary across them.

Willett et al. continue to claim: "The argument that spatial variation maps into temporal variation was based on an intuitive argument (Figure 2) that was never tested. The reason why this argument fails is that there is no temperature history that can fit multiple data that have the same closure temperature, but different ages" (lines 1149-1151). Both statements are

- 480 false. Regarding the first statement, Schildgen et al. (2018) tested and demonstrated the spatial correlation bias for several realistic field scenarios, but Willett et al. dismissed those tests because they inferred that differences in the boundary conditions of the thermal model used to predict the ages (Pecube) and the model used to invert the ages (GLIDE) creates the spurious accelerations. However, this predicted effect is insignificant, based on the tests we present here (Figs. 1, 3 and 4). Therefore, Willett et al.'s dismissal of Schildgen et al.'s (2018) synthetic tests that demonstrate the spatial correlation bias is
- 485 unwarranted, as is the statement that Schildgen et al.'s (2018) argument was "never tested". If the second statement were

true, then the use of age-elevation profiles to infer exhumation histories would be impossible. In reality, it is impossible to fit multiple ages with the same closure temperature only if those samples are found at identical elevations (or more precisely, identical distances travelled since closure). Given that samples are rarely reported from identical elevations, and that elevation uncertainties are typically of the order of several tens of meters in any case, this limit to the feasibility of finding solutions from regions experiencing differing exhumation histories is not nearly as restrictive as Willett et al. imply.

490

495

Willett et al. argue that with a single thermochronometer, it can be very difficult to resolve a temperature history. They state "*This is why all sites identified by Herman et al. (2013) as having sufficient resolution, have ages from more than one thermochronometer*" (lines 1193-1194). This statement is incorrect. Five out of the 32 sites deemed to show sufficient resolution by Herman et al. (2013) included data from only a single thermochronometer. Sites that comprised only apatite fission-track data include Aconquija, the Mérida Andes, the Kyrgyz Tien Shan and the western Pamir, and only apatite (U-Th)/He data were included in the inversion for southern Peru.

A.2 The "Chicken or Egg" debate

Willett et al. state: "we have established that there are no spurious accelerations in erosion, only genuine ones" (line 1227).
While intriguing, this statement is wholly unsupported, and also contradicted by the many times that the authors emphasize
how the resolution cut-off value used by Herman et al. (2013) was inappropriate. Willett et al.'s conclusion that a resolution cut-off of 0.5 is more appropriate would eliminate 90% of the results reported by Herman et al. (2013), which includes 25 out of the 32 "resolved" locations of exhumation-rate changes in the late Cenozoic. Even this higher cut-off does not address the spatial correlation biases, model errors, and operator errors that compromise the results presented in Herman et al. (2013). Thus, "genuine" accelerations in erosion rate characterise only a small minority of the cases put forward by Herman

505 et al. (2013); to be precise, Schildgen et al. (2018) argued that "genuine" accelerations in erosion were identified in seven out of 32 regions (which are not the same as those with resolution > 0.5).

Willett et al. note that "Given the target timeframe of the last 6 Myr, young ages are needed and this gives a bias ... toward high erosion rates, but it does not follow that this leads to a bias towards recent acceleration" (lines 1249-1251). As we argued in section 2.3, a bias does follow, if resolution is better in the most recent time bin, and a prior erosion-rate value is

510 selected that is lower than the median erosion rate in that most recent time bin. We believe that such a reversion-to-the-prior bias is still dwarfed by the spatial correlation bias based on our analyses and those of Herman et al. (2013), but without being able to examine the results of the tests that Herman et al. (2013) performed with alternative prior values in detail, it is difficult to make a conclusive statement in this regard, as the two types of biases are intertwined.

Willett et al. suggest on lines 1253-1254 that in Schildgen et al. (2018), "complicating factors including the difficulty of 515 establishing cause and effect in a system with feedback were not discussed." In fact, Schildgen et al. (2018) discussed this difficulty in detail regarding the St. Elias range in Alaska and the Kyrgyz Tien Shan, both of which were inferred to show accelerations that can be linked to changes in climate and/or tectonics. For several of the other sites where Schildgen et al. (2018) concluded tectonics was the main driver for increases in erosion rates, the rationale behind that interpretation, which is largely based on the detailed studies of the authors who originally published the data, was explained. These original

520 studies often included more sophisticated 2D or 3D thermo-kinematic modelling of the data and independent geological support. But, in many of the locations with purported increases in exhumation according to Herman et al. (2013), there is no climate versus tectonics debate either because the accelerations noted by Herman et al. (2013) are erroneous, or because the accelerations appear real, but are very localized and limited to individual fault blocks that were exhumed due to local fault-geometry and stress-field configurations.

- 525 Willett et al.'s claim that in the literature review of Schildgen et al. (2018), "the approach used was to search recent literature for evidence of active tectonics and if found, they attributed not just young ages, but also recent acceleration, to tectonics" (lines 1255-1256) is an oversimplification and mischaracterization of the detailed analysis presented in the Supplementary Information of Schildgen et al. (2018). Willett et al. accuse Schildgen et al. (2018) of a "confirmation bias" (lines 1310-1339) that takes the form of neglecting to discuss the difficulty of distinguishing between tectonic and climatic
- 530 forcing of exhumation in a landscape, and neglecting to discuss a number of papers that purportedly contradict the interpretations of the causes of changes in exhumation rates. However, the examples the authors give for this bias (from comments we refer to above and next) are inaccurate portrayals of what is in Schildgen et al. (2018).

Willett et al. state: "In addition, although many previous studies using a variety of other interpretation methods found results that support Herman et al. (2013) (e.g., Zeitler et al., 1982; Ehlers et al., 2006; Thiede and Ehlers, 2013; Michel et al.,

- 535 2018; Vernon et al., 2008; Shuster et al., 2005; Sutherland et al., 2009; Thomson et al., 2010a,b; Avdeev et al., 2011; Shuster et al., 2011; Ballato et al., 2015; Bracciali et al., 2016), none of these studies swayed an interpretation away from their "spurious" assessment" (lines 1326-1330). Although we would not claim that Schildgen et al. (2018) cited every relevant paper in a world-spanning, but abbreviated review of 195 papers, several of these papers mentioned above by Willett et al. were indeed cited and discussed by Schildgen et al. (2018), and those interpretations were used to infer
- 540 spurious, tectonic or glacial causes of increases. To give just a few examples, Ballato et al. (2015) was cited as evidence for a tectonically driven increase in exhumation in the Alborz mountains; Avdeev and Niemi (2011) was cited to support the interpretation of a tectonic driver for uplift in the Greater Caucasus; Shuster et al. (2011) was cited as evidence for localized glacial incision in Fiordland; and Thomson et al. (2010) was cited in support of some of the erosion-rate increases in the Apennines being real and related to tectonics.

**545 References**

555

- Adams, B. A. and Ehlers, T. A.: Deciphering topographic signals of glaciation and rock uplift in an active orogen: a case study from the Olympic Mountains, USA, Earth Surf. Proc. Landf., 42, 1680–1692, doi: 10.1002/esp.4120, 2017.
- Avdeev, B. and Niemi, N. A.: Rapid Pliocene exhumation of the central Greater Caucasus constrained by low-temperature thermochronometry, Tectonics, 30, TC2009, doi: 10.1029/2010TC002808, 2011.
- 550 Baker, V.R.: Hypotheses and geomorphological reasoning. In Rhoads, B.L. and Thorn, C.E. (eds.): The Scientific Nature of Geomorphology: Proceedings of the 27th Binghampton Symposium in Geomorphology, Wiley, Chichester, p. 57-85, 1996.
  - Ballato, P., Landgraf, A., Schildgen, T. F., Stockli, D. F., Fox, M., Ghassemi, M. R., Kirby, E. and Strecker, M. R.: The growth of a mountain belt forced by base-level fall: Tectonics and surface processes during the evolution of the Alborz Mountains, N Iran, Earth Planet. Sci. Lett., 425, 204–218, doi: 10.1016/j.epsl.2015.05.051, 2015.
  - Bermúdez, M. A., van der Beek, P. and Bernet, M.: Asynchronous Miocene-Pliocene exhumation of the central Venezuelan Andes, Geology, 39, 139–142, doi: 10.1130/G31582.1, 2011.
    - Beucher, R., van der Beek, P., Braun, J. and Batt, G. E.: Exhumation and relief development in the Pelvoux and Dora-Maira massifs (western Alps) assessed by spectral analysis and inversion of thermochronological age transects, J. Geophys.
- 560 Res., 117, F03030, doi: 10.1029/2011JF002240, 2012.
  - Butler, R. W. H.: Tectonic evolution of the Himalayan syntaxes: the view from Nanga Parbat, Geol. Soc. London Spec. Publ., 483, 215–254, doi:10.1144/SP483.5, 2019.

- Crowley, J. L., Waters, D. J., Searle, M. P. and Bowring, S. A.: Pleistocene melting and rapid exhumation of the Nanga Parbat massif, Pakistan: Age and P–T conditions of accessory mineral growth in migmatite and leucogranite, Earth Planet. Sci. Lett., 288, 408–420, doi:10.1016/j.epsl.2009.09.044, 2009.
- Egli, D., Mancktelow, N. and Spikings, R.: Constraints from 40Ar/39Ar geochronology on the timing of Alpine shear zones in the Mont Blanc-Aiguilles Rouges region of the European Alps, Tectonics, 36, 730–748, doi: 10.1002/2016TC004450, 2017.
- Ehlers, T. A., Willett, S. D., Armstrong, P. A. and Chapman, D. S.: Exhumation of the central Wasatch Mountains, Utah: 2.
- 570 Thermokinematic model of exhumation, erosion, and thermochronometer interpretation, J. Geophys. Res., 108, 2173, doi: 10.1029/2001JB001723, 2003.
  - Fox, M., Herman, F., Willett, S. D. and May, D. A.: A linear inversion method to infer exhumation rates in space and time from thermochronometric data, Earth Surf. Dynam., 2, 47–65, doi: 10.5194/esurf-2-47-2014, 2014.
- Fox, M., Herman, F., Willett, S. D. and Schmid, S. M.: The exhumation history of the European Alps inferred from linear
  inversion of thermochronometric data, Am. J. Sci., 316, 505–541, doi: 10.2475/06.2016.01, 2016.
  - Fuller, C. W., Willett, S. D., Fisher, D. and Lu, C. Y.: A thermomechanical wedge model of Taiwan constrained by fissiontrack thermochronometry, Tectonophysics, 425, 1–24, doi: 10.1016/j.tecto.2006.05.018, 2006.

Glotzbach, C., Bernet, M. and van der Beek, P.: Detrital thermochronology records changing source areas and steady exhumation in the Western European Alps, Geology, 39, 239–242, doi: 10.1130/G31757.1, 2011a.

- 580 Glotzbach, C., van der Beek, P. A. and Spiegel, C.: Episodic exhumation and relief growth in the Mont Blanc massif, Western Alps from numerical modelling of thermochronology data, Earth Planet. Sci. Lett., 304, 417–430, doi: 10.1016/j.epsl.2011.02.020, 2011b.
  - Herman, F., Braun, J. and Dunlap, W. J.: Tectonomorphic scenarios in the Southern Alps of New Zealand, J. Geophys. Res., 112, B04201, doi: 10.1029/2004JB003472, 2007.
- 585 Herman, F., Copeland, P., Avouac, J.-P., Bollinger, L., Mahéo, G., Le Fort, P., Rai, S., Foster, D., Pêcher, A., Stüwe, K. and Henry, P.: Exhumation, crustal deformation, and thermal structure of the Nepal Himalaya derived from the inversion of thermochronological and thermobarometric data and modeling of the topography, J. Geophys. Res., 115, B06407, doi: 10.1029/2008JB006126, 2010.

Herman, F., Cox, S. C. and Kamp, P. J. J.: Low-temperature thermochronology and thermokinematic modeling of

- 590 deformation, exhumation, and development of topography in the central Southern Alps, New Zealand, Tectonics, 28, TC5011, doi: 10.1029/2008TC002367, 2009.
  - Herman, F., Seward, D., Valla, P. G., Carter, A., Kohn, B., Willett, S. D. and Ehlers, T. A.: Worldwide acceleration of mountain erosion under a cooling climate, Nature, 504, 423–426, doi: 10.1038/nature12877, 2013.

Jamieson, R. A. and Beaumont, C.: On the origin of orogens, Geol. Soc. Am. Bull., 125, 1671-1702, doi: 10.1130/B30855.1,

595

2013.

565

- Jiao, R., Herman, F. and Seward, D.: Late Cenozoic exhumation model of New Zealand: Impacts from tectonics and climate, Earth Sci. Rev., 166, 286–298, doi: 10.1016/j.earscirev.2017.01.003, 2017.
- Montgomery, D. R.: Valley formation by fluvial and glacial erosion, Geology, 30, 1047–1050, doi: 10.1130/0091-7613(2002)030<1047:VFBFAG>2.0.CO;2, 2002.
- 600 Robert, X., van der Beek, P., Braun, J., Perry, C., Dubille, M. and Mugnier, J. L.: Assessing Quaternary reactivation of the Main Central thrust zone (central Nepal Himalaya): New thermochronologic data and numerical modeling, Geology, 37, 731–734, doi: 10.1130/G25736A.1, 2009.

Rolland, Y., Rossi, M., Cox, S. F., Corsini, M., Mancktelow, N., Pennacchioni, G., Fornari, M. and Boullier, A. M.: 40Ar/39Ar dating of synkinematic white mica: insights from fluid-rock reaction in low-grade shear zones (Mont Blanc

- Massif) and constraints on timing of deformation in the NW external Alps, Geol. Soc. London Spec. Publ., 299, 293– 315, doi: 10.1144/SP299.18, 2008.
  - Rossi, M. and Rolland, Y.: Stable isotope and Ar/Ar evidence of prolonged multiscale fluid flow during exhumation of orogenic crust: Example from the Mont Blanc and Aar Massifs (NW Alps), Tectonics, 33, 1681–1709, doi: 10.1002/2013TC003438, 2014.
- 610 Schildgen, T. F., Ehlers, T. A., Whipp, D. M., Jr., van Soest, M. C., Whipple, K. X. and Hodges, K. V.: Quantifying canyon incision and Andean Plateau surface uplift, southwest Peru: A thermochronometer and numerical modeling approach, J. Geophys. Res., 114, F04014, doi: 10.1029/2009JF001305, 2009.
  - Schildgen, T. F., van der Beek, P. A., Sinclair, H. D. and Thiede, R. C.: Spatial correlation bias in late-Cenozoic erosion histories derived from thermochronology, Nature, 559, 89–93, doi: 10.1038/s41586-018-0260-6, 2018.
- 615 Schneider, D. A., Zeitler, P. K., Kidd, W. and Edwards, M. A.: Geochronologic constraints on the tectonic evolution and exhumation of Nanga Parbat, western Himalaya syntaxis, revisited, J. Geol., 109, 563–583, doi: 10.1086/322764, 2001.

Shuster, D. L., Cuffey, K. M., Sanders, J. W. and Balco, G.: Thermochronometry reveals headward propagation of erosion in an alpine landscape, Science, 332, 84–88, doi: 10.1126/science.1202357, 2011.

- Shuster, D. L., Ehlers, T. A., Rusmore, M. E. and Farley, K. A.: Rapid glacial erosion at 1.8 Ma revealed by 4He/3He thermochronometry, Science, 310, 1668–1670, doi: 10.1126/science.1118519, 2005.
- Sutherland, R., Gurnis, M., Kamp, P. J. J. and House, M. A.: Regional exhumation history of brittle crust during subduction initiation, Fiordland, southwest New Zealand, and implications for thermochronologic sampling and analysis strategies, Geosphere, 5, 409–425, doi: 10.1130/GES00225.1, 2009.
  - Thiede, R. C. and Ehlers, T. A.: Large spatial and temporal variations in Himalayan denudation, Earth Planet. Sci. Lett., 371-

625 372, 278–293, doi: 10.1016/j.epsl.2013.03.004, 2013.

- Thomson, S. N., Brandon, M. T., Reiners, P. W., Zattin, M., Isaacson, P. J. and Balestrieri, M. L.: Thermochronologic evidence for orogen-parallel variability in wedge kinematics during extending convergent orogenesis of the northern Apennines, Italy, Geol. Soc. Am. Bull., 122, 1160–1179, doi: 10.1130/B26573.1, 2010.
- Tippett, J. M. and Kamp, P. J. J.: Fission track analysis of the Late Cenozoic vertical kinematics of continental Pacific crust,
  South Island, New Zealand, J. Geophys. Res., 98, 16119–16148, doi: 10.1029/92JB02115, 1993.
- Treloar, P. J., Rex, D. C., Guise, P. G., Wheeler, J., Hurford, A. J. and Carter, A.: Geochronological constraints on the evolution of the Nanga Parbat syntaxis, Pakistan Himalaya, Geol. Soc. London Spec. Publ., 170, 137–162, doi: 10.1144/GSL.SP.2000.170.01.08, 2000.
- Valla, P. G., Shuster, D. L. and van der Beek, P. A.: Significant increase in relief of the European Alps during midPleistocene glaciations, Nature Geosci., 4, 688–692, doi: 10.1038/ngeo1242, 2011.
  - Willenbring, J. K. and Jerolmack, D. J.: The null hypothesis: globally steady rates of erosion, weathering fluxes and shelf sediment accumulation during Late Cenozoic mountain uplift and glaciation, Terra Nova, 28, 11–18, doi: 10.1111/ter.12185, 2016.
    - Willett, S., Beaumont, C. and Fullsack, P.: Mechanical model for the tectonics of doubly vergent compressional orogens,

640 Geology, 21, 371–374, doi: 10.1130/0091-7613(1993)021<0371:MMFTTO>2.3.CO;2, 1993.

Willett, S. D., Fisher, D., Fuller, C., Yeh, E.C. and Lu, C.Y.: Erosion rates and orogenic-wedge kinematics in Taiwan inferred from fission-track thermochronometry, Geology, 31, 945–948, doi: 10.1130/G19702.1, 2003.

- Willett, S.D., Herman, F., Fox, M., Stalder, N., Ehlers, T.A., Jiao, R., Yang, R.: Bias and error in modelling thermochronometric data: resolving a potential increase in Plio-Pleistocene erosion rate. Earth Surf. Dyn. Discussions, doi: 10.5194/esurf-2020-59.
- 645

Zeitler, P. K.: Cooling history of the NW Himalaya, Pakistan, Tectonics, 4, 127–151, doi: 10.1029/TC004i001p00127, 1985.

- Zeitler, P. K., Johnson, N. M., Naeser, C. W. and Tahirkheli, R.: Fission-track evidence for Quaternary uplift of the Nanga Parbat region, Pakistan, Nature, 298, 255–257, doi: 10.1038/298255a0, 1982.
- Zeitler, P. K., Koons, P. O., Bishop, M. P., Chamberlain, C. P., Craw, D., Edwards, M. A., Hamidulah, S., Qasim Jan, M.,
- 650 Khan, M. A., Khattak, M. U. K., Kidd, W. S. F., Lackie, R. L., Meltzer, A. S., Park, S. K., Pêcher, A., Poage, M. A., Sarker, G., Schneider, D. A., Seeber, L. and Shroder, J. F.: Crustal reworking at Nanga Parbat, Pakistan: Metamorphic consequences of thermal-mechanical coupling facilitated by erosion, Tectonics, 20, 712–728, doi: 10.1029/2000TC001243, 2001.

---

## Editor Comment (EC1) · Richard Gloaguen (Editor) · 18 Sep 2020

I encourage scientists of our community to actively contribute to this controversial discussion. Please ensure that the review process occur in a good manner (impartial, independent and fair) by keeping your posts factual.

---

## Referee Comment (RC1) · Anonymous Referee #1 · 28 Sep 2020

This manuscript seems to have been written as a detailed rebuttal of the conclusions of the paper by Schildgen et al. (2018) which questioned the accuracy of the linearised inversion method of Fox et al. (2014) of thermochronological data. Specifically Schildgen et al. (2018) suggested that the methodology encouraged the incorrect interpretation of spatial variability as temporal variability, and thereby creating a systematic bias towards an apparent temporal acceleration in exhumation. No doubt the authors are motivated by the fact that the earlier paper, published in Nature I see, brings into question much published work by the present authors.

My overall reaction is that the present manuscript contains some useful information in

that it sets out all of the resolution and uncertainty issues associated with the method of Fox et al. (2014).

The present manuscript quantitatively describes through theory and example the sources of uncertainty and bias in the Fox et al. (2014) approach, including those arising from theory errors (due to forward modelling) data errors and the limited resolution caused by data sampling and the physics of the data-model relationship. Again this is largely in the context of combatting assertions by Schildgen et al. (2018), nevertheless I think this exposition is in principal useful to have in the literature.

At the same time the manuscript is excessively long (my version is 78 pages) and an arduous and at times repetitive read in parts. I question how many readers will understand it all in its present form. There is a good argument to move a lot of this material to an appendix or supplementary material without loss of the main points. I realize that the authors are probably intent on getting this full detail into the literature, but I think having it all in the main body of a paper will certainly diminish its likely impact. I suggest the Bayesian Inference tutorial material could be moved to an Appendix or left out and the results cited as they are required. In addition do we really need to see all numerical tests? again use of appendices would help. Much of the underlying theory of Bayesian inference appears in the literature cited, but there are also many books outside of the Earth Sciences literature, admittedly two popular ones, Menke and Tarantola are cited here and so perhaps better use of specific results in those books could be used here.

Some of the writing is clumsy, an example is on line 251 '...an inversion scheme based on inverse problem theory which minimizes Gaussian errors in observations, and a Bayesian parameter model in which model parameters are assumed to have Gaussian distributions about a prior value'. Some more conciseness would be a help here and in many other places. Note that in Bayesian inference we don't actually 'Minimize Gaussian errors in observations' rather a Likelihood function p(d | m) expresses the probability of observing the data given a model. It may or may not take a Gaussian form,

indicating all departures from fitting the data perfectly are expected to be described by a Gaussian PDF. Its maximization occurs in Maximum Likelihood techniques not Bayesian Inference. One might consider Maximizing the Posterior PDF as one way of characterising that PDF but this differs from Likelihood maximisation when the prior is not uniform. Of course when the data model relationship is linear and Likelihood and Prior are Gaussian then the posterior PDF is also Gaussian.

While I have not studied the Fox et al. (2014) paper in any detail it is clear from the account here that it is a linearised Bayesian inversion method combining a prior with a Likelihood to estimate a Gaussian approximated posterior probability distribution. This approach is well known and understood, especially in the geophysical literature for many decades. The works of Tarantola and Valette being a prime embodiment of the approach. The present manuscript gives a summary of the linearised theoretical results as they pertain to quantification of resolution and model covariance matrices and their interpretation. It appears to me to be a competent summary of linearised inversion within a Bayesian framework, my comments above notwithstanding. I do feel that some references to the latest editions of some well established books such as Menke (2012, rather than 1984) or Aster, Borchers and Thurber (2018, 3rd edition) could assist the reader in this regard, as all of the relevant theory is set out there. The paper references relevant to linear discrete parameter estimation, or which this is an example, are, correctly, to the original works from the 1970s-1980s which is good to see.

From L430 onward the authors' analyse the Normalised erosion rate metric, NR, eqn. (14) (lines 425-475). This is a nonlinear function of the model parameters recovered in an inversion. I note that there is a detailed critique of this term, as it was important to Schildgen et al. (2018) to support some of their arguments. A discussion is given of Expectation of a ratio (Line 440). This expectation identity relies on the terms x and x being independent which they are clearly not in eqn. (14), since both involve e1 and e2. There must be a cross term affecting the Expectation of theFrom L430 onward ratio
which is ignored here. This is an error.

The authors continue on and obtain an approximation of the variance of NR using a Taylor expansion, as well as discussing its bias and its complicated behaviour. Quantities such as variance and expected value of some property of the model parameters while analytical, are just moments of what is in reality a probability distribution over NR constrained by the data. Since the point of Bayesian inversion is to determine a Posterior probability distribution of the model parameters then by drawing from that (Gaussian) distribution and calculating NR provides a posterior ensemble of NR values from which a complete (non Gaussian) PDF can be inferred. A better (more complete) approach would be to map out that PDF in full rather than drawing inferences from estimates such as variance, which are at best only partially characterise its behaviour. How well can a variance describe a high skewed distribution anyway? As I understand it this is the objective of the authors, i.e. trying to quantify the uncertainty and bias in the NR term, in which case it is better to examine the actual distribution, if its accessible, and I think here it must surely be. The PDFs give the full picture, low order moments such as variance and expectation do not.

The authors are correct in their assertions (L1305-1310) that the Bayesian solution will tend to go to the prior when data influence is poor, which means have large data covariances. This is well known and a fundamental property of Bayesian inference. However it is not the case that the posterior will tend to the prior when data, or averaged data (in this case), are inconsistent. It is not entirely clear to me that this is what the authors are suggesting, but I got the impression it might be implied by their discussion of resolution and the influence of the prior (multiple sections across the manuscript). This seems to be a pertinent point in their rebuttal of the Schildgen results. Inconsistent data (e.g. tight constraints on a subset of model parameters which do not agree) is quite a different thing from weak data constraints (e.g. due to large data noise). Only the weak constraint case does the posterior tend to the prior, not the inconsistent data case.
A toy example makes this clear. Consider a single prior variable with Gaussian uncertainty. Imagine that new data are a direct observation of this variable. The weak data case is when measurements occur with large errors relative to the Gaussian prior. They will have little affect on the prior mean, i.e. posterior tends to the prior. The inconsistent data case is when measurements with errors comparable to the Gaussian prior are widely separated. Here the posterior mean may be far from the prior mean. As I say its not entirely clear to me whether the authors have mistook one for the other in their arguments, but even if they haven't, I think it best to explicitly clarify this difference in the manuscript so that there is no confusion in the readers mind.

How is the closure temperature estimation from Dodson (1973) applicable here? Isn't this a rather approximate method in itself? How relevant is it to low temperature thermochronological data such as fission track and He? As I understand it there is a continuous response, in terms of annealing of diffusion over a range of temperatures, which would be a significant proportion (say 50%) of the estimated closure temperature. In this case the measured age is not directly related to the time that the system became closed (i.e. started to accumulate fission tracks or He). Shouldn't one ideally solve the full FT annealing and He diffusion equations to find when and at what depth the system became closed? This does not appear to be done here. No doubt if it were this would create a more nonlinear inversion problem which may be more difficult to handle with the linear/linearized inversion approaches used here? These issues should probably be discussed and justified.

Ultimately any inverse model result is conditional on the assumptions (e.g. model resolution matrix depends on the assumed linear relationship between model and data). I agree that using different forward models to generate synthetic data and perform an inversion introduces unquantified 'theoretical errors' in an inversion test. Quantifying such errors however is usually difficult, especially systematic errors.

The question of spatial vs temporal variability in inference results, which is at issue as one of the central arguments between the present authors' and those of Schildgen et

al. (2018). From what I can tell the original work of Fox et al. collected and smoothed over data sets spatially by introducing a Gaussian prior imposing a spatial (smoothing) covariance. Much is discussed on this point here and whether this introduces bias and encourages spatial variability to be mis-interpreted as temporal variability. The present authors obviously argue not, however its not clear to me why use of global spatial prior smoothing of Fox et al. is even necessary. It seems a rather backward step in itself, compared to more sophisticated 'data adaptive' spatial smoothing approaches popular in Bayesian methods these days. I'm thinking here of Bayesian spatial sampling known as Partition modelling which has been around for more than twenty years, See Denison et al. (2002 for a summary). That type of approach allows sharp local variations to be recovered if they are supported by the data, e.g. by sampling data across a discontinuity. This is much more preferable than simply imposing global spatial correlation via a prior. I see Partition Modelling has already been applied to in low temperature Thermo-chronology problems by Stephenson et al. (2006) almost than a decade and a half ago. This paper or any related work is not cited or discussed. In that sense I see a lot of energy here expended here on defending the validity of what appears to be a somewhat outdated framework.

My impression is that the manuscript is detailed defence of the Fox et al. (2014) inversion approach which may be useful for readers to see in some form. However it is unnecessarily long and could probably start at page 35 after an introduction. A manuscript 50% shorter would be more appropriate and I expect have more impact.

Minor points:

I can see no content in eqn. (8)? - It appears blank to me.

References: Menke W. (2012) Geophysical Data Analysis: Discrete Inverse Theory, 3rd ed. Aster, R. Borchers B. & Thurber C. (2018). Parameter Estimation and Inverse Problems - 3rd Edition, Elsevier

D.G.T. Denison, N.M. Adams, C.C. Holmes, D.J. Hand, Bayesian partition modelling,

[Figure]

Comput. Stat. Data Anal. 38 (4) (2002) 475– 485.

Stephenson, J. and Gallagher, K. and Holmes, C. C., Low temperature thermochronology and strategies for multiple samples 2: Partition Modelling for 2D/3D distributions with discontinuites, EPSL, 2006, 241, 557-570.

**ESurfD**

---

## Referee Comment (RC2) · Anonymous Referee #2 · 5 Oct 2020

**1    Overview**

Willett et al. (present study; *W+PS*) argue that the analyses and conclusions of Herman et al. (2013; *H+13*) remain valid, despite the critiques of Schildgen et al. (2018; *S+18*) and Willenbring and Jerolmack (2016; *WJ16*).

But what is the conclusion of Herman et al. (2013)? Is their conclusion that some mountainous, glaciated regions were exhuming faster between 0-2 Ma compared with 4-6 Ma? Is their conclusion that most glaciated mountains were exhuming faster between these times? Most mountains? Or that there was a net increase in erosion rates

globally?

The central argument of *H+13* / *W+PS* is that several regions have yielded low-T thermochronology data which support high-resolution reconstruction of exhumation from 6 Ma to present. When binned in 2 Ma intervals, these reconstructions show increased exhumation rate between two time periods, 4-6 Ma and 0-2 Ma, an increase coincident with the onset Northern Hemisphere glaciations.

**2    Schildgen et al., 2018 critique**

*S+18* argued that erosion rate increases at these 'high-resolution' sites were often spurious, the result of *H+13* inadvertently grouping samples with very different geomorphic or tectonic histories. For example, a 20 Ma apatite (U-Th)/He (hereafter 'AHe') age taken from atop a plateau might constrain the "long term" exhumation rate, while a 1 Ma AHe age from a glacial valley bottom constrain the more recent exhumation rate. Even if the plateau and valley were eroding at different but constant erosion rates, they could be accidentally combined to calculate a shared exhumation history. The outcome of this 'spatial correlation bias,' *S+18* argued, was an apparent but false increase in erosion rate.

*W+PS* counter that the inversion routine of *H+13* (GLIDE; Fox et al., 2014) does not suffer from this effect. First, samples may only pass through the cooling temperature of each thermochronometer once and therefore disparate ages from the same thermochronometer cannot be grouped by GLIDE. Second, when low-T thermochronology data are simulated for a rapidly uplifting block adjacent to a slowly uplifting footwall, both rapid and slow exhumation are correctly inverted for and no spurious acceleration is observed, even though the rapid block has better calculated resolution. Finally, *W+PS* argue that *S+18* incorrectly compared ages from simulations with a fixed basal temperature (Pecube) and with a flux basal temperature (GLIDE). This mistake, *W+PS*

argue, accounts for the false increase in exhumation rate presented by *S+18*.

**3   Willenbring and Jerolmack, 2016 critique**

*WJ16* argued that the methodology of *H+13* is biased because only sites with rapid exhumation will precisely resolve a recent change in exhumation. Even if GLIDE perfectly identifies regions where thermochronology ages resolve a recent increase in erosion rate (a point contested by *S+18*), there will be many regions which have experienced constant or lower erosion rates that are cannot be resolved and are therefore under-represented.

In order to be collected, rocks must exhume up through the crust to Earth's surface after passing through the relevant closure temperatures at depth. The time required for this transit imposes a limit on which erosion rates can be measured across Earth's surface for any given time interval. Critically, the minimum resolvable exhumation rate for a given thermochronometer increases nonlinearly toward the present and the distribution of measurable erosion rates becomes increasingly truncated. *WJ16* argue that this effect biases the analyses of *H+13*, that any apparent increase in erosion rate toward the present actually signifies a progressive clipping of the measurable erosion rates for a given time interval.

*W+PS* mention this critique but grossly misrepresent the *WJ16* argument. Here is the original *WJ16* argument:

> "[W]e plot a variation of an idea of Anders et al. (1987) that describes two clipped distributions of sedimentation rate data both in the maximum depth of a population sampled and in the precision possible for the youngest ages...the conceptual analogy is useful."

and

> "The essence of the argument is that slower rates of exhumation are progressively clipped from the measured age distribution of rocks as one approaches the modern because of the precision of the technique. The reason is that thermochronometers measure the time since achieving an associated closure depth; rate is simply the measured time divided by this depth. If erosion rates have not been high enough to bring rocks to the surface, the rates associated with those buried rocks cannot be measured."

And here is the response to this critique in *W+PS*:

> "...Willenbring and Jerolmack (2016) misinterpreted the meaning of the limit to resolution, or what they called "precision". They took this concept from measurements of sediment layer thickness [the 'idea of Anders et al., (1987)' referenced in *WJ16* argument above], which can be biased by the inability to measure layer thicknesses below a specific precision (Anders et al., 1987).
> "Thermochronometric data are fundamentally different in that an age does not represent an estimate of rate at a point or across a closed interval of time such as represented by a stratigraphic layer, but rather, represents the integral of the exhumation rate...from the time of closure to the present day. Thus, rather than being unsampled, the region [from the youngest closure age until present] is actually the most heavily sampled part of parameter space. The better definition of the "limit to resolution" is that no change of exhumation rate can be resolved between this limit and time zero, because there is no sampling internal to this part of the parameter space, but the average rate across this interval is sampled and can be determined."

This response misconstrues the legitimate criticism raised by *WJ16*.

More troubling, this limitation of only having access to rapid exhumation measurements for the time period in question (0 - 2 Ma) is never dealt with in a systematic way later in the manuscript. Given that GLIDE will assign high resolution values when "the time interval is resolved in the neighborhood of the point of interest" and will lower resolution values when "contamination enters from the other time intervals," it is unsurprising that the regions for which there is good resolution since 2 Ma are largely regions which are exhuming very rapidly.

**4   Sensitivity analysis**

What remains to be demonstrated is whether the methodology of *H+13* / *W+PS* could resolve constant or decreasing erosion rates within a synthetic test. The analyses in Section 4 of *W+PS* indicate that constant but spatially distinct long-term erosion rates can be resolved, but similar tests are not provided for a recent change in exhumation rate.

If not all possible recent changes in exhumation can be resolved by the *H+13* / *W+PS* methodology, we are left with the original question of how to interpret the *H+13* observation that some regions appear to exhume faster during $0 - 2$ Ma compared with $4 - 6$ Ma. Presumably, some regions have also recently experienced slower exhumation and some have remained constant. Within even a single orogen this seems plausible.

**5   The problem of still buried rocks**

ll. 185-189. "[I]f one regresses age against depth (Figure 2d) with a moving depth window, one obtains the correct, unbiased regional mean erosion rate (Figure 2e, Average

1)."

This statement and the corresponding figure, *W+PS* Fig. 2d (reproduced here in Fig. 1), neglect the fact that a significant portion of rocks which passed through the closure depth after a recent change in exhumation rate will not yet have reached Earth's surface. *W+PS* Fig. 2d illustrates only a special case, where erosion is spatially varied but constant through time. This is unfortunate and misleading, because the question at hand is whether transient erosional signals (i.e., the onset of Northern Hemisphere glaciations) can be measured (in the midst of spatial variability).

If an instantaneous change in exhumation rate occurred recently, it will only be observable if rocks: a) began beneath the closure depth and b) were exhumed from that depth to the surface to be collected. This exhumation distance imposes a limit on which recent exhumation rates can be measured (Fig. 1).

The following thought experiment crudely estimates the magnitude of this 'still buried rock' bias effect (Fig. 2). *H+13* evaluated the difference in global erosion rates between the time periods of $4 - 6$ Ma and $0 - 2$ Ma. So, it is useful to consider what would happen if global erosion rates changed significantly and instantaneously at 2 Ma. Apatite $(U - Th)/He$ is the lowest temperature system considered in *H+13* and would close at a depth of approximately 2 km, depending on the local exhumation rate and geothermal gradient. At 2 Ma, all rocks below this closure depth would begin their ascent according to the new surficial erosion rate. Therefore, a geologist collecting surficial samples today could only possibly collect samples which rose at a rate of (2 km) / (2 Ma) or greater. All other rocks would still be buried. This effect is illustrated in Fig. 2 for a range of times at which erosion rates could have instantaneously changed.

Extending this exercise, we could assume that if global erosion rates changed 2 Ma, they became similar to erosion rates observed today, a reasonable null hypothesis (whether global erosion accelerated or decelerated is immaterial for the exercise).

A global compilation of cosmogenic 10Be erosion rate estimates by Portenga and Bierman (2011) can serve as a primitive estimate of how global erosion rates are distributed today (Fig. 2). (Obviously, there are many caveats to this simplification, but the broad range of erosion rates represented and the relative rarity of erosion rates greater than 1 km/Ma should be uncontroversial.) In this scenario, only a small fraction of rocks from that 2 Ma erosion rate change would have travelled to the surface. The large majority would still be in transit.

From this exercise, two observations stand out. First, only terrains experiencing very rapid exhumation will inform the thermochronology record of erosion rate changes at 2 Ma. Second, it is much more likely to observe accelerations in exhumation than to observe constant or decelerating exhumation. To avoid confusion, note the following distinction. The approach of *H+13* / *W+PS* will register low erosion rates during the period of 2 Ma to present owing to old AHe and AFT ages (how 'resolved' low rates are depends on the GLIDE algorithm). What the *H+13* / *W+PS* approach will not register, however, are any decelerations down to rates below about 1km/Ma.

We must take as a null hypothesis the idea that global net erosion rate change was zero from 6 Ma until present. For the reasons outlined above, it is my view that the *H+13* / *W+PS* methodology is inherently incapable of resolving unchanging net erosion rates globally. The authors have not taken seriously this critique, initially outlined in *WJ16*, and have not demonstrated their ability to resolve a constant erosion rate or a decreasing erosion rate.

**Figure captions**

**Fig. 1:** *W+PS* Fig. 2d (left panel) is meant to illustrate that one can obtain an unbiased estimate of regional erosion by regressing cooling rates as a function of closure depth (rather than as a function of time). On the right-hand panel, it is shown that, even when regressing against closure depth, apparent cooling rates are inherently biased owing

to the fact that rocks exhuming below some minimum resolvable cooling rate (dot-dash line) will not be exposed at Earth's surface.

**Fig. 2:** For a closure depth of 2 km, approximate to apatite (U-Th)/He, the minimum resolvable exhumation rate is shown for an erosion rate change some time before present. For reference, these rates are compared with a global compilation of erosion rates from the CRONUS database, as presented by Portenga and Bierman, 2011 (right-hand panel). For example, if at 2 Ma (star), global erosion rates changed to the distribution shown on the right, only a small fraction of these rates would be represented at the surface within 2 Ma (shaded region); the majority of erosion rates would be 'archived' within rocks still in transit to the surface.

[Figure]

**Fig. 1.**

[Figure]

**Fig. 2.**

---

## Author Comment (AC2) · 5 Oct 2020

The comment offered by van der Beek et al. is long. There are many issues in their comment that we will clarify in subsequent responses, but we begin with a couple of the more important points here. We do this in the spirit of an open discussion format offered by this journal and for the benefit of the broader ESurf community so as to avoid additional confusion in the thermochronometer community concerning the application of inverse models to thermochronometer data.

[Figure]

Response to comments on Model error in geotherm calculations

The most important point to address is the existence and importance of the errors made in the geotherm calculation in Schildgen et al. (2018) and demonstrated in Willett et al. (2020). In the section starting at line 84, van der Beek et al. (2020) acknowledge that they made the error, but claim that it is not a significant error. We do not see where they have demonstrated that this error is insignificant, and do not find any reason to take results of an analysis with acknowledged errors, over that conducted in our paper, for reasons that we will discuss here.

The fundamental problem is that one cannot use a steady state geotherm in a forward model to generate ages, invert them with a transient thermal model and expect to obtain a meaningful result. Regardless as to how one calibrates the two, a steady geotherm cannot approximate a transient geotherm over a significant period of time. It seems that we now agree on this point. However, van der Beek et al. argue that the steady solution based on a 30 km fixed temperature boundary condition is actually more accurate than the fixed gradient model included in Glide. This issue is not important to the error of mismatching forward and inverse models, but it is important to the thermochronometry community who need to accurately calculate geotherms, so we will digress for one (long) paragraph to discuss this issue.

We agree that there are settings where crustal thickening or underthrusting leads to downward advection of heat. There are also extensional systems, in which tectonic advection leads to upward advection of heat. Erosion leads to upward advection of heat. Any change of crustal thickness leads to changes in heat production (by crustal thickening or thinning) and thermal resistance. All of these processes lead to transience of the thermal field. However, the timescale of relaxation of this transience is determined by the thickness and heat content of the thermal lithosphere, which extends through the thermal boundary layer of the upper mantle to the asthenosphere. This has been well established since the first papers in plate tectonics (Williams and von Herzen, 1974; Sclater and Francheteau, 1970; McKenzie (1978); England and
Thompson; 1984; Furlong and Chapman, 2013). We have seen nothing in the subsequent 50 years to change this observation and tens of thousands of papers that have accepted and built upon the concept of the thermal lithosphere, so the claim that a base of lithosphere boundary condition has "exaggerated transience" (line 126) is not supported in the literature. There are studies that have chosen to approximate lithospheric transience with a thermal boundary condition at a shallower depth in the lithosphere, often because of computational limitations, but this has been (or should have been) recognized as an approximation. This approximation for a basal boundary condition can be either a constant temperature or constant gradient (also referred to as a constant flux condition based on Fourier's Law) and can be applied at a material point, e.g. at the base of the crust, which moves up or down with time, or at a spatial point, e.g. at a constant depth. A constant gradient has the advantages that the time to a new steady state is longer than the fixed temperature approximations, and is therefore closer to that of the full physical problem. The constant gradient and material point boundaries have the advantage that advective heating or cooling through the boundary is approximated, by changing temperature in the former, and changing position in the latter. The worst approximation is a constant temperature applied at a constant depth. The response time for a temperature boundary condition applied at a fixed, and shallow, depth is much shorter than the lithosphere problem and the fixed temperature suppresses all advective heat flux across this boundary, so that energy is not conserved at the boundary. Depending on the tectonic setting, and the timescale of interest, these approximations for a basal boundary condition provide different levels of accuracy. van der Beek et al argue that because one setting (underthrusting during contraction) has a downward component to its advection, an isothermal boundary at a constant and very shallow depth (e.g. 30 km) gives a better representation of the temperature field than a model that has no approximation to the thermal timescale, but includes only the erosional component to the advection. With crustal thickening there is a downward component to advection. Neither Glide nor PECUBE nor the 1-D model of Schildgen et al. (2018) include this downward advection, so all have a positive error

in temperature. van der Beek et al. argue that because the fixed 30-km-temperature boundary condition introduces a negative temperature error, i.e. it is much too cold, these two errors will offset each other to obtain a more accurate solution. We would like to see such a fortuitous outcome demonstrated, or supported with references. We are not aware of any previous studies documenting this in the geothermics community. In the van der Beek et al. comment, there is no model or error analysis offered and therefore no assurance that the negative error is not two or ten times larger than the positive error. Nor is there any reason to presume that spatial distributions in these errors will match. van der Beek et al. do not address extensional settings, where these errors would add constructively, or non-tectonic settings, where the fixed temperature condition provides the only error. In most convergent settings, the downward advection is more than compensated for by the increase in heat production, so orogens tend to have high lower crustal temperatures (Pope and Willett, 1998; Beaumont et al., 2004; review by Furlong and Chapman, 2013). Only subduction forearcs are characterized by lower than average crustal temperatures. Our model of a constant flux applied at a large depth will correctly represent the timescale of the thermal lithosphere and correctly includes advection due to isostatic uplift in response to erosion, but neglects changes in heat production and neglects advection due to vertical velocity gradients within the lithosphere that are negative for crustal thickening and positive for crustal thinning. Where erosion rates are large with respect to the vertical strain rate integrated over the lithospheric column, the model will be accurate.

To return to the more relevant issue, van der Beek argue that their geotherm error cannot be the sole error in the model because a model with no spatial variation in erosion rate does not result in an inferred acceleration of erosion (their Figure 1), as predicted by Figure 3 of Willett et al. (2020). However, this statement is not correct. The geotherms in Figure 3 are for a constant exhumation rate of 1 mm/yr. By including several tens of ages, incorrectly calculated, as input to the inverse model, the model they generated no longer recovers the correct erosion rates or the correct geotherm, which is coupled to the erosion rate. The geotherms in van der Beek et al., Figure 1
are not identical to the ones shown in our Figure 3 and these differences can easily account for the decceleration observed in the van der Beek et al result. We see no evidence that this model is anything other than exactly as we stated – in error because of the errors in the age calculation. To clarify the manuscript, we will change the caption to Figure 3, to express this caveat, to state: Schildgen et al. (2018) used synthetic ages calculated using a steady state geotherm similar to the blue curve, inverted them using temperatures predicted from a transient geotherm similar to the red curves, and concluded that the failure to recover the correct exhumation rate demonstrated a problem with spatial correlation in the GLIDE inversion method.

The second point made by van der Beek et al. is that the 16 to 28% errors that they do obtain are too small to be important, because these errors are "within the range of typical uncertainty associated with quantitative inferences of exhumation rates from thermochronology data" (line 114). There is no evidence that these errors are negligible, and it is unlikely that they will be. First, this is a single example and there is no assurance that another example will not give 40% or 80% error. In any case, we would argue that 20% error is over any significance level. These models are control experiments, whose sole purpose is to control for, and thus identify, error. They are not intended to simulate typical conditions with typical uncertainty (see discussion below). We remind the reader that the model includes a vertical, dip-slip fault with 36 km of exhumation, a feature that does not exist on Earth, so these are not typical conditions. The point of these models is to systematically control sources of error, in order to identify their relative importance (see our paper, Section 4.0 and 4.1). In fact, if an error cannot be eliminated entirely, the model loses its purpose, as errors in an outcome will remain ambiguous as to their source.

For this reason, we continue to see no reason to consider relevant any of the models of the van der Beek et al comment or Schildgen et al. (2018), based on PECUBE ages analysed through GLIDE. They contain errors in the ages of up to 100% and these translate into errors of tens of percent in erosion rate (at a minimum), and of unknown

magnitude and direction in acceleration. Even if there are other errors, such as spatial correlation bias, these can never be separated from the geotherm errors, so these models do not meet the intent of their construction, to identify the source and relative magnitude of errors. In contrast, the models in our paper have zero age error and have been constructed to isolate specific errors. Further support for our position is offered in the following section.

Response to: Bias to the prior vs spatial correlation bias

The second important point that we need to address is the way in which the synthetic models are constructed and what they show. This is first addressed in van der Beek et al 's comment section 2. They present several models (Figs 1 and 2) with large errors that they argue show combinations of resolution and correlation bias. However, these models also contain the geotherm errors discussed above. They also add random errors to the data, which adds a fourth source of error to the problem. Adding another dimension to the error is opposite to what these control experiments are intended to do – isolate, not compound, errors. Having four potential sources of error makes it impossible to identify which is responsible for the observed error. In contrast, the models presented in our paper isolate errors, in particular by removing the geotherm error, and not adding random noise. The comparable models in our paper show none of the same errors, even when data from both sides of the fault are included (e.g. Figs. 6, 14). We have not run the analysis on data from only one side of the fault, but if there is no error in the combined data, there will be no error in the simpler case. The simplest conclusion is that all error in the van der Beek et al. comment and Schildgen et al. (2018) models is due to these geotherm errors, possibly with a component due to the added noise.

Rather than accept this conclusion, van der Beek et al. argue that we have "cleverly designed" (Line 163) the data set in order to avoid the errors produced by their models. Figure 5 of our paper shows the model data sets that we generated. We attach here a modified version of this figure that includes the data distribution of the Alps in a similar

format for comparison, and will be included in the revised paper.

The first point to make is that comparing the model data sets to the Alpine data in the detail that van der Beek et al. do in lines 172 to 180 is not appropriate. We could refer to the extensive discussion in our paper regarding the purpose of these models (Section 4.0-4.1), but it is easier to quote directly from van der Beek et al.'s comment (line 338): we "used it [the synthetic test] simply to test for the occurrence of a spatial correlation bias across a densely sampled and strong gradient in thermochronologic ages, not to attempt a realistic simulation of the European Alps". We understood this, elaborated extensively on the purpose of the tests in our paper and agree completely with this statement. But recognizing this purpose, counting numbers of co-located points, looking at elevations and numbers of ages from specific chronometers serves no purpose. If we are not attempting a realistic simulation of the Alps, there is no reason to match details of the data characteristics, particularly while ignoring the actual values of the ages. The model includes a vertical dip-slip fault with 36 km of exhumation over 36 Million years, a feature not present in the Alps, so a model which matches the sample locations, elevations and thermochronometer type, would, and does, fail to match the ages. We also should point out there was no "design" element to the creation of these data sets, in the sense that we did not test them, modify values or in any way iterate until we found a desired result. It makes no sense that we would do so, given that we also included data sets, B, C, and E, which all have poor distributions and resolution, so were clearly not designed with some hidden intent. The purpose of these models is to contrast high-resolution and low-resolution examples, and thereby isolate resolution errors. We made no pretenses about this fact. Data sets A and D are intended to be high resolution tests and are explicitly constructed to do this.

Data sets A and D are intended to show high resolution data; the Alps have this feature in common with these synthetic data, but it is the commonality in ages, and thus resolution, that is important, not specifics of the metadata. As much of our paper argues, resolution is defined by the values of the ages and their distance from the closure
isotherm. Because we are interested in the 6 to 0 Ma time frame, this requires dense age coverage of this time interval, and if we also want to avoid geotherm advection errors in the fully coupled model, and maintain its independence from the prior, we require some age coverage of the preceding time interval. The External Alps data have very good coverage over this time window, even with only 3 thermochronometers, because AFT ages are spread across a range of 2 to 14 Ma and ZFT ages are typically between 8 and 20 Ma. However, the synthetic model with its high geothermal gradient and high erosion rate, has no ZFT ages over 8 Ma and no AFT ages over 6 Ma, even with the greater elevation range used for our synthetic data. To obtain a high-resolution model test with this uplift function, it is necessary to push back the age range, either by changing the closure temperature of the existing chronometers, increasing the elevation range, or by adding a higher temperature system. We chose to do the latter two. The fact that there are no muscovite 40Ar/39Ar data in the actual Alps is not relevant; as pointed out by van der Beek et al. themselves (see quote above); we are not trying to model the Alps, we are attempting to model a "densely sampled" example. We cannot have a dense sampling if we do not have ages in the appropriate time interval. As we discuss in section 4.2, as a high-resolution test, it would be completely appropriate to add many more data and many more systems, even approaching an infinite number of data. The point of this model is to eliminate the resolution errors, leaving only the spatial correlation error, and following standard practice in error analysis, we are justified in adding as many data as necessary to accomplish this. In fact, we have been restrained, and limited the number of data generated to something similar to the natural Alps (new Figure 5).

The analysis shown by van der Beek et al. in their Figure 3 has removed the point of the resolution test. van der Beek et al. make two analyses of our high-resolution model of Figure 14. First, they look at the earlier timesteps of the model. Because these timesteps are not the target of the experiment, we did not generate ages over these intervals, they are poorly resolved and there is a correspondingly larger resolution error. Second, they remove all the age data from the 8 Ma to 14 Ma time window, finding that

there are subsequently errors in this timeframe and some of the younger timesteps. There is no revelation in this result. By removing all the key ages, they have converted a high data-density model into a low data-density model and thereby reintroduced the resolution errors that this model was intended to remove. This has nothing to do with spatial correlation bias or any other error. The model response to increasing resolution error is complex because of the coupling of resolution errors to the model errors associated with the geotherm. As we explain in our paper (section 4.5), with the thermal coupling, resolution errors are compounded, which is why it is important to have age coverage prior to the timeframe of interest, so as to minimize advection errors. Also, as discussed at several places in our paper, the temporal resolution metrics do not reflect these geotherm errors. We also note that these geotherm errors are much higher than any natural settings because of the extreme advection due to 36 Ma of erosion at 1 mm/yr. This synthetic test is very challenging to our model as it would be to any other. We could go back and redesign a different synthetic uplift model with lower erosion rates, closer to what is actually observed in the Alps, so it would not be necessary to introduce 40Ar/39Ar data to obtain high resolution, but this could be perceived as designing the test in our favour, so we have retained the uplift function of Schildgen et al. (2018).

As a final point, we strongly object to the casual dismissal of models using the true solution as a prior model (line 158). van der Beek et al. state that because the true solution is not known, it cannot be used as the prior in practice, and so these models are not relevant. However, much like the discussion above, that does not acknowledge their own point, that this is "not to attempt a realistic simulation of the European Alps", or to demonstrate how the model performs under typical real-world conditions. We are attempting to learn something about the method, not trying to simulate the way in which the model is applied in normal circumstances. We remind the reader that the model includes a vertical, dip-slip fault with 36 km of exhumation, a feature that does not exist on Earth, so there is nothing typical about this exercise. These experiment results (our Figs. 8 and 13) are actually quite remarkable. We have taken a model

result with large errors (take any figure with errors in our paper – they all respond the same way), changed one number in the inversion parameters (the prior erosion rate in the NW corner of the model), and the errors are gone – not reduced, but effectively gone. This requires an explanation, not casual dismissal. Suppose for a moment errors are a mix of resolution errors and spatial correlation bias, which is a model error. There is no theoretical reason for model errors to be related to the prior model in a Bayesian inversion. In fact, it goes against the very principle of model errors, that they are built into the parameterization and thus unavoidable through the addition of data or changing of inversion details, including parameter priors. In explaining spatial correlation bias, neither Schildgen et al. (2018) nor Willenbring and Jerolmack (2016) made any statement that the bias applies only to Bayesian models with an incorrect prior. Nor should they; this would make no sense. An averaging error should apply to all models that include that averaging. On the other hand, theory (Eqn. 9 in Willett et la., 2020) shows that resolution errors in a Bayesian model are eliminated if the prior and true solution are equal. In this experiment, model (spatial averaging) errors should be retained, but resolution errors eliminated. The outcome is: all errors are eliminated. The conclusion reached from this is that all errors in the GLIDE inversion are resolution errors. The fact that this is also the conclusion of the analysis comparing high and low resolution models provides additional confirmation - theory, and two control experiments align with a common outcome: there is no spatial correlation bias as defined by Schildgen et al. (2018). van der Beek et al. provide no explanation for this experiment. We anticipate a response that this is a rhetorical difference and that spatial correlation bias is a type of resolution error. However, Schildgen et al. (2018) never state this and all their treatment and analysis imply that they regard it as a model error. If they now suggest that spatial correlation bias is a resolution error, they must accept the implications. In particular, resolution errors go to zero with high data density. Many examples, e.g. Alps, are argued by Schildgen et al. (2018) to have both high data density (van der Beek comment, line 338) and to be dominated by spatial correlation bias (Schildgen et al., 2018). These cannot both be true if spatial correlation bias is a resolution error.

The main point of this discussion is that we see no evidence of anything wrong in our synthetic tests. Nor anything incorrect in the methodology of generating the data. We see no need to modify our paper, nor even how this would be done without removing the essential point of the error analysis.

References Cited:

McKenzie, D., 1978, Some remarks on the development of sedimentary basins: Earth And Planetary Science Letters, v. 40, no. 1, p. 25–32, doi: 10.1016/0012-821X(78)90071-7. Beaumont, C., Jamieson, R., Nguyen, M., and Medvedev, S., 2004, Crustal channel flows: 1. Numerical models with applications to the tectonics of the Himalayan-Tibetan orogen: Journal Of Geophysical Research-Solid Earth, v. 109, no. B6, p. B06406, doi: 10.1029/2003JB002809. Furlong, K.P., and Chapman, D.S., 2013, Heat Flow, Heat Generation, and the Thermal State of the Lithosphere: Annual Review of Earth and Planetary Sciences, v. 41, no. 1, p. 385–410, doi: 10.1146/annurev.earth.031208.100051. Pope, D., and Willett, S.D., 1998, Thermal-mechanical model for crustal thickening in the central Andes driven by ablative sub-duction: Geology, v. 26, no. 6, p. 511–514. Sclater, J.G., and Francheteau, J., 1970, The Implications of Terrestrial Heat Flow Observations on Current Tectonic and Geo-chemical Models of the Crust and Upper Mantle of the Earth: Geophysical Journal of The Royal Astronomical Society, v. 20, no. 5, p. 509–542, doi: 10.1111/j.1365-246X.1970.tb06089.x. Williams, D.L., and Herzen Geology, Von, R.P. Heat loss from the Earth: new estimate, doi: 10.1130/0091-7613(1974)2<327:HLFTEN>2.0.CO;2.

Figure 5: Synthetic age data sets for model bias and resolution tests, comprising two zones with differing erosion rate. Colors represent ages from different ther-mochronometer systems corresponding in closure temperature to: green: AHe, blue: AFT, yellow: ZHe, red: ZFT and black: muscovite 40Ar/39Ar. Data Sets A, B, C were generated using a constant geothermal gradient. Data sets D, E were generated using the GLIDE transient thermal model with a flux boundary condition. For reference, (f) shows the measured distribution for the full Alps and for the external Alps.

**ESurfD**
[Figure]

**Fig. 1.**

---

## Short Comment (SC2) · 8 Oct 2020

**Further discussion of Bias and Error in modelling thermochronometric data by Willett et al.**

Peter van der Beek[1], Taylor Schildgen[1,2], Rasmus Thiede[3], Hugh Sinclair[4]

[1]Institute for Geoscience, Potsdam University, Potsdam, 14476, Germany
[2]GFZ German Research Centre for Geosciences, Telegrafenberg, Potsdam, 14473, Germany
[3]Institute for Geoscience, Christian Albrecht University Kiel, Kiel, 24118, Germany
[4]School of Geosciences, The University of Edinburgh, Edinburgh, EH8 9XP, UK

Willett et al. focus their response on two aspects of our initial comment: (1) the role of the different geotherms in the forward and inverse models and (2) the resolution and data density of the synthetic models. We will provide some additional clarification on these aspects below. Before we do so, however, we feel it is important to underline once more that Schildgen et al. (2018) and Willett et al. pursue fundamentally different goals. Schildgen et al. (2018) posed a *geological* question: whether the "worldwide acceleration of mountain erosion under a cooling climate" since 6 Ma inferred by Herman et al. (2013) from thermochronologic datasets was robust. In contrast, Willett et al. aim to identify different sources of error in the GLIDE inversions of thermochronologic ages for erosion rates, thereby posing a *methodological*, model-framed question.

As stated in our initial comment, a detailed analysis of the potential sources of error within the GLIDE inversion procedure used by Herman et al. (2013) was outside the scope of the Schildgen et al. (2018) analysis. Schildgen et al. (2018) did not set out to test whether the model *could* make robust predictions of erosion history, but whether it *had done so* in the Herman et al. (2013) analysis. While Willett et al.'s new analysis is a useful exercise, it does not address the question of whether the model application to real-world data leads to accurate predictions of erosion histories. As Willett et al. state in their response: "*we are attempting to learn something about the method, not trying to simulate the way in which the model is applied in normal circumstances*". Therefore, it is unclear (and Willett et al. do not clarify) how their new analyses validate the Herman et al. (2013) results. In fact, validating the application of such models to natural systems is inherently impossible (Oreskes et al., 1994): even if Willett et al. would have shown that the inversion is capable of retrieving accurate exhumation histories in the conditions they provided (and the reanalysis of some of Willett et al.'s synthetic tests in our initial comment shows that it does not), these results do not address the analysis performed by Herman et al. (2013).

**1 Variable geotherm calculation**

While the discussion on the choice of thermal boundary conditions in numerical models in Willett et al.'s response is interesting and itself worthy of debate, the issue at stake in this exchange is whether the difference in geotherms as calculated in Pecube and GLIDE causes the spurious accelerations in the synthetic models of Schildgen et al. (2018) and our initial comment. Before we go into this, one point needs to be corrected: in their response, Willett et al. appear to imply that Pecube systematically uses a "steady-state" geotherm to calculate thermochronologic ages, whereas GLIDE would use a "transient" geotherm. However, all of the synthetic models that are being discussed here, whether they were designed by Schildgen et al. (2018) or Willett et al.,

have temporally constant exhumation rates and *should* therefore *tend towards* steady-state geotherms, in both Pecube and GLIDE. The difference between the models lies in the steady-state shape of the geotherm and the time taken to reach steady state (see Willett et al.'s Fig. 3).

Willett et al. dismiss the synthetic models in Schildgen et al. (2018) and in our initial comment because of the difference in boundary conditions when calculating the geotherm, writing: "*all error in the van der Beek et al. comment and Schildgen et al. (2018) models is due to these geotherm errors*". But Willett et al. provide no support for the assertion that differences in boundary conditions produce spurious accelerations, neither in their manuscript nor in the response to our comment. Nor do they address why spurious accelerations are present in inversions using the high-density synthetic data generated by Willett et al., where there are no differences in geotherm calculation between the forward and the inverse models. We reiterate the observations on which we based our rebuttal of Willett et al.'s dismissal:

- Figure 1 in our initial comment shows that when the inferred cause of accelerations in our synthetic models (what we term the spatial correlation bias) is removed, the models do not predict accelerating erosion rates in the last few Myr, despite the difference in geotherm calculation between the forward and inverse models;
- Figure 3 in our initial comment shows that Willett et al.'s models with a transient geotherm (but ages calculated with the same thermal boundary conditions in the forward and inverse model) show spurious accelerations in regions of spatially variable exhumation rates, either at earlier times than shown in Willett et al. or in the last few Myr when the high-temperature ages are removed;
- Figure 4 in our initial comment shows that Willett et al.'s models with a *fixed* geotherm similarly produce spurious accelerations in regions of spatially variable exhumation rates.

Our conclusion from these tests is thus the opposite of what is suggested by Willett et al.: the geotherm errors cannot explain the spurious accelerations observed in these synthetic tests. We provided some tentative explanations for why this may be the case, but we did not argue that one or the other model provides a "*more accurate*" solution to the geotherm problem; we acknowledge that the thermal boundary conditions in both Pecube and GLIDE are imperfect.

**2 Resolution and Data Density**

In their response, Willett et al. frequently invoke "high" versus "low" resolution and data density, but without quantifying these classifications. The first argument in Willett et al.'s response is that the "*commonality of ages*" between the Alpine dataset and their "high-resolution" synthetic datasets A and D makes the results from these synthetic tests applicable to the Alps. We have already refuted this argument in our original comment (lines 210-222).

Second, Willett et al. argue that looking at earlier time-steps of the model is unwarranted, writing: "*Because these timesteps are not the target of the experiment, we did not generate ages over these intervals, they are poorly resolved and there is a correspondingly larger resolution error*". This statement is incorrect: Dataset D includes ~30 mica $^{40}Ar/^{39}Ar$ ages within the 8-10 Ma time-window (Fig. 5 of the Willett et al. manuscript, or Fig. 1 of the response). Moreover, we could have taken the 10-12 Ma time-window as our starting point, which contains >50 mica $^{40}Ar/^{39}Ar$ ages in both datasets A and D. The predicted erosion rates are lower in the 10-12

Ma time-window than in the 8-10 Ma time-window (leading to larger accelerations when compared to later time-windows) but the resolution is similar (Figure 1).

Third, Willett et al. argue that by removing the hypothetical mica $^{40}$Ar/$^{39}$Ar data from the datasets A and D, we have "*converted a high-density model into a low-density model and thereby reintroduced resolution errors*". We note, however, that (1) since all the synthetic mica $^{40}$Ar/$^{39}$Ar ages are >8 Ma, the synthetic tests in which they are removed contain exactly the same number of data within the critical 6-0 Ma time-windows as the original tests; (2) as already noted in our initial comment, the resolution values for the recent time-windows are nearly identical between the two tests (compare panels a-c and g-i in Figs. 3 and 4 of our initial comment). Given the similarity of data and resolution values within the critical time-windows, we do not understand on what grounds Willett et al. characterize one model as "high-density" and the other as "low-density".

As a final point, we provide some clarification of why Willett et al.'s models that use the true solution as the prior (Figs. 8 and 13 of the Willett et al. manuscript) are irrelevant. In Bayesian theory, if the posterior probability distribution of the parameter values (i.e., in this case, the predicted spatial-temporal distribution of erosion rates) is equal to the prior probability distribution (i.e., the input prior erosion rates), then the available data have not contributed to constraining the solution. In this case, the prior erosion rates are equal to the true erosion rates, which have been used to predict the synthetic data. Therefore, for each point in space and time, the model's "initial guess" (the prior), is correct and perfectly reproduces the data. The data cannot influence this problem, and the only thing we learn from this test is that there are no obvious errors in the code. This is the explanation for why the errors disappear. From a practical viewpoint, this test is irrelevant because we cannot know *a-priori* what the exhumation history of a particular region is (i.e., there is no way to know the prior, it is necessarily a guess) and if we did, there would be no point collecting data in such a region because we wouldn't learn anything new from that data.

**References**

Herman, F., Seward, D., Valla, P. G., Carter, A., Kohn, B., Willett, S. D. and Ehlers, T. A.: Worldwide acceleration of mountain erosion under a cooling climate, Nature, 504, 423–426, doi: 10.1038/nature12877, 2013.

Oreskes, N., Shrader-Frechette, K. and Belitz, K.: Verification, validation, and confirmation of numerical models in the earth sciences, Science, 263, 641–646, doi: 10.1126/science.263.5147.641, 1994.

Schildgen, T. F., van der Beek, P. A., Sinclair, H. D. and Thiede, R. C.: Spatial correlation bias in late-Cenozoic erosion histories derived from thermochronology, Nature, 559, 89–93, doi: 10.1038/s41586-018-0260-6, 2018.

Willett, S.D., Herman, F., Fox, M., Stalder, N., Ehlers, T.A., Jiao, R., Yang, R.: Bias and error in modelling thermochronometric data: resolving a potential increase in Plio-Pleistocene erosion rate. Earth Surf. Dyn. Discussions, doi: 10.5194/esurf-2020-59.

[Figure]

**Figure 1. Modified version of Figure 3 in our initial comment, showing synthetic results for earlier time-windows. Input exhumation rates are 1 mm/yr NW of the fault (thick black line) and 0.25 mm/yr SE of the fault. Dataset D of Willett et al. (shown in a; symbols and colours as in Fig. 3 of our initial comment) and a prior erosion rate ($e_p$) of 0.35 mm/yr were used. (a-c) Predicted temporal evolution in erosion rates using all the data (equivalent to Willett et al.'s Figure 14): (a) predicted erosion rates for the 6-4 Ma time bin; (b) predicted erosion rates for the 2-0 Ma time bin; (c) normalized difference in erosion rates, only shown where the resolution in each time bin is > 0.25. Contours in plots (a-c) show the predicted resolution. (d-i) Predicted temporal evolution during earlier time bins for this inversion: (d) predicted erosion rates for the 10-8 Ma time bin; (e) predicted erosion rates for the 6-4 Ma time bin (same as a); (f) normalized difference in erosion rates, only shown where the resolution in each time bin is > 0.25; (g) (d) predicted erosion rates for the 12-10 Ma time bin; (e) predicted erosion rates for the 6-4 Ma time bin (same as a); (f) normalized difference in erosion rates, only shown where the resolution in each time bin is > 0.25. Note the significant increase in erosion rates that precedes the decrease shown by Willett et al.**

---

## Author Comment (AC3) · 15 Oct 2020

Response to Anonymous Referee #2

REVIEWER: 1. Overview: what is the conclusion of Herman et al. (2013)? Is their conclusion that some mountainous, glaciated regions were exhuming faster between 0-2 Ma compared with 4-6 Ma? Is their conclusion that most glaciated mountains were exhuming faster between these times? Most mountains? Or that there was a net increase in erosion rates globally?

RESPONSE:

[Figure]

We find this a rather odd question to ask. It is with reference to a paper published 7 years ago, not the one currently under review, and it is a question that we believe was answered implicitly and explicitly many times in both papers. Herman et al. (2013) and Willett et al. (2020) make no estimate of the mean global erosion rate or its change in time. There is no claim to this effect in either manuscript. There would not be such a focus on resolution and its estimation, if we were not trying to determine which regions have the resolution to make an estimation of changes in erosion rate. The maps in Herman et al. (2013) and Willett et al. (2020) (Fig. 25) show precisely where the analysis establishes sufficient resolution and therefore where it applies. It does not apply elsewhere and the conclusions are limited to the specific sites on those figures. These are all sites where erosion rates are high enough that thermochronometric ages are young and resolution is established (Willett et al. (2020), Figure 1). If it is grey on the map, there is no estimate made. We do not argue that the well-resolved regions are representative of the Earth as a whole. We can add this statement to the current paper if desired, but it is not clear as to why we should have to do this since we never claimed to provide a global estimate. In terms of a testable null hypothesis: regions that have erosion rates high enough that thermochronometric data can provide cooling rate constraints over the last 6 Ma have had no change in erosion rate over this time.

REVIEWER:

2 Schildgen et al., 2018 critique

RSEPONSE:

This is a fair assessment of our paper.

REVIEWER:

3 Willenbring and Jerolmack, 2016 critique

WJ16 argued that the methodology of H+13 is biased because only sites with rapid exhumation will precisely resolve a recent change in exhumation. Even if GLIDE perfectly

**ESurfD**
identifies regions where thermochronology ages resolve a recent increase in erosion rate (a point contested by S+18), there will be many regions which have experienced constant or lower erosion rates that are cannot be resolved and are therefore under-represented.

RESPONSE:

This is true, but only relevant if we are trying to estimate the average global erosion rate. As stated above, we are not trying to obtain a global average.

REVIEWER:

"The essence of the argument is that slower rates of exhumation are progressively clipped from the measured age distribution of rocks as one approaches the modern because of the precision of the technique. The reason is that thermochronometers measure the time since achieving an associated closure depth; rate is simply the measured time divided by this depth. If erosion rates have not been high enough to bring rocks to the surface, the rates associated with those buried rocks cannot be measured."

And here is the response to this critique in W+PS:

"...Willenbring and Jerolmack (2016) misinterpreted the meaning of the limit to resolution, or what they called "precision". They took this concept from measurements of sediment layer thickness [the 'idea of Anders et al., (1987)' referenced in WJ16 argument above], which can be biased by the inability to measure layer thicknesses below a specific precision (Anders et al., 1987). "Thermochronometric data are fundamentally different in that an age does not represent an estimate of rate at a point or across a closed interval of time such as represented by a stratigraphic layer, but rather, represents the integral of the exhumation rate...from the time of closure to the present day. Thus, rather than being unsampled, the region [from the youngest closure age until present] is actually the most heavily sampled part of parameter space. The better definition of the "limit to resolution" is that no change of exhumation rate can be resolved
between this limit and time zero, because there is no sampling internal to this part of the parameter space, but the average rate across this interval is sampled and can be determined."

This response misconstrues the legitimate criticism raised by WJ16.

RESPONSE:

We don't agree that we are misconstruing WJ16. We understand what they said and our paper is directly responding to the passage quoted by the reviewer. The problem is that what WJ16 say in the first paragraph: "If erosion rates have not been high enough to bring rocks to the surface, the rates associated with those buried rocks cannot be measured." is, in our view, not correct. The premise and analysis of WJ16 is based on this misunderstanding of the nature of thermochronometric ages. If the premise is wrong, there is no point in engaging with the predictions. This is what the above paragraph from our paper is addressing. An age integrates the full exhumation path and therefore it is NOT necessary to bring up a rock from the closure depth to the surface in order to measure a change. This is more or less the same point brought up by the reviewer in section 5 on buried rock bias, so we will address this further in response to that point.

REVIEWER:

4. Sensitivity analysis

What remains to be demonstrated is whether the methodology of H+13 / W+PS could resolve constant or decreasing erosion rates within a synthetic test. The analyses in Section 4 of W+PS indicate that constant but spatially distinct long-term erosion rates can be resolved, but similar tests are not provided for a recent change in exhumation rate.

RESPONSE:

We agree that we have not investigated in detail the case where there is a true change

in erosion rate. The paper is already long and we feel it is more important to establish the principles of resolution and bias to provide context, and to address the problem of the "false positives", that is to show that the increases that H+13 find are not artifacts of the methodology. If necessary, future work could show this, but there is not a strong argument that it is harder to resolve an increase than to resolve a constant rate, provided the resolution criteria are selected to resolve either.

REVIEWER:

5 The problem of still buried rocks ll. 185-189. "[I]f one regresses age against depth (Figure 2d) with a moving depth window, one obtains the correct, unbiased regional mean erosion rate (Figure 2e, Average 1)." This statement and the corresponding figure, W+PS Fig. 2d (reproduced here in Fig. 1), neglect the fact that a significant portion of rocks which passed through the closure depth after a recent change in exhumation rate will not yet have reached Earth's surface. W+PS Fig. 2d illustrates only a special case, where erosion is spatially varied but constant through time. This is unfortunate and misleading, because the question at hand is whether transient erosional signals (i.e., the onset of Northern Hemisphere glaciations) can be measured (in the midst of spatial variability).

If an instantaneous change in exhumation rate occurred recently, it will only be observable if rocks: a) began beneath the closure depth and b) were exhumed from that depth to the surface to be collected. This exhumation distance imposes a limit on which recent exhumation rates can be measured (Fig. 1).

The following thought experiment crudely estimates the magnitude of this 'still buried rock' bias effect (Fig. 2). H+13 evaluated the difference in global erosion rates between the time periods of 4 – 6 Ma and 0 – 2 Ma. So, it is useful to consider what would happen if global erosion rates changed significantly and instantaneously at 2 Ma. Apatite (U – Th)/He is the lowest temperature system considered in H+13 and would close at a depth of approximately 2 km, depending on the local exhumation rate and geothermal

gradient. At 2 Ma, all rocks below this closure depth would begin their ascent according to the new surficial erosion rate. Therefore, a geologist collecting surficial samples today could only possibly collect samples which rose at a rate of (2 km) / (2 Ma) or greater. All other rocks would still be buried. This effect is illustrated in Fig. 2 for a range of times at which erosion rates could have instantaneously changed.

RESPONSE:

We are happy to discuss the thought experiment offered by the reviewer. However, we think this framing is not correctly describing the situation. Yes, rocks that closed at 2 Ma, but experienced a slow erosion rate, are still buried. But there is still a rock at the surface. That rock still has an age. An apatite (U-Th)/He age of this rock has only two possibilities; it is less than 2 Ma or greater than 2 Ma. If it is less than 2 Ma, it accurately measures the current erosion rate; higher closure-temperature ages would then correctly resolve the change in rate, provided there is at least one age over 2 Ma. If the surface apatite (U-Th)/He age is older than 2 Ma, it closed prior to the acceleration, but its age reflects the integrated rate of exhumation since this time, including the 2 Ma at a higher rate (Fig.1a, red curve). Relative to a rock that did not experience an increase in erosion rate, but continued to exhume at the original rate, all ages at the accelerating site would be younger by a constant amount, and this is true for any erosion rate, not just high rates. The same is true for a rock that experienced a decrease in erosion rate (Fig.1a, blue curve); the age integrates the travel time from the closure depth, and so reflects any change in exhumation rate post-closure. Reviewer 2 argues that no changes in rate can be detected for rates less than the black dashed line (unexposed rock limit) in Fig.1, but this is clearly not the case. Rates inferred from these ages using the two mineral method are shown in Figure 1b. The ages do not perfectly resolve the chang, but both increases and decreases are detectable. The fact that part of the zone from the closure depth to the surface is still buried is not the limitation. Stating that a rock from the closure depth must come to the surface since a change in rate occurred, as is shown in Reviewer 2's Fig. 1b in order to detect it

is overly restrictive. The example in our Figure 1 shows that a 4 Ma apatite He age would still detect an increase or a decrease in erosion rate, but with half (ignoring the non-linear relationship between age and erosion rate) the magnitude and spread over time. The earlier rate is still properly resolved.

What an older age cannot do is resolve precisely the timing and magnitude of the change in erosion rate. It is detected, but poorly resolved and thus averaged with the earlier rate. The older the age, the worse the resolution, as discussed in our paper. If the ages are too old, resolution will be so poor that data with measurement error will not resolve it and the H+13 analysis would reject the site; but this limit is much lower than the 1 mm/yr estimated by the reviewer. Given old ages, most analyses will underestimate the magnitude and overestimate the onset time of an increase in erosion rate. Note that this bias is towards a constant rate of erosion, not a false increase. This was the basis for our statement on lines 196-198. This critique is also true for the reviewer's Figure 2. We don't see that we have misrepresented the problem or what W+P stated; nor have we failed to engage with the criticism. We are simply stating in the current manuscript that they premised the problem incorrectly, and we stand by our original discussion based on the logic presented above.

REVIEWER:

From this exercise, two observations stand out. First, only terrains experiencing very rapid exhumation will inform the thermochronology record of erosion rate changes at 2 Ma. Second, it is much more likely to observe accelerations in exhumation than to observe constant or decelerating exhumation. To avoid confusion, note the following distinction. The approach of H+13 / W+PS will register low erosion rates during the period of 2 Ma to present owing to old AHe and AFT ages (how 'resolved' low rates are depends on the GLIDE algorithm). What the H+13 / W+PS approach will not register, however, are any decelerations down to rates below about 1km/Ma.We must take as a null hypothesis the idea that global net erosion rate change was zero from 6 Ma until present. For the reasons outlined above, it is my view that the H+13 / W+PS

methodology is inherently incapable of resolving unchanging net erosion rates globally. The authors have not taken seriously this critique, initially outlined in WJ16, and have not demonstrated their ability to resolve a constant erosion rate or a decreasing erosion rate.

RESPONSE:

Again, we have to disagree with most of this conclusion. It is mostly directed at the issue addressed in the first paragraph – as if we are trying to estimate a global erosion rate. As we are not estimating global erosion rates, most of this and the 'null hypothesis' as stated above are not relevant. We have effectively pre-selected the sites we studied as having high erosion rates (not 1 mm/yr – that is much too high), and we are testing the null hypothesis that these sites individually and collectively have no change in erosion rate.

There is a hint of a true bias implicit to this final criticism, even though the quantification in Figs 1 and 2 is not correct. Sites with high erosion rates will be more likely to have experienced a recent acceleration than deceleration; sites with low modern rates are more likely to have experienced a deceleration in the recent past, and because low rate sites are removed, and because erosion rates cannot be negative, this does create a bias in the analysis suite of Herman et al. (2018). This bias does not depend on the rate or the burial argument of Reviewers Figures 1 and 2, but rather on the frequency and magnitude of erosion rate changes in orogenic belts. For a given modern erosion rate, there is some probability as to what the erosion rate would be at any point in the past, i.e. 2 Ma, and this probability will be skewed because of non-negativity and truncation of low rates. We believe this skewness will be small because tectonic-driven erosion rate changes are heavily damped by geomorphic processes and isostasy (Whipple and Meade, 2004) and rapid climate change is mostly cyclic, but we acknowledge there are few data and no systematic treatment of this problem. This is a bias on the median change, i.e. the mean or median of the distribution of Figure 25 in Willett et al. (2020), and if this distribution were closer to zero-median, we would have investigated it in

Herman et al. (2013). However, this is not a bias on the result at any individual site, so is not the main topic of the current manuscript or part of the critique by Schildgen et al. (2018).

In summary, we think this review is mostly arguing a point that is not really the subject of this paper: whether the high erosion rate mountain belts of the world are representative of the Earth as a whole. We have not made any statement that this is true and are happy to clarify that once more in the introduction of this paper. If one interprets the hypothesis of Willenbring and Jerolmack (2016) as being applicable on a local site bias, we contend it is still not correct because it does not recognize the integral nature of a cooling age since closure, and the preselection implicitly done by Herman et al. (2013). We do accept that there could be a bias in the median value of the global analysis of Herman et al. (2013) due to the loss of some sites where deceleration has led to the sites dropping below the resolution cutoff, which would bias the median of the distribution in Fig. 24, but this bias does not affect the result at any individual site. We can add a paragraph to section 6.4 acknowledging this effect.

Ref

Whipple, K., and Meade, B., 2006, Orogen response to changes in climatic and tectonic forcing: Earth and Planetary Science Letters, v. 243, no. 1-2, p. 218–228, doi: 10.1016/j.epsl.2005.12.022.

[Figure]

**Fig. 1.** Resolution of an increase or decrease in erosion rate at 2 Ma, where all ages close prior to the change. (a) Depth time path and ages. (b) Inferred erosion rates from the mineral pair method.

---

## Referee Comment (RC4) · Anonymous Referee #3 · 22 Oct 2020

The contribution by Willett et al. is perplexing and disappointing. The author list is a who's who of accomplished thermal modelers, but the result is long yet incomplete, pedantic yet sloppy, and preoccupied. The paper is essentially a protracted comment on Schildgen et al. (2018), whom they want to counter on seemingly every point, down to the colors of circles on a schematic diagram. It's plainly overdone. At 43 pages of text without contributing much that is new, it comes across as an attempt to overwhelm the reader into submission. I am hoping the authors recognized, or even intended, the irony of calling van der Beek et al.'s initial 15-text-page response "long." In terms of

[Figure]

style, Willett et al. has much that is unacceptable. In its frequent digressions, it guesses at thought processes and motives by Schildgen et al. (2018) and Willenbring and Jerolmack (2016) instead of just sticking to the facts. It puts words into their mouths, refuting hypotheses they did not pose. It is hard to avoid the conclusion that the entire affair has become partly personal, and that this is tainting the thinking and judgement of the authors.

Figure 2, and the discussion surrounding it, are not well done, and actually seem to harm the case the authors are trying to make rather than setting the stage for it. To begin with, the figure itself is confusing, as depth and age axes are flipped without annotation (e.g. age rises to the right in some cases, to the left in others). Point i went from being the oldest one in 2d and 2e to about the same age as point d in 2f – it appears they also crossed themselves up by flipping an age axis. The text often refers to the wrong part of the figure (I think they mean 2d in line 161, not 2e; 2e in Line 171, not 2b; and the first 2d in line 182 should be 2e). In the text, the reader needs to essentially open Schildgen et al. (2018) and Willenbring and Jerolmack (2016) to follow the authors' attempts to translate those papers' approaches to the figure, and some of those translations appear selective in a pernicious way (e.g., lines 178-180). The overall example itself is so oversimplified as to also be confusing, and possibly self-defeating. The authors seem to assert that their Average 1 is best, or "unbiased", but it's hard to justify extrapolating uplift rates backward in time to before the oldest ages in the faster-uplift regions (points c and f). One could just as easily, and certainly more conservatively, say that there is no evidence for earlier erosion in those regions, making Average 3 preferable – or, better yet, stopping the attempt to calculate a regional average uplift rate at age c and not going further back in time. Average 1 actually imparts an assumption (bias?) of spatial correlation, saying that the subregions defined by abc and def were also exhuming at time i, despite the data having no information from those subregions indicating that this was so, only proximity to the ghi region. One would need to look at the detailed geology to defend such a claim. . .

It is also confusing to call the same model unbiased (line 171) and biased (line 395). The difference in boundary conditions used between models for synthetic tests is certainly unfortunate, but the authors did not demonstrate that it had a large effect on the comparisons in Schildgen et al. (2018); the counter-example provided by van der Beek et al. believably indicates that it did not affect their conclusions. The point of including a geotherm assuming 36 Ma of 1 mm/yr erosion in Figure 3 is not evident; are they claiming that Schildgen et al. (2018) went that far? The subsequent examples from GLIDE are also deceptive, though I imagine unintentionally so. As van der Beek et al. point out, the examples demonstrating the ability of GLIDE to correctly reproduce erosion rates across a sharp interface used far more advantageous data than Herman et al. (2013) used to derive their conclusions for the Alps. When the data better match what Herman et al. (2013) actually used (Fig. 10-12), the model produced spurious accelerations, at the resolution levels used by Herman et al. (2013), and without boundary condition mismatch. There is also an odd tendency to proclaim victory and move on, when the data don't appear to match the words. The authors try to construct a chain of QEDs but instead leave a trail of question marks. As one example, in the Fig. 7 test, the estimated erosion rate in the SE is about twice the true one (to the best of my ability to read their color bar), but they call it a good match. This is also the only model where the SE region is resolved according to both "resolution" and reduced variance metrics. Shouldn't this be worrisome?

Similarly, the "perfect prior" tests (Fig 8, 13) do indeed seem trivial. By equation 3, if zc = A e(prior), then e(post)=e(prior). It's not clear if the equality is strictly true – that would depend on whether any noise was added to the synthetic data. Willett et al. don't mention adding noise, so I'm assuming they did not. If that's indeed the case, then in fact none of the tests they present had to withstand routine and inevitable data scatter, making them all suspect for purposes of verifying the robustness of the method. Evaluating the resolution of a statistical method using only noiseless synthetic data would be oddly incomplete.

As a result of this all, I came away still not knowing when GLIDE results are robust, which promised to be a primary contribution of the paper. This became particularly evident at line 1170: "Resolution remains a relative measure, and determining a precise confidence level a priori is not possible, but can be estimated based on spatial patterns, relationship to sample locations, fit to the age data and sensitivity to the prior." In other words, run a big complex computation, and then manually inspect the results to see if you think they actually fit and are justified, nudging thresholds on a case-by-case basis as necessary. This blurs the boundary between quantification and interpretation, and opens the door for arguments about motivated reasoning.

Even as a review and interrogation of GLIDE fidelity, the paper is a bit disappointing. The GLIDE topographic correction assumes that topography does not change through time, but the authors mention that in two of their high-resolution cases (Taiwan, southern New Zealand) all topography developed recently. What is the effect of presuming pre-existing topography that wasn't there? Other potential model errors are discussed in passing (line 401-407) in a somewhat oversimplified way. One omitted assumption is that all thermochronometers of the same name have the same closure temperature, which is incorrect. Could their model be artificially accelerating late cooling by assuming that all apatite loses helium at the rate of Durango apatite (Tc = 70C for dT/dt = 10C/Myr; Farley 2000), as opposed to low-radiation-damage apatite (Tc = 55C; Shuster et al., 2006)? This is unexplored.

A puzzling part of the back and forth is the failure of anyone to do a simple test, which is to model the data NW and SE of the Penninic Line independently. Van der Beek et al. sort of do this in their Figure 1, albeit with synthetic data for the purpose of testing boundary condition changes. If the isolated models show no acceleration, and the combined model does show it, then that's a pretty good indication that the acceleration signal came from combining data across a major structure, presumably a red flag. If one or both of the isolated models do show acceleration, then that's an indication that the signal is at least partially independent of spatial correlation across a suspect interface. Fox et al. (2014) sensibly state, with appropriate caution, that their method is best used "for regional studies where the... exhumation rates are smooth in space and are not strongly affected by surface-breaking faults. This latter complication can be easily accounted for, where these are well-identified, by building them into the correlation structure. In such cases, samples from either side of a fault could follow independent exhumation histories." Even easier than customizing the correlation structure is just running separate models.

The authors are essentially claiming that there is no need to follow their own advice if they use more demanding (yet fungible) resolution limits. Even after 10 figures with synthetic model results, their case is not convincing.

One gets the feeling that the authors simply do not want to say in so many words that Schildgen et al. (2018) are basically correct that most of the data do not support the conclusions by Herman et al. (2013). Insofar as the Herman et al. (2013) claim was based on the overwhelming weight of a global data set showing the same signal everywhere, and given that the authors now admit that the majority of those data in fact do not have the necessary resolving power, one is left to wonder why the argument needs to be so ferocious.

There is potentially useful information in this paper, where the authors reassess resolution and how it applies to their original data sets, although it would be better if the tests used more realistic synthetic data in terms of both noise and comparability to the actual available data. If the authors cut down the paper by »50% by getting rid of the argumentation against Schildgen et al. (2018), concentrate on a more thorough exploration of possible biases and errors in GLIDE modeling of the higher-data-density areas analyzed by Herman et al. (2013), that could help clarify how much the remaining data say. The paper is far from that point, though.

Alternatively, the paper could be published almost as-is (the errors surrounding Figure 2 really should be fixed, tests in Fig 8&13 clarified or redone, etc.) and appear together

with van der Beek et al.'s responses, and everyone can just move on. I don't think the authors would be well served by this, but it would provide some closure.

---

## Editor Comment (EC4) · Richard Gloaguen (Editor) · 22 Oct 2020

This review is a modified version in which some wordings, that were a bit antagonistic, have been amended at the request of the editorial team. This review remains very critical and maybe overly so to some readers. I kindly ask my peers to focus on the content of the review.

---

## Editor Comment (EC5) · Richard Gloaguen (Editor) · 22 Oct 2020

A decision on this submission will be made shortly. I am waiting for a last information and a technical change in a review. Due to the controversy this submission has engendered the decision will be first communicated to the editorial team to ensure agreement.

---

## Editor Comment (EC6) · Richard Gloaguen (Editor) · 23 Oct 2020

I have received several requests concerning the comments and reviews. Maybe the following infos will enlighten you.

1- This manuscript has been submitted to Copernicus within a two-stage protocol. First, the manuscript is posted as a discussion paper (ESurfD) after a technical check and an overall evaluation by the editors. If positively evaluated by peers the manuscript is then accepted as a full paper in ESurf. During the first stage, both registered peers and nominated reviewers can post comments WITHOUT oversight from the editors. Posts and reviews can only been removed if they do not adhere to the Copernicus policies

upon request. It implies that both reviewers and peers should contribute with respect and politeness and focus on the scientific debate.

2- Reviews should be dispassionate, fair and unbiased. The editor should not interfere. In the present case, I, as handling editor can only ask the reviewers to comply. I can not and will not tell them what they should write or how.

3- The decision concerning the submission will be made SOLELY based on a scientific basis. Basis for the decision is primarily the reviews but comments will also be taken in to accounts. This is a decision on the submitted manuscript submitted to ESurfD. Nothing else.
* * *

---

## Author Comment (AC4) · 24 Oct 2020

RESPONSE: This review is rather heavy on opinion and somewhat lighter on criticisms that are backed up with observation, but we will nonetheless attempt to address what points we can. We will ignore the first paragraph as this is opinion that falls outside the scope of peer review and contains no backing support, not even reference to sections in our paper to permit us to respond.

As to the factual criticisms, our general impression is that the reviewer must have been intimidated by the length of the paper, and so not read it very completely, as nearly all the requested additions are already in the paper. At a minimum, justifications and

explanations for specific reviewer-comments are already provided in the text, so we draw attention to these, but see little need for additions or major modifications to the paper.

REVIEWER: Figure 2, and the discussion surrounding it, are not well done, and actually seem to harm the case the authors are trying to make rather than setting the stage for it. To begin with, the figure itself is confusing, as depth and age axes are flipped without annotation (e.g. age rises to the right in some cases, to the left in others). Point i went from being the oldest one in 2d and 2e to about the same age as point d in 2f – it appears they also crossed themselves up by flipping an age axis.

RESPONSE: The letter labels in part (e) on some points are reversed. Points are plotted correctly. We will correct the figure, but there is no effect on the meaning.

REVIEWER: The text often refers to the wrong part of the figure (I think they mean 2d in line 161, not 2e; 2e in Line 171, not 2b; and the first 2d in line 182 should be 2e).

RESPONSE: We will correct the latter two typos in revision; line 161 is correct.

REVIEWER: In the text, the reader needs to essentially open Schildgen et al. (2018) and Willenbring and Jerolmack (2016) to follow the authors' attempts to translate those papers' approaches to the figure, and some of those translations appear selective in a pernicious way (e.g., lines 178-180).

RESPONSE: Line 178 is pointing out that Schildgen et al. plotted data points with the lowest elevation as having the largest distance above the closure isotherm. We don't know how this would occur. Without explanation as to why they did this, we cannot reproduce this figure or address it. That is all we are pointing out. We are not sure how this is regarded as pernicious on our part.

REVIEWER: The overall example itself is so oversimplified as to also be confusing, and possibly selfdefeating. The authors seem to assert that their Average 1 is best, or "unbiased", but it's hard to justify extrapolating uplift rates backward in time to before

the oldest ages in the faster-uplift regions (points c and f). One could just as easily, and certainly more conservatively, say that there is no evidence for earlier erosion in those regions, making Average 3 preferable – or, better yet, stopping the attempt to calculate a regional average uplift rate at age c and not going further back in time. Average 1 actually imparts an assumption (bias?) of spatial correlation, saying that the subregions defined by abc and def were also exhuming at time i, despite the data having no information from those subregions indicating that this was so, only proximity to the ghi region. One would need to look at the detailed geology to defend such a claim. . .

RESPONSE: We are not advocating for any average or arguing that any one is better than another. The purpose of this figure is explained on lines 224-242: it is to demonstrate that there are many ways to calculate averages or to parameterize models. To demonstrate that any specific method of averaging has bias is not to prove that ALL treatments of thermochronometric data are biased. We don't see any way in which this figure undermines this argument. We took the thought experiment directly from Willenbring and Jerolmack and find it a useful illustration. It provides a more intuitive alternative to equations and numerical models.

REVIEWER: It is also confusing to call the same model unbiased (line 171) and biased (line 395).

RESPONSE: Line 395 is not referring to the same bias. An average can be unbiased with respect to a regional average rate and still be biased towards a local rate. We will modify the text at this location to make it clearer.

REVIEWER: The difference in boundary conditions used between models for synthetic tests is certainly unfortunate, but the authors did not demonstrate that it had a large effect on the comparisons in Schildgen et al. (2018); the counter-example provided by van der Beek et al. believably indicates that it did not affect their conclusions.

RESPONSE: So, an error of 100% in ages (our figure 4) is "unfortunate" but "doesn't

have a large effect"? This will be reassuring news to many thermochronometry labs who are currently under the impression that generating data with 100% error would not be acceptable. See our response of Oct 5. There is no acceptable error in a control experiment. We contend that all error in the Schildgen/van der Beek models is due to this error and they have not provided the counterargument by calculating the ages correctly and recalculating an inverse model. Our models with comparable resolution, but no geotherm error have no false acceleration. We consider this an effective demonstration. The argument has (correctly) moved on to "what is comparable resolution?", but the Schildgen models and the initial van der Beek comment models are demonstrably wrong for reasons explained in the original Willett et al. manuscript and our response to van der Beek's comment. As such, these models should not be part of the discussion.

REVIEWER: The point of including a geotherm assuming 36 Ma of 1 mm/yr erosion in Figure 3 is not evident; are they claiming that Schildgen et al. (2018) went that far?

RESPONSE: This is exactly what Schildgen et al did.

REVIEWER: The subsequent examples from GLIDE are also deceptive, though I imagine unintentionally so. As van der Beek et al. point out, the examples demonstrating the ability of GLIDE to correctly reproduce erosion rates across a sharp interface used far more advantageous data than Herman et al. (2013) used to derive their conclusions for the Alps. When the data better match what Herman et al. (2013) actually used (Fig. 10-12), the model produced spurious accelerations, at the resolution levels used by Herman et al. (2013), and without boundary condition mismatch.

RESPONSE: This misrepresents the intent and outcome of the modeling. The modeling is broken out into a series of tests. One of the series of tests is to investigate resolution by running a series of models from high resolution to low resolution. None of the tests is intended to model the Alps or to mimic the precise characteristics of the Alpine data, an impossible task in any case. This point is acknowledged by van der

Beek et al (comment Sept 17 line 338). The resolution models contain a spectrum of data sets that likely span the resolution of the actual Alps data; only the lowest resolution models fail, and these models have less than a third of the number of data that are available in the Alps. Figure 9 shows a case where a model with even fewer data, far fewer than used by Herman et al. (2013), accurately reproduces the input parameters, because the distribution in age is advantageous. The models and current text are explicit in pointing out this intent.

Our paper contains an introductory section (4.1) explaining this point, but the reviewer has not acknowledged or referred to this explanation, but rather is repeating the content of the van der Beek et al comment of Sept 17, including the false statement that the Alpine data look more like our synthetic data set C, rather than A (see our modified figure given in comment of Oct 5). Fig 10-12 have fewer data than the Alps and are distributed systematically younger. This is largely irrelevant in any case as the model erosion rate function is not that of the Alps and resolution is determined in part by the erosion rate function.

REVIEWER: There is also an odd tendency to proclaim victory and move on, when the data don't appear to match the words. The authors try to construct a chain of QEDs but instead leave a trail of question marks. As one example, in the Fig. 7 test, the estimated erosion rate in the SE is about twice the true one (to the best of my ability to read their color bar), but they call it a good match. This is also the only model where the SE region is resolved according to both "resolution" and reduced variance metrics. Shouldn't this be worrisome?

RESPONSE: There is a point to every model. Success is determined by the intent and outcome of a model. Not incidental characteristics. The model of Fig 7 has a 100 km correlation length to make the point that there is little sensitivity to the correlation length. Solution is too smooth, so the SE erosion rate is too high, but there is no false acceleration, which is the subject of the paper. As discussed in our paper (e.g. section 6.1), the resolution scales with correlation length, so if one used a 100 km correlation

length, a different cut-off value should be used. So, no, this is not worrisome.

REVIEWER: Similarly, the "perfect prior" tests (Fig 8, 13) do indeed seem trivial. By equation 3, if $zc = A e(prior)$, then $e(post)=e(prior)$. It's not clear if the equality is strictly true – that would depend on whether any noise was added to the synthetic data. Willett et al. don't mention adding noise, so I'm assuming they did not. If that's indeed the case, then in fact none of the tests they present had to withstand routine and inevitable data scatter, making them all suspect for purposes of verifying the robustness of the method. Evaluating the resolution of a statistical method using only noiseless synthetic data would be oddly incomplete.

RESPONSE: We state explicitly (line 485) that we will not consider noise in the data. We could do so, but the reviewer has already made the point that the paper is long as is. We have, of course, run models with noise in the data – it decreases resolution, but can be compensated for by data density, and plays no important role in questions of model bias. A well-designed error analysis does not mix sources of error; it isolates them.

REVIEWER: As a result of this all, I came away still not knowing when GLIDE results are robust, which promised to be a primary contribution of the paper. This became particularly evident at line 1170: "Resolution remains a relative measure, and determining a precise confidence level a priori is not possible, but can be estimated based on spatial patterns, relationship to sample locations, fit to the age data and sensitivity to the prior." In other words, run a big complex computation, and then manually inspect the results to see if you think they actually fit and are justified, nudging thresholds on a case-by-case basis as necessary. This blurs the boundary between quantification and interpretation, and opens the door for arguments about motivated reasoning.

RESPONSE: This is the reality of all estimation problems. Every field that deals with data analysis has had to deal with similar problems. What surprises us most is that so many in thermochronometry don't seem aware that this is normal in both forward and

inverse modeling. Note that reviewer 1 questioned the need for a long, pedagogical introduction, but subsequent comments and reviews have demonstrated that there is need.

REVIEWER: Even as a review and interrogation of GLIDE fidelity, the paper is a bit disappointing. The GLIDE topographic correction assumes that topography does not change through time, but the authors mention that in two of their high-resolution cases (Taiwan, southern New Zealand) all topography developed recently. What is the effect of presuming pre-existing topography that wasn't there?

RESPONSE: Agreed -topographic evolution is problematic, but this is an unfaced problem in all modeling of thermochron data. All thermo-kinematic models make similar assumptions. The justification is that the lowest T systems are the most sensitive to topography, and are the youngest ages, so topography at time of closure should be close to the modern for the most sensitive ages. We note that there is discussion of this in Fox et al., (2014). We can add reference to this paper at the location the reviewer mentions.

REVIEWER: Other potential model errors are discussed in passing (line 401-407) in a somewhat oversimplified way. One omitted assumption is that all thermochronometers of the same name have the same closure temperature, which is incorrect. Could their model be artificially accelerating late cooling by assuming that all apatite loses helium at the rate of Durango apatite (Tc = 70C for dT/dt = 10C/Myr; Farley 2000), as opposed to low-radiation-damage apatite (Tc = 55C; Shuster et al., 2006)? This is unexplored.

RESPONSE: Agreed – it is unexplored. We acknowledge this point and state it explicitly on line 565. It was discussed in Fox et al. (2014) and Herman et al. (2018). In fact, it is likely to be much more important than any spatial correlation effects. This could be explored at the expense of lengthening the paper, which raises other objections.

REVIEWER: A puzzling part of the back and forth is the failure of anyone to do a simple test, which is to model the data NW and SE of the Penninic Line independently. Van der

Beek et al. sort of do this in their Figure 1, albeit with synthetic data for the purpose of testing boundary condition changes. If the isolated models show no acceleration, and the combined model does show it, then that's a pretty good indication that the acceleration signal came from combining data across a major structure, presumably a red flag. If one or both of the isolated models do show acceleration, then that's an indication that the signal is at least partially independent of spatial correlation across a suspect inter- face. Fox et al. (2014) sensibly state, with appropriate caution, that their method is best used "for regional studies where the: : : exhumation rates are smooth in space and are not strongly affected by surface-breaking faults. This latter compli- cation can be easily accounted for, where these are well-identified, by building them into the correlation structure. In such cases, samples from either side of a fault could follow independent exhumation histories." Even easier than customizing the correlation structure is just running separate models.

RESPONSE: This is an excellent idea. Which is why we did it. This experiment is described on lines 928-933 with results shown in Figure 19. It gives a conclusive result; it is a shame the reviewer seems not to have read this part of the paper.

REVIEWER: The authors are essentially claiming that there is no need to follow their own advice if they use more demanding (yet fungible) resolution limits. Even after 10 figures with synthetic model results, their case is not convincing.

RESPONSE: There is a justification given (section 4.1); we took the challenge of at- tempting to model the most difficult case possible: a vertical fault with 13 km of relative displacement, a feature that does not exist on Earth. If we can model this successfully, and many of the models we showed do model this successfully, the kind of smooth variation in erosion rate that is typical of most places on Earth (including the Alps) would be far easier. More importantly, the point of any modeling exercise is not to show success or failure; it is to understand why a model succeeds or fails and thereby better understand the model and better predict its behaviour elsewhere, under easier conditions.

REVIEWER: One gets the feeling that the authors simply do not want to say in so many words that Schildgen et al. (2018) are basically correct that most of the data do not support the conclusions by Herman et al. (2013). Insofar as the Herman et al. (2013) claim was based on the overwhelming weight of a global data set showing the same signal everywhere, and given that the authors now admit that the majority of those data in fact do not have the necessary resolving power, one is left to wonder why the argument needs to be so ferocious.

RESPONSE: We do not see where we admit that the majority of the data do not have the resolving power. Perhaps the reviewer could point this out and reference line numbers in our manuscript.

REVIEWER: There is potentially useful information in this paper, where the authors reassess resolution and how it applies to their original data sets, although it would be better if the tests used more realistic synthetic data in terms of both noise and comparability to the actual available data.

RESPONSE: We do not see where the reviewer has engaged with (or even read) the arguments as to why noise is omitted (to isolate errors) or how it is impossible to match original data sets (if you don't know and implement the "real" erosion rates, you cannot match the ages and the locations at the same time), so we do not find this advice particularly helpful. We have made a serious effort to make a systematic error analysis, rather than to generate a few random models with "realistic" data, which we find an unproductive methodology. This point has come up in a number of comments, so we are currently preparing a response on the general issue of the proper use of models as hypothesis tests.

REVIEWER: If the authors cut down the paper by Âż50% by getting rid of the argumentation against Schildgen et al. (2018), concentrate on a more thorough exploration of possible biases and errors in GLIDE modeling of the higher-data-density areas analyzed by Herman et al. (2013), that could help clarify how much the remaining data say.

The paper is far from that point, though. Alternatively, the paper could be published almost as-is (the errors surrounding Figure 2 really should be fixed, tests in Fig 8&13 clarified or redone, etc.) and appear together with van der Beek et al.'s responses, and everyone can just move on. I don't think the authors would be well served by this, but it would provide some closure.

RESPONSE: We have selected a long-format, open discussion, and open access journal for this manuscript to provide a comprehensive analysis of bias and uncertainties associated with the result of Herman et al. and the critique of it posed by Schildgen et al. The topic addressed in both these papers is of high prominence in the thermochronology and surface processes communities and warrants a thorough evaluation. While our manuscript is long, the length of the manuscript is allowed by this journal. Furthermore, we maintain that a manuscript of this length is needed to comprehensively advance the science on this topic. If readers are vested in this topic they will read it. We feel it is important to cover both the contrasting results of Schilgen et al and Herman et al. and present a systemic analysis of the GLIDE modeling approach. Based on the above arguments, we respectfully disagree with this reviewer's request to reduce manuscript length by 50%.

---

## Author Comment (AC5) · 25 Oct 2020

Preface to our Response: We specifically chose ESurf as the journal for submitting the Willett et al., manuscript (this paper) so that an open scientific discussion could occur around the contrasting results of Herman et al., (2013) and Schildgen et al., (2018). While this comments by van der Beek et al., and our responses, may appear as a heated exchange, we would like to emphasize for the broader scientific community reading this that it is our intent to have a healthy scientific discussion on this topic so that the strengths, weaknesses, and caveats associated with the inversion of thermochronometric data are clear. With that in mind, we respond in the following to the

latest comment by van der Beek et al.

COMMENT: Willett et al. focus their response on two aspects of our initial comment: (1) the role of the different geotherms in the forward and inverse models and (2) the resolution and data density of the synthetic models. We will provide some additional clarification on these aspects below. Before we do so, however, we feel it is important to underline once more that Schildgen et al. (2018) and Willett et al. pursue fundamentally different goals. Schildgen et al. (2018) posed a geological question: whether the "worldwide acceleration of mountain erosion under a cooling climate" since 6 Ma inferred by Herman et al. (2013) from thermochronologic datasets was robust. In contrast, Willett et al. aim to identify different sources of error in the GLIDE inversions of thermochronologic ages for erosion rates, thereby posing a methodological, model-framed question.

RESPONSE: We don't really see a difference in the goals of Schildgen et al. (2018) and Willett et al. (this paper). Schildgen et al. made no new GLIDE models of erosion rates from the data of Herman et al., so they did not test "robustness", which is defined as consistency of results in the face of data or assumption errors. Probably van der Beek et al. meant to say "accuracy", although their subsequent paragraph argues that this is impossible to do, but whatever the goals, Schildgen et al. (2018) was in effect was a comment on Herman et al., and demonstration of the accuracy of the GLIDE inversions is the centerpiece of both papers.

COMMENT: As stated in our initial comment, a detailed analysis of the potential sources of error within the GLIDE inversion procedure used by Herman et al. (2013) was outside the scope of the Schildgen et al. (2018) analysis. Schildgen et al. (2018) did not set out to test whether the model could make robust predictions of erosion history, but whether it had done so in the Herman et al. (2013) analysis. While Willett et al.'s new analysis is a useful exercise, it does not address the question of whether the model application to real-world data leads to accurate predictions of erosion histories. As Willett et al. state in their response: "we are attempting to learn something about

the method, not trying to simulate the way in which the model is applied in normal circumstances". Therefore, it is unclear (and Willett et al. do not clarify) how their new analyses validate the Herman et al. (2013) results. In fact, validating the application of such models to natural systems is inherently impossible (Oreskes et al., 1994): even if Willett et al. would have shown that the inversion is capable of retrieving accurate exhumation histories in the conditions they provided (and the reanalysis of some of Willett et al.'s synthetic tests in our initial comment shows that it does not), these results do not address the analysis performed by Herman et al. (2013).

RESPONSE: This is an interesting perspective on the approach taken by both Schidgen et al.(2018) and our current paper, and we will return to this issue in a later response. However, we would like to point out that the paragraph above states that "validating the application of such models to natural systems is inherently impossible (Oreskes et al., 1994)", whereas the same paragraph states "Schildgen et al. (2018) did not set out to test whether the model could make robust predictions of erosion history, but whether it had done so in the Herman et al. (2013) analysis." In other words, Schildgen et al. set out to do what Oreskes et al. (1994) argued is inherently impossible.

In fact, Oreskes is correct, it is not possible to confirm or refute a model directly, simply because one never knows the true answer. Schildgen et al. do not compare the results of Herman et al (2013) to the true erosion rate history; they don't know the true erosion rate history. They compare the Herman et al. results to a different set of inferences that they derived by extraction from published literature. Herman et al (2013) and Schildgen et al. (2018) are essentially alternative models with differing results. To judge between them, it is necessary to assess each to determine if one or the other is more likely to come up with the correct answer. This is why we (Willett et al., this paper) have conducted a complete error analysis of Herman et al. to try to determine where errors might arise. We would do the same and make an in-depth analysis of the model of Schildgen et al. (2018), but because their paper included no reproducible

tests (other than the tests of GLIDE, which were done with the wrong geotherm), we cannot conduct an error analysis. Hence our focus remains on the Herman et al. (2018) model.

COMMENT: 1 Variable geotherm calculation

Willett et al. dismiss the synthetic models in Schildgen et al. (2018) and in our initial comment because of the difference in boundary conditions when calculating the geotherm, writing: "all error in the van der Beek et al. comment and Schildgen et al. (2018) models is due to these geotherm errors". But Willett et al. provide no support for the assertion that differences in boundary conditions produce spurious accelerations, neither in their manuscript nor in the response to our comment. Nor do they address why spurious accelerations are present in inversions using the high-density synthetic data generated by Willett et al., where there are no differences in geotherm calculation between the forward and the inverse models.

RESPONSE: There are no spurious accelerations to explain in the high density synthetic data models (Figures 5, 6, 7, 8).

COMMENT: We reiterate the observations on which we based our rebuttal of Willett et al.'s dismissal: • Figure 1 in our initial comment shows that when the inferred cause of accelerations in our synthetic models (what we term the spatial correlation bias) is removed, the models do not predict accelerating erosion rates in the last few Myr, despite the difference in geotherm calculation between the forward and inverse models;

RESPONSE: The model shown in Figure 1 of the comment has large errors due to the wrong geotherm (Willett et al., Fig. 4). We don't know in detail how this affects the inverse model result – acceleration or deceleration, given the complex feedbacks on the thermal model, triggered by data errors. Adding and removing data with large errors can easily flip errors between acceleration or deceleration, simply as a function of the errors. We don't think it is our responsibility to do the error analysis on Schildgen

et al.'s or van der Beek et al.'s models. We demonstrated errors of up to 100% in age. Errors in erosion rates are in the tens of percent - probably higher where erosion rates are low. This is unacceptable by any possible standard we could come up with. We are not inclined to spend additional time analysing models with such large errors, where any interpretation is tainted by speculation regarding the effect of these errors. Our Figures 5,6,7,14) show similar models with no geotherm errors and there is no false acceleration.

COMMENT: • Figure 3 in our initial comment shows that Willett et al.'s models with a transient geotherm (but ages calculated with the same thermal boundary conditions in the forward and inverse model) show spurious accelerations in regions of spatially variable exhumation rates, either at earlier times than shown in Willett et al. or in the last few Myr when the high-temperature ages are removed;

RESPONSE: We did explain these errors - in detail. We explained this in our last response and stand by that statement. These are resolution errors, combined with a low value of the prior. It is always possible to keep removing data, until a model is poorly resolved. Or find some part of a model in space or time that is unresolved. In the early timesteps, there are no ages in the fast uplifting region, so there will be smoothing of the low uplift rates into the high uplift zone. These will remain in the early history until a sufficient number of high uplift-zone ages enter into the problem. Until then, the problem is defined as poorly resolved.

COMMENT: • Figure 4 in our initial comment shows that Willett et al.'s models with a fixed geotherm similarly produce spurious accelerations in regions of spatially variable exhumation rates. Our conclusion from these tests is thus the opposite of what is suggested by Willett et al.: the geotherm errors cannot explain the spurious accelerations observed in these synthetic tests. We provided some tentative explanations for why this may be the case, but we did not argue that one or the other model provides a "more accurate" solution to the geotherm problem; we acknowledge that the thermal boundary conditions in both Pecube and GLIDE are imperfect.

RESPONSE: Not sure we understand this comment. The use of a steady geotherm to generate ages, inverted with a transient geotherm, explains the spurious accelerations in all models from Schildgen et al and Figures 1 and 2 from the comment by van der Beek et al. These models are, in our view, invalidated by these large errors.

If this refers rather to model errors involving advection amplification of resolution errors, it is correct that this amplification is important only in the full thermal models. However, there are still resolution errors in the constant gradient models shown in van der Beek et al., Figure 4. These also arise from removing data or showing early (under-resolved) timesteps. With higher resolution these errors vanish (van der Beek et al. comment, Figure 4a,b,c). This was our stated conclusion- these figures are consistent with our conclusions, along with the other dozen models we constructed. At high resolution, there is no error; at low resolution, there are errors, but the nature and direction of the errors are not generalizable. Again, we are not sure why we are responsible to explain every model van der Beek et al. can produce. We presented a large and consistent set of models in our paper, which provide more insight than isolated examples of poorly-defined phenomenon with guesses as to causal relationships. For example, our data set C is very similar to van der Beek's Figure 4. It did have large errors and false accelerations. These disappeared when the prior was increased (Willett et al., Figures 10-12) or when the age distribution of the data was more favorable (Figure 9).

The issue of how high and low resolution are defined, we take up below.

COMMENT: 2 Resolution and Data Density

In their response, Willett et al. frequently invoke "high" versus "low" resolution and data density, but without quantifying these classifications. The first argument in Willett et al.'s response is that the "commonality of ages" between the Alpine dataset and their "high-resolution" synthetic datasets A and D makes the results from these synthetic tests applicable to the Alps. We have already refuted this argument in our original comment (lines 210-222).

RESPONSE: There is a simple definition. High resolution is obtained when resolution errors vanish. If one can continue to add data and the answer changes, the initial case was not high resolution. By removing data until errors appear, van der Beek et al. have found the limit between high resolution and low(er) resolution.

In fact, "high" and "low" resolution are relative and context specific. The purpose of the models in Figures 6, 7, 8, 13, 14, is to differentiate between model and resolution errors. This is a standard first test in error analysis and is absolutely necessary. If one does not know if errors are model or data based (resolution), it is impossible to understand any additional analysis. Justification and explanation for the model test is given in Lines 583 to 605 of our paper. To complete this test, it is fully justified to use an infinite number of data, so high resolution is defined to be anything that fully eliminates resolution errors up to and including an infinite number of data.

We stand by our original statement, by removing data or looking at earlier timesteps van der Beek et al have sidestepped the purpose of the model test by changing a high resolution data set into a low resolution data set.

We don't see relevance of comparison to the Alpine data. This was not our justification.

COMMENT: Second, Willett et al. argue that looking at earlier time-steps of the model is unwarranted, writing: "Because these timesteps are not the target of the experiment, we did not generate ages over these intervals, they are poorly resolved and there is a correspondingly larger resolution error". This statement is incorrect: Dataset D includes ~30 mica 40Ar/39Ar ages within the 8-10 Ma time-window (Fig. 5 of the Willett et al. manuscript, or Fig. 1 of the response). Moreover, we could have taken the 10-12 Ma time-window as our starting point, which contains >50 mica 40Ar/39Ar ages in both datasets A and D. The predicted erosion rates are lower in the 10-12 Ma time-window than in the 8-10 Ma time-window (leading to larger accelerations when compared to later timewindows) but the resolution is similar (Figure 1).

RESPONSE: See point above. Age counts are not an exclusive measure for resolution.

First, timesteps must be bracketed by ages, not simply sampled. Second, there are many other factors such as spatial position relative to other ages. If we could just count ages, why would we go through the numerous exercises revolving around resolution determination?

We also point out that the spatial distribution of data in these models based on Data set A, particularly with the Argon data removed, is very poor in the sense that data are clustered with the majority of ages on either side positioned near to the fault, meaning they are separated by less than 1 correlation length. Also given that the fault displacement and the erosion rate contrast is greater than anywhere observed on Earth, we would not speculate as to how many ages define "well" or "poorly" resolved. We accepted these model conditions from Schildgen et al.'s test, but it is not appropriate to contend that these are "typical" of any natural examples. The only extrapolatable conclusion from these tests is that there exists a threshold for data density, above this there are no errors, below this, there are errors. Where this threshold is for the model is easy to establish; where this threshold is for a natural example, such as the Alps, cannot be determined by direct comparison with the models because the erosion rate function and data/age distribution are different.

We also note that we discussed this potential situation in our paper (lines 1179-1186) and acknowledge that it is possible to find a situation where a false increase in erosion rate derived from the prior or nearby data might not be recognized. But this requires a very particular set of circumstances. The fact that van der Beek et al. have modified our model until these circumstances are met is nothing we did not already acknowledge; the important point is how likely is that these circumstances would appear in nature without recognition.

COMMENT: Third, Willett et al. argue that by removing the hypothetical mica 40Ar/39Ar data from the datasets A and D, we have "converted a high-density model into a low-density model and thereby reintroduced resolution errors". We note, however, that (1) since all the synthetic mica 40Ar/39Ar ages are >8 Ma, the synthetic tests in which

they are removed contain exactly the same number of data within the critical 6-0 Ma time-windows as the original tests;

RESPONSE: Same point as above. If errors appear by removing data, these are resolution errors by definition. If there is a model where errors disappear by adding data, the first model was not high resolution. All ages older than 6 Ma contribute to the estimate and resolution of the 6 to 0 Ma window, not just the ages that fall into the particular timestep.

COMMENT: (2) as already noted in our initial comment, the resolution values for the recent time-windows are nearly identical between the two tests (compare panels a-c and g-i in Figs. 3 and 4 of our initial comment). Given the similarity of data and resolution values within the critical time-windows, we do not understand on what grounds Willett et al. characterize one model as "high-density" and the other as "low-density".

RESPONSE: Based on the existence of resolution errors. This is not circular because we recognize them as resolution errors because they vanish as more data are added or as the prior goes nearer to the true value (compare Willett et al., Figs 10 and 12). This is why one model has little value – it is only through multiple and contrasting models that one understands the sources of errors and behavior of a method. It would be great to have a binary criterion, so one can black-box an analysis, but this does not exist for most estimation problems. The values of posterior resolution and how well they reflect the situation are a separate issue that we will address later.

Most of these comments by van der Beek et al seem to be conflating the multiple purposes of the model tests. There are two purposes, carefully laid out by the organization of our paper:

Establish whether or not there are model errors or only resolution errors. Section 4.3. Figures 6, 7, 8, 13,14. There are no significant errors (including no false accelerations) showing that errors are indeed resolution errors.

Having established that errors are resolution errors, the next steps are:

to determine how many data, in what configuration and over what age range are adequate to resolve an erosion rate history. This is the subject of section 4.4 or our paper and the models in figures 6, 7, 9, 10,11, 12, 14, and 15. to determine how well posterior metrics predict accurate solutions. See line 646 in our paper.

van der Beek et al seem to not recognize the dual purpose and so argue that there is something artificial in the data construct of data set A, when in fact, we probably should have included many hundred more data and more thermochronometric systems to properly address objective (1). We tried to include the minimum necessary number of data in the first experiments, so as to address objective (1), but retain their relevance to Objective (2) (see line 651). It is therefore no surprise that as soon as data are removed the model falls into the "lower" resolution state.

Objectives (2) and (3) are much more complicated questions and it is nearly impossible to generalize a simple answer, which is why we have spent much of our paper discussing how one does this in practice (e.g. section 6.1). We have constructed a range of data sets (Fig. 5), some of which are clearly at the low end of resolution, in order to make a sincere effort to show the range of possible outcomes, including demonstrating failure of our model (e.g. Figure 10). We are happy to continue and expand this discussion, but it should be done on the basis that the extensive suite of models we have presented, which are only a subset of the hundreds of models that we have run over the years, are a legitimate portrayal of the behaviour of our models. Rather than engage with this suite of models and outcomes, van der Beek et al. continue to try to argue that we are covering up some universal failure of our models by perturbing the successful models until they fail. We don't find that this approach is productive.

We are preparing another response in which we can go into more depth as to how one determines resolution in practice and as to why counting ages in specific timesteps is

inadequate, although we thought we covered this rather thoroughly in section 6.1.

COMMENT: As a final point, we provide some clarification of why Willett et al.'s models that use the true solution as the prior (Figs. 8 and 13 of the Willett et al. manuscript) are irrelevant. In Bayesian theory, if the posterior probability distribution of the parameter values (i.e., in this case, the predicted spatial-temporal distribution of erosion rates) is equal to the prior probability distribution (i.e., the input prior erosion rates), then the available data have not contributed to constraining the solution. In this case, the prior erosion rates are equal to the true erosion rates, which have been used to predict the synthetic data. Therefore, for each point in space and time, the model's "initial guess" (the prior), is correct and perfectly reproduces the data. The data cannot influence this problem, and the only thing we learn from this test is that there are no obvious errors in the code. This is the explanation for why the errors disappear. From a practical viewpoint, this test is irrelevant because we cannot know a-priori what the exhumation history of a particular region is (i.e., there is no way to know the prior, it is necessarily a guess) and if we did, there would be no point collecting data in such a region because we wouldn't learn anything new from that data.

RESPONSE: Thanks for the clarification. However, this is not quite correct. Glide is a linear method; there is no initial guess that is to be improved upon. What there is, is a joint probability function for each parameter that reflects a balance between (1) staying near the prior; (2) being consistent with the data (ages); and (3) remaining within the constraints of the model (in this case, the geotherm and spatial correlation structure). By setting the prior equal to the true solution, we eliminate the need to balance fitting the ages and staying close to the prior, but not the third constraint. We still need to balance (1) and (2) against (3). This trade-off is expressed in Willett et al., current paper, Eqn. 12. There is a spatial correlation structure and it does impose averaging; we don't question this. The question is how large is this effect. This is not a test for errors in the code; this is a test for errors in the model, for example, excess smoothness.

---

## Referee Comment (RC5) · Anonymous Referee #4 · 26 Oct 2020

My immediate impression is that this is a serious, high quality study that is of interest to publish (bearing in mind that it's not really something within my current field of research). The paper appears to be largely inspired by the Schildgen et al. nature paper in which the authors argued that inferred late Cenozoic increases in erosion rates might be due to a spatial correlation data bias. The Willett paper addresses this issue with an extensive look into errors inherent to interpreting thermo-chronological data. Based on what appears to be detailed analysis, they show that there is no systematic bias of the nature suggested by Schildgen et al.. I agree the Willett paper is contro-

versial in the sense that it won't please the authors of the Schildgen paper. However, I also think that the results published by Schildgen deserve to be critically analysed and questioned and this seams like a good attempt at achieving this. This paper will definitely generate interest and it will undoubtedly rekindle debate on the influence of late Cenozoic climate change on erosion rates.

---

## Editor Comment (EC7) · Richard Gloaguen (Editor) · 26 Oct 2020

This review appeared as editor comment (EC) in an earlier version. To allow a more transparent review process, we asked the reviewer to post it again as an official review comment (RC).

---

## Author Comment (AC6) · 12 Nov 2020

There is a general issue in this debate as to the objectives, goals and hypotheses of the various papers involved. In several places in comments (ESURF-2020-59-SC1, SC2) van der Beek et al. have tried to differentiate the goals of the Schildgen et al. (2018) analysis and the Willett et al. (ESURF 2020-59) goals and analysis. For example:

COMMENT (ESURF 2020-59 SC2, suppl., line 13): Schildgen et al. (2018) posed a geological question: whether the "worldwide acceleration of mountain erosion under a cooling climate" since 6 Ma inferred by Herman et al. (2013) from thermochronologic datasets was robust**. In contrast, Willett et al. aim to identify different sources of error

in the GLIDE inversions of thermochronologic ages for erosion rates, thereby posing a methodological, model-framed question.

Later, van der Beek et al. stated:

COMMENT (Esurf-2020-59-SC1-supplement, line 57) : In other words, Schildgen et al. (2018) did not set out to test whether the model could make robust** predictions of erosion history, but whether it had done so in the Herman et al. (2013) analysis."

These comments indicate that Schildgen et al. (2018), took as a given, that they could determine what is the true erosion rate history at any given site, compare this to Herman et al. (2013) results, thereby checking not only the accuracy of the Herman et al. predictions, but also the source of the errors. Both of these are strong claims. In general, it is not possible to "know" any given erosion rate history as this is a model quantity (see discussion in ESURF-2020-59-AC5), and to assign a source or cause to error is rarely a simple process, requiring extensive error analysis, which is why this is the focus of our paper. This raises an important aspect of our paper, the critique of Schildgen et al. (2018), in which we argued that Schildgen et al. did not present a reproducible methodology, capable of testing these hypotheses (ESURF2020-59, line 1310). Van der Beek et al. suggest that we are asking a different, methodological, question, rather than a geologic question, but we would argue that we are investigating the same question, but, we are providing the hypothesis testing that is not present in Schildgen et al. (2018). We would like to elaborate on that point here to explain why this distinction is important.

The basic scientific method involves four stages. These include: (1) observation, (2) hypothesis formulation, (3) testing of the hypothesis with new observations or experiments, and, (4) conclusion, including possible hypothesis revision.

Schildgen et al. (2018), and now van der Beek et al., made a number of observations (stage 1) regarding age distributions, tectonic processes, or previous models, and from this, they formulated a number of hypotheses (stage 2). Their hypotheses include a

direct interpretation of the data, for example if ages reflect exclusively tectonic drivers or not, but mostly their hypothesis involves statements of errors in the Herman et al analysis. For example, in the comment by van der Beek et al.:

COMMENT: (Esurf-2020-59-SC1-supplement, line 57) "A spatial correlation bias, as defined by Schildgen et al. (2018), creates spurious accelerations in erosion rates when data from areas with spatially variable exhumation histories are inappropriately combined. Although Schildgen et al. (2018) concluded that the spatial correlation bias was a common problem with the Herman et al. (2013) results, it was not the only problem discussed. Other sources of error in the Herman et al. (2013) results arose from model errors, notably (1) assuming vertical rock exhumation in regions where lateral advection of rocks is important, and (2) ignoring the impacts of changes in topography on thermochronometer age patterns; as well as from operator errors, such as (3) the inclusion of samples reheated by volcanic flows or hydrothermal fluids, and (4) the inclusion of partially reset or unreset samples from sedimentary rocks in the inversions. A detailed analysis of the potential sources of error within the GLIDE inversion procedure was outside the scope of the Schildgen et al. (2018) analysis, which focused rather on the implications and robustness of the Herman et al. (2013) results. In other words, Schildgen et al. (2018) did not set out to test whether the model could make robust predictions of erosion history, but whether it had done so in the Herman et al. (2013) analysis."

The numbered list in this quote from van der Beek et al. defines a set of hypotheses (Stage 2). Stage (3) of the scientific method would require a set of tests to demonstrate that these potential errors are, in fact, errors, and if so, how important they are, i.e. are they significant errors, and important enough to reverse the conclusions of Herman et al. (2013)? However Schildgen et al. (2018) contains no tests. There are no tests for the importance of non-vertical rock exhumation. There are no tests for the effects of transient topography. There are no tests for the effect of inclusion of hydrothermally-influenced ages. There are no tests for the effect of including partially

reset ages. There are no tests for any specific field sites using a modified data set, having removed questionable data. There are no tests for robustness** of the Herman et al (2013) results, which would require rerunning these models with modified data or model control parameters. The complete lack of hypothesis testing means that it is impossible to either confirm or refute any of the hypotheses made above, or to judge the validity of any conclusions in Schildgen et al. (2018). One might accept that published work had already proven the existence of some of the data errors, but most of the list above are model errors (non-vertical paths, etc.) which might have published models to demonstrate process, but are not proven to be significant for specific examples. And in all cases, the impact of each of these models or data errors on the Herman et al. GLIDE models needs to be tested, and this was not done. It is completely possible that all model and data errors are critical or are negligible; without testing, one cannot say.

Contrast this to the way that van der Beek et al. (ESURF-2020-59-SC1) responded to the realization that they made a geotherm error in Schildgen et al. Figures ED 3-6, as identified in our manuscript (Willett et al., ESURF-2020-59). They immediately made a model test and announced that even though all their ages were wrong by up to 100%, it was a negligible error given the low precision of most inferred erosion rates, and therefore all their models were still valid. If a 10 to 100% error in every individual age of a data set results in a negligible effect on a Herman et al. (2013) GLIDE model result, doesn't it seem important to check the significance of all the data or model errors listed above? Schildgen et al. (2018) did not think so; it was sufficient to hypothesize them, assume that they were critical errors, i.e. fatal to the Herman et al. (2013) conclusions, and move directly to step (4), their own conclusions.

As we discussed (Esurf 2020-59-AC1), the test van der Beek et al. conducted for the models of Schildgen et al. Figs ED 3, 5, 6) was not valid. No error is valid for a control test, certainly not 30%, but the important point, methodologically, is that discovery of model errors or data errors (what van der Beek et al. call operator errors*) should be tested. In fact, we have no problem with the first two steps of the Schidgen et

al. (2018) analysis. It is completely appropriate to identify potential errors and model inadequacies of Herman et al. (2013). We welcome critical assessments of all our work. We would welcome future work in which the importance of these errors is tested. What we object to is a set of very strong conclusions based on hypotheses with no testing.

When we state that Schildgen et al. (2018) have no reproducible methodology, we are referring to this lack of reproducible hypothesis testing (stage 3). The three models that were presented in Schildgen et al. Figs ED 3, 5, 6) were a test, but were conducted incorrectly as described in Willett et al.(ESURF2020-59). Once these tests are removed, there is no remaining hypothesis testing and therefore no basis for any of their conclusions.

The question of reproducibility also applies to the Schildgen et al.'s (2018) assessment of a "spurious" result in Herman et al. (2013). We argued that this term is undefined and van der Beek et al. attempted to clarify this with the following comment:

COMMENT (esurf-2020-59-SC1-supplement lines 409-419): "Schildgen et al. (2018) used the term "spurious" simply in its generally accepted meaning of "false" or "fake", describing in detail why any given acceleration was deemed "spurious" for each region. In addition to inappropriate combination of data, the reasons also included models errors and operator errors that are not considered by Willett et al., such as (1) inappropriate assumptions of purely vertical exhumation in regions where lateral rock advection plays an important role (e.g., Southern Alps of New Zealand, Olympics, Apennines, Taiwan); (2) inappropriate assumptions of no change in surface morphology where such changes were shown to be critical for understanding thermochronometer age patterns (e.g., Aconquija, Fiordland, western European Alps, southern Peru, Bolivia, Coast Range); (3) inappropriate inclusion of samples that were reported to have been reheated by volcanic flows (e.g., San Juan Mountains, southern Peru) or hydrothermal fluids (Eritrea); and (4) inappropriate inclusion of partially reset or unreset data from sedimentary rocks (e.g., Taiwan, New Zealand). Some of the spurious increases may

have arisen due to a reversion to the prior in some cases, but in reality, the reason for the spurious acceleration does not matter so much as the fact that it is fake. By strictly limiting the definition of "spurious", the authors have sidestepped addressing the true extent of problems in the Herman et al. (2013) inversion results. Uniform application of a model that takes no account of changes in surface morphology, rock-exhumation pathways, or tectonic features to a global dataset that includes many data points unrelated to exhumation is bound to fail in some places. For the results presented by Herman et al. (2013), we conclude that it has failed in the majority of cases."

So the clarification states that spurious means whatever the authors want it to mean, again selecting from their list of untested hypotheses, reinforcing our argument that this is an irreproducible designation. Furthermore, by mixing data errors with model errors with, what are really model interpretations (tectonic vs climate changes or topographic transients), they render this assessment meaningless as a testable quantity. We will ignore the statements regarding "false" or "fake", as these are meaningless words in a scientific context.

Willett et al. (ESURF-2020-59) is focussed on the methodological aspects of GLIDE and its response to spatial gradients in age because this is the only aspect of the Schildgen et al. (2018) study that even attempted any hypothesis testing. It is not because we are "sidestepping" the other criticisms; it is because these criticisms are hypotheses with no testing, either in Schildgen et al. (2018) or the van der Beek et al. comments. If they are ever tested and shown to be important, we will address them, or, given valid tests, we would accept the results.

In addition to not testing the hypotheses regarding errors in the Herman et al. (2013) analysis, Schildgen et al.'s (2018) own assessment of individual site erosion rates and their cause is neither tested, nor reproducible. We challenge anyone to reproduce Figure ED 2 or Table 1. We are not able to, and assert that it is not possible. The results of Schildgen et al, (2018) are qualitative, subjective interpretations. One might agree with their interpretation or not agree with it, but it does not constitute a reproducible

result.

The conclusions of van der Beek et al.'s comment provide an appropriate summary of the logic of both Schildgen et al. (2018) and their comment

COMMENT (ESURF-2020-59-SC1, line 435) : "We also reaffirm the conclusions from Schildgen et al. (2018) that a great majority of the results reported by Herman et al. (2013) are unreliable due to a combination of spatial correlation biases, model error, and operator* error. Reversion to the prior erosion rate may have also led to spurious results in some sites, but the spatial correlation bias is likely the most common issue in areas that were not significantly affected by model errors in GLIDE (e.g., assumption of vertical exhumation pathways and assumption of no changes in topography through time) or operator errors".

We note that these conclusions are identical to their hypotheses, but without a single test in the interim between hypothesis and conclusion. Note that, even if the Schildgen et al. (Figures ED 3, 5, 6) tests were valid, they would have provided only an example that a spatial correlation bias might exist for their one selected data set and one selected erosion rate function, not that it does exist in any or "most" of the Herman et al results. The tests were designed as existence tests, not for significance in any of the natural examples. As for the other types of errors in their list, no tests were attempted. Spatial correlation bias is concluded, not only to be important, but to be "the most common" error. Even if they had established error, how can one conclude that a specific error is the most common, when there are no tests presented even for its existence, let alone frequency, and no tests for the existence, significance, or frequency of the alternative errors?

Based on the above concerns we raised over the Schildgen et al., (2018) manuscript and the comment by van der Beek et al., we stand by the results of Herman et al., (2013) and supporting analysis provided in our manuscript (ESURF2020-59). We stand by our original paper, that identified the lack of hypothesis testing and do not

see that we have misrepresented anything in our critique of Schildgen et al. (2018). We explained this argument in our paper at various points (e.g. paragraph at line 1310); we could expand this argument as we do above, but the paper is already long on criticism and we think the point is made effectively. We agree we have paid less attention to some of the secondary errors hypothesized by Schildgen et al., and we are willing to consider them further, once there are proper tests of the significance of these errors. Until then, we think the focus on the spatial correlation bias is appropriate.

*"operator errors" refer to failure to follow documented procedure, like landing an airplane in a field, when procedure calls for an airport. What van der Beek et al. describe are "data errors".

**"robust" refers to model insensitivity to small errors in the data, i.e. adding or removing some data gives the same result. What van der Beek et al. refer to is "accuracy".

---

## Author Comment (AC7) · 7 Dec 2020

Response to comment esurf-2020-59-SC1

We made several responses to the major points of this comment in our author comments esurf-2020-59-AC2, esurf-2020-59-AC5, esurf-2020-59-AC6 In this comment, we will assure completeness by making a point by point rebuttal, but we provide references to the earlier responses.

We will put this response into a supplement, primarily so that we can more easily format the response for clarity.

[Figure]

**ESurfD**

Interactive
comment

Changes to the manuscript to respond to this, and the other, comments will follow in a separate author comment.

Please also note the supplement to this comment:
https://esurf.copernicus.org/preprints/esurf-2020-59/esurf-2020-59-AC7-supplement.pdf

[Figure]

**Supplement:**

**Response to Comment SC1**

Comment by van der Beek et al in red. Our response in black.

Disparities between the findings of Herman et al. (2013) and earlier interpretations of exhumation rates and patterns in many of the regions where they saw increases led Schildgen et al. (2018) to explore the robustness of the Herman et al. (2013) results. Several of these earlier studies employed thermo-kinematic modelling to quantitatively interpret the data, including some of our own studies (e.g., Bermúdez et al., 2011; Beucher et al., 2012; Glotzbach et al., 2011a; 2011b; Robert et al., 2009; Schildgen et al., 2009; Thiede and Ehlers, 2013) and some by co-authors of the Herman et al. (2013) paper (e.g., Ehlers et al., 2003; Fuller et al., 2006; Herman et al., 2007; 2010; 2009; Willett et al., 2003). Schildgen et al.'s (2018) approach was to examine the spatial patterns of the inferred erosion rates, compare these to the original data and mapped major structures, and review the literature to assess the original interpretations of these data. The approach was thus essentially abductive, which has been shown to be an efficient and even desirable logical approach in geomorphology (e.g., Baker, 1996). Schildgen et al. (2018) also developed synthetic tests to better understand what was likely driving the results of Herman et al. (2013). Schildgen et al. (2018) focused only on the predictions that Herman et al. (2013) deemed well resolved; only these points were shown on the maps of Schildgen et al. (2018).

Van der Beek et al. list a large number of studies with the implication that these somehow contradict the results of Herman et al. (2013). This is not the case. Many of these studies found an increase in erosion rates over the last 5 Ma. Others were modeling studies that were not parameterized in a way that was capable of establishing a change in erosion rate over the last 5 Ma. None of these studies can differentiate between tectonic and climate-change induced changes in exhumation rate. All of these studies include interpretations or models that are not inherently better than the approach taken by Herman et al. (2013). To use these results to "test" Herman et al. (2013) requires posing an objective test (See our comment AC-6). This was not done by Schildgen et al. (2018). They made a subjective selection as to which aspects of previous studies they took and which they ignored, and in every case where there was a discrepancy, they concluded that Herman et al. was wrong and the alternative was right.

A spatial correlation bias, as defined by Schildgen et al. (2018), creates spurious accelerations in erosion rates when data from areas with spatially variable exhumation histories are inappropriately combined. Although Schildgen et al. (2018) concluded that the spatial correlation bias was a common problem with the Herman et al. (2013) results, it was not the only problem discussed. Other sources of error in the Herman et al. (2013) results arose from model errors, notably (1) assuming vertical rock exhumation in regions where lateral advection of rocks is important, and (2) ignoring the impacts of changes in topography on thermochronometer age patterns; as well as from operator errors, such as (3) the inclusion of samples reheated by volcanic flows or hydrothermal fluids, and (4) the inclusion of partially reset or unreset samples from sedimentary rocks in the inversions. A detailed analysis of the potential sources of error within the GLIDE inversion procedure was outside the scope of the Schildgen et al. (2018) analysis, which focused rather on the implications and robustness of the Herman et al. (2013) results. In other words, Schildgen et al. (2018) did not set out to test whether the model could make robust predictions of erosion history, but whether it had done so in the Herman et al. (2013) analysis.

We addressed this point directly in AC-5 and AC-6. The approach of Schidgen et al (2018) is predicated on the idea that they could determine the "correct" answer from the literature in order to "test" Herman et al. (2013). No quantitative tests were ever applied.

In the following, we will first focus on three main points made by Willett et al.:

1. Model error due to variable geotherm calculation: We show that, although the geotherm calculation method in GLIDE is based on a poor choice of boundary conditions, Willett et al.'s dismissal of Schildgen et al.'s (2018) synthetic tests based on this difference is a red herring;

We addressed this in AC2 extensively.

2. Bias to the prior versus the spatial correlation bias: We show that bias to the prior erosion rate is an additional source of bias towards acceleration in the Herman et al. (2013) results, but new synthetic tests presented here imply it is less of a problem than the spatial correlation bias;

We addressed this in AC2.

3. Biased post-processing operator and resolution: We argue that Willett et al.'s criticism of the use of postprocessing operators and lack of regard for resolution is misdirected at Schildgen et al. (2018), should instead be directed at Herman et al. (2013), and has no impact on the Schildgen et al. (2018) analysis.

We don't see the basis of this statement. With the replacement of Figures 2 and 3 of Herman et al. by Figures 24, 25 of esurf2020-59, there are no ratios and thus no ratio bias at all in Herman et al., so this statement is unfounded. Most of esurf2020-59 is resolution analysis, so we also see no basis for the second part of this statement. Schildgen et al. made no quantitative statistical analysis of resolution and provide no evidence that they used the Herman et al analysis other than using

one value as an analysis cut-off. Most importantly, they interpreted NR maps to infer the existence of errors, e.g. spatial correlation bias, without any knowledge of the error mapped into the NR.

We then continue with (4) a few comments on the spatial variability in thermochronometric ages and (5) on the additional field examples presented by Willett et al. We finish (6) with comments on the definition of "spurious" as used by Schildgen et al. (2018). In an appendix, we provide some line-specific comments on the Discussion 80 section of Willett et al., which is replete with mistakes and mischaracterizations of the work presented by themselves, the work of Schildgen et al. (2018) and that of Herman et al. (2013).

We will respond to these below.

**1 Model error due to variable geotherm calculation**

Willett et al. claim that due to the differing boundary conditions between the model Schildgen et al. (2018) used to predict synthetic ages, Pecube (with a basal temperature boundary condition), and the model that performs the inversion, GLIDE (with a basal heat-flux boundary condition), "the inversion will infer an increase in erosion rate with time in order to fit these ages" (line 553)...

We responded to this in AC2 and again in AC5. We stand by our contention that the large errors (due to their choice of a different, constant-T, boundary condition) in Schildgen et al.'s synthetic data models render them invalid. They have not corrected this error to show that they obtain the same result. They did show that their age errors produced errors of up to 28% in the inferred erosion rates of the high erosion region and did not check the low erosion rate region. (see AC2).

**2 Bias to the prior versus spatial correlation bias**

**2.1 Bias to the prior versus spatial correlation bias in synthetic tests**

Willett et al. argue that the problematic results presented by Herman et al. (2013) are affected by a Bayesian bias to the prior erosion rate rather than a "spatial correlation bias". This argument is partly semantic; the inversion interpolates both temporally and spatially between incomplete data, which when done incautiously, introduces both types of bias. The spatial correlation bias causes spurious accelerations in inferred erosion due to the combination of areas that experienced rapid exhumation (and hence have young ages) with slowly exhumed areas that yield older ages. The Bayesian bias to the prior, in contrast, returns the (input) prior erosion-rate estimate when the data do not constrain the erosion rate for a given time window. We illustrate the effects of these biases by a set of additional synthetic tests. We have run these tests in the same way as in Schildgen et al. (2018); as demonstrated in Section 1, the different calculation of the geotherm between the forward and the inverse model does not significantly affect these synthetic tests.

These models have errors in the ages of 10s to over 100%. Van der Beek et al make no demonstration that the geotherm error "does not significantly affect these tests". Their argument is only that because the error is not intuitive in its second derivative (acceleration), it must not be significant. This is not a valid argument. They have not corrected the error and shown that the model produces the same result. We see no reason to respond to models whose predictions are ambiguous due to this large error. We presented an extensive suite of models equivalent to these, but without geotherm errors. We think the discussion should be restricted to these models.

**2.2 Spatial correlation bias in the synthetic tests presented by Willett et al.**

At this point, we can ask why our synthetic results are so different from those of Willett et al. The answer to this question is twofold: removal of the spurious erosion-rate increase in Willett et al.'s inversions is achieved using spatially variable prior erosion rates and thus independent a-priori knowledge of this spatial distribution (their Fig. 8), the careful design of input data that have an idealized spatial and temporal distribution (their Figs. 6, 7, 9, 14 and 15), or both (their Fig. 13). The models of Willett et al. with a more realistic data distribution (i.e. Set C; their Figs. 10-12) show exactly the accelerations we expect (partly counteracted in Fig. 12 by the reversion to a very high prior erosion rate), and illustrate the dominance of the spatial correlation bias over the reversion to the prior bias similar to our Fig. 2. Although Willett et al. claim that their dataset A (and D, which has the same data distribution) "roughly corresponds in pattern to the Alpine data set" (line 581), it is in fact carefully designed to produce the desired result. In particular:

• Datasets A/D contain five thermochronological systems (and Sets B/E contain four), whereas the real Alps dataset only includes three systems and is dominated by apatite and zircon fission-track ages (290 out of 309 data, the remaining 19 being apatite (U-Th)/He ages). Most noticeably, 30% of the ages in Willett et al.'s Sets A/D (and 22% in Sets B/E) are mica 40Ar/39Ar ages. Neither Herman et al. (2013) nor Fox et al. (2016) used any mica 40Ar/39Ar cooling ages in their inversions, for the simple reason that these do not exist. The few 40Ar/39Ar dates available for this part of the Alps are crystallisation ages for minerals in fault zones (e.g., Egli et al., 2017; Rolland et al., 2008; Rossi and Rolland, 2014), and thus not representative of regional cooling related to exhumation.

• The datasets by Willett et al. are characterised by a very high number of co-located data (120 out of 176 data locations in Sets A/D combine two thermochronometers, in general associated with mica 40Ar/39Ar; 15 have three or more thermochronometers). In the real dataset, in contrast, only 48 out of 251 data locations have two thermochronometers (all are apatite and zircon fission-track), and five locations have three (with additional apatite (U-Th)/He).

• All datasets by Willett et al. consist of "perfect" ages, as predicted by the forward model. In contrast, we added a random scatter of up to 10% to our synthetic ages in order to better reflect imperfect natural data.

• Sample locations in Sets A/D are more heavily weighted toward high elevations than the real data: 38% of the data from

**180 the NW zone are from locations >2000 m, whereas only 28% of the real data are from such high elevations. The data locations in these datasets also have a much wider spatial spread than the real data....**

We responded to this comment in AC2 and again in AC5 and stand by those comments. To summarize, the difference between the models of van der Beek et al and our paper is that our models use ages that are calculated in a manner that is internally consistent with the inverse model and van der Beek et al. do not do the same. In the cases where van der Beek et al use a decimated version of our data, they have turned the high-resolution-models into low-resolution models. Our paper carefully presents a range of resolution test cases including low-resolution cases for contrast; van der Beek et al., have replaced this suite with a set of only low-resolution models, or shifted the analysis to parts of the model with low resolution, largely defeating the purpose of the models.

**2.3 Bias in reversions to the prior in the Herman et al. (2013) results**

Willett et al. argue that, when "reversion to the prior" occurs, such a result will not have a "generalizable tendency toward acceleration" (lines 1128-1129). Although we agree with this general statement, in the application of GLIDE by Herman et al. (2013), reversions to the prior erosion rate will likely create spurious accelerations. This is because, as originally argued by Willenbring and Jerolmack (2016), all of the "resolved" regions include sites of rapid exhumation and therefore young thermochronological ages, which will tend to increase resolution in the most recent time bin. To demonstrate this point, we illustrate the distribution of resolution values from the results of Herman et al. (2013) in two different time bins, 2-0 Ma and 6-4 Ma (Fig. 5 a, b) and the distribution of erosion rates in those time bins (Fig. 5 c, d). These plots illustrate, first, the tendency of the data to be better resolved in the 2-0 Ma time bin (median resolution of 0.48) compared to the 6-4 Ma time bin (median of 0.33). Moreover, 75% of all data points in the 6-4 Ma time bin have a resolution lower than 0.4, whereas only 32% of data points in the 2-0 Ma time bin have a resolution lower than 0.4. For co-located points, 90% show a higher resolution in the 2-0 Ma time bin compared to the 6-4 Ma time bin (Fig. 5 b). Hence, reversion to the prior, which Willett et al. suggest affects most of the results with resolution below 0.4, will much more likely affect points in the 6-4 Ma time bin than in the 2-0 Ma time bin. Moreover, most of the erosion rates in the 2-0 Ma time bin are higher than the prior of 0.35 mm/yr: the median erosion rate in the 2-0 Ma time bin is 0.48 mm/yr, and 80% of the erosion rates in the 2-0 Ma time bin have an erosion rate higher than 0.35 mm/yr (Fig. 5 c, d). Given the relatively high erosion rates that characterize the 2-0 Ma time bin, reversion to the prior in the older time bin will commonly produce spurious accelerations in exhumation.

We don't really see the point of this figure. Younger timesteps will always be better resolved. This is normal. Ages provide an integral constraint on erosion rates, therefore there are always more ages constraining younger timesteps (see AC3). We do not know where we stated that bias to the prior becomes important at resolution values of 0.4, but didn't intend to. One cannot simply take one value of resolution and state that bias to the prior is now significant. There are too many other factors involved and there is no reason to expect a one-to-one relationship between the temporal resolution parameter and bias-to-the-prior. For example, younger timesteps that have no ages in them, but do have older ages (as in Fig 1 "Low Resolution" region) will have lower values of resolution but less dependence on the prior as they must be consistent with the older ages. The strong dependence on the prior is for regions with no data or timesteps so old that there are no ages sampling the interval. This figure of van der Beek is mixing these different spatial and temporal resolution characteristics together and speculating that there is a one to one relationship between resolution and bias. This appears to be another attempt to use a fixed value of resolution to justify overly broad conclusions. There is no analysis or test provided by van der Beek et al. to show that any of this speculation is true. Herman et al. (2013) tested for this problem by running inversions with different priors and found no effect, thereby demonstrating that this argument is false.

The test by Herman et al. (2013) to explore the effect of the prior on the global compilation is an interesting counterexample to the argument by Willett et al. that most of the lower resolution results are affected by this reversion to the prior. Herman et al. (2013) reported that when choosing a prior erosion rate of 0.7 or 1.0 mm/yr (the latter of which is higher than many of the inferred erosion rates in the 2-0 Ma time bin), they still see a predominance of accelerations in their inversions. Whereas Herman et al. (2013) used this result to argue for the robustness of their conclusions, we argue instead that it illustrates that Herman et al.'s (2013) results are not dominated by a Bayesian prior bias, but rather they are predominantly affected by a spatial correlation bias, which creates spurious accelerations, but is less affected by the choice of the prior.

Our synthetic tests (Figs. 1 and 2) show that bias to the prior exists, but in most cases, it is overpowered by the spatial correlation bias. The fact that the results from Herman et al. (2013) are insensitive to the prior erosion rate is not a sign of robustness of the results, but rather the pervasiveness of the spatial correlation bias.

The synthetic tests (Figs 1 and 2 of van der Beek et al.) have large data errors and should not be considered valid (AC2).

The final statement (also on line 247 of comment) demonstrates the stretched logic of the van der Beek et al. analysis: they consider two possibilities: (1) Herman et al. are in error due to bias to the prior, or (2) Herman et al. are in error due to spatial correlation bias. Elimination of (1) is therefore proof of (2). A third option, that Herman et al. are correct, is simply not considered, and therefore is not tested. This is not a valid analysis.

**3 Biased post-processing operator and resolution**

Willett et al.'s insistence that it is the post-processing operator, i.e. plotting normalized erosion-rate differences, rather than

the inversion that creates the bias, is difficult to understand. Increased erosion rates through time appear as such, whether they are visualized from direct comparison of time-bin maps or plotted as differences, ratios, or normalized differences. The magnitude of the metric changes in each case, but the sign (positive or negative) does not. In contrast to the analysis of Herman et al. (2013), the interpretations of Schildgen et al. (2018) focused on the sign of the change, not its magnitude, and on whether that change is reasonable considering (1) the spatial and temporal (age) distribution of the data, and (2) previous, more detailed analyses that include independent geologic data and more appropriate modelling of the data (e.g., with changing topography and/or lateral components of rock advection, and exclusion of data unrelated to exhumation). The ratios of erosion rates used by Herman et al. (2013), namely the erosion rate from the more recent time bin divided by that of the earlier time bin, are problematic, as the values tend to blow up when the earlier erosion rate is small. Although the normalized difference used by Schildgen et al. (2018) is arguably also imperfect, it has the benefit of tracking fractional changes in exhumation rates (i.e., a change from 0.5 to 1.0 mm/yr or from 0.05 to 0.1 mm/yr both result in a value of 0.5), and being symmetric (i.e., changes in erosion rates from 1.0 to 0.5 mm/yr or from 0.5 to 1.0 mm/yr yield respective normalized differences of -0.5 and 0.5).

We don't understand where van der Beek et al come up with this statement that Herman et al focussed on the magnitude of the signal, whereas Schildgen et al. focussed on the sign. Herman et al. quantified their results, but summarized them as showing an "increase in mountain erosion rates since about 6 Myr ago, by nearly a factor of two for the Pleistocene compared to the Plocene. " (Herman et al., 2013, page 3). "Nearly a factor of two" is hardly a quantitative estimate, because we recognized that what was important was the sign, and that the area where Herman et al resolved erosion rates was small. Herman et al. did nothing further with this number ("nearly two"). Herman et al concluded that these regions experienced an increase in erosion rate, not an increase in erosion rate of precisely some value. Their conclusions would be unchanged if it were smaller or larger, provided it remains above 1, i.e. the direction of change is unchanged. We would thus argue that Herman et al. also focussed on sign. We will add a new paragraph to our paper summarizing the conclusions of Herman et al., since several reviewers also questioned what these conclusions were.

One cannot modify a quantity with a non-linear operator and not track the error through the calculation. With a linear operator, this is simple and things like thresholds map directly, but division is not a linear operator. This is basic error analysis.

Importantly, much of the Schildgen et al analysis was an interpretation of the normalized difference maps. The assessment of the Herman et al. inversion results, including all the Schildgen et al. statements regarding the presence of spatial correlation bias was based on visual inspection of these maps. Schildgen et al. contains no spatial analysis of the Herman et al. (2-13) results; it has only an interpretation and that interpretation is based on spatial patterns. There are numerous examples where the interpretation directly refers to the location of the maximum values of NR. If they used other aspects of the Herman et al analysis, they give no indication, but they did publish twenty-some normalized ratio maps and no other portrayal of the Herman et al. results such as resolution maps or posterior variance. This undoubtedly gave a distorted view that impacted interpretation. This is easily seen from the figures of our paper. Consider Figure 17 as an example. Schildgen et al. took the results of Figures 17 a-i, reduced them to Figure 17j, omitted all resolution and variance information, taking only a single contour of the continuous resolution map as a binary quality criterion and made an interpretation. As a summary of the information in Figures 17 a-i, Figure 17j is neither complete, nor accurate, and omitting the information of Figure 17k makes it worse as it leaves the impression of largely uniform values of NR across the region with no measure of relative guality. How much did this influence Schildgen et al.s interpretation? We don't know, because an interpretation is not reproducible, but the examples we gave in our paper (section 5.3) give much circumstantial evidence that they made direct interpretation of the exaggerated regions of these maps, at the expense of the well-resolved parts. We can check our text that we have not overstated this, but we don't think it is unfair to make an interpretation of their analysis. We note that their "proof" of the existence of a spatial correlation bias, as described in their paper, was given exclusively as an interpretation of these maps.

We note that Willett et al. (1) make a mistake in their description of the normalised difference NR (their Eq. 14 returns a negative number in case of an acceleration); and (2), more critically, in their analysis of the resolution of NR, they modify the definition of NR to maximise its value and minimise its resolution by dividing by e2 instead of max(e1, e2), as done by Schildgen et al. (2018) (line 467 of Willett et al.; note that in their definition e1 > e2 in case of an acceleration). By altering the definition of the NR in their error analysis, Willett et al. are analysing a metric that mixes Herman et al.'s (2013) original ratio and Schildgen et al.'s (2018) normalised difference.

Equation 14 is reversed in definition of e1 and e2; this is a typo in the equation, not a difference in the calculation and we will correct the equation. All our figures have the correct sign. Thank you for drawing this to our attention.

The second "error" is not an error or modification of the definition. We state on line 438 that we assume that e2 is the larger quantity for the variance analysis between lines 438 and 480, because we did not want to double the number of equations we would need to write, as one needs to do with use of the max{} function. This is explained again on line 467, where we indicate that we "drop the absolute value for simplicity" This does not affect the generality of the analysis, just the brevity. For the results shown in the paper, both NR and the variance of NR use the definition as in Schildgen et al.

The absolute differences now advocated by Willett et al. are problematic in their own right, as areas with high erosion rates tend to dominate any global "signal". Moreover, Willett et al. report mean values from their histograms of erosion-rate differences, neglecting to take into account how the dominance of extreme values is exacerbated by the use of the mean instead of the median as a measure of central tendency in these positively skewed distributions (their Fig. 24). For example,

the median difference for a resolution cutoff of 0.5 is 0.42 mm/yr, whereas the mean is 0.65 mm/yr (Fig. 24c in Willett et al.). We can further illustrate the problem with absolute differences by considering the impact of removing one region of rapid erosion rates from the compilation. Again using a resolution cut-off of 0.5, after excluding results from Taiwan, where extreme reported increases in erosion rates are related to the inclusion of unreset thermochronometer data in the inversion (Schildgen et al., 2018), the mean drops from 0.65 to 0.37 mm/yr and the median drops from 0.42 to 0.39 mm/yr. However, these numbers still cannot be taken at face value, as the results still suffer from spatial correlation biases, reversion-to-prior biases, model error, and operator error.

We don't see the significance of these comments. There are no consequences of using the mean or the median as nothing is done with this number in either Herman et al. (2013) or esurf-2020-59. The difference between them is also not important for any of the issues under discussion or for the conclusions of Herman et al. (2013) or Esurf-2020-59.

The thermochronometry community of Taiwan have been discussing for years which ages are reset and which are not, with no resolution of this question. If Van der Beek et al have finally solved this problem, it would be considerate if they reported this to this community along with an explanation for how they determined it.

The last statement is undemonstrated speculation.

Oddly, Willett et al. claim that the analysis of Schildgen et al. (2018) was focused on areas of poorly resolved results and that Schildgen et al. (2018) "never address resolution". We reiterate from Schildgen et al. (2018) that the analysed results were those that passed the threshold defined by Herman et al. (2013) as "well resolved", i.e., with a resolution > 0.25, and were used by these authors to support a "worldwide increase in erosion rates". Willett et al. show a welcome new appreciation for the importance of better resolved results (lines 1000-1003), but the repeated suggestion that Schildgen et al. (2018) misdirected attention to areas of poorly resolved results is unfounded.

Within each area with resolution greater than 0.25, there is a range of resolutions, from better to worse. Schildgen et al. report only a single interpretation per site (Figure ED2, Table 1), so if they made an assessment of well-resolved and poorly-resolved areas within each site for values > 0.25, they did not report it. They lumped them into one outcome. We gave several examples where Schildgen et al, (2018) (see quotes in section 5.3) use the values from the relatively poorly resolved periphery to characterize all the results at a site. Furthermore, We don't know what Schildgen et al. (2018) used in their interpretation as it is an irreproducible interpretation, not a quantitative analysis, so we have tried to limit our criticism to those cases where they explicitly described the signal on the periphery of the resolved region as evidence for spatial correlation bias, and where the argument would be difficult to make had they focused on the best-resolved regions. They show twenty-some maps of normalized erosion rate. None of these include the resolution information, only a threshold cut-off. All regions with resolution above 0.25 appear to be treated equally in their interpretation.

The spurious increases documented in Schildgen et al. (2018) and here (Figs. 2 – 4) comprise from best to least resolved areas, and all are above the "well resolved" threshold of 0.25 used in Herman et al. (2013). Nevertheless, we agree that focusing on more highly resolved results is better practice, although doing so does not eliminate spurious increases (Schildgen et al., 2018 and Section 2 above). The implications of the selected cut-off resolution value are substantial: as noted in Schildgen et al. (2018), increasing the resolution threshold to 0.5, which characterizes the "well resolved" region in the Alps that Willett et al. prefer to focus on, would eliminate 90% of the "resolved" erosion ratios reported by Herman et al. (2013), and would comprise

data from only seven distinct regions, as shown in Willett et al.'s Fig. 25. One of those regions (central Himalaya) comprises a single resolved point that shows a decrease in exhumation rates through time, not an increase; five regions (Wasatch Mountains, Western Alps, Northern Apennines, Taiwan, Fiordland) suffer from spatial correlation bias, model or operator errors and sometimes a combination of these, as discussed in the Supplementary Information of Schildgen et al. (2018); in three regions (Coast Mountains of British Columbia, External Massifs of Western Alps, Fiordland) glacial valley incision has been previously inferred (Shuster et al., 2005; 2011; Valla et al., 2011; see below). Restriction of the results to these few locations precludes any attempts at generalization to a global scale.

We don't see what objective criterion is being used to establish what constitutes a "global scale". In fact, Herman et al. (2013) and esurf-2020-59 make no claims as the "global" significance of their results. "Global applies to the distribution of the data, and does not suggest that the increase is applicable to every point on the globe. This is stated in Herman et al. (2013), but we will add a section to ESURF 202-59 explaining the conclusions of Herman et al., as this has come up a number of times. Results are presented for what they are. If there are only seven regions, there are only seven regions. We present the results and readers can determine how significant they are. We don't see by what criteria van der Beek et al can judge and declare how many sites are enough; this is subjective.

Furthermore, the hypothesis of Schildgen et al. is not: "Herman et al included too much area in their analysis". It is "Herman et al.s method has a spatial correlation bias, such that all their results are wrong". If it were the first, we would not be writing this paper. Van der Beek et al. seem to be adding this argument as backup to their actual hypothesis.

Note that claims regarding Wasatch Mountains, Alps, Apennines, Taiwan and Fiordland are unsupported by any models or hypothesis testing (see AC6).

**4 Spatial variability of thermochronometer ages**

Willett et al. expend considerable effort in critiquing cartoons by Schildgen et al. (2018) that have no vertical or age scale, and only a rudimentary horizontal scale (lines 793-831; 1320-1322 in Willett et al.). We do not see much merit in this discussion, but note that Willett et al. appear to confuse isotherms (surfaces of equal temperature, which are sketched into the cartoons of their Figure 16 a-c) and isochrones (surfaces of equal thermochronologic age, which were drawn in the cartoons of Schildgen et al. (2018), reproduced in Willett et al.'s Figure 16 d-f).

There is no confusion *on our part* regarding isotherms or isochrons. We are drawing isochrones as we state in our paper. van der Beek et al. seem to think we are drawing isotherms because, even after we have pointed out their errors, they have not returned to the original literature to see what the original models show and how they differ from what is portrayed in Schildgen et al Figure 1. Our isochrones are consistent with the literature cited in our paper, whereas we don't know the origin of the Schildgen et al. (2018) versions. The attribution in Schildgen et al. was imprecise; they write only "discussed in the main text or supplement", and we suspect that there is no source, as we know the literature in this field very well and are not aware of examples as shown in their figure. There are two points in this comparison. First, many tectonic settings are characterized by spatially constant erosion rates, not gradients. Second, the underlying hypothesis of Schildgen et al. is that they could "test" the results of Herman et al. (2013) by a careful reading and analysis of the literature, yet they were unable to accurately summarize even the simple models of their Figure 1 without introducing modifications sympathetic to their arguments. For example, in a separate publication in the same year, Schildgen and van der Beek (Fission track thermochronology, Chapter 19, Malusa and Fitzgerald, eds., 2018) published a wedge kinematic model taken from Willett and Brandon (2002), and it is portrayed accurately. The equivalent wedge cartoon in Schildgen et al. (Nature, 2018) is represented differently, with zones of constant age shown by Willett and Brandon replaced by continuous gradients in age. If confirmation bias comes into even cartoon representations of kinematic models, it is also likely to come into the more unconstrained interpretations of natural settings.

This also serves as a good demonstration that thermo-kinematic models are not inherently better than other analysis methods. The kinematic model that (Schildgen et I. Figure 1b) envision for a thrust ramp was implemented in their test case of Figure ED5. This kinematic model includes normal sense shear across the entire thrust ramp. We have never seen observations or models suggesting such a deformation mode, primarily because it fails to balance cross-sectional area. If used to predict thermochronometric ages, it would give an answer, but an incorrect one. Yet Schildgen et al. (2018) throughout their paper treat thermo-kinematic models as unquestionably correct, thereby "testing" the Herman et al. results, whereas they are simply another model, and are subject to serious errors in cases where the practitioner has misconceptions regarding, for example, ramp kinematics or geotherms.

The point that Willett et al. appear to be trying to make is that thermochronologic ages will be spatially constant over length scales larger than the correlation lengthscale used in the GLIDE inversions in most tectonic settings. To assess this point, we illustrate some original data that inspired the cartoons (Fig. 6). For the Wasatch Mountains (Fig. 6a), the AHe and ZFT ages shown in Ehlers et al. (2003) increase steadily with distance from the range-bounding Wasatch Fault, whereas AFT ages show only a slight increase until 17 km from the fault, where they start increasing more rapidly with distance. Likewise, in the Southern Alps of New Zealand (Fig. 6b), all thermochronometer ages increase rapidly with distance from the Alpine Fault, starting from distances of 10-20 km from the fault (e.g., Herman et al., 2009). Importantly, in both these cases, the ages show significant variation over length scales of ~30 km, which was the correlation length scale used in Herman et al. (2013), making these settings prime examples of where that inversion was affected by the spatial correlation bias.

We don't see the significance of showing these figures. The Wasatch data are exactly as we describe them; once corrected for elevation (not done by van der Beek et al) there is almost no gradient over the region with double dating. The point about the Alpine Fault data was not that there is no gradient, but that it is not a simple thrust ramp. All studies that could fit the thermochronometry data required more complex kinematics. The final statement that these are "affected by the spatial correlation bias" is hypothesis without testing (see AC6).

**5 Comments on natural examples**

**320 5.1 European Alps**

Willett et al. focus much of their analysis and discussion on the European Alps, presumably because, as they claim, the Alps "play an important role in the study of Schildgen et al. (2018)" (line 863). We consider the European Alps to be no more important than any of the other 32 locations where Herman et al. (2013) reported resolved changes in late-Cenozoic erosion rates. Schildgen et al. (2018) opted to highlight the Alps as one of three examples in the main text due to the very high density of available data, which gives GLIDE the best chance of performing well. Although Willett et al. have gone to great lengths to demonstrate the reality of the erosion-rate increase in the western external Alps, they neglect to consider that increased erosion in the External Crystalline Massifs, to the north and west of the Penninic thrust front, is limited to localized valley incision, as has been demonstrated with both apatite 4He/3He studies (Valla et al., 2011) and detailed thermo-kinematic modelling that includes a temporally evolving topography (Glotzbach et al., 2011b). The latter study also demonstrated how, when steady-state topography was assumed, the pattern could be mistaken for a generalized increase in exhumation rate, as happens in the GLIDE inversions. This difference is not trivial, as presuming a regional increase in exhumation rather than localized valley incision has major implications for sediment flux to the oceans, carbon cycle impacts, and landscape evolution.

This comment raises a point that we did not discuss in our paper, the significance of incised valleys, but as van der Beek et al mention it at several points in their comment (lines 301, 328, 378) it is perhaps more important than we originally assessed and is worth adding a discussion to our paper.

This point is important because van der Beek et al. and Schildgen et al (2018) have the relevance of this observation effectively backwards. Mountainous regions experiencing an increase in erosion rate often respond by initial incision of primary river valleys as the largest rivers respond to the increase in local base level fall or faster incision with increased river discharge. If erosion rates increase due to valley glacier incision, this can be even more pronounced. Deeply incised valleys are thus evidence for a recent increase in erosion rate. The surrounding hillslopes, ridges, or lower-order river valleys might not have the same increase in erosion rate, given that most landscape response times are millions to tens of millions of years, but eventually they catch up to valley incision. Thermochronometric data from the valleys will detect the increase in erosion rate and, depending on topographic form and data distribution, will measure the valley incision rate, including its increase. However, they may or may not detect the lack of change on surrounding high elevation regions that have not yet had time to respond, which is what Glotzbach et al. (2011) demonstrated. The regional mean erosion rate is the average of the erosion rate in valleys and surrounding ridges, so if the ridge erosion rate remains constant and the valleys have accelerated erosion, the average erosion rate of the region has also increased. The thermochronometric ages reflect this increase in erosion rate and accurately measure the increase in rate of the valleys, although the estimate of the regional mean change (including valleys and ridges) might be overestimated. The Glide analysis of Herman et al. (2013) accurately shows this increase in erosion rate, and the conclusions of Herman et al (2013) that this is a region that has had accelerated erosion rate is correct. If there is a potential overestimation of the regional rate, this would still not change the sign of the change, which is an increase in erosion rate.

In spite of this, van der Beek et al. and Schildgen et al. (2018) argue that the existence of deep valleys refutes the Herman et al. (2013) results and that the existence of these valleys implies that the thermochron-derived increase is "spurious" in areas including the Alps, the Olympics, Patagonia and Fjordland, New Zealand (Schildgen et al., 2018, Table 1). Their argument seems to be that because the volume of sediment removed through valley incision is smaller than for spatially uniform erosion, the result is somehow invalid. We see no justification for this argument. Herman et al. made only a qualitative comparison to sediment flux, but made no argument that the raw numbers coming out of the inversion were representative of either global or regional sediment fluxes (AC3). The important conclusions of Herman et al. were that erosion rates in mountainous regions increased and that the increase was measurable by thermochronometric data. With deep valley incision, we not only have the high erosion rates and high relief needed for thermochronometry to accurately measure changes in valley erosion rate, we also have the incidental geomorphic evidence for a recent increase in erosion rate (i.e. deep valleys, less eroded high topography). In other words, we have the needed conditions to resolve an erosion rate increase and independent evidence that such an increase has occurred. Schildgen et al.'s (2018) assessment of "spurious" is based entirely on a hypothetical misuse of the Herman et al. results by some future studies that might interpret these results in terms of sediment volume. The thermochronometric data, and the geomorphology support an increase in erosion rate. Schildgen et al's (2018) analysis finds support for an increase in erosion rate (recall that they claim their analysis is focussed only on sign). Herman et al. (2013) find an increase in erosion rate. And yet these cases are classified as "spurious". This highlights another case where Schildgen et al.'s (2018) summary Table 1 and Supplement Figure 2 have attached the label of "spurious" to sites and analyses for such disparate reasons as to render the categorization meaningless.

Incidentally, on lines 939-940, Willett et al. state "Finally, we address the geologic evidence of Schildgen et al. (2018)'s hypothesis that the external and internal Alps are separated by an active normal fault along the Penninic Line." Although Willett et al. delve into considerable detail in the following paragraph to argue against this hypothesis, Schildgen et al. (2018) never suggested this. Willett et al. appear to have misunderstood the aim of the synthetic test presented in Schildgen et al. (2018): it was used simply to test for the occurrence of a spatial correlation bias across a densely sampled and strong gradient in thermochronologic ages, not to attempt a realistic simulation of the European Alps.

The part about the Pennenic Line as a normal fault is partially correct; we were in part responding to conference talks and even the pre-review versions of the Schildgen et al (2018) paper which we saw and not only suggested, but explicitly stated that the Penninic Line was a normal fault. This material seems to have been omitted from the final published version of Schildgen et al., so we will check that we are not commenting on statements no longer made by the authors. However, the main reason this issue is important is because Schildgen et al. in both model and interpretation of the data assume that there is a sharp boundary between the external and internal Alps. There is no support for this, with multiple sub-5 Ma ages south of the fault, and geochronologic evidence that the Penninic Line has not been active since the mid-Miocene. Observed ages can easily be fit by a broad doming uplift of the Alps, not a vertical fault. This is one reason why the ages in their "Alps model" is very different from the real Alps.

The aim of the synthetic test has been discussed in AC2 and AC5. van der Beek et al (lines 166-182) go into great detail as to how our model was constructed incorrectly because the synthetic data differ from that of the Alps, and now it is claimed this was not meant to be a realistic simulation of the Alps. Both cannot be true. In any case, we understand perfectly the point of any synthetic data models and have extensive discussion in our paper in the introduction to section 4. The long introduction to our modeling section is in part because Schildgen et al. did not define the goals for their models, and so did not differentiate between model errors, resolution errors, bias to the prior or spatial correlation bias so their models were were not designed, nor sufficient, to test for any specific errors, and we want to be careful not to repeat this error.

**5.2 Nanga Parbat**

It is difficult to assess the inversions Willett et al. present for Nanga Parbat, because they do not discriminate between ages inside and outside the massif. All apatite and zircon fission-track ages within the massif are < 3.4 Ma (Treloar et al., 2000; Zeitler, 1985; Zeitler et al., 1982), and all such ages outside the massif are > 3 Ma (see the Supplementary Information of Schildgen et al., 2018). Willett et al. now add mica 40Ar/39Ar data, but it is unclear if Herman et al. (2013) used those data in their inversions. Herman et al. (2013) did not report any such data, and the resolution values they report for Nanga Parbat are considerably lower than the resolution values shown by Willett et al. (e.g., maximum resolution in the 6-4 Ma time bin of 0.4, rather than ca. 0.6 shown in Willett et al.; and maximum values in the 4-2 Ma time bin of 0.5, rather than ca. 0.7 shown in Willett et al.), implying that the inversion presented in Herman et al. (2013) used less data. Mica 40Ar/39Ar data from the core of the massif are  $\leq 4$  Ma, with one exception along the Indus River (Treloar et al., 2000; Zeitler et al., 2001); older ages

are only encountered within the massif-bounding shear zones. Mica 40Ar/39Ar ages outside the massif are > 10 Ma without exception, and they are mostly > 20 Ma. Moreover, careful interpretation of mica 40Ar/39Ar data from Nanga Parbat is required, as excess Ar is a commonly reported problem, and several reported ages are crystallisation ages rather than cooling ages (Schneider et al., 2001). If any of these complications in the data were considered by Willet et al., they are not reported, raising the possibility of operator error.

Despite there being no information from in-situ data within the massif prior to 4 Ma, and there being no ages

---

## Author Comment (AC8) · 17 Dec 2020

Reply to comment esurf-2020-59-SC1

Although our paper (ESURF-2020-59) presents a number of models to illustrate the various errors in the GLIDE inversion, including resolution errors, model errors and their manifestations as spatial correlation bias or bias to the prior, van der Beek et al argue that these models are not adequate to provide a conclusive result and therefore offer their own examples (esurf-2020-59-SC1, Figures 1 and 2). However, these models were constructed using synthetic ages calculated using an incorrect geotherm, inconsistent with the thermal model included in the GLIDE model. We argued that

this error is large enough to invalidate these models. Van der Beek et al. disagreed, arguing that the difference in boundary conditions was too small to account for the errors observed in their models (esurf-2020-59-SC1, lines 99-116). Relying on this contention, they presented an experiment in which they compared spatially uniform to spatially non-uniform data, where the difference was argued to illustrate the magnitude of the spatial correlation bias. From their comment, line 139:

COMMENT: "We illustrate the impact of the Bayesian prior bias alone with inversions that include data only from the NW side of the fault (Fig. 1 a-i) or only from the SE side (Fig. 1 j-l). As discussed above, when including only data from the rapidly but constantly exhuming NW side, the model returns constant rates or minor decelerations since 6 Ma (Fig. 1 a-i). The absolute rates depend on the employed prior, especially toward the edges of the model, where the resolution is lowest. A similar result (no acceleration) is obtained when only inverting data from the more slowly exhuming SE side of the fault, although in that scenario, the low resolution (< 0.25 everywhere) implies that both the 6-4 Ma and 2-0 Ma time bins largely revert to the prior erosion rate (Fig. 1 j-l). We illustrate the combined effects of the Bayesian prior bias and the spatial correlation bias for the synthetic western Alps case by including data from both sides of the fault and varying the uniform prior erosion rate (Fig. 2). These tests show that spatially variable exhumation rates add substantial bias to the inversion results (compare Fig. 1 and. Fig. 2). The bias in Fig. 2 contains elements of the Bayesian prior bias; the two biases are impossible to disentangle in areas of spatially variable exhumation. The compounded effects of both, nevertheless, are clear: accelerations occur when data from both sides of the fault are included in the inversion, regardless of the chosen prior erosion rate, suggesting a predominance of the spatial correlation bias. Furthermore, we see that by running an inversion with data from both sides of the fault, the resolution is higher on both sides compared to the resolution found when using data from only the NW or the SE (compare resolution 5 contours in Figs. 1 and 2). Thus, data are being combined from both sides to "better" constrain the exhumation history of each side, and that more highly resolved result produces the spurious increase we call the

spatial correlation bias."

Although our models with correctly-calculated ages show little of the error they suggest is a spatial averaging effect, we thought it useful and more conclusive to reproduce their model juxtaposing the full data set and the half data set, but using ages calculated with the correct geotherm. Figure 1 below shows a reproduction of this experiment. There are three models in this experiment. All use a new data set that will be described in our revised paper, but is sparser than our data set A (esurf-2020-59, Fig. 5), but with data more uniformly distributed in space. The uniform spatial distribution is needed to distinguish between spatial averaging, which is maximum near the fault, from temporal averaging which is more uniform in space. A constant geothermal gradient model is used to eliminate geotherm model errors. The erosion rate function is identical to the Alpine model of Schildgen et al. (2018) and the van der Beek et al comment. We show all time steps from 0 Ma to 16 Ma to avoid the criticism that we provide preferential reference to specific timesteps. Ages in the low-erosion rate region go back to 30 Ma. The experiment compares an inversion using only the age data from the fast-uplifting side of the fault (NW) (Model 3) to models with data from both sides of the fault (Models 1, 2), with the argument that the difference between the two must be due to spatial averaging. The full data set models are shown in the left four columns. Two prior erosion rates are used, 0.35 mm/yr (Model 1) and 1.0 mm/yr (Model 2). Resolution and reduced variance do not depend on the prior erosion rate, so are applicable to both Model 1 and 2. The reduced data model (Model 3), using only the ages from the high erosion rate zone, is shown in the right three columns. We do not show the corresponding model with a prior of 1.0 mm/yr because it returns exactly and uniformly the true erosion rate of 1.0 mm/yr.

Models 1 and 2 are very similar to our models in esurf-2020-59, for example, Figures 6 and 9, but here are illustrated for a longer timespan, covering 8 time intervals. The high erosion rate region is well-resolved and accurate from 6 Ma to 0 Ma. At earlier times, there are significant errors in the model with the low prior, but very little error in

the model with the high prior. This indicates that errors are a consequence of low resolution and bias to the prior, as confirmed by the low values of resolution, little reduction of the variance, and the sparsity of ages falling into or before these timesteps. The low-erosion rate region has low resolution everywhere, but is still reasonably accurate across all time intervals, as a consequence of having many old ages, which provide an integral constraint on the erosion rate.

Comparison between these models can be used to distinguish between low resolution errors with bias to the prior, and spatial correlation bias. Comparison of Model 1 and Model 2 demonstrates sensitivity to the prior model and thus resolution errors. Comparison between Models 1 and 3 shows spatial correlation errors. The solutions for the high-erosion rate region of Models 1 and 3 are very similar. The same first three time intervals are well-resolved and accurate. Solution accuracy begins to deteriorate at earlier timesteps, but both erosion rate and resolution in these two models are almost indistinguishable. There are differences, but these are limited to the immediate proximity of the fault (less than one correlation length). The region affected by spatial smoothing is indicated by red dashed lines in time interval 10-8 Ma. As in other models of our paper, the spatial correlation errors are largest where there are no data near the fault, indicating that the spatial correlation is not averaging age data so much as interpolating empty space between the data. Some spatial averaging error is there, but is small and limited to a smoothing effect directly across the fault. This error is also visible in the high prior model, again limited to immediate vicinity of the fault. There are much larger errors in the low erosion rate region, but these are resolution errors in Model 3, introduced by the removal of all data from this region.

These models are very different from the models shown in Figures 1 and 2 of the comment of van der Beek et al. Although the data sets are not equivalent in resolution (this is impossible to do), the data differences in terms of number and location of ages are not significant. The primary difference is that the van der Beek data contain errors due to the incorrect geotherm calculation. van der Beek et al. argued that this error is

not significant, but this is clearly not the case. Their models contain large errors that cover most of their model domain on both sides of the fault, even for the case of ages restricted to the high erosion rate domain. We do not reproduce these errors. The errors that we do observe are mostly resolution errors and correspond well with low values of the resolution metric, and the small smoothing effect across the fault. The difference in result is conclusive; the models in Schildgen et al. (2018) and van der Beek et al. (esurf-2020-59-SC1) are dominated by their geotherm error and cannot be interpreted as demonstrating any other effect such as the spatial correlation bias. The correct quantification of "spatial correlation bias" is shown by the difference between columns 1 and 5 in our Figure 1, and is small and local to the fault.

This figure will be included in our revised paper, although to avoid lengthening the paper, we will put it in a supplement.
* * *
[Figure]

Fig. 1. Glide Models comparing data only from high uplift area and all data

[Figure]

**Fig. 2.** Fig.1 continued

---

## Author Comment (AC9) · 6 Jan 2021

This manuscript seems to have been written as a detailed rebuttal of the conclusions of the paper by Schildgen et al. (2018) which questioned the accuracy of the linearised inversion method of Fox et al. (2014) of thermochronological data. Specifically Schildgen et al. (2018) suggested that the methodology encouraged the incorrect interpretation of spatial variability as temporal variability, and thereby creating a systematic bias towards an apparent temporal acceleration in exhumation. No doubt the authors are motivated by the fact that the earlier paper, published in Nature I see, brings into question much published work by the present authors. My overall reaction is that the present manuscript contains some useful information in that it sets out all of the resolution and uncertainty issues associated with the method of Fox et al. (2014).

The present manuscript quantitatively describes through theory and example the sources of uncertainty and bias in the Fox et al. (2014) approach, including those arising from theory errors (due to forward modelling) data errors and the limited resolution caused by data sampling and the physics of the data-model relationship. Again this is largely in the context of combatting assertions by Schildgen et al. (2018), nevertheless I think this exposition is in principal useful to have in the literature. At the same time the manuscript is excessively long (my version is 78 pages) and an arduous and at times repetitive read in parts. I question how many readers will understand it all in its present form. There is a good argument to move a lot of this material to an appendix or supplementary material without loss of the main points. I realize that the authors are probably intent on getting this full detail into the literature, but I think having it all in the main body of a paper will certainly diminish its likely impact. I suggest the Bayesian Inference tutorial material could be moved to an Appendix or left out and the results cited as they are required. In addition do we really need to see all numerical tests? again use of appendices would help. Much of the underlying theory of Bayesian inference appears in the literature cited, but there are also many books outside of the Earth Sciences literature, admittedly two popular ones, Menke and Tarantola are cited here and so perhaps better use of specific results in those books could be used here.

RESPONSE: This is a fair assessment of both content and motivation. We will give this section of the paper a hard edit. We will shift some of the models to an appendix, but prefer not to move the theory out of the main paper as it is specific to the Fox et al methodology and therefore essential to the points of the paper. Also, it isn't that long. We have consolidated some of the models, for example combining 3 figures into 1, where the resolution and variance plots were identical. We are less concerned with readability and impact as thoroughness. As any perusal of the comments shows,

there is a great deal of misunderstanding and lack of familiarity with concepts that are perhaps well established in geophysics, but not in thermochronometry. If we were writing for a different community, we would not include the inference description or discussion of error sources, but for this paper and this target audience we prefer to lose casual readers put off by the length, but help interested readers understand better how parameter inference is used and what the various analysis tools are, in order to better understand the outcome, how to interpret it and to avoid the sort of "black-box" approach we too often see with modeling tools that enter the community without full awareness of limitations.

Some of the writing is clumsy, an example is on line 251 '...an inversion scheme based on inverse problem theory which minimizes Gaussian errors in observations, and a Bayesian parameter model in which model parameters are assumed to have Gaussian distributions about a prior value'. Some more conciseness would be a help here and in many other places. Note that in Bayesian inference we don't actually 'Minimize Gaussian errors in observations' rather a Likelihood function $p(d \mid m)$ expresses the probability of observing the data given a model. It may or may not take a Gaussian form, indicating all departures from fitting the data perfectly are expected to be described by a Gaussian PDF. Its maximization occurs in Maximum Likelihood techniques not Bayesian Inference. One might consider Maximizing the Posterior PDF as one way of characterising that PDF but this differs from Likelihood maximisation when the prior is not uniform. Of course when the data model relationship is linear and Likelihood and Prior are Gaussian then the posterior PDF is also Gaussian.

RESPONSE: We clarified the text at this point. We have developed the method, largely assuming Gaussian distributions where maximum likelihood methods and Bayesian Methods are effectively identical, but it is correct that this is not true in general and we should state it correctly.

While I have not studied the Fox et al. (2014) paper in any detail it is clear from the account here that it is a linearised Bayesian inversion method combining a prior with

a Likelihood to estimate a Gaussian approximated posterior probability distribution. This approach is well known and understood, especially in the geophysical literature for many decades. The works of Tarantola and Valette being a prime embodiment of the approach. The present manuscript gives a summary of the linearised theoretical results as they pertain to quantification of resolution and model covariance matrices and their interpretation. It appears to me to be a competent summary of linearised inversion within a Bayesian framework, my comments above notwithstanding. I do feel that some references to the latest editions of some well established books such as Menke (2012, rather than 1984) or Aster, Borchers and Thurber (2018, 3rd edition) could assist the reader in this regard, as all of the relevant theory is set out there. The paper references relevant to linear discrete parameter estimation, or which this is an example, are, correctly, to the original works from the 1970s-1980s which is good to See.

RESPONSE: We have updated some references, including the Menke book.

From L430 onward the authors' analyse the Normalised erosion rate metric, NR, eqn. (14) (lines 425-475). This is a nonlinear function of the model parameters recovered in an inversion. I note that there is a detailed critique of this term, as it was important to Schildgen et al. (2018) to support some of their arguments. A discussion is given of Expectation of a ratio (Line 440). This expectation identity relies on the terms x and x being independent which they are clearly not in eqn. (14), since both involve e1 and e2. There must be a cross term affecting the Expectation of theFrom L430 onward ratio which is ignored here. This is an error.

RESPONSE: This is a good point - we concur. In fact this paragraph is not really needed as we don't use this result for anything, so we will simply remove it as part of the shortening and editing process.

The authors continue on and obtain an approximation of the variance of NR using a Taylor expansion, as well as discussing its bias and its complicated behaviour. Quantities such as variance and expected value of some property of the model parameters while analytical, are just moments of what is in reality a probability distribution over NR constrained by the data. Since the point of Bayesian inversion is to determine a Posterior probability distribution of the model parameters then by drawing from that (Gaussian) distribution and calculating NR provides a posterior ensemble of NR values from which a complete (non Gaussian) PDF can be inferred. A better (more complete) approach would be to map out that PDF in full rather than drawing inferences from estimates such as variance, which are at best only partially characterise its behaviour. How well can a variance describe a high skewed distribution anyway? As I understand it this is the objective of the authors, i.e. trying to quantify the uncertainty and bias in the NR term, in which case it is better to examine the actual distribution, if its accessible, and I think here it must surely be. The PDFs give the full picture, low order moments such as variance and expectation do not.

RESPONSE: We agree that this could be done - it would require numerical sampling of posterior distributions given the covariances, and would be limited in that we have only Gaussian posteriors, i.e. only moments, as the inversion is analytical, but it would solve the inverse Gausssian distribution problems. However, the purpose of this section is mostly to argue that one should not use ratio quantities such as the NR. We hope that the case is made and we will not see this quantity in the literature again, in which case a more sophisticated analysis of the error is not needed, so we prefer to not put in additional work for an analysis tool that we hope will now disappear.

The authors are correct in their assertions (L1305-1310) that the Bayesian solution will tend to go to the prior when data influence is poor, which means have large data covariances. This is well known and a fundamental property of Bayesian inference. However it is not the case that the posterior will tend to the prior when data, or averaged data (in this case), are inconsistent. It is not entirely clear to me that this is what the authors are suggesting, but I got the impression it might be implied by their discussion of resolution and the influence of the prior (multiple sections across the manuscript).

This seems to be a pertinent point in their rebuttal of the Schildgen results. Inconsistent data (e.g. tight constraints on a subset of model parameters which do not agree) is quite a different thing from weak data constraints (e.g. due to large data noise). Only the weak constraint case does the posterior tend to the prior, not the inconsistent data case.

A toy example makes this clear. Consider a single prior variable with Gaussian uncertainty. Imagine that new data are a direct observation of this variable. The weak data case is when measurements occur with large errors relative to the Gaussian prior. They will have little affect on the prior mean, i.e. posterior tends to the prior. The inconsistent data case is when measurements with errors comparable to the Gaussian prior are widely separated. Here the posterior mean may be far from the prior mean. As I say its not entirely clear to me whether the authors have mistook one for the other in their arguments, but even if they haven't, I think it best to explicitly clarify this difference in the manuscript so that there is no confusion in the readers mind.

RESPONSE: We agree with this point. However, we are almost always referring to the sparse data case, not the inconsistent data case. This is most important for the numerous cases of very sparse data where one wants to differentiate between bias to the prior and interpolation of very distal data. The inconsistent data case could come up, but this is more the "how are data averaged" question as discussed in our Figure 2. We will add a comment to this effect and discuss how data residuals can be used to test for data-model consistency.

How is the closure temperature estimation from Dodson (1973) applicable here? Isn't this a rather approximate method in itself? How relevant is it to low temperature thermochronological data such as fission track and He? As I understand it there is a continuous response, in terms of annealing of diffusion over a range of temperatures, which would be a significant proportion (say 50%) of the estimated closure temperature. In this case the measured age is not directly related to the time that the system became closed (i.e. started to accumulate fission tracks or He). Shouldn't one ideally solve the

full FT annealing and He diffusion equations to find when and at what depth the system became closed? This does not appear to be done here. No doubt if it were this would create a more nonlinear inversion problem which may be more difficult to handle with the linear/linearized inversion approaches used here? These issues should probably be discussed and justified.

RESPONSE: This is an important point for those cases where there is a complex cooling history, or where the first-order kinetics are a less-than-optimal approximation (fission track annealing). We present this as a potential model error (lines 405-410, line 593), and this is discussed in earlier publication (Fox et al. 2014), but to do more would require an entire new study and paper, particularly for those aspects that destroy the linearity of the GLIDE model. The same is true for the other model error, temperature, where multidimensional aspects of advection or mantle heat flow differences are not included. We expect this is even more important than the kinetics, and acknowledge this problem under model errors, but investigation would require much new work.

Ultimately any inverse model result is conditional on the assumptions (e.g. model resolution matrix depends on the assumed linear relationship between model and data). I agree that using different forward models to generate synthetic data and perform an inversion introduces unquantified 'theoretical errors' in an inversion test. Quantifying such errors however is usually difficult, especially systematic errors. The question of spatial vs temporal variability in inference results, which is at issue as one of the central arguments between the present authors' and those of Schildgen et al. (2018). From what I can tell the original work of Fox et al. collected and smoothed over data sets spatially by introducing a Gaussian prior imposing a spatial (smoothing) covariance. Much is discussed on this point here and whether this introduces bias and encourages spatial variability to be mis-interpreted as temporal variability. The present authors obviously argue not, however its not clear to me why use of global spatial prior smoothing of Fox et al. is even necessary. It seems a rather backward step in itself, compared to more sophisticated 'data adaptive' spatial smoothing approaches popular in Bayesian

methods these days. I'm thinking here of Bayesian spatial sampling known as Partition modelling which has been around for more than twenty years, See Denison et al. (2002 for a summary). That type of approach allows sharp local variations to be recovered if they are supported by the data, e.g. by sampling data across a discontinuity. This is much more preferable than simply imposing global spatial correlation via a prior. I see Partition Modelling has already been applied to in low temperature Thermo-chronology problems by Stephenson et al. (2006) almost than a decade and a half ago. This paper or any related work is not cited or discussed. In that sense I see a lot of energy here expended here on defending the validity of what appears to be a somewhat outdated framework.

RESPONSE: Other approaches are possible, and we have investigated some, including breaking correlation structure (Stalder et al., 2020). The spatial correlation approach is one of the simplest. The question of its utility or whether some new approach is needed depends somewhat on how common discrete fault offsets are in thermochronometry. Our experience is that they are surprisingly rare. Their importance is often exaggerated by the sparsity of data and the fact that there are mapped faults everywhere, so for any sampled ages that vary in space, there is a fault between them, but that does not prove that that fault was active in order to offset the ages. Any fault that moved prior to closure of thermochronometric ages is irrelevant to those ages, so most mapped faults have no effect on kinematics and age offset. This point is often missed in interpretations. The simplest test for the effect of cross-fault averaging is simply to invert suites of ages independently on each side. We did this for the Alps and in the revised manuscript, for the Nanga Parbat example. In each case there was no effect on the result, indicating no cross-fault averageing.

My impression is that the manuscript is detailed defence of the Fox et al. (2014) inversion approach which may be useful for readers to see in some form. However it is unnecessarily long and could probably start at page 35 after an introduction. A manuscript 50% shorter would be more appropriate and I expect have more impact.

RESPONSE: As discussed, we will edit the intro, but as other reviewers and commenters have indicated, there are still issues arising that need explaining and we don't think that our target audience is going to pick up a book on inverse theory and figure it out on their own. If we can provide a readable introduction for a community with less experience in parameter estimation, it could be seen as helpful and we would rather err on the side of completeness, so as to avoid the sort of criticisms that we are getting now.

Minor points:

I can see no content in eqn. (8)? - It appears blank to me.

RESPONSE: Oops - correct - we lost an equation in formatting somewhere.

References: Menke W. (2012) Geophysical Data Analysis: Discrete Inverse Theory, 3rd ed. Aster, R. Borchers B. & Thurber C. (2018). Parameter Estimation and Inverse Problems - 3rd Edition, Elsevier D.G.T. Denison, N.M. Adams, C.C. Holmes, D.J. Hand, Bayesian partition modelling, r Comput. Stat. Data Anal. 38 (4) (2002) 475– 485. Stephenson, J. and Gallagher, K. and Holmes, C. C., Low temperature thermochronology and strategies for multiple samples 2: Partition Modelling for 2D/3D distributions with discontinuites, EPSL, 2006, 241, 557-570.

---

## Author Response (AR2)

Dear Editors,

Please find attached a (hopefully) final version of our paper, esurf-2020-59.

The request for this revision was to reduce or minimize the references to the work by Schildgen and van der Beek. We have thought hard about how to do this, but do not see a way to do this fully. Our paper is part of a debate and is in fact designed to counter a paper (Schildgen et al., 2018) that was very detailed in its criticism of our earlier work. To not respond with equal specificity would greatly weaken our paper. The logic and the structure of paper are built around the points of criticism, so to show the response without explaining why we are doing this, and without making a direct comparison between approaches would only confuse readers.

We can however see the point that direct conflict and criticism can be reduced, so we have done two things:

(1) we have gone back through the paper and removed several lines or even paragraphs that were direct referrals to something in the Schildgen et al. paper. There were a number of places where this cleaned up the content.

(2) The section "6.3 Problems with Schildgen et al. ", we assume was the most problematic in the view of the AE, and we assume that it was this section that he recommended to move to a Supplement. We have considered the options and have opted to simply delete the entire section. Although a summary in a discussion is often useful, one in a Supplement is just repetitive, so we will trust that the interested reader can make the connections based on the content of the main body of the paper and we will forego the summary. Some important references were moved elsewhere.

We agree that this makes the tone of the paper less confrontational, and hope this will satisfy the concerns of the editors.

Regards,

Sean Willett (and co-authors)

---

## Author Response (AR3)

Author Response:

Following a request from the AE, we have restructured the paper, moving figures 3 and 4 to the discussion and concentrating all responses to Schildgen et al. (2018) critique to a single section in the discussion.